# Astaxanthin: Past, Present, and Future

**DOI:** 10.3390/md21100514

**Published:** 2023-09-28

**Authors:** Yasuhiro Nishida, Pernilla Christina Berg, Behnaz Shakersain, Karen Hecht, Akiko Takikawa, Ruohan Tao, Yumeka Kakuta, Chiasa Uragami, Hideki Hashimoto, Norihiko Misawa, Takashi Maoka

**Affiliations:** 1Fuji Chemical Industries, Co., Ltd., 55 Yokohoonji, Kamiich-machi, Nakaniikawa-gun, Toyama 930-0405, Japan; 2AstaReal AB, Signum, Forumvägen 14, Level 16, 131 53 Nacka, Sweden; pernilla.berg@astareal.se (P.C.B.); behnaz.shakersain@astareal.se (B.S.); 3AstaReal, Inc., 3 Terri Lane, Unit 12, Burlington, NJ 08016, USA; khecht@astarealusa.com; 4First Department of Internal Medicine, Faculty of Medicine, University of Toyama, 2630 Sugitani, Toyama 930-0194, Japan; takikawa@med.u-toyama.ac.jp; 5Graduate School of Science and Technology, Kwansei Gakuin University, 1 Gakuen-Uegahara, Sanda 669-1330, Japan; taoruohan@kwansei.ac.jp (R.T.); yumeka-kakuta@kwansei.ac.jp (Y.K.); chiasa.uragami@kwansei.ac.jp (C.U.); hideki-hassy@kwansei.ac.jp (H.H.); 6Research Institute for Bioresources and Biotechnology, Ishikawa Prefectural University, Suematsu, Nonoichi-shi 921-8836, Japan; n-misawa@ishikawa-pu.ac.jp; 7Research Institute for Production Development, 15 Shimogamo-morimoto-cho, Sakyo-ku, Kyoto 606-0805, Japan

**Keywords:** astaxanthin, microalgae, mitochondria, SDGs, anti-aging, slow-aging, commercial production

## Abstract

Astaxanthin (AX), a lipid-soluble pigment belonging to the xanthophyll carotenoids family, has recently garnered significant attention due to its unique physical properties, biochemical attributes, and physiological effects. Originally recognized primarily for its role in imparting the characteristic red-pink color to various organisms, AX is currently experiencing a surge in interest and research. The growing body of literature in this field predominantly focuses on AXs distinctive bioactivities and properties. However, the potential of algae-derived AX as a solution to various global environmental and societal challenges that threaten life on our planet has not received extensive attention. Furthermore, the historical context and the role of AX in nature, as well as its significance in diverse cultures and traditional health practices, have not been comprehensively explored in previous works. This review article embarks on a comprehensive journey through the history leading up to the present, offering insights into the discovery of AX, its chemical and physical attributes, distribution in organisms, and biosynthesis. Additionally, it delves into the intricate realm of health benefits, biofunctional characteristics, and the current market status of AX. By encompassing these multifaceted aspects, this review aims to provide readers with a more profound understanding and a robust foundation for future scientific endeavors directed at addressing societal needs for sustainable nutritional and medicinal solutions. An updated summary of AXs health benefits, its present market status, and potential future applications are also included for a well-rounded perspective.

## 1. Introduction

Astaxanthin (AX), a captivating red-orange pigment belonging to the carotenoid family, has garnered tremendous attention in recent years owing to its extraordinary physical properties, biochemical characteristics, and physiological effects. This remarkable compound has emerged as a promising contender in the realm of human health and well-being, prompting a surge in scientific research. In fact, the number of PubMed-indexed publications on astaxanthin has soared exponentially, skyrocketing from a mere 29 papers in 2001 to a staggering 414 papers in 2022, marking a fourteen-fold increase over the past decade. To date, more than 3500 papers on “astaxanthin” have been indexed in the PubMed database (Figure 1). 

The global landscape of AX research is undergoing a notable shift, with countries worldwide, including Japan, spearheading significant advancements. Initially relegated to a modest role as a coloring agent in the aquaculture and poultry industries, AX has experienced a remarkable transformation. As the fisheries, poultry, and livestock sectors underwent structural changes, the use of AX in commercial feeds expanded exponentially. Simultaneously, novel applications in human health have propelled AX to new heights of commercial significance. Intriguingly, recent studies have begun to unveil the potential of AX in addressing health challenges stemming from societal shifts. Thus, AX holds the potential to become one of the rare natural products capable of fulfilling diverse needs within human society as we forge ahead into the future.

According to available market research and forecasts, the market for AX and its end products is currently estimated to be valued between USD 647.1 million and USD 1633.7 million in 2021. Furthermore, revenue projections suggest that the market is expected to reach USD 965 million to USD 3200 million by 2026 [1,2,3]. This indicates a projected compound annual growth rate (CAGR) ranging from 8% to 16%. The rapid expansion of AX can be better comprehended by delving into its discovery, current applications, and how the latest research and market trends are shaping its future.

This review article also provides a comprehensive exploration of the historical journey up to the present behind the discovery of AX, shedding light on its chemical and physical features, its distribution in organisms and biosynthesis, and furthering its often-overlooked significance in human culture. Unlike previous publications, we delve into the rich tapestry of AXs impact on various aspects of human culture, unveiling its profound connections and implications. Furthermore, we delve into the potential contributions of AX in addressing future social and environmental challenges, underscoring its versatility and potential for positive transformation. By bridging the past, present, and future, this article offers a unique perspective on the multifaceted role of AX in shaping our world.

## 2. Nature and Cultural Aspects of Astaxanthin

### 2.1. Astaxanthin; Chemistry, History of Discovery and Structural Investigation

#### 2.1.1. Astaxanthin; Chemical Structure and its Properties

Astaxanthin (3,3′-dihydroxy-β,β-carotene-4,4′-dione; AX) is a carotenoid with a chemical formula of C_40_H_52_O_4_ and a molecular structure that includes two hydroxyl and two carbonyl groups. AX exhibits an orange to deep-red color due to the presence of 13 conjugated double bonds. It is important to note that in its crystalline form, astaxanthin takes on a glossy black-purple color. The molecular structure of AX is symmetrical, with two chiral carbons at the 3 and 3′ positions of both terminal β-ionone groups, giving rise to three possible optical isomers (stereoisomers): (3*S*,3′*S*), meso (3*R*,3′*S*), and (3*R*,3′*R*)-AX. Additionally, due to the presence of nine double bonds in the polyene moiety, there can theoretically be 512 geometric isomers. While most naturally occurring AX is in the all-trans configuration, 9-, 13-, and 15-*cis* isomers have also been identified (Figure 2).

In addition to the free form, where no hydroxyl group modifications are present, AX also occurs naturally in a form where hydroxyl groups are modified by fatty acid esters. In animals, AX may be present in a protein complex, while in bacteria, it can be found as glycosides [4,5]. The detailed roles of these processes are discussed in other sections, specifically in Section 2.1.6 for fatty acid esters, Section 2.1.8 for carotene proteins, and Section 2.2.1 for glycosides. The physical and major spectral properties of AX are described in Appendix A.

#### 2.1.2. Astaxanthin; Discovery and History of Structural Investigation

Investigations into astaxanthin (AX) began soon after the initial discovery of carotenoids. The study of carotenoids dates back to the early nineteenth century, when carotenoids were first found and extracted from paprika (in 1817), saffron (in 1818), annatto (in 1825), carrots (in 1831), and autumn leaves (in 1837) [5,6]. In those early years, carotenoid structures were still largely unknown, and their characterization was primarily based on their solubility and light absorption properties. AX seems to be the same as that initially called crustaceorubin by the British naturalist Henry Nottidge Moseley in 1877 and by Marian Isabel Newbigin in 1897 [7].

The early 20th century marked a major turning point in carotenoid analysis with the invention of chromatography, a revolutionary biochemical technique that became a staple in the chemistry of natural organic compounds. In 1906, Tswett successfully separated carotenes, xanthophylls, and chlorophylls from green leaves using column chromatography for the first time. Subsequently, the 1930s became known as the “golden age” of carotenoid structure elucidation. During this period, Karrer and Kuhn characterized eight carotenoids, including β-carotene, which they discovered to be a precursor of vitamin A. Their remarkable achievements earned them the Nobel Prize in Chemistry [8]. They also elucidated the structures of lutein, zeaxanthin, and AX [5,6].

At that time, carotenoid structural studies were conducted using elemental analysis and oxidative degradation reactions with strong oxidizers like KMnO_4_. However, these techniques were not sensitive and required several grams of carotenoids in crystalline form for analysis. In the 1970s, significant improvements in analytical instrumentation, including the introduction of various spectroscopic and separation techniques such as MS, ^1^H-NMR, ^13^C-NMR, and HPLC, revolutionized the analysis of carotenoids. These advancements made it possible to analyze smaller samples more effectively. As a result, over 600 carotenoids found in nature have been structurally elucidated [4,9].

The structure of AX was elucidated relatively early in the history of carotenoid structural determination. In 1933, Kuhn and Lederer isolated two carotenoids from the shell and eggs of the lobster species *Astacus gammarus* (now known as *Homarus gammarus*) and named them “astacin” (now known as astacene) and “ovoester” [10]. In 1937, Stern and Salomon isolated a protein complex called “ovoverdin” from lobster, and in 1938, Kuhn and Sörensen further characterized “ovoverdin” and identified “ovoester” as a xanthophyll carotenoid, renaming it “astaxanthin” [11,12]. The name “asta” is derived from “Astacus” the genus name of the lobster. Kuhn and Sörensen demonstrated that AX exhibited behavior consistent with ovoester based on its melting point (215.5–216 °C) and elemental analysis. Additionally, astacin was determined to be an oxidized artifact of AX. Based on these findings, the structure of AX was determined to be 3,3′-dihydroxy-β,β-carotene-3,3′-dione [11]. In 1933, von Euler et al. isolated the red pigment “salmon acid” from salmon muscle [13], and in 1973, Khare et al. showed that salmon acid was identical to AX based on MS and ^1^H-NMR spectral data [14]. The search for natural sources of AX during the 1948–1950s led to its extraction from flamingo wings [15], grasshoppers, and other insects [16], as well as from the flower petals of the *Adonis* plant [17]. These early works on the isolation and identification of AX were published in “Nature,” one of the most prominent scientific journals, which highlights the great interest in natural pigments at that time. Subsequently, as described in Section 2.2, AX was found to be widely distributed in microorganisms, algae, and animals. Since 1970, AX, like other carotenoids, has been characterized using various spectroscopic techniques, including MS, ^1^H-NMR, and ^13^C-NMR [4]. Furthermore, X-ray crystallography was conducted in the 2000s [18,19]. Figure 3 shows an Oak Ridge Thermal Ellipsoid Program (ORTEP) diagram of a single molecule obtained from a single crystal of all-trans AX.

#### 2.1.3. The History of Astaxanthin Research in Japan

Today, AX research, including studies related to its biotechnology, is being conducted globally. However, in its early phases until the 1990s, it was predominantly carried out in Japan, where Japanese researchers made significant contributions to the field. Let us examine the history of AX research in Japan.

In Japan, AX research initially began in the field of fisheries. In the 1970s, Matsuno et al. conducted extensive research on carotenoids found in various aquatic animals, approaching the subject from the perspectives of natural product chemistry and comparative biochemistry (for detailed information, refer to other reviews [20,21,22]). Furthermore, Hata, Katayama, et al. studied the pathway of AX production in goldfish (colored varieties of *Carassius auratus*), nishikigoi (colored varieties of *Cyprinus carpio*), and Japanese tiger prawn (*Marsupenaeus japonicus*, formerly *Penaeus japonicus*). They proposed the pathway of AX biosynthesis in these aquatic animals, considering the structures of various metabolic intermediates, using dietary carotenoids such as zeaxanthin and β-carotene [23,24,25]. Kitahara, Hata, Hatano, Ando, et al. also contributed research on the metabolism of AX in salmon [26,27,28,29].

In the 1980s, Matsuno, Fujita, and Miki et al. made further discoveries in AX research. They revealed the reductive metabolism of AX to tunaxanthin through their studies on the coloration of marine fish, such as sea bream and yellowtail, and the metabolism of carotenoids in their eggs [30,31]. Additionally, Miki et al. reported that the administration of AX to aquaculture fish improved egg quality, hatching, development, and growth of fry. These findings were based on their studies of AX dynamics in fish eggs [20].

In the 1990s, Miki et al. (1991) made a significant discovery regarding the antioxidant activity of AX. They found that AX exhibited a much stronger (more than 100 times) capability in quenching singlet oxygen than that of α-tocopherol (vitamin E) and that AX scavenged free radicals, superior to other examined carotenoids, including β-carotene and zeaxanthin, as well as α-tocopherol [32]. Furthermore, Nishino et al. conducted research on the anti-carcinogenic effects of AX and other carotenoids [33,34]. Prior to the 1990s, AX had primarily been studied in the field of fisheries. However, these findings sparked research on its applications in medicine and human health. Meanwhile, in 1993, the Marine Biotechnology Institute in Japan discovered an AX-producing marine bacterium, later identified as belonging to the genus *Paracoccus* [35]. Subsequently, in 1995, AX biosynthesis genes, including a novel key gene for the ketolation reaction [36], were isolated from this *Paracoccus* strain, and their functions were clarified by Misawa et al. [37], followed by the isolation of the key gene from the green alga *Haematococcus pluvialis* [38]. This breakthrough led to the development of current research not only in AXs biosynthesis but also in metabolic engineering and synthetic biology for AX production. The pioneering studies conducted in the 1990s laid the foundation for AX and carotenoid research as it stands today.

Since 2000, AX has garnered significant attention in the field of preventative healthcare, particularly in relation to various lifestyle-related diseases. It has also gained recognition in the cosmetics industry for its anti-photooxidation and skin-aging effects. As a result, there have been a growing number of studies conducted worldwide on AX. In Japan, AX research and applications have been particularly prevalent in the field of ophthalmology. Considerable clinical evidence has accumulated regarding the effects of AX on eyestrain (asthenopia) [39,40,41,42,43,44,45,46,47]. Based on these research findings, several functional food products have been launched in Japan. These products will be discussed in more detail in Section 3, focusing on their industrial uses.

#### 2.1.4. Astaxanthin; Optical Isomers

In Section 2.1.1, it is mentioned that there are three possible optical isomers of AX. In the past, when AX was extracted from lobsters by Kuhn and Sörensen, it was considered optically inactive since it displayed minimal optical rotation [11]. 

In 1975, Liaaen-Jensen et al. successfully obtained optically active AX from the green alga *Haematococcus pluvialis* strain NIVA-CHL 9, now referred to as *H. lacustris* strain NIVA-CHL 9. In this study, *H. lacustris* will be referred to as *Haematococcus* algae, unless otherwise specified [48]. For further details, please see Section 2.2.2. Liaaen-Jensen et al. reduced the isolated AX from *Haematococcus* algae using NaBH_4_, and the resulting product exhibited a circular dichroism (CD) spectrum consistent with (3*R*,3′*R*)-zeaxanthin. Consequently, it was determined that AX derived from *Haematococcus* algae possesses a stereoconfiguration of (3*S*,3′*S*) [49,50]. Please note that the “*R*” and “*S*” nomenclature rules for absolute configuration were followed, where the hydroxyl group connecting the chiral carbon at the 3,3′ position to the chiral center is oriented upward (HO►), designating zeaxanthin as “*R*” and AX as “*S*”. For further information regarding nomenclature rules, please refer to Appendix A.

In 1976, Andrewes et al. made a significant discovery regarding AX obtained from *Phaffia* yeast, specifically *Phaffia rhodozyma* (currently known as *Xanthophyllomyces* dendrorhous). They observed that the CD spectrum of AX from this yeast was completely opposite to that of (3*S*,3′*S*)-AX, indicating that the AX from *Phaffia* yeast exclusively adopts the (3*R*,3′*R*) conformation [51]. This finding prompted us to conduct a meticulous comparison of the CD spectra of AX obtained from various marine animals. The observed differences in intensities (Δε) indicated that the AX from marine animals is a mixture of optical isomers. Appendix A presents the CD spectra of the three stereoisomers of AX for reference.

In 1979, Vecchi and Müller successfully separated racemic AX into three optical isomers using high-performance liquid chromatography (HPLC) in a normal-phase system. They employed diastereomeric esters of di-(−)-camphanate for this purpose [52]. Through this method, they were able to separate the optical isomers of AX obtained from lobster, shrimp, salmon, and starfish. The analysis revealed that AX in shrimp, salmon, and starfish comprised a mixture of the three optical isomers (3*R*,3′*R*), meso, and (3*S*,3′*S*). Furthermore, Vecchi and Müller directly separated the racemic AX into the three optical isomers using HPLC with a commercially available column called Sumichiral OA-2000. This column utilized an optically active stationary phase known as *N*-3,5-dinitrobenzoyl-D-phenylglycine (refer to Appendix A) [53]. By employing this method, Vecchi and Müller confirmed the existence of three stereoisomeric forms of AX in various marine animals.

#### 2.1.5. Astaxanthin; Geometric Isomers

Most natural AX exists as a mixture of geometrical isomers, including small amounts of 9-*cis*, 13-*cis*, and 15-*cis* forms, along with the all-trans form (Figure 2). In 1980, Roche’s group isolated ten geometric isomers of AX by HPLC, and their UV-VIS and ^1^H-NMR spectral data were reported [54]. Very recently, Yao et al. reported studies on the Raman spectra of the isomers of AX using density functional theory (DFT) calculations [55]. They confirmed that the theoretically calculated Raman spectra accurately reproduced the experimentally recorded Raman spectra of the all-trans, 9-*cis*, and 13-*cis* isomers of AX. They expanded the theoretical studies to the other isomers of AX (15-*cis*, 9,9′-*cis*, 9,13-*cis*, 9,13′-*cis*, 9,15-*cis*, 13,13′-*cis*, and 13,15-*cis* isomers) and proposed the assignment of the vibrational modes. They also discussed the stability of the isomers by comparing the theoretically predicted relative energies and estimated that the ratio of the all-*trans* configuration is approximately 70%, while 9-*cis* and 13-*cis* isomers each account for about 10%. The other isomers make up less than 2% under thermal equilibrium conditions.

Recently, Honda et al. introduced effective methods to generate geometrical isomers of AX in a thermally dependent process [56,57]. To date, there have been few reports on the physiological activities related to the geometric isomers of AX [58]. From a physicochemical perspective, several properties of cis isomers, including absorption maxima, solubility in solvents, and antioxidant activity, have been shown to differ from those of the all-*trans* form [4,54,58,59,60]. Specifically, the recent reports by Honda et al. and others indicate that certain geometric isomers may have higher bioavailability in rodents and are expected to have clinical applications in the near future [58,60,61].

#### 2.1.6. Astaxanthin Fatty Acid Esters

In addition to the free form, where no hydroxyl group modifications are present, AX also occurs naturally in a form where hydroxyl groups are modified by fatty acid esters (details of the distributions are described in Section 2.2). AX exists in both mono- and di-ester forms, with the ester moieties commonly composed of saturated fatty acids ranging from C12 to C18. Esterified derivatives of AX with highly unsaturated fatty acids such as eicosapentaenoic acid (EPA) and docosahexaenoic acid (DHA) have also been reported in marine animals. For example, AX in *Haematococcus* algae occurs mainly as a series of monoesters with C16 to C18 fatty acids [62,63,64], while in krill it occurs as diesters with highly unsaturated fatty acids such as DHA and EPA [65,66]. Therefore, when quantification of AX is required, it is often calculated from the absorbance value based on the absorption coefficient of the free form as a tentative quantification value. For more accurate quantification, saponification should be applied, and the free AX content should be quantified by HPLC. In other words, the value obtained by converting all of the esterified AX into its free form is often used as the AX concentration. The reasons for this and the details of the analytical methods are discussed individually in Section 2.4.2. One of the most important concerns is that AX can be readily converted to “astacene” (3,3′-dihydroxy-2,3,2′,3′-tetradehydro-β,β-carotene-4,4′-dione) through oxidation in alkaline solutions in the presence of oxygen (Appendix A) [67]. Therefore, to accurately quantify AX, esterified AX must first undergo alkaline saponification under anoxic conditions or enzymatic treatment with cholesterol esterase from *Pseudomonas* sp. [67,68]. The enzymatic treatment is generally more convenient as the hydrolysis reaction proceeds without artifacts under normal oxygen levels.

#### 2.1.7. Astaxanthin Aggregates

AX is strongly lipophilic, as indicated in Appendix A. Similar to many other carotenoids, AX is believed to undergo self-aggregation in hydrated polar solvents, resulting in the formation of aggregates [69,70]. The water concentration in the AX-solvent mixture influences the morphology of these aggregates and significantly impacts their photophysical properties. Spectroscopic analysis reports suggest that the absorption spectrum of AX aggregates is either blue-shifted (H-aggregate) or red-shifted (J-aggregate) compared to that of the monomer, reflecting the conditions during aggregation [71,72,73]. 

In the J-aggregate, astaxanthin molecules are arranged from head to tail, forming a relatively relaxed aggregate. Conversely, the H-aggregate exhibits a tighter “card-pack” stacking of polyene chains, which are somewhat aligned in parallel to each other [74] (Figure 4). Notably, the formation of H-aggregates is a unique characteristic of carotenoids possessing a hydroxyl group on the terminal cyclohexene ring. Introducing an O-R group in place of the hydroxyl group inhibits aggregation, thereby strongly suppressing aggregation in the ester form of AX [75].

Aggregates often exhibit significant differences in behavior compared to non-aggregated forms of biomolecules. These changes in physical properties can have a significant impact on their biological activity, particularly their pharmacological activity. For instance, H-aggregates of carotenoids demonstrate higher photostability in aqueous solutions compared to monomers; however, their radical scavenging activity and ability to quench singlet molecular oxygen are much lower than those of the monomers [75,76]. Incorporating biomacromolecules and amphiphilic compounds such as DNA, proteins (e.g., bovine serum albumin), and polysaccharides (e.g., arabinogalactan chitosan) can stabilize the formation of AX aggregates with biomacromolecules [75,77,78]. Recent studies have reported successful incorporation of H- and J-aggregates into DNA/chitosan co-aggregates and the preparation of complex nanosuspensions containing these two types of aggregates [77]. AX aggregates, which are typically unstable, can be stabilized by incorporating them into hydrophobic microdomains of these polymers, such as DNA/chitosan complexes, even in the absence of EtOH/water solvents. Highly aggregated molecular complexes have demonstrated different behavior in terms of radical scavenging activity compared to monomers in simple aqueous polar solvents [77]. This difference may be attributed to the newly formed intermolecular hydrogen bonds with the biopolymer and the presence of a π-π conjugated structure in the intermolecular association. These factors may explain the variation in antioxidant activity between H- and J-aggregates due to their different electron transport capacities [77]. However, there are still many unresolved aspects regarding the physiological activity of aggregates, and further investigations are required.

Recently, researchers have isolated five distinct forms of AX aggregates that allow for the adjustment of intermolecular coupling between AX molecules. Time-resolved absorption spectroscopic studies with sub-30 fs time-resolution have been conducted on these aggregates [79]. Each form of AX aggregate is capable of undergoing intermolecular singlet fission, with rates of triplet generation and annihilation that can be linked to the strength of intermolecular coupling. This finding challenges the conventional model of singlet fission in linear molecules [80], as it demonstrates that the triplet state of AX is directly formed from the initial ^1^B_u_^+^ (S_2_) photoexcited state through an ultrafast singlet fission process. This discovery highlights the potential use of AX aggregates, particularly the H-aggregate, as photoprotectors in biological systems. The H-aggregate of AX exhibits a significant hypsochromic shift in absorption, extending into the UV spectral region, compared to that of the monomer. Consequently, the H-aggregate of AX efficiently absorbs light in the UV—blue spectral range. Upon photoexcitation, the H-aggregate of AX can safely dissipate its energy as heat through the triplet excited state, which is formed via the ultrafast fission process. The significance of aggregates in natural systems is further discussed in Section 2.1.8.

#### 2.1.8. Carotenoproteins: Astaxanthin-Protein Complexes

AX has hydroxyl groups at the C3 and C3′ positions and carbonyl groups at the C4 and C4′ positions. Therefore, it exhibits a high affinity for certain proteins, such as albumin, and can readily form pigment-protein complexes.

In many marine animals, AX is present in the form of protein-bound complexes. One of the most well-known examples of AX-protein complexes is seen in the blue, purple, and yellow hues of crustacean exoskeletons, which are predominantly derived from the AX-protein complex [81]. For more information on carotenoid-protein interactions in aquatic organisms, refer to other reviews [81].

The relationship between the structure of AX and color has been extensively studied, particularly in the case of “crustacyanins”, which contribute to the blue to purple coloration of lobster shells belonging to the species *Homarus gammarus* and *H. americanus*. Crustacyanins are members of the lipocalin superfamily of proteins, as deduced from the amino acid sequence of their subunits, which are hydrophobic ligand-binding proteins [82,83,84,85,86]. The multimeric α-crustacyanin (with a maximum absorption wavelength of approximately 630 nm) and dimeric β-crustacyanin (with a maximum absorption wavelength of approximately 580–590 nm), isolated from lobster shells, exhibit blue to purple colors. The structure of α-crustacyanin has been investigated through CD spectra and X-ray crystallographic analysis of its substructure, β-crustacyanin. α-crustacyanin is a large macromolecule with a molecular weight of approximately 320 kDa, consisting of eight pairs of heterodimeric β-crustacyanin units, which are themselves composed of heterodimers formed by two apocrustacyanins. Apocrustacyanins comprise five subunits: A_1_, C_1_, and C_2_, each with a molecular weight of approximately 21 kDa, and A_2_ and A_3_, each with a molecular weight of approximately 19 kDa. In lobsters (*Homarus gammarus* ), the major subunits of β-crustacyanin are A_2_ and C_1_ [83]. Consequently, there are 16 molecules of free AX within α-crustacyanin, as each apocrustacyanin associates stoichiometrically with an equal amount of AX. X-ray crystallographic analysis of β-crustacyanin has revealed three characteristics resulting from the binding of AX: elongation of the chromophore due to the 6-s-trans planar structure, hydrogen bonding between the C4, C4′ keto group and water, as well as histidine residues, and the close interactions of the two chromophores (AX). Since the usual maximum absorption wavelength (λ_max_) of the AX monomer is around 470 nm, both β- and α-crustacyanins exhibit a strong bathochromic shift in their absorption spectra as a result of the conformational change of the chromophore within the protein, resulting in a purple and blue color, respectively [84]. Further bathochromic shifts of up to 45 nm can be observed due to aggregation effects during the association of β-crustacyanin with α-crustacyanin in lobster shells. In crustaceans, the combination of apoprotein subunits varies among species and mutations, contributing to the variation in the coloration of crustacyanins [86,87]. 

The relationship between the structure of AX and color has been extensively studied, particularly in the case of “crustacyanins”, which contribute to the blue to purple coloration of lobster shells belonging to the species *Homarus gammarus* and *H. americanus*. Crustacyanins are members of the lipocalin superfamily of proteins, as deduced from the amino acid sequence of their subunits, which are hydrophobic ligand-binding proteins [82,83,84,85,86]. The multimeric α-crustacyanin (with a maximum absorption wavelength of approximately 630 nm) and dimeric β-crustacyanin (with a maximum absorption wavelength of approximately 580–590 nm), isolated from lobster shells, exhibit blue to purple colors. The structure of α-crustacyanin has been investigated through CD spectra and X-ray crystallographic analysis of its substructure, β-crustacyanin. α-crustacyanin is a large macromolecule with a molecular weight of approximately 320 kDa, consisting of eight pairs of heterodimeric β-crustacyanin units, which are themselves composed of heterodimers formed by two apocrustacyanins. Apocrustacyanins comprise five subunits: A_1_, C_1_, and C_2_, each with a molecular weight of approximately 21 kDa, and A_2_ and A_3_, each with a molecular weight of approximately 19 kDa. In lobsters (*Homarus gammarus* ), the major subunits of β-crustacyanin are A_2_ and C_1_ [83]. Consequently, there are 16 molecules of free AX within α-crustacyanin, as each apocrustacyanin associates stoichiometrically with an equal amount of AX. X-ray crystallographic analysis of β-crustacyanin has revealed three characteristics resulting from the binding of AX: elongation of the chromophore due to the 6-s-trans planar structure, hydrogen bonding between the C4, C4′ keto group and water, as well as histidine residues, and the close interactions of the two chromophores (AX). Since the usual maximum absorption wavelength (λ_max_) of the AX monomer is around 470 nm, both β- and α-crustacyanins exhibit a strong bathochromic shift in their absorption spectra as a result of the conformational change of the chromophore within the protein, resulting in a purple and blue color, respectively [84]. Further bathochromic shifts of up to 45 nm can be observed due to aggregation effects during the association of β-crustacyanin with α-crustacyanin in lobster shells. In crustaceans, the combination of apoprotein subunits varies among species and mutations, contributing to the variation in the coloration of crustacyanins [86,87]. 

Another carotenoprotein that forms the exoskeleton in lobsters, similar to crustacyanin, is crustochrin. Crustochrin exhibits a yellow hue with hypsochromically shifted bands, having a maximum absorption wavelength of 400–410 nm. This protein contains approximately 20 astaxanthin molecules and demonstrates typical exciton-exciton interactions through natural H-aggregates (see Section 2.1.7), with the chromophores arranged in a stack-of-cards formation [88,89]. Interestingly, these two distinct groups of proteins (crustacyanins and cristochrins), in terms of color hue, have been found to localize differently within the lobster exoskeleton [86,90]. This characteristic localization will be described in Section 2.2.4. Recent studies indicate that crustacyanins are restricted to Malacostraca crustaceans but are widely distributed within this group. These crustacean-specific genes are divided into two distinct clades within the lipocalin protein superfamily. The fact that the crustacyanin gene family emerged early in the evolution of Malacostraca crustaceans suggests that this protein played a significant role in the evolutionary success of this group of arthropods [86,91]. Crustaceans, in particular, are known for their diverse species-specific shell colors and patterns, and these proteins are believed to be involved in functions such as protection through cryptic coloration, reproduction, and communication [92,93].

Moreover, in crustaceans, AX-binding proteins are not limited to α- or β-crustacyanin alone but also exist as complexes bound to “ovoverdins” and other proteins in crustaceans. Ovoverdins, reported as the pigment responsible for the dark green color (λ_max_; ca. 465–470 and 660–670 nm) of lobster ovaries and eggs [12,94,95,96], are a complex of AX (mostly in the free form) and lipovitellin, which is a predominant glycolipoprotein found in the yolk of egg-laying organisms [97]. Corresponding to their λ_max_, AX may bind to two distinct sites: one might be a weak non-specific association, and the other is a specific stoichiometric association with lipoproteins. Additionally, there are at least two different molecular-weight proteins (ca. 700 kDa and 600 kDa) [96]. In the latter, ovoverdin seems to form a multimer of four subunits consisting of a, b, c, and d, according to the SDS-PAGE results [96].

Similar AX-protein complexes have also been reported to form in other organisms. For example, a blue AX-protein complex called “velellacyanin” has been isolated from the blue mantle of the blue-colored “by-the-wind-sailor” jellyfish *Velella velella* [82,98]. There are two types of this protein, named V600 (λ_max_; ca. 600 nm) and V620 (λ_max_; ca. 620 nm), respectively, based on their maximum absorption wavelengths. The molecular weight of each is >300 kDa [82,98]. The velellacyanins are multimeric formations of multiple subunits and form a helical structure. The quaternary structures of velellacyanin and the N-terminal peptide sequence of the subunit comprising V600 reveal similarity to apocrustacyanin C [99,100,101]. In fact, immunocross-reactivity showed reactivity with polyclonal antibodies for not only apocrustacyanin C but also apocrustacyanin A [102]. However, it currently remains unclear whether these velellacyanin apoproteins belong to the lipocalin superfamily, and future studies, including their origin and evolutionary position, are expected.

In echinoderms, two well-known carotenoproteins, as shown below, have been partially characterized by X-ray structural and CD spectral analyses; however, no phylogenetic or functional analysis of the proteins, including detailed genetic background, has yet been available. The common starfish *Asterias rubens* has “asteriarubin”, a purple-blue carotenoprotein, also present [103,104]. Asteriarubin (λ_max_; ca. 570 nm) is approximately 43 kDa and comprises four subunits with a molecular weight of approximately 11 kDa each. The major carotenoids in this protein are AX and its acetylenic and dehydro analogues, such as 7,8-dideoxyhydroastaxanthin and 7,8,7′,8′-tetradehydroastaxanthin. These carotenoids are metabolites of AX found in echinoderms and are described in Section 2.2.4. Interestingly, the amino acid sequence shows no homology to the apocrustacyanin subunits [81,103]. Reconstitution studies revealed similarities between the binding requirements of asteriarubin and crustacyanin; however, the tetrameric asteriarubin contains only one carotenoid molecule, and the CD spectrum shows no exciton splitting. Therefore, in this case, molecular aggregation and carotenoid-protein chromophore interactions were considered unlikely to be determinants of the bathochromic shift [81]. This slight bathochromic shift may be attributed to the absence of exciton effects compatible with extended π conjugation due to hydrogen bonding between the terminal polar group and the protein and the co-planarity of the ring [104]. Another type of carotenoprotein in echinoderms, the vivid blue skin of calcified starfish called “blue star”, *Linckia laevigata*, has a blue carotenoprotein called “linckiacyanin” (λ_max_; 395, 612 nm), with (3*S*,3′*S*)-AX as the major carotenoid [105]. Although the molecular weight of linckiacyanin is quite large (>103 kDa), the main glycoprotein subunit is small, at only approximately 6 kDa. However, the minimum molecular weight of the native subunits (approximately 16 kDa) means that there are at least 200 carotenoid molecules per molecule of linkyacyanin [105]. Since linkyacyanin showed no cross-reactivity with polyclonal antibodies for the β-crustacyanin subunit [102], it is possible that linkyacyanin is a distinct family member from the lipocalin superfamily.

In Asia, it is easy to find vivid pink egg clumps on the surface of rice plants and on the walls of aqueducts for rice fields. This pigment is known as “ovorubin”. Ovorubin is a carotenoprotein found in the perivitelline fluid that surrounds the embryo of the fertilized egg, which is an accessory gland of the female reproductive tract of the South American freshwater snail *Pomacea canaliculata* (Gastropoda: Ampullariidae). Ovorubin is described as a large red AX-binding glycoprotein of approximately 330 kDa [106], and the binding of (3*S*,3′*S*)-AX and their fatty acid esters to ovorubin results in a small bathochromic shift (20–30 nm) to λ_max_ 510 nm [81]. The protein is also an oligomer composed of three subunits of approximately 28, 32, and 35 kDa and is a very high-density glycosylated lipo-carotenoprotein (VHDL) with phospholipids, sterols, and carotenoids as ligands. It is highly glycosylated. This protein provides the egg with resistance against sun radiation and oxidation of lipids [107] and is thought to play an important physiological role in the storage, transport, and protection of carotenoids during snail embryogenesis [108]. In addition to ovorubin, there is another carotenoprotein called “alloporin” (λ_max_; 545 nm) found in the soft coral *Allopora californica*. Alloporin is approximately 68 kDa and comprises four subunits with a molecular weight of approximately 17 kDa each. It has an equal molar of (3*S*,3′*S*)-AX bound to it [109,110]. This seems to be similar to asteriarubin.

The authors eagerly anticipate a future where the mysteries behind the physical-chemical properties and mechanisms of the chromophores found within these remarkable AX-containing carotenoproteins are unraveled. Aiming to shed light on the intricate details and functions of these chromophores, their characterization holds the key to unlocking a deeper understanding of the captivating structures and extraordinary roles played by these carotenoproteins. The journey to uncover their secrets promises to be a captivating exploration into the realms of science and discovery.

All photosynthetic plants utilize carotenoid-binding proteins as an important component of their photosynthetic function. For example, in chloroplast thylakoid membranes, pigment molecules such as carotenoids and chlorophyll function within the light-harvesting protein complex (LHC), an antenna pigment-protein complex bound to the photosystem, to achieve extremely high-efficiency light harvesting. The photosystem II supercomplex (PSII) is also a pigment-protein complex that catalyzes water splitting and oxygen-evolving reactions in photosynthesis, converting light energy into chemical energy. The PSII core complex is composed of more than 20 subunits and contains approximately 35 chlorophylls (Chl) and 12 β-carotene molecules, as well as other oxidation-reduction cofactors required for electron transfer [111,112]. PSII is especially sensitive to light-induced damage (photodamage) among the components of photosynthesis because it is an extremely oxidative reagent with a high enough potential for the light-excited P680 Chl molecule, which is utilized in the MnCaO_5_ cluster to split water. If the rate of photodamage exceeds the rate of repair under excessively intense light conditions, a phenomenon called photoinhibition of PSII occurs. Photoinhibition of PSII is caused by reactive oxygen species (ROS) that are generated during the excitation energy transfer and electron transport processes. In particular, the repair system of photodamaged PSII is sensitive to various ROS. To reduce the effects of photoinhibition of PSII, plants have developed systems to quench or suppress the production of ROS. Non-photochemical quenching (NPQ) is one of these defense mechanisms in plants. One of the most important mechanisms is the involvement of carotenoids, which directly quench singlet oxygen generated by excited chlorophyll, or xanthophylls, which enhance heat dissipation of excess light energy through the xanthophyll cycle. Thus, it is clear that carotenoids play a pivotal role in oxygen-evolving photosynthesis. These carotenoids are present in membrane proteins in the thylakoid membrane or directly in the thylakoid membrane [111,112], while some coexist with water-soluble proteins. One of those groups is the orange carotenoid proteins, which are widely distributed in cyanobacteria [113]. However, there is no direct evidence that AX in the specific protein complex is involved in naïve plants’ photosynthetic function; however, certain green plants have AX-binding proteins as a photoprotector. For instance, the microalgae *Coelastrella astaxanthina* Ki-4 (*Scenedesmaceae* sp. Ki-4), isolated from the asphalt surface in mid-summer, benefits from a water-soluble AX-binding carotenoprotein called “AstaP” [114]. AstaP (AstaP-orange1) is a secreted protein that exhibits thermally stable ^1^O_2_ quenching activity [114], which is induced through exposure to strong light. The deduced N-terminal amino acid sequence of AstaP reveals that it represents a new class of carotenoid-binding proteins homologous to the fasciclin family proteins, with extensively N-glycosylated regions [114]. A related green alga, *Scenedesmus* sp. Oki-4N, has three comparable AstaP orthologs. However, AstaP-pinks has no glycosyl residues, while AstaP-orange2 has a glycosylphosphatidylinositol (GPI) anchor motif and a higher isoelectric point (pI = 3.6–4.7), which is significantly different from the original AstaP-orange1 (pI = 10.5) [115]. Orthologues of AstaP have also been found in diverse green algae, including *Chlamydomonas reinhardtii* and *Chlorella variabilis*, which are also induced by light irradiation [116]. These results are unique examples of how the use of water-soluble AX in photosynthetic organisms is a novel strategy to protect cells from severe photooxidative stress. The native form of AstaP (AstaP-orange1) from *C. astaxanthina* Ki-4 binds to AX quite specifically, whereas the expression of a correctly folded recombinant protein in *E. coli* and the evaluation of its binding to the protein revealed that it is also capable of binding to other xanthophylls and carotenes [117]. Recombinant AstaP is a ~20 kb water-soluble protein that accepts carotenoids in acetone solution or embedded in biological membranes. It then has the property of forming carotenoid-protein complexes with apparently equal stoichiometry [117].

Recently, recombinant plants have been developed that possess ketocarotenoids. These ketocarotenoids are typically lacking or completely absent from the photosynthetic system found in higher plants. However, it appears that they have been successfully integrated into the photosynthetic system. The high accumulation of AX and other ketocarotenoids has been found to impact growth, CO_2_ assimilation, and photosynthetic electron transfer in transgenic plants. Moreover, studies have demonstrated that ketocarotenoids act specifically on the thylakoid membrane and, more specifically, on PSII [118,119].

Interestingly, transplastomic lettuce, which has been genetically modified in its chloroplast genome, has been found to predominantly accumulate AX [120]. Despite having low levels of naturally occurring photosynthetic carotenoids [111], this lettuce exhibits normal growth similar to that of non-modified plants. Initially, the quantum yield of PSII in this lettuce is low under normal growth conditions; however, it becomes comparable to control leaves under higher light intensities. In AX-accumulating lettuce, in addition to β-carotene, echinenone and canthaxanthin are bound to the PSII monomer, while the normal binding of photosynthetic carotenoids is absent. This lack of normal carotenoid binding affects the assembly, photophysical properties, and function of PSII. However, the repair mechanisms in AX-accumulating lettuce enable the maintenance of PSII function despite photodamage. The high antioxidant capacity of AX, its esters, and other ketocarotenoids accumulated in thylakoid membranes is believed to provide protection against reactive oxygen species (ROS) generated during oxygenic photosynthesis [111].

When carotenoids and xanthophylls are highly accumulated, they can influence the physical properties of the lipid membrane itself, such as fluidity and permeability to small molecules [121]. In AX-accumulating lettuce, the high accumulation of xanthophylls may prevent the permeation of singlet oxygen produced by PSII and enable efficient quenching within the lipid bilayer. Specifically, AX and carbonyl derivatives of lutein have been observed to adhere to the surface of the PSII core complex, indicating their effective quenching of singlet oxygen (^1^O_2_). Essentially, the apparent photosynthetic capacity of this lettuce may be attributed to the antioxidant effect of AX and its derivatives, compensating for the absence of essential naturally occurring carotenoids [111].

The Silkworm, *Bombyx mori* larvae, possesses a carotenoid-binding protein (BmCBP) in their silk glands. This protein has an apparent molecular weight of 33 kDa and binds carotenoids in a 1:1 molar ratio. Lutein constitutes ninety percent of the bound carotenoids, although it also binds to other carotenoids [122]. In its cytoplasmic form, BmCBP acts as a non-internalized lipophorin receptor, binding to lutein and functioning as a “transport carrier” [123]. The deduced amino acid sequence of BmCBP indicates its affiliation with the steroidogenesis acute regulatory protein (StAR) family, featuring a distinctive structural element known as the StAR-related lipid transfer domain. This domain facilitates lipid translocation and recognition [122]. BmCBP demonstrates the ability to bind various carotenoids, including dietary AX, and transport them to the silkworm cocoon [124]. In recent years, significant advancements have been made, including the successful heterologous expression of BmCBP in *E. coli* while maintaining its functions [125]. Furthermore, the crystal structure of BmCBP has been determined through X-ray structural analysis, and the binding site for xanthophylls has been identified [126].

In recent years, carotenoid-binding proteins, including AstaP from the carotenoid binding protein (CBP) group, have emerged as promising antioxidant nanocarriers with potential applications in biomedicine [127]. The binding ability of AstaP to carotenoids also indicates its potential industrial utility in the recovery and concentration of carotenoids from crude extracts [125].

AX-protein complexes have also been observed in fish. In salmon, for instance, muscle AX exhibits specific binding to the surface of actomyosin, a protein present in skeletal muscles. Interestingly, the binding between this protein and AX does not appear to be stereoselective, meaning it does not show preference based on the spatial arrangement of AX molecules [128,129].

In general, animals are unable to synthesize carotenoids de novo, with a few exceptions, as mentioned in Section 2.2.4. Therefore, recent research has highlighted the involvement of transport proteins in the absorption of carotenoids from the intestinal tract and their transfer to various tissues. These transport proteins likely share functions with other carotenoids and lipids, such as cholesterol. However, the precise binding modes between these transport proteins and AX remain unclear, although it is presumed that the binding is relatively loose. It is still uncertain whether the binding occurs in a stoichiometric manner or not. Additionally, enzymatic degradation of carotenoids takes place in various tissues. This degradation involves reactions catalyzed by carotenoid cleavage oxygenases, resulting in the formation of apocarotenoids. A well-known example is the conversion of β-carotene into vitamin A retinoids by the enzyme BCO1. The detailed roles of these processes are discussed in other sections, specifically in Section 2.3.4 and Section 4.1 for mammals.

#### 2.1.9. Astaxanthin as a Powerful Antioxidant

AX is believed to exert its antioxidant activity through both direct quenching and scavenging of reactive chemical species, such as reactive oxygen species (ROS) and reactive nitrogen species (RNS). Additionally, it employs indirect mechanisms by inducing a group of antioxidant enzymes in biological systems. However, this review specifically focuses on the ROS-scavenging mechanism of AX. For further information on AXs role in the biological environment (e.g., plasma membrane and carrier proteins like HDL/LDL) and its induction of antioxidant enzymes, such as SODs, GSTs, GPXs, and Catalase, via the activation of the Nrf2/PGC-1α pathway, as well as its suppression of the production of pro-inflammatory cytokines by inhibiting the NFκB or JAK/STAT pathway, please refer to the author’s other comprehensive review (see Sections 1, 2.1 and 2.2 of [130]).

##### Quenching Singlet Oxygen

Singlet molecular oxygen (^1^O_2_) is generated from the ground-state triplet molecular oxygen (^3^O_2_: ^3^∑_g_^−^) through photochemical processes in biological systems [131,132,133,134,135,136,137,138,139,140,141]. Approximately ^1^O_2_ exists in two singlet states with different spins (^1^∑_g_^+^ and ^1^∆_g_), with ^1^∆_g_ being the lowest excited singlet state. The former has a very short lifetime, while the latter, although short-lived, has a longer lifetime than the former. Therefore, the term “singlet oxygen” generally refers to the ^1^∆_g_ state. The lifetime of singlet oxygen is also strongly influenced by the surrounding environment. Additionally, there is a very short-lived dimol molecule (O_2_(^1^∆_g_)-O_2_(^1^∆_g_)) that forms as a result of the reaction between two singlet oxygen molecules. This dimol molecule can be detected through luminescence in the red spectral region, which corresponds to twice the energy of O_2_(^1^∆_g_) emission in the infrared spectral region around 1270 nm.

Under normal environmental conditions, the electric dipole transition of oxygen molecules from the ground triplet state (^3^Σ_g_^−^) to the lowest electronically excited singlet state (^1^Δ_g_) has an extremely low transition probability. This transition is forbidden due to considerations of spin angular momentum, orbital angular momentum, and parity. As a result, singlet oxygen is typically generated through interaction with photosensitizers such as porphyrins and chlorophylls. Additionally, it is believed that singlet oxygen can be generated in the absence of light through the Haber-Weiss reaction involving superoxide (O_2_^•–^) and hydrogen peroxide (H_2_O_2_). Furthermore, an autocatalytic reaction involving the cyclization of peroxyl radicals can also produce singlet oxygen via a tetraoxide intermediate. Therefore, the Russell mechanisms facilitate the generation of singlet oxygen from lipid peroxyl radicals. Enzymatic reactions, such as those catalyzed by myeloperoxidase in monocytes, can also lead to the production of singlet oxygen [131,138,139,141]. 

The production of singlet oxygen is indeed harmful to biological tissues. This is due to its ability to readily oxidize and modify lipids, proteins, and nucleic acids, which are vital for biological functions, thereby causing their loss of function through Diels-Alder reactions. Consequently, singlet oxygen has been implicated in several diseases. For instance, it has been strongly suggested that singlet oxygen is involved in light-exposed skin and ocular tissues, contributing to conditions such as skin aging, skin cancer, Porphyria, Smith-Lemli-Opitz syndrome, glaucoma, cataracts, and age-related macular degeneration. Moreover, even in the absence of light exposure, singlet oxygen’s involvement is strongly suspected in the onset or exacerbation of diabetes mellitus and bronchial asthma [142].

Carotenoids, such as lycopene, β-carotene, lutein, and AX, are highly efficient quenchers of singlet molecular oxygen (^1^O_2_). Their reaction rate constants, typically around 10^10^ M^−1^ s^−1^, approach the limit of diffusion control in solvents [76,143]. The quenching mechanism of ^1^O_2_ by carotenoids has been recently analyzed through quantum dynamics calculations and ab initio calculations [144]. Theoretical studies suggest that the ground-state singlet carotenoid (^1^Car) and ^1^O_2_ molecules can form a weakly bound complex, facilitated by the donation of electron density from the carotenoid’s highest occupied molecular orbital (HOMO) to the π_g_^*^ orbitals of ^1^O_2_. The quenching of ^1^O_2_ is governed by a Dexter-type superexchange mechanism involving charge transfer states (Car^•+^/O_2_^•−^). Quantum dynamics calculations demonstrate that the quenching of ^1^O_2_ by carotenoid/O_2_ complexes occurs rapidly, within sub-picosecond timescales, due to strong electronic coupling. This theoretical study highlights the crucial role of carotenoid cation radical species (Car^•+^) in achieving efficient ^1^O_2_ quenching. Notably, AX is known for its high activity and stability against ^1^O_2_ quenching [143]. 

Experimental evidence supports the theoretical understanding that as the polyene chain length of carotenoids increases, i.e., the conjugated π-electron system becomes more extended, the HOMO level of carotenoids decreases, and their ^1^O_2_ quenching activity becomes stronger. For instance, Conn and Edge et al. conducted an evaluation of the singlet oxygen scavenging activity of various β-carotene and lycopene analogs with different lengths of conjugated π-electron systems. They extended the conjugated double bonds from 7,7′-dihydro-β-carotene (*n* = 7) to dodecapreno-β-carotene (*n* = 19), and the experimental results demonstrated that the ^1^O_2_ quenching activity increased as the length of the conjugated double bond was increased (refer to Table 1 and Appendix A) [145,146]. 

AX (*n* = 13) possesses two expanded π-electron systems due to the presence of C4, C4′ diketo groups connected to the *n* = 11 conjugated polyene of C40 carotenoids. Additionally, AX has polar hydroxyl groups at both ends (refer to Figure 2). These molecular frameworks might specifically contribute to the expression of AXs superior antioxidant activity while maintaining its molecular stability.

Comparisons of the activity of cis-isomers, such as β-carotene, with the all-*trans* geometrical isomer have been reported. Specifically, in terms of ^1^O_2_ scavenging activity, the all-*trans* configuration is known to exhibit the highest activity, followed by 15-*cis* and 9-*cis* isomers. This indicates that the rate constant for deactivating ^1^O_2_ by the *cis*-isomers decreases as the *cis*-bond of β-carotene moves away from the center of the molecule, resulting in less efficient ^1^O_2_ quenching compared to the all-*trans* isomers [145]. Time-resolved resonance Raman studies have shown that all β-carotene isomers share a common triplet state, characterized by a twist in the central carbon-carbon double bond compared to the ground state. This structural variation may have an impact on the conjugated system [147,148].

In biological model membranes, such as phospholipid liposomes, several studies have investigated the effectiveness of AX in inhibiting ^1^O_2_-induced peroxidation of phospholipid membranes. However, the observed activity of AX in these studies sometimes falls below expectations, such as exhibiting a lower reaction rate constant compared to β-carotene, which can vary depending on the detection system [149]. Nevertheless, when evaluated based on endoproducts as outcomes, AX has been found to exert effective protective actions against phospholipid membrane peroxidation and cellular damage induced by photosensitization in the presence of photosensitizers [150,151,152,153,154]. 

The reaction between AX and singlet oxygen is primarily attributed to physical quenching, as described previously. However, a small fraction of AX does form reactive products with ^1^O_2_. According to Nishino et al., the major products generated from this reaction are endoperoxides, specifically astaxanthin 5,6-endoperoxide or astaxanthin 5,8-endoperoxide. These endoperoxides represent ^1^O_2_ adducts formed at the C=C bonds of the β-end group [155].

**Table 1 marinedrugs-21-00514-t001:** Singlet oxygen quenching activity of astaxanthin: comparison with common antioxidants [k(10^9^M^−1^s^−1^)].

^1^O_2_ Generator	EDN *			EDN *		NDPO_2_ *	EP *
Reference	[156]			[157]		[158]	[159]
Detection	Luminescence			Luminescence		Luminescence	Absorbance of DPBF
Solvent	CDCl_3_	CDCl_3_/ CD_3_OD (2:1)	DMF/ CDCl_3_ (9:1)	CDCl_3_	CDCl_3_/ CD_3_OD (2:1)	EtOH/CHCl_3_ /D_2_O (50:50:1)	EtOH/CHCl_3_/D_2_O (50:50:1)
**1. Carotenoids**							
**Astaxanthin**	**2.2**	**1.8**	**5.4**	**2.2**	**1.8**	**24.0**	**11.7**
Canthaxanthin	2.2	1.3	2.0	-	1.2	21.0	
Zeaxanthin	2.0	0.73	3.4	1.9	0.12	10.0	11.2
β-Cryptoxanthin	2.0	0.27	1.7	-	-	6.0	7.0
**β** **-Carotene**	**2.2**	**0.28**	**1.1**	**2.2**	**0.049**	**14.0**	**10.8**
**Lycopene**	**3.0**	**1.4**	**3.4**	**-**	**-**	**31.0**	**14.0**
Capsanthin	-	-	-	-	-	-	12.1
Lutein	0.61	0.26	2.1	0.8	-	8.0	8.1
α-Carotene	0.66	0.23	0.93	-	-	19.0	10.0
Fucoxanthin	0.29	0.075	0.97	-	0.005	-	-
Tunaxanthin	-	-	-	0.15	-	-	-
**2. Vitamin C**							
L-Ascorbic acid	-	-	0.00089	-	-	-	-
**3. Vitamin E**							
α-Tocopherol	0.02	0.0039	0.049	-	-	0.28	0.13
β-Tocopherol	-	-	-	-	-	0.27	0.093
γ-Tocopherol	-	-	-	-	-	0.23	0.084
δ-Tocopherol	-	-	-	-	-	0.16	0.041
Trolox	-	-	0.010	-	-	-	0.042
**4. Polyphenols/other phenolic antioxidants**							
α-Lipoic acid	0.056	0.038	0.072	-	-	0.13	0.0019
Ubiquinone-10	0.0019	0.0021	0.0068	-	-	-	0.062
BHT	-	-	0.004	-	-	-	-
Caffeic acid	-	-	0.0023	-	-	-	0.00069
Ferulic acid	-	-	-	-	-	-	0.00027
CurcuminI	-	-	0.0036	-	-	-	-
(-)-EGCG	-	-	0.0096	-	-	-	0.0051
Gallic acid	-	-	0.0023	-	-	-	-
Pyrocatechol	-	-	0.0055	-	-	-	-
Pyrogallol	-	-	0.0055	-	-	-	-
Quercetin	-	-	0.0018	-	-	-	-
Resveratrol	-	-	0.0018	-	-	-	-
Sesamin	-	-	0.0012	-	-	-	-
Capsaicin	-	-	0.0021	-	-	-	-
Probucol	-	-	0.00044	-	-	-	-
Edaravon	-	-	0.0067	-	-	-	-

* The ^1^O_2_ was generated in a dark reaction by thermodissociation from the respective endoperoxide. EDN, 1,4-dimethylnapthalene endoperoxide; NDPO_2_, of 3,3′-(1,4-naphthylene) dipropionate endoperoxide; EP, 1-methylnaphthalene-4-propionate endoperoxide; DPBF, 1,3-diphenylisobenzofuran; DMF, *N*,*N*-dimethylformamide; (−)-EGCG, (−)-epigallocatechin gallate.

##### Scavenging of Free Radicals and Inhibition of Lipid Peroxidation

Carotenoids can occasionally display pro-oxidant activity by oxidizing other lipids instead of acting as antioxidants, especially under high oxygen partial pressure or in the absence of other antioxidants. However, in comparison to other carotenoids, AX has shown minimal pro-oxidant activity in both simple solvent-based systems and evaluation systems that involve phospholipid membranes [143,160]. This is in contrast to lycopene, which is often assessed for its antioxidant properties.

Studies employing pulse radiolysis and time-resolved spectroscopy have revealed that carotenoids engage with free radicals through different mechanisms [161,162,163,164,165]. The preliminary products formed during the interaction between carotenoids and free radicals indicate that electron transfer and radical addition are kinetically favored reactions. These reactions result in the oxidation of the carotenoid to its radical cation or the generation of carotenyl adduct radicals, such as [R-Car]^•^ [166]. 

Carotenoids possess the necessary reactivity to function as antioxidants, and their reaction rates with free radicals are comparable to those of polyunsaturated fatty acids reacting with the same oxidants [143]. When comparing the reactivity of carotenoids with free radicals, it is observed that carotenoid radical cations can be reduced by α-, β-, and γ-tocopherol, while lycopene and β-carotene can reduce δ-tocopherol radicals [161,167]. β-Carotene^•–^ transfers an electron to oxygen but not vice versa, while lycopene undergoes reversible electron transfer with O_2_^•–^ due to its more positive electronegativity resulting from its planar geometry. Among the common carotenoids, AX radical cations are the most easily reduced. This implies that AX radical cations can be reduced by other carotenoids, like lycopene. Additionally, the interaction between carotenoids and other antioxidants may significantly contribute to their antioxidant activity *in vivo*. Skibsted and colleagues demonstrated that AX radical cation is effectively reduced by polyphenols such as isoflavonoids, in addition to the antioxidants mentioned earlier [168].

Interestingly, AX exhibits lower reducing abilities compared to other carotenoids. When reacting with CCl_3_OO^•^, AX does not directly form a radical cation but instead undergoes an addition of a radical, which subsequently decays to form the cation radical [165]. Despite this, AX and canthaxanthin demonstrate greater effectiveness as antioxidants compared to β-carotene or zeaxanthin in retarding hydroperoxide formation during azo-initiated lipid peroxidation in homogeneous methyl linoleate/AMVN systems. However, the rates of AMVN-induced oxidation of AX and canthaxanthin are slower than those of β-carotene and zeaxanthin [169].

These findings indicate that AX and other carotenoids, when consumed in combination with multiple antioxidants rather than individually, may have a better ability to inhibit lipid peroxidation through the interaction of antioxidant networks. This could provide an explanation for the suggestion that aggressive supplementation of synthetic β-carotene at high doses may actually increase the risk of lung cancer in smokers and asbestos-exposed workers [170,171]. On the other hand, simultaneous intake of green and yellow vegetables, which contain multiple carotenoids and antioxidant vitamins, has been associated with a reduced risk of cancer [172,173,174]. Therefore, understanding the reactivity of AX with reactive oxygen species (ROS) and free radicals, as well as the physicochemical properties of its reaction intermediates, is crucial for a comprehensive understanding of the true antioxidant activity of AX.

##### Structures of Radical Cation and Dication of Astaxanthin as Predicted Based on DFT Calculations and Resonance Raman Spectroscopy

The results from pulse radiolysis and time-resolved spectroscopy studies demonstrate that carotenoids possess the necessary reactivity to function as antioxidants through kinetically favored reactions, including electron transfer and radical addition when interacting with free radicals. Various tocopherols can reduce carotenoid radical cations, and lycopene can undergo reversible electron transfer with O_2_^•–^ due to its positive electronegativity. AX and canthaxanthin exhibit higher antioxidant effectiveness compared to β-carotene or zeaxanthin, despite slower rates of AMVN-induced oxidation.

Resonance Raman spectroscopy, along with theoretical calculations, has been utilized to investigate the molecular structures of radical species of AX, revealing significant changes in bond orders and vibrational modes. Figure 5 displays the steady-state absorption spectra of AX in acetone with different amounts of FeCl_3_ solutions (1 mM acetone solution) added. The addition of FeCl_3_ solution oxidizes AX, resulting in the formation of a radical cation peaking around 850 nm and a dication peaking around 700 nm, depending on the amount of FeCl_3_ solution added. Resonance Raman spectra of AX and its radical species were recorded and compared with the DFT calculations in Figure 6. The DFT calculations accurately replicate the ground-state (S_0_) Raman spectrum, while the resonance Raman spectrum of the radical species can be seen as a combination of the calculated Raman spectra of the radical cation and the dication of AX. This is due to the fact that the resonance Raman spectrum of the ground (S_0_) state species was recorded using 532 nm laser light, which resonates with the S_0_ → S_2_ absorption of AX, while the resonance Raman spectrum of the radical species was recorded using 808 nm laser light, which is in resonance with both the radical cation and dication of AX. The agreement between the theoretically calculated Raman spectra and the experimentally observed resonance Raman spectra enables a detailed discussion of the molecular structures of the radical species of AX. As an example, Figure 7 illustrates the bond lengths of the ground (S_0_) state, radical cation, and dication of AX. It is noteworthy that the bond alterations evident in the ground (S_0_) state species undergo dramatic changes in the radical cation species, with all C=C double bonds tending to elongate and all C-C single bonds tending to shrink. This suggests that the bond orders of all the C=C and C-C bonds in the conjugated polyene approach 1.5. In the case of the dication species, there is a striking reversal of bond alterations in the central part of the polyene chain, with C=C double bonds becoming C-C single bonds and C-C single bonds becoming C=C bonds. These significant changes in bond orders are accurately reflected in the vibrational modes, which were predicted based on DFT calculations. The theoretically predicted molecular structures of the radical species of AX, supported experimentally through resonance Raman spectroscopy, serve as a valuable tool for exploring the functions of the radical species of AX in biological systems.

Carotenoids exhibit the necessary reactivity to function as antioxidants through favorable reactions with free radicals, including electron transfer and radical addition. AX and canthaxanthin demonstrate higher antioxidant effectiveness compared to β-carotene or zeaxanthin, despite slower rates of AMVN-induced oxidation. Resonance Raman spectroscopy and theoretical calculations are employed to investigate the molecular structures of radical species of AX, uncovering significant changes in bond orders and vibrational modes. The agreement between the theoretically calculated Raman spectra and the experimentally observed resonance Raman spectra enables a detailed discussion of the molecular structures of the radical species of AX, providing valuable insights into their functions in biological systems. Furthermore, although AX often forms fatty acid esters in nature, it has been reported that there is no essential difference in the first oxidation potential between AX and its n-octanoic mono- and diesters. This suggests that the AX esters have similar scavenging rates for ^•^OH, ^•^CH3, and ^•^OOH radicals compared to AX itself [175].

##### Biochemical Aspects of AX Properties against ROS

The intracellular localization and ROS scavenging activity of AX were presented in an earlier review [130].

To summarize, inflammation, whether acute or chronic, generates ROS and often leads to oxidative stress *in vivo*. Important actions of AX include the inhibition of nuclear translocation of NFκB, which promotes inflammatory responses, and the activation of Nrf2, a transcription factor of a group of anti-inflammatory enzymes. In parallel, oxidative stress caused by ROS can be reduced by improving mitochondrial function, which is a major source of ROS *in vivo* [130,176]. 

In conclusion, AX could exert its typical effects *in vivo* by inducing multifaceted antioxidant activity beyond the antioxidant activity derived from the chemical properties of the compound itself. Their typical efficacies are shown in Section 3.

### 2.2. Astaxanthin; Distribution, Derivatives and Optical Structure in Nature

AX is found in a considerable number of organism species, which cover a wide taxonomic variety that ranges from bacteria to several eukaryote kingdoms, i.e., fungi (yeasts), algae, higher plants, and animals. Table 2 provides information on the approximate amounts and isomers of AX in major species. With the exception of animals, organisms that possess AX are capable of synthesizing it de novo through either the non-mevalonate pathway (also known as the MEP pathway) or the mevalonate pathway. The non-mevalonate pathway involves the conversion of pyruvate and glyceraldehyde 3-phosphate into 1-deoxy-D-xylulose-5-phosphate (DXP) and 2-C-methyl-D-erythritol-4-phosphate (MEP) [177]. Further details on these pathways can be found in Section 2.3.

#### 2.2.1. Bacteria and Archaea

Among bacteria, the genus *Paracoccus*, belonging to the class α-Proteobacteria (Alphaproteobacteria) in the phylum Proteobacteria, has been found to produce (3*S*,3′*S*)-AX [178,179,180]. Additionally, other bacteria such as *Brevundimonas*, *Sphingomonas*, and Altererythrobacter species have also been reported to have the ability to produce AX [181,182,183,184,185,186]. In these bacteria, AX biosynthesis occurs from β-carotene through oxygenation reactions catalyzed by two enzymes called CrtZ and CrtW. The two enzymes are β-C3-hydroxylase [β-carotene (β-carotenoid) 3,3′-hydroxylase] and β-C4-ketolase [β-carotene (β-carotenoid) 4,4′-ketolase (oxygenase)], and facilitate multistep hydroxylation and ketolation (oxygenation) reactions at the C3 and C4 positions of the β-ionone ring (β-end group), respectively (Figure 8) [37]. The CrtZ-type β-C3-hydroxylase responsible for the hydroxylation of the C3 and C3′ positions of the β-ionone rings generates a single stereo configuration, resulting in the production of (3*S*,3′*S*)-AX with a specific absolute optical configuration [187,188].

It was interestingly found that the phylum Cyanobacteria, which are photoautotrophic bacteria, possess a distinct β-C3-hydroxylase, designated CrtR, which shows moderate homology not to CrtZ but to CrtW [189]. Moreover, cyanobacteria were shown to retain a distinct β-C4-ketolase, designated CrtO, that shows significant homology to CrtI (phytoene desaturase) [190], in addition to CrtW. Thus, their presence could theoretically lead to the formation of AX. However, due to the substrate specificity of these enzymes, major carotenoids are not AX but its early-stage precursors such as 3′-OH-echinenone, echinenone, and zeaxanthin, as well as other cyanobacterium-related carotenoids such as myxol glycosides. Thus, AX is ordinarily not present or one of trace amounts of carotenoids in cyanobacteria [191,192], while previous reports, despite some debate regarding analytical methods and accuracy, described the presence of AX as a constitutive carotenoid in this phylum [193]. 

The reason is attributed to the extremely low or no reactivity of cyanobacterial CrtR towards a substrate with a 4-keto-β-end group. CrtR is ordinarily likely to mediate the synthesis of myxol 2′-fucoside by its β-C3-hydroxylase activity, while the *Synechocystis* sp. PCC 6803 CrtR can convert β-carotene into zeaxanthin, as shown by functional analysis using *E. coli* [189,191]. Similarly, functional analysis with *E. coli* demonstrated that cyanobacterial CrtO also exhibits very low or no reactivity towards a substrate with a 3-hydroxy-β-end group, since CrtO handles echinenone synthesis from β-carotene by its β-C4-ketolase activity (Figure 8) [194]. Moreover, it was found that cyanobacterial CrtW enzymes generally retain much lower activity for such a substrate, compared with *Paracoccus* and *Brevundimonas* CrtW proteins [195]. As a side note, a recent study reported that the introduction of the *crtW* and *crtZ* genes from *Brevundimonas* sp. SD212 into *Synechococcus* sp. PCC 7002, a type of cyanobacteria, resulted in the production and enhancement of AX productivity [191].

Although these AX products occur mainly in their free form in these bacteria, it has been reported that *Agrobacterium aurantiacum* (properly *Paracoccus* sp. strain N81106) produces a glycosylated AX, i.e., AX monoglucoside [196]. Additionally, *Sphingomonas astaxanthinifaciens* and *S. lacus* PB304 also contain AX dirhamnoside and AX dideoxyglucoside, respectively [197,198,199]. The enzyme gene that forms these glycosides, *crtX*, has been found in the complete gene clusters of *Paracoccus* sp. strain N81106 [37,200] and the genus *Sphingomonas*. Such an activity of CrtX was suggested using *Pantoea ananatis* CrtX with an AX-producing recombinant *E. coli* [201]. However, functional analysis using E. coli has not confirmed the ability of the putative gene of *Sphingomonas* to mediate such glycosylation reactions [188,198]. Furthermore, the presence of AX has also been implicated in the phyla Actinomycetota and Deinococcota (Deinococcus-Thermus); however, reliable structural analysis of AX and its biosynthetic genes in these phyla is still lacking. Further details regarding the bacterial AX biosynthetic pathway are discussed in Section 2.3.

It is indeed plausible that AX may play a protective role in cells exposed to intense sunlight and high levels of natural radiation. Bacteria that produce AX, such as those found in ocean surfaces, coastal areas, and hot springs, inhabit environments where they are subjected to these harsh conditions [181,182,183,184,196,202]. AX, with its antioxidant properties, has the potential to scavenge free radicals generated by UV radiation and protect cells from oxidative damage. Furthermore, AXs ability to absorb and dissipate excess light energy may also contribute to cellular photoprotection. These mechanisms suggest that AX could serve as a natural defense mechanism against the damaging effects of sunlight and radiation on these bacteria.

The presence of AX in halophilic archaea, which thrive in high-salinity environments where other organisms cannot survive, has been suggested [203,204]. In particular, studies on *Halobacterium salinarum* R1 have shown that depletion of the *CYP174A1* gene, which codes for a cytochrome P450 (CYP; P450), resulted in decreased production of AX. It is noteworthy that the genome of *H. salinarum* does not contain genes encoding CrtZ-type or CrtR-type β-C3-hydroxylase, CrtW-type β-C4-ketolase, or CrtO-type β-C4-ketolase, which may be involved in AX production in other organisms [204]. This suggests that CYP174A1 may have a role in the biosynthesis of AX in these halophilic archaea. However, the presence and distribution of AX in archaea as a whole have yet to be fully confirmed, and further research is needed to elucidate this aspect. 

#### 2.2.2. Eukaryotes; Fungi and Protozoa

Certain colored fungi are capable of producing carotenoids, including xanthophylls [205]. However, the production of AX (astaxanthin) in fungi is quite limited and is currently only observed in the genus *Xanthophyllomyces*. The yeast *Phaffia rhodozyma*, which is the anamorph of *Xanthophyllomyces* dendrorhous, was initially isolated from exudates of deciduous broadleaf trees at high altitudes in the northern hemisphere, where it is exposed to intense UV radiation [206,207]. In recent years, similar species belonging to the genus *Xanthophyllomyces* have also been discovered in the southern hemisphere, such as in Patagonia, South Australia, and Tasmania [208,209,210]. The differences in host trees and the evolutionary distances between these *Xanthophyllomyces* species from the northern and southern hemispheres suggest that they have been strongly influenced by the ancient continental separation of the Earth, which is of great interest from an Earth science perspective [211]. 

*Phaffia* yeast, including *Xanthophyllomyces* species, produce (3*R*,3′*R*)-AX in its free form [51]. The biosynthesis pathway of AX in *Phaffia* yeast is catalyzed by specific monooxygenase enzymes belonging to the P450 family, such as CrtS/Asy, along with a reductase enzyme. This pathway differs significantly from the metabolic pathway observed in bacteria [212,213]. It is believed that AX plays a crucial cytoprotective role for *Phaffia* yeast in adapting to these harsh environments. The biosynthesis of AX in *Phaffia* yeast is likely induced by redox imbalances, particularly those involving the NAD(P)H/NAD(P)+ couple and the oxidative environment [214]. 

Overall, the production of AX in fungi, particularly in *Phaffia* yeast, is an intriguing phenomenon, and its biosynthesis pathway and protective role in challenging environments are subjects of scientific interest and investigation.

Several zooplankton and phytoplankton species have the ability to synthesize AX de novo. The genera *Nannochloropsis* in the class Eustigmatophyceae, *Aurantiochytrium* in the class Labyrinthulea, and *Euglena* and *Trachelomonas volvocina* in the class Euglenophyceae, as well as certain arthropods like copepods and krill, have been reported to have the potential for AX production [215,216,217,218,219] (Table 2). These findings suggest the possibility of industrial-scale production of AX from these organisms. With advances in biotechnology and cultivation techniques, there is increasing interest in harnessing the AX production capacity of these zooplankton and phytoplankton species for commercial purposes. Industrial production of AX from these natural sources could provide a sustainable and renewable supply of this valuable compound for various applications. However, further research and optimization are required to fully exploit their potential and scale up the production process effectively.

#### 2.2.3. Eukaryotes; Algae and Higher Plants

It is worth noting that several green algae are known to accumulate extremely high concentrations of AX in cells under high sunlight, high salinity, and starvation conditions (Figure 9). The most well-known example is the green algae of the genus Hemacotococcus, particularly *H. lacustris* (Gir.-Chantr.) Rostaf. (=*H. pluvialis* Flot.). Please note that this document supports the official scientific name of *H. lacustris* (Gir.-Chantr.) Rostaf., as per the comprehensive opinion of Ota et al. [220,221]. 

Descriptions of *Haematococcus* algae (=*H. lacustris*), its biology, and life cycles emerged from the works of the German botanist Julius von Flotow in 1844 and the American botanist Tracy Elliot Hazen in 1899. Although early botanists described a “blood red pigment” produced by this algae [222], it was later determined that this was AX [49,223]. Nevertheless, since the description of “*H. pluvialis* Flot.” can also be frequently found in numerous publications, this species is referred to as “*Haematococcus* algae” in this paper to avoid misunderstandings, unless there are exceptions in specifying the strain name. Recent molecular genetics re-classification has revealed that *H. lacustris* strains are highly genetically diverse [224], and other species previously assigned to the *Haematococcus* genus may be reclassified as a separate genus, possibly leaving only one species in this genus, *H. lacustris* [48]. The properties and industrial production of this alga are presented in detail in Section 3.3.

In *Haematococcus* algae, (3*S*,3′*S*)-AX is present in the form of a fatty acid ester bonded to the hydroxyl group at the C3, C’3 position, with the main component being a monoester. Other specific species of green algae belonging to the genera *Acutodesmus, Asterarcys, Bracteacoccus, Botryococcus, Chlamydomonas, Chlorella (Chromochloris), Chlorococcum, Coelastrella, Monoraphidium, Neochloris, Protosiphon, Sanguina, Scenedesmus, Scotiellopsis, Tetraedron*, and *Vischeria* have also been reported to accumulate AX mainly in esters and/or free form, as well as in protein binding forms [114,225,226,227,228,229,230,231,232,233,234,235,236,237,238,239,240,241,242]. Another example is the fatty acid esters of AX diglucoside, which have been reported from a low-temperature-tolerant algae (*Chlamydomonas nivalis*) that grows on snowfields and glaciers in the Alps and polar regions worldwide [229]. Interestingly, a very recent report isolated *Dysmorphococcus globosus*-HI, belonging to the family Phacotaceae within the order *Chlamydomonadales*, class Chlorophyceae, from the Himalayan region of northern India, and it was found that AX accounts for about 56% of the dry intracellular weight. However, reproducibility and further studies on this report seem to be needed [243]. Additionally, many other phytoplankton also produce or accumulate carotenoids, including AX [244,245]. Therefore, it is likely that the accumulation of carotenoids in higher animals through the food chain is a result of the presence of carotenoids in plankton [246].

**Figure 9 marinedrugs-21-00514-f009:**
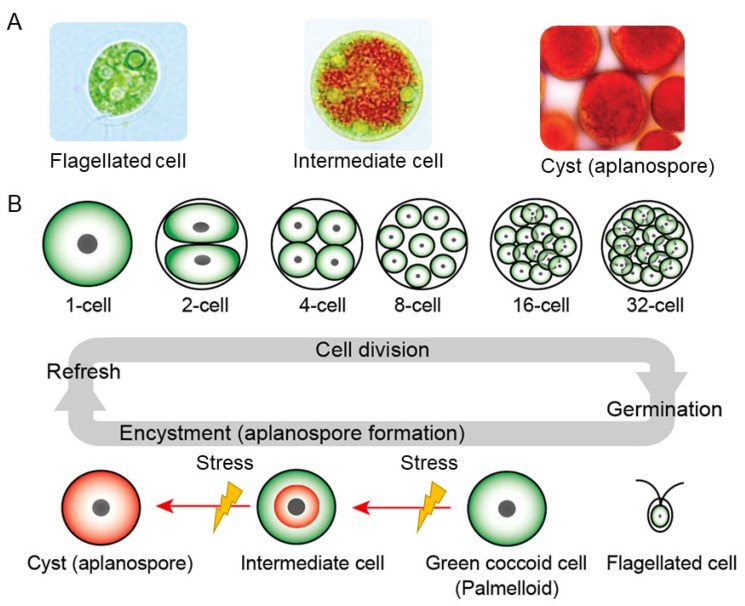
Life cycle of *Haematococcus* algae. (**A**) Three different cell morphologies of a typical *Haematococcus* algae. (**B**) Life cycle: When old cultures are transplanted into fresh medium, coccoid palmelloid cells undergo cell division to form flagellated cells within the mother cell wall. After germination, flagellated cells settle and form palmelloid cells. Environmental stress such as strong light, nutrient depletion, and/or high salinity accelerates the accumulation of astaxanthin during encystment. This figure was reproduced with some additional information, citing ref [247] under the terms of the Creative Commons Attribution License.

In these algae, AX provides three biological advantages to the algal cells: (1) stabilization of quiescent cells. Some of these algae form non-motile, extra-long-life, environmentally resistant cysts called aplanospores under severe environmental stress, as described above. These cysts are quiescent cells similar to plant seeds, with decreased chloroplast volume and a high accumulation of oil droplets containing high concentrations of AX in the cytoplasm [223,247,248,249,250,251]. In the case of *Haematococcus* algae, mature cyst cells (also often described as aplanospores [221,252] or akinetes [224]) exhibit a deep scarlet color (Figure 9). The biological mechanism of this unique accumulation in oil droplets is shown in detail in Section 2.3.1, which provides an example of the biosynthesis of AX in *Haematococcus* algae. AX is believed to protect the cells and their stored components, such as lipids and DNA, from the surrounding severe environment, including intense light exposure [253]. (2) In snow algae, AX surrounds and masks the chloroplasts in the algae, thereby promoting the ability to utilize red light wavelengths, which are much more effective for CO_2_ uptake than green or blue light [254]. (3) Similar to the putative role in (2), it has been reported that in *Haematococcus* algae, AX in green-red non-motile cyst cells, often called palmelloids before becoming scarlet-colored mature quiescent cyst cells, is localized in the cell center (the nuclear periphery) under normal light conditions but rapidly diffuses to the cell periphery when exposed to strong light. In cells where AX has completely moved to the periphery, a thin layer of AX is identified just beneath the cell wall, and if these cells are placed in the dark, AX redistributes to the center of the cell (the nuclear periphery) [250,255]. It has been hypothesized that this translocation may be facilitated by actin filaments [255]. In all cases (1) to (3), AX likely performs a similar physiological function in terms of protecting photosynthetic organs and intracellular components from intense light. Other organisms, such as *E. sanguinea*, which accumulates AX fatty acid esters [217,256], and *C. astaxanthina* Ki-4, which produces a complex of AstaP and AX [257], s show similar responses that accumulate AX derivatives under intense light, suggesting that AX plays the same role in a wide variety of microalgal species.

In terrestrial higher plants, carotenoids serve as accessory pigments in photosynthesis. They capture light energy as antennae pigments and also quench reactive oxygen species (ROS) generated by photosynthetic reaction centers under intense light [112] (see also Section 2.1.7). These primary carotenoids are typically located in the thylakoid membranes of chloroplasts and are essential for efficient photosynthetic reactions. They are biosynthesized in chloroplasts and are present within chloroplasts as well [258,259]. However, terrestrial higher plants often accumulate carotenoids in chromoplasts or lipid globules that are not directly involved in photosynthesis. Examples include the carotenoids found in fruits, flowers, and roots of many plants, such as lycopene in tomatoes and β-carotene in carrot roots. These carotenoids are known as secondary carotenoids [5,259]. AX is rarely found as a major carotenoid involved in photosynthetic function in higher plants, as mentioned previously. Therefore, AX is considered a secondary carotenoid rather than a primary carotenoid essential for photosynthesis. Some earlier studies reported the presence of AX in specific higher plants [260]. However, it was later confirmed that the reddish carotenoids found in autumn leaves were actually rhodoxanthins and their metabolites [261,262]. Currently, the only terrestrial plants where AX has been reliably identified as a major carotenoid are the reddish petals of certain species belonging to the genus *Adonis* of the Ranunculaceae family. In these plants, (3*S*,3′*S*)-AX is present in the form of fatty acid diesters [17,263,264,265]. Nevertheless, there remains a possibility that higher plants containing AX will be discovered in the future, as undiscovered plant species may still exist worldwide.

#### 2.2.4. Eukaryotes; Animals

With a few exceptions in certain arthropods, all animals lack the ability to produce carotenoids de novo [266]. Therefore, the AX that exists is either obtained from the diet or metabolically converted from precursor carotenoids. 

The coloration of invertebrates and poikilothermic vertebrates primarily depends on the types of chromatophores present in their integuments (i.e., skin and exoskeleton) (see Appendix A). While there are various types of chromatophores, both carotenoids and phenolic pteridines contribute to the red and yellow color patterns. Specifically, red chromatophores, known as erythrophores, contain ketocarotenoids like AX. Research on animal coloration, focusing on pigments, has predominantly examined ornamental color signals that indicate animal maturity and provide reproductive advantages. Recent hypotheses propose that oxidative stress plays a crucial role in maintaining the honesty of condition-dependent carotenoid-based signaling [267,268,269]. This is due to the potential of carotenoids to serve as honest indicators of phenotypic quality and, consequently, as targets for resource allocation trade-off hypotheses. In many animals, the amount of pigment available *in vivo* is limited, and its allocation may involve a trade-off between the animal’s mature phenotype and various essential biological functions, such as the immune system and antioxidant defense [270,271].

Concurrently, pigments have been extensively linked to prey-predator relationships [272,273,274,275]. Numerous studies have demonstrated that prey species exhibit vibrant colors as a means to signal the presence of predator defenses, thus serving as a mechanism to deter predators. In other words, predators advertise their defenses by being conspicuous, which enhances predator recognition and avoidance learning through uniform signaling. Long-wavelength color patches, such as red, orange, and yellow, are recognized as effective components of many visual warning signals, particularly when combined with black [276]. The red color derived from ketocarotenoids, including AX, holds significant prominence, especially for mammals like us.

##### In Case of Invertebrates

As mentioned above, AX contributes to the colorful body color and internal organs of invertebrates. Information on the distribution and conversion of carotenoids in aquatic organisms is detailed in Matsuno’s comprehensive review [21]. Similarly, another review on stereoisomers of hydroxyl groups at the C3 and C’3 positions of AX in aquatic animals is detailed in Matsuno et al. [277]. Therefore, in this review, we only provide information on AX in particular. Some examples of these are shown below.

In luminal animals, AX and “nor”-carotenoids, such as 2-norastaxanthin and actinioerythrin (Appendix A), are present in sea anemones [278], where the carbon at the 2 or 2′ position of AX is detached. AX has also been reported in jellyfish and corals [279,280]. It is taken up from crustacean-like zooplankton as a dietary source [280,281].

In mollusks, (3*S*,3′*S*)-AX has been found in coiled mollusks such as the spindle snail (*Fusinus perplexus*) and the apple snail (*Pomacea canaliculata*) [21]. Triton’s trumpet also contains AX, which is produced through the oxidative metabolic conversion of β-carotene and (3*R*,3′*R*)-zeaxanthin from starfish [281]. It is a mixture of three optical isomers that are taken up by the starfish when they feed on it. In bivalves, (3*S*,3′*S*)-AX is present as a trace component along with pectenolone in the ovaries of the Yesso scallop (*Mizuhopecten yessoensis*) [281,282]. Additionally, some Yesso scallops have orange adductor muscles [283]. Moreover, various colorful shells are found in the noble scallop (*Mimachlamys crassicostata*, synonym: *Chlamys nobilis*), with some individuals having golden shells and scallops with golden inner tissues [284]. These individuals have a high accumulation of carotenoids. The reasons for the difference in coloration of these bivalves are discussed in Section 2.3.3. Interestingly, it has been reported that these golden individuals have higher expression of antioxidant enzymes and improved immunological responses, as well as increased resistance to low-temperature stress compared to their brown counterparts [285,286,287]. AX is present in the viscera and ovaries of cephalopods (squids and octopuses) as a mixture of three stereoisomers [288].

In arthropods, the optical isomer composition of AX can be divided into three groups: (1) consisting of only one isomer or more than 90% of the dominant isomer; (2) mostly containing two isomers with a trace amount or a small proportion of another isomer; and (3) a mixture of all three isomers, with two at similar levels and the third in larger or smaller quantities (Table 2). While the majority of arthropods rely on dietary carotenoids, certain crustaceans, such as shrimps, prawns, and crabs, can convert β-carotene into AX through oxidative reactions involving echinenone, canthaxanthin, and adonirubin. With the exception of group (1), the AX in these cases consists of a mixture of three optical isomers due to the likely lack of stereospecificity in introducing hydroxyl groups at the C3 and C3′ positions [281]. In other arthropods, the conversion to AX appears to utilize β-carotene and zeaxanthin as substrates. Acari and copepods predominantly produce (3*S*,3′*S*)-AXs through this process [289,290]. However, it is noteworthy that Antarctic krill exhibits significantly higher levels of AXs with the (3*R*,3′*R*) stereo configuration [277,291]. Further details are provided in Section 2.3.3. Additionally, these AXs can exist in free, monoester, and diester forms. 

Furthermore, some AXs are present as protein complexes called crustacyanins [83]. Detailed information on crustacyanins was provided in Section 2.1.5. In lobsters (*Homarus gammarus*), crustacyanins, including β-crustacyanin (blue) and its octamer, α-crustacyanin (purple), and crustochrin (a yellow analog of these proteins), are found in the exoskeleton in the form of free AX associated stoichiometrically with apoproteins. Although the exoskeletons of these organisms are covered by a thick cuticular layer of epidermis, the localization of these characteristic carotenoproteins differs. In a study involving whitened American lobsters (*H. americanus*) fed a carotenoid-deficient diet, the accumulation of AX esters initially occurs in the epidermis, which then migrates and accumulates crustacyanins, resulting in the formation of a thick cuticle. Finally, multiple crustacyanin molecules are stacked like plates within the epicuticular layer to form crustochrin [292]. This study also demonstrated that UV irradiation promotes the accumulation of these carotenoproteins in the exoskeleton, suggesting that these pigments may serve as either cryptic coloration or protection against UV radiation [292]. On the outer surface of the carapace, where crustochrin is present, AX exists as an H_1_-aggregate, exhibiting a typical hypsochromic shift and absorbing light primarily in the UV region. Therefore, crustochrin itself exhibits a UV-shielding effect. Furthermore, the H_1_-aggregate of AX is considered less reactive with other compounds since there is no intermediate state for UV-induced triplet formation, and singlet fission occurs directly from the photoexcited state of ^1^Bu with an ultrafast time scale, which may be beneficial for the survival of lobsters from a physicochemical perspective [79] (see Section 2.1.6). An immunohistological study conducted on rock lobsters (*Panulirus cygnus*) also revealed the presence of cluster cyanine subunits not only in the cuticle layer of the exoskeleton but also in the epithelial layer. Moreover, *P. cygnus* is characterized by various patterned areas on its shell, including fine red and white spots and horizontal white stripes along the length of the shell. Immunohistochemical staining showed that crustacyanins were restricted to the colored areas on the carapace and the corresponding epidermis. These results suggest that the shell color pattern is formed through the expression of crustacyanins in the epithelium and their incorporation into the exoskeleton, rather than solely the absence of carotenoid chromophores [86].

In a study of the characterization and metabolism of carotenoproteins during embryonic development of the lobster *H. gammarus*, larvae contained carotenoproteins of three unknown structures (blue, red, and yellow) that were thought to be metabolized and transferred by an unknown factor (referred to as “AX”) from ovoverdin, the major carotenoprotein in egg yolk. Additionally, the egg yolk carotenoids were identified as free AX and adonirubin, while during tissue formation, AX was esterified. Therefore, the diester of AX was found to be the major carotenoid in embryos and larval tissues. This AX diester was found to be bound to red carotene proteins. The increase in esterified AX suggests that an early enzymatic mechanism leading to the acylation reaction occurs during embryogenesis [90].

AX has also been reported in insects, including grasshoppers [16,293,294] (Table 2). In most cases, these AX compounds are derived from their diet within the food chain. However, there are a few exceptions among arthropods, such as spider mites (Acari; e.g., *Tetranychus urticae*) [295,296] and aphids [297,298,299], which have the ability to synthesize carotenoids de novo. These organisms possess orthologous genes for enzyme proteins, namely *CrtYB* and *CrtI*, which are involved in the carotenoid biosynthetic pathway. It is believed that these genes originated in fungi and were acquired through lateral gene transfer [266,296,300] (for more details, refer to Section 2.3.3).

In echinoderms, it has been reported that AX exists as a mixture of three stereoisomers in starfish, sea cucumbers, and sea urchins [301,302,303]. From starfish, AX and its triple-bonded derivatives, namely 7,8-didehydroastaxanthin and 7,8,7′,8′-tetradehydroastaxanthin (refer to Appendix A), have been identified [280,304]. Previously, 7,8-didehydroastaxanthin and 7,8,7′,8′-tetradehydroastaxanthin were referred to as asterinic acid, named after the scientific name of the starfish species (*Asterias rubens*) [304,305].

In protochordates, the presence of (3*S*,3′*S*)-AX has been reported in sea squirts (*Halocynthia roretzi*), along with other major carotenoids [306].

##### In Case of Vertebrates

In fish, the presence of AX depends on the food chain, which means it is the result of absorption, accumulation, or metabolism from the diet. The red color on the body surface of many white meat fish species, apart from red meat fish, is attributed to AX and serves as the basis for the characteristic body color of each species. The origin of these body colors can be divided into the following patterns: (1) Salmonidae, including salmon and trout, which contain AX in their skeletal muscles; and marine white meat fishes with a red body surface, such as red snapper; and (2) carp fishes, which can metabolically convert AX from zeaxanthin. The former accumulates AX derived from crustaceans, which are their major food source, resulting in a mixture of three stereoisomers of AX [281]. The latter metabolically converts (3*R*,3′*R*)-zeaxanthin as a precursor, leading to the presence of only (3*S*,3′*S*)-AX [24,307]. Due to these characteristics, AX is often used as a coloring agent for snapper and salmon, while zeaxanthin is used as a coloring agent for goldfish (colored varieties of *Carassius auratus*) and Nishikigoi (colored varieties of *Cyprinus carpio*) [307]. Moreover, marine fish are frequently fed zeaxanthin as a coloring agent (Appendix A). The metabolism and conversion to and from AX will be discussed in detail in Section 2.3.3.

In the case of salmonid fish, AX is believed to originate from minuscule arthropods referred to as zooplankton. These zooplankton serve as prey for salmonid fishes, which in turn accumulate the pigments derived from them (refer to Table 2). Furthermore, there are significant differences in carotenoid composition between land-locked and migratory forms of salmonids due to variations in their dietary habits. Among salmonid fishes, the highest concentrations of AX have been reported in Sockeye salmon, and the ratio of AX to total carotenoids is also high in their muscles and eggs [308] (Table 2). 

Additionally, numerous fish species, particularly those belonging to the red-fleshed and salmonid categories, undergo metabolic conversion of AX into yellow xanthophylls like tunaxanthin or salmoxanthin via a reductive pathway. These yellow pigments are then deposited and stored in their integument (skin). Further elaboration on these processes will be provided in the forthcoming Section 2.3.3.

In amphibians, the presence of AX has been reported in newts and frogs/toads. In these organisms, AX exists in both free and ester forms (Table 2). It is believed that AX, along with other carotenoids, plays a significant role in warning coloration against predators and in the nuptial coloration that indicates individual maturity. Carotenoids responsible for body coloration are thought to be present in the dermis as xanthophores and erythrophores, as well as in liver tissue [309]. However, their specific forms in other tissues remain unclear. Similar to red-skinned fish, AX is believed to be present in erythrospores found in integument tissues. Since many amphibians feed on insects, it is possible that some carotenoids derived from arthropods, mainly carotenes, accumulate intact or undergo oxidative conversion to become AX. Unfortunately, many of the studies on this topic are relatively old and were conducted before the prevalence of HPLC. With a few exceptions, such as the Japanese newt studies, they provide limited information [309,310]. Therefore, further instrumental analysis is required to determine the precise form in which AX exists. Recently, it was reported that certain frogs may possess enzymatic genes that facilitate the conversion of C4-ketocarotenoids, such as AX, through oxidative metabolism [294] (for more details, refer to Section 2.3.3).

In reptiles, certain lizards and snakes display vibrant ornamental colors on their skin as a sign of maturity. These colors often include shades of yellow and red, which are attributed to the accumulation of high concentrations of carotenoids and phenolic pigments, such as pteridines. These orange to red pigments are typically found in the form of ketocarotenoids and, in some cases, contain high levels of AX (Table 2). In reptiles, these ketocarotenoids accumulate in the dermis within erythrophores [311,312,313]. Similar to reptiles, avian species also exhibit red-orange ornamental colors derived from C4-ketocarotenoids like AX [314]. In birds, AX has been reported in numerous species [314], most notably contributing to the body coloration of flamingos [15]. Interestingly, the ratio of AX to total carotenoids in flamingo feathers is relatively low, with canthaxanthin being the most abundant ketocarotenoid in many species. However, all species still contain significant amounts of AX, particularly in the legs of American flamingos [315]. While in most bird species, AX is visually identifiable in feathers, beaks, and legs, it has also been found in the skin beneath the plumage and other body tissues (e.g., liver, blood, etc.). As with other animals, there exists a trade-off between carotenoid ornamentation of the body surface and other physiological functions due to the limited dietary supply of carotenoids. These ketocarotenoids were initially believed to indicate individual maturity and be involved in defense against oxidative stress and regulation of immunity. However, the results have been inconsistent, and meta-analysis suggests that their presence reflects individual quality rather than consistently reducing oxidative stress [269]. The metabolism of these pigments is discussed in Section 2.3.3.

Considering the localization of AX in avian tissues, it is believed to coexist with collagen matrices and lipids in the subcutaneous tissue, dermis, and epidermis. Unlike reptiles, birds lack specific chromatophores other than melanocytes in their skin tissue, including erythrophores (see Appendix A). In feathers, AX is presumed to be present within the keratin matrices. Interestingly, AX is also found in the retinas of various bird species and turtles. It has been hypothesized that AX, along with other xanthophylls, may serve to protect retinal cells, particularly longwave-sensitive (LWS) cone cells, by acting as cut-off filters against blue light, especially in situations with intense light exposure [316,317,318]. Birds exhibit well-defined cone cells in the retina with oil droplets that contain specific carotenoids, including AX (R-type). These carotenoids are believed to have a wavelength-cutting effect on light, enhancing sensitivity in their respective wavelength ranges [316,317,318]. In LWS cone cells, the dominant optical isomers of AX are (3*S*,3′*S*)-AX [319]. In the majority of bird species, AX and other ketocarotenoids are presumed to be present in the retina, even in those without AX pigmentation in their plumage, with a few exceptions found in phylogenetic groups adapted to low-light environments, such as penguins and owls [320,321,322]. Penguins and owls inhabit nocturnal or deep-diving habitats and may have evolved colorless cone oil droplets to avoid interference with their vision, as colored droplets could impact their visual perception [318]. The functional role and evolutionary significance of these cone oil droplets are discussed in greater detail in other reviews (refer to [318,323]).

In addition to color patterns, some animals utilize polarized light patterns as a means of communication. The biological polarizers found in nature rely on physical interactions with light, including birefringence, differential reflection, and scattering. Interestingly, among invertebrates, a species called the marine stomatopod crustacean, specifically the mantis shrimp (*Odontodactylus scyllarus*), possesses a unique biological polarizer based on the linear dichroism of carotenoid molecules in its antennal scale. These creatures are believed to have a fundamentally different color recognition system from ours and to perceive and experience the world of colors in a distinct manner [324]. In the mantis shrimp, AX is deposited on the antennal scales, and the presence of AX in this dichroic compound allows the scales to polarize light. Within these antennal scales, AX exists as an optical mixture. By utilizing AX as a dichroic material in their antennal scales, mantis shrimps are able to manipulate the polarization component of light rather than its color, generating signals that vary with the direction of polarization [325]. Hence, AX plays a crucial role in the visual function of diverse organisms, including those employing polarized light communication.

In mammals, studies analyzing blood carotenoids have demonstrated that humans are capable of absorbing AX. Additionally, mice and rats have been observed to transfer AX to various tissues, including adipose tissue, liver, skeletal muscle, brain, and others [250,326]. More extensive information on these findings can be found in Section 4.1.3.

To summarize, animals discussed in Section 2.2.3 and Section 2.2.4 accumulate AX on their body surfaces, utilizing it for breeding purposes or as cryptic/warning coloration. AX is also found in abundance in eggs. It is speculated that AX may offer protection to eggs against light and reactive oxygen species (ROS) and is believed to be involved in reproductive functions, hatching, post-hatching survival, and growth. Further details and hypotheses regarding these aspects can be found in Section 3 and Section 5.

#### 2.2.5. Astaxanthin Content in Various Organisms

The content of AX per 100 g in various organisms is as follows: lobster/crayfish—0.1–0.3 mg in the whole body, Sockeye salmon muscle and eggs—2.0–3.8 mg, krill—2.0–4.0 mg, *Phaffia* yeast—20–1000 mg [327], dry biomass of *Paracoccus*—20–2000 mg, and mature cyst cells of *Haematococcus* algae—up to 9800 mg [328,329] (Table 2).

Among the reported species, *Haematococcus* algae has the highest AX content by a significant margin. Additionally, the purity of AX as a carotenoid is also high in *Haematococcus* algae. Therefore, the current industrial production of AX, apart from chemical synthesis, primarily relies on *Haematococcus* algae.

As a result, most of the evidence supporting the use of AX as a dietary supplement for humans is based on *Hematococcus* algae. Other sources of AX, whether biological or synthetic, have limited applications in human consumption but are widely used in animal feed. Additionally, some products, such as krill oil, are valued for their pharmaceutical benefits derived from other components, like omega-3 fatty acids, rather than solely relying on the effects of AX. Consequently, in this article, the focus on human use of AX will primarily revolve around *Hematococcus* algae.

**Table 2 marinedrugs-21-00514-t002:** Astaxanthin: Summary of its presence in various organisms and its origin.

Taxon	Scientific Name	Common Name	Astaxanthin				Reference
			Form ^†^	Stereoisomer (3*R*,3′*R*, *meso*, 3*S*,3′*S*)	Content (mg/100 g)	Origin	
**Bacteria, Prtoteobacteria, Alphaproteobacteria**							
***Paracoccus carotinifaciens*** (w/. mutation)		PanaFerd^-^AX	Free form	3*S,*3′*S*	**2180** (50.2% of total Car)	*De novo* synthesis	[329,330]
*Paracoccus* sp. strain N81106 (NBRC 101723) (*Agrobacterium auranticum)* (w/. mutation)		N/A	Free form and glycoside	3*S,*3′*S*	~800 (63.2% of total Car)	*De novo* synthesis	[331]
*Brevundimonas* sp. M7 (w/. mutation)		N/A	Free form **	3*S,*3′*S ***	130	*De novo* Synthesis **	[186]
*Sphingomonas astaxanthinifaciens* TDMA-17		N/A	Free form	3*S,*3′*S ***	96.0 (34.3% of total Car)	*De novo* Synthesis **	[182]
*Paracoccus haeundaensis* KCCM 10460 (Co-culture w/. Lactic Acid Bacteria)		N/A	Free form	3*S,*3′*S ***	82.1	*De novo* synthesis	[332]
*Paracoccus bogoriensis* BOG6T (DSM16578, LMG2279)		N/A	Free form	3*S,*3′*S*	40 (10.8% of total Car)	*De novo* synthesis	[183]
*Brevundimonas* spp.		N/A	Free form **	3*S,*3′*S ***	2.8~36.5	*De novo* Synthesis **	[186]
*Sphingomicrobium astaxanthinifaciens* CC-AMO-30B		N/A	Free form	3*S,*3′*S ***	4.0	*De novo* Synthesis **	[185]
*Brevundimonas* sp. strain SD212 (NBRC 101024)		N/A	Free form	3*S,*3′*S*	N/A (9.9% of total Car)	*De novo* synthesis	[181]
**Archaea**							
*Halobacterium salinarium* NRC-1		N/A	Free form ^‡^	N/D	26.5 (*c.a.*73% of total Car)	*De novo* Synthesis *	[203]
*Haloarcula hispanica* ATCC 33960		N/A	Free form ^‡^	N/D	1.7 (*c.a.*1.3% of total Car)	*De novo* *Synthesis**	[203]
**Eukaryota, Fungi**							
*Xanthophyllomyces dendrorhous* (ATCC SD 5340)		*Phaffia* Yeast	Free form	3*R,*3′*R*	723.5~1247.8 (*c.a.* 73% of total Car)	*De novo* synthesis	[327,333]
**Eukaryota, Plantae**							
***Adonis amurensis*** (Reddish flower varieties)		Amur adonis, pheasant’s eye	Fatty acid esters	3*S,*3′*S ***	**~3310** (in upper red part of petal) (*c.a.* 70% of total Car)	*De novo* Synthesis **	[334]
*Adonis annua*		Autumn pheasant’s eye, blooddrops	Fatty acid esters	3*S,*3′*S*	120~1000 (in dry petal) (*c.a.* 75% of total Car)	*De novo* Synthesis **	[17,50]
*Adonis aestivalis*		Summer pheasant’s eye	Fatty acid esters	3*S,*3′*S*	166 (in wet petal) (87.4% of total Car)	*De novo* synthesis	[265]
**Eukaryota, Plantae, Chlorophyta**							
** *Haematococcus lacustris* **		*Haematococcus* *pluvilialis*, *Haematococcus* algae	Fatty acid esters	3*S,*3′*S*	**~9800** (in red cyst) (>90% of total Car)	*De novo* synthesis	[64,328]
*Neochloris wimmeri* CCAP-213/4		N/A	Fatty acid esters	3*S,*3′*S ***	~1920 (*c.a* 85% of total Car)	*De novo* Synthesis **	[227,228,232]
*Asterarcys quadricellulare* PUMCC 5.1.*1*		N/A	N/A	3*S,*3′*S***	~1550 (*c.a* 13% of total Car)	*De novo* Synthesis **	[236]
*Protosiphon botryoides* SAG-731/1a		N/A	Fatty acid esters	3*S,*3′*S***	~1430 (*c.a* 80% of total Car)	*De novo* Synthesis **	[227,228]
*Scotiellopsis oocystiformis* SAG-277/1		N/A	Fatty acid esters	3*S,*3′*S ***	~1090 (*c.a* 70% of total Car)	*De novo* Synthesis **	[227,228]
*Chlorococcum* sp.		N/A	Fatty acid esters	3*S,*3′*S ***	~*c.a.* 700 (*c.a.* 32% of total Car)	*De novo* Synthesis **	[335,336,337]
*Chlorella zofingiensis* SAG-211/14		Chlorella	Fatty acid esters	3*S,*3′*S ***	~680 (*c.a.* 75% of total Car)	*De novo* Synthesis **	[227,228]
*Scenedesmus vacuolatus* *SAG-211/15*		N/A	Fatty acid esters	3*S,*3′*S ***	~270 (40–50% of total Car)	*De novo* Synthesis **	[227,228]
*Chlamydocapsa* spp. Strain 101-99/R2		N/A	N/A	3*S,*3′*S ***	~44.4 (20.3% of total Car)	*De novo* Synthesis **	[338]
*Neochloris oleoabundans* UTEX#1185		N/A	Fatty acid esters	3*S,*3′*S ***	N/A	*De novo* Synthesis **	[232]
*Dysmorphococcus globosus-HI*		N/A	Free form/ Fatty acid esters	3*S,*3′*S ***	~517,090??	*De novo* Synthesis **	[243]
**Eukaryota, Chromista, Bigyra, Labyrinthulomycetes**							
*Aurantiochytrium* sp. RH-7A-7 (w/. mutation)		Labyrinthulomycetes	N/A	3*S,*3′*S ***	-470 (*c.a.* 85% of total Car)	*De novo* Synthesis *	[218]
*Thraustochytrium* sp. CHN-3 (FERM P-18556)		Labyrinthulomycetes	Free form **	3*S,*3′*S ***	~280 (~60% of total Car)	*De novo* Synthesis *	[339]
*Aurantiochytrium* sp. KH-10		Labyrinthulomycetes	Fatty acid esters/ Free form	3*S,*3′*S ***	~81 (28% of total Car)	*De novo* Synthesis *	[340]
*Thraustochytrium* sp. CHN-1		Labyrinthulomycetes	Free form	3*S,*3′*S*	50 (*c.a.* 50% of total Car)	*De novo* Synthesis *	[341,342]
**Eukaryota, Chromista, Gyrista** **, Eustigmatales**							
*Nannochloropsis gaditana* strain *S4* (w/. mutation)		*Nannochloropsis*	Free form	3*S,*3′*S*?	~219 (14.4% of total Car)	*De novo* Synthesis	[219]
*Nannochloropsis oculata*		*Nannochloropsis*	Free form	3*S,*3′*S*?	3.4 ng/10^6^ cells	*De novo* Synthesis	[343]
*Nannochloropsis* *salina*		*Nannochloropsis*	Free form	3*S,*3′*S*?	9.6 ng/10^6^ cells	*De novo* Synthesis	[343]
**Eukaryota, Excavata, Euglenozoa**							
*Euglena sanguinea*		Euglena	Fatty acid esters/free	3*S,*3′*S*	~1.9 (80% of total Car)	*De novo* Synthesis	[256,344]
*Trachelomonas volvocina*		Euglena	Fatty acid esters/ Free form	3*S,*3′*S **	N/A	*De novo* Synthesis	[215]
**Animals (Invertebrate), Coelenterata**							
*Velella velella*		By-the-wind sailor (Jerry fish)	Free form	Mixtures of stereoisomers	N/A	Accumulated from dietary Crustaceans	[98]
*Aurelia aurita*		(Jerry fish)	Fee form/ Fatty acid esters (minor)	N/A	12.2 (*c.a.*67% of total Car)	N/A	[279]
*Metridium senile* var*. fimbriatum*		Frilled anemone (Sea anemone)	Fatty acid esters (in ovary)	Mixtures of stereoisomers	N/A	Oxidative metabolite of *β*-carotene	[345,346]
*Corynactis californica*		Strawberry anemone (Sea anemone)	Fatty acid esters	N/A	N/A	Oxidative metabolite of *β*-carotene	[347]
**Animals (Invertebrate), Mollusca** **, Gastropoda**							
*Clione limacina*		Sea angel	Free form	3*S,*3′*S*	0.051 (1.1% of total Car)	Oxidaive metabolite of zeaxanthin	[348]
*Paedoclione doliiformis*		Sea angel	Free form	3*S,*3′*S*	0.8 (5.5% of total Car)	Oxidaive metabolite of zeaxanthin	[348]
*Semisulcospira libertina*		Terestorial Snail (Kawanina in Japanese)	Free form	3*S,*3′*S*	0.2 (6.5% of total Car)	Oxidaive metabolite of zeaxanthin	[349]
*Fushinus perplexu*		Spindle shell	Free form	3*S,*3′*S*	0.2 (4.0% of total Car)	Oxidative metabolite of *β*-carotene	[350]
*Pomacea canaliculata*		Apple snail	Free form	3*S,*3′*S*	5.0 in gonad, 2.31 in egg (~75% of total Car)	Oxidative metabolite of *β*-carotene	[351]
**Animals (Invertebrate), Mollusca** **, Cephalopoda**							
*Octopus vulgaris*		Common octopus	Fatty acid esters/ Free form	Mixtures of stereoisomers (46:22:32)	3.2 in liver (*c.a*.80% of total Car)	Accumulated from dietary crustaceans	[288]
*Watasenia scintillans*		Firefly squid	Fatty acid esters/ Free form	Mixtures of stereoisomers (40:6:54)	5.0 in liver (>90% of total Car)	Accumulated from dietary crustaceans	[288]
**Animals (Invertebrate), Mollusca, Polyplacophora**							
*Placiphorella japoonica*		Chiton	Free form	Mixtures of stereoisomers (5:3:2)	1.25 (~34% of total Car)	Oxidative metabolite of *β*-carotene	[352]
*Acanthochitona defilippii*		Chiton	Free form	3*S,*3′*S*	1.55 in gonad (~4.0% of total Car)	Oxidative metabolite of *β*-carotene	[352]
*Liolophura japonica*		Chiton	Free form	3*S,*3′*S*	0.8 in viscera (~10% of total Car)	Oxidative metabolite of *β*-carotene	[352]
**Animals (Invertebrate), Echinodermata**							
*Peronella japonica*		Sea urchin	Free form	Mixtures of stereoisomers (3:7:90)	~3.0 in gonad (*c.a.*43% of total Car)	Oxidative metabolite of *β*-carotene	[301]
*Asteria pectinifera*		Starfish	Free form	Mixtures of stereoisomers (50:25:25)	~1.35 (% of total Car)	Oxidative metabolite of *β*-carotene	[305]
*Asterias amurensis*		Starfish	Free form	Mixtures of stereoisomers (48:25:27)	~4.64 (% of total Car)	Oxidative metabolite of *β*-carotene	[305]
**Animals (Invertebrate), Arthropoda, Crustacea, Decapoda (Lobsters, rock lobsters and crawfishes)**							
*Procambarus clarkii*		Louisiana crawfish	Fatty acid ester/ Free form	Mixtures of stereoisomers	7.9–19.8 in carapace	Oxidative metabolite of *β*-carotene	[353]
*Pontastacus leptodactylus* (*Astacus leptodactylus*)		Turkish crayfish	Fatty acid esters /Free form	Mixtures of stereoisomers **	5.0 in carapace 0.13 in muscle 0.98 in intestine (82.5% of total Car)	Oxidative metabolite of *β*-carotene **	[354]
*Panulirus japonicus*		Japanese Spiny Lobster (Ise-ebi)	Free form/ Fatty acid esters	Mixtures of stereoisomers (20:20:56)	3.3 in carapace (65% of total Car)	Oxidative metabolite of *β*-carotene	[355]
**Animals (Invertebrate), Arthropoda, Crustacea, Copepoda**							
*Tigriopus californicus*		Red marine copepod	Free form (major)	3*S*,3′*S* (major)	~423	Oxidative metabolite of *β*-carotene	[290,356]
**Animals (Invertebrate), Arthropoda, Crustacea, Eucarida**							
*Euphausia superba*		Antarctic krill	Fatty acid esters/ Free form	3*R*,3′*R* (Major, ~70%)	~566 in eye	Oxidative metabolite of *β*-carotene ?	[291,357]
*Euphausia* *pacifica*		Pacific krill (Isada)	Fatty acid esters/ Free form	3*R*,3′*R* (Major)	~ 252 in eye	Oxidative metabolite of *β*-carotene ?	[357,358]
**Animals (Invertebrate), Arthropoda, Crustacea, Decapoda (Prawns and shrimps) *****							
*Pandalus borealis*		Atlantic shrimp (Northern prawn)	Fatty acid esters/ Free form	Mixtures of stereoisomers (25:52:23)	~28.48 in carapace	Oxidative metabolite of *β*-carotene	[62,359,360]
*Penaeus japonicus*		Japanese tiger prawn (Kuruma-ebi)	Fatty acid esters/ Free form	Mixtures of stereoisomers (12:40:48)	~13 in carapace	Oxidative metabolite of *β*-carotene	[361,362]
*Penaeus semisulcatus*		Green tiger prawn	Fatty acid esters/ Free form	Mixtures of stereoisomers (19:44:57)	~15.6 in carapace	Oxidative metabolite of *β*-carotene	[362]
*Penaeus monodon*		Black tiger prawn	Fatty acid esters/ Free form	Mixtures of stereoisomers (16:43:41)	~7.3 in carapace	Oxidative metabolite of *β*-carotene	[362]
*Litopenaeus vannamei*		Whiteleg shrimp	Fatty acid esters/ Free form	Mixtures of stereoisomers (23:44:32)	~5.8 in carapace	Oxidative metabolite of *β*-carotene	[362]
*Metapenaeus joyneri*		Shiba shrimp	Fatty acid esters/ Free form	Mixtures of stereoisomers (14:46:40)	~3.3 in carapace	Oxidative metabolite of *β*-carotene	[362]
**Animals (Invertebrate), Arthropoda, Crustacea, Decapoda , Brachyura (Crabs) *****							
*Chionoecetes japonicus*		Red snow crab (Beni-zuwai crab)	Fatty acid esters/ Free form **	Mixtures of stereoisomers **	~23 in carapace (with demineralization treatment)	Oxidative metabolite of *β*-carotene?	[363]
*Chionoecetes opilio*		Snow crab (Zuwai crab)	Fatty acid esters/ Free form	Mixtures of stereoisomers?	~11.9 in carapace (~91.7% of total Car)	Oxidative metabolite of *β*-carotene?	[364]
*Callinectes sapidus*		Blue crab	Fatty acid esters/ Free form	Mixtures of stereoisomers?	~9.8 (with demineralization treatment)	Oxidative metabolite of *β*-carotene?	[365]
*Cancer pagurus*		Brown crab	Fatty acid esters/ Free form	Mixtures of stereoisomers (56:24:20)	0.37 in carapace	Oxidative metabolite of *β*-carotene?	[366,367]
**Animals (Invertebrate), Arthropoda, Crustacea, Decapoda (Others) *****							
*Paralithodes brevipes*		Hanasaki crab	Fatty acid esters/ Free form	Mixtures of stereoisomers (26:9:6)	~2.4 in carapace (~39.9% of total Car)	Oxidative metabolite of *β*-carotene	[368]
*Paralithodes camtschaticus*		Red king crab	Fatty acid esters/ Free form	Mixtures of stereoisomers (45–55:7–19:27–48)	~0.35 in carapace (~97% of total Car)	Oxidative metabolite of *β*-carotene	[277,369]
*Cervimunida princeps*		Squat lobster	Fatty acid esters/ Free form	Mixtures of stereoisomers (26:9:65)	~ 0.45 in carapace (~100% of total Car)	Oxidative metabolite of *β*-carotene	[369]
*Upogebia major*		Japanese mud shrimp	Fatty acid esters/ Free form	Mixtures of stereoisomers (72:21:7)	~ 0.25 in carapace (~100% of total Car)	Oxidative metabolite of *β*-carotene	[369]
*Birgus latro*		Coconut crab	Fatty acid esters/ Free form	Mixtures of stereoisomers (9:41:50)	~ 0.3 in carapace (~96% of total Car)	Oxidative metabolite of *β*-carotene	[370]
*Asellus aquaticus*		Isopoda	Free form/ Fatty acid esters?	N/A	~0.52 (~37.5% of total Car)	Oxidative metabolite of *β*-carotene?	[371]
*Pleuroncodes planipes*		Red crab langostilla	Fatty acid esters/ Free form	Mixtures of stereoisomers (3–4:1:3–4)	N/A	Oxidative metabolite of *β*-carotene?	[372]
**Animals (Invertebrate), Arthropoda, Arachnida, Acari**							
** *Balaustium murorum* **		**Red velvet mite**	Free form/ Fatty acid esters	3*S,*3′*S ***	**~61,530 mg/** 100 g protein (60% of total Car)	Oxidative metabolite of zeaxanthin *(De novo* Synthesis **)	[373]
*Panonychus citri*		Citrus red mite	Fatty acid esters	3*S,*3′*S ***	~263 mg/ 100 g protein (42.5% of total Car)	*De novo* Synthesis **	[374]
*Tetranychus kanzawai*		Kanzawa spider mite	Fatty acid esters	3*S,*3′*S ***	Undefined	*De novo* Synthesis	[375]
*Tetranychus urticae*		Two-spotted spider mite	Fatty acid esters	3*S,*3′*S ***	Undefined	*De novo* Synthesis	[296,376]
*Eylais hamata*		Hydracarina	Free form/ Fatty acid esters (minor)	N/A	12.2 (*c.a.*67% of total Car)	N/A	[377]
*Eylais extendens*		Hydracarina	N/A	N/A	Undefined (*c.a.*70% of total Car)	N/A	[378]
*Schizonobia sycophanta*		Parasite mite	Fatty acid ester	3*S,*3′*S*	Undefined (30% of total Car)	*De novo* Synthesis **	[49,289]
**Animals (Invertebrate), Arthropoda, Arachnida, Araneae**							
*Trichonephila clavata*		Arachnida spider	??	Mixtures of stereoisomers (2:1:1)	0.02 (1.9% of total Car)	Oxidative metabolite of *β*-carotene	[299]
**Animals (Invertebrate), Arthropoda, Insecta**							
*Locusta migratoria*		Migratory locust	Free form	Mixtures of stereoisomers (2:1:1)	0.25 in brown form (12.5% of total Car)	Oxidative metabolite of *β*-carotene	[16,293]
*Aiolopus thalassinus tamulus*		Grasshopper	Free form	Mixtures of stereoisomers (2:1:1)	0.09 in brown form (3.0% of total Car)	Oxidative metabolite of *β*-carotene	[293]
*Schistocerca gregaria*		Desert locust	Free form	Mixtures of stereoisomers *	N/A	Oxidative metabolite of *β*-carotene	[16]
**Animals (Vertebrate), Fish (Salmonidae)**							
***Oncorhynchus nerka*** (Wild, anadromous form)		**Sockeye salmon**	Free form (muscle/egg)/ Fatty acid esters (skin)	Mixtures of stereoisomers (depending on AX source)	**2.6–3.8 in muscle** in egg (% of total Car)	Accumulated from dietary crustaceans	[308]
*Oncorhynchus nerka* (Wild, non-anadromous form)		Kokanee salmon	Free form (muscle/egg)/ Fatty acid esters (skin)	Mixtures of stereoisomers (depending on AX source)	−0.8 in muscle 0.4–2.8 in skin −1.1 in golnad (−94% of total Car)	Accumulated from dietary crustaceans	[379,380]
*Oncorhynchus kisutch*		Coho salmon	Free form (muscle/egg)/ Fatty acid esters (skin)	Mixtures of stereoisomers (depending on AX source)	1.0–2.1 in muscle in egg (% of total Car)	Accumulated from dietary crustaceans	[308]
*Salvelinus alpinus* (Wild)		Arctic char	Free form (muscle/egg)/ Fatty acid esters (skin)	Mixtures of stereoisomers (depending on AX source)	0.86 in muscle in egg (−30% of total Car)	Accumulated from dietary crustaceans	[308,381]
*Salmo salar* (Wild)		Atlantic salmon	Free form (muscle/egg)/ Fatty acid esters (skin)	Mixtures of stereoisomers (depending on AX source)	0.6–0.8 in muscle in egg (% of total Car)	Accumulated from dietary crustaceans	[308,382]
*Oncorhynchus keta*		Chum salmon	Free form (muscle/egg)/ Fatty acid esters (skin)	Mainly 3*S,*3′*S* (in ovary)	0.1–0.5 in muscle −0.7 in egg 0.1 in skin (male) (4.8–90% of total Car)	Accumulated from dietary crustaceans	[28,29,308,383]
*Oncorhynchus gorbuscha*		Pink salmon	Free form (muscle/egg)/ Fatty acid esters (skin)	Mixtures of stereoisomers	0.4–0.7 in muscle in egg (% of total Car)	Accumulated from dietary crustaceans	[308]
*Oncorhynchus tshawytscha*		Chinook salmon	Free form (muscle/egg)/ Fatty acid esters (skin)	Mixtures of stereoisomers	0.54 in muscle in egg (% of total Car)	Accumulated from dietary crustaceans	[308]
*Oncorhynchus masou* (Wild, anadromous form)		Masu salmon	Free form (muscle/egg)/ Fatty acid esters (skin)	Mixtures of stereoisomers	0.3–0.8 in muscle 0.03–0.8 in skin 0.7–1.7 in egg (1.9–80% of total Car)	Accumulated from dietary crustaceans	[308,380,384]
*Oncorhynchus masou ishikawae* (Wild, anadromous form)		Red-spotted masu salmon	Free form (muscle/egg)/ Fatty acid esters (skin)	Mixtures of stereoisomers	0.2 in muscle trace in skin N/D in egg (1.9–68.5% of total Car)	Accumulated from dietary crustaceans	[380]
*Oncorhynchus masou rhodurus*		Biwa trout	Free form (muscle/egg)/ Fatty acid esters (skin)	Mixtures of stereoisomers	0.2 in muscle −0.1 in skin (3.2–58.3% of total Car)	Accumulated from dietary crustaceans	[380]
*Oncorhynchus mykiss* (Wild, pigmented phenotype)		Rainbow trout	Free form (muscle/egg)/ Fatty acid esters (skin)	Mixtures of stereoisomers	trace in muscle 0.8 in skin trace in egg (1.9–42.3% of total Car)	Accumulated from dietary crustaceans	[385]
**Animals (Vertebrate), Fish (Non-Salmonidae )**							
*Sebastolobus macrochir*		Broadbanded thornyhead (Kichiji rockfish)	Fatty acid esters (skin)	N/A	26 in skin (>90% of total Car)	Accumulated from dietary crustaceans or Oxidative metabolite of *β*-carotene/ zeaxanthin?	[386]
*Plectropomus leopardus*		Coral trout (Suziara)	Fatty acid esters/ Free form (skin)	Mixtures of stereoisomers (13:7:80)	19.5 in skin (84.8% of total Car)	Accumulated from dietary crustaceans or Oxidative metabolite of *β*-carotene/ zeaxanthin?	[387]
*Epinephelus fasciatus*		Blacktip grouper (Akahata)	Fatty acid esters (skin)	N/A	2.27 in skin (74% of total Car)	Accumulated from dietary crustaceans or Oxidative metabolite of *β*-carotene/ zeaxanthin?	[386]
*Beryx splendens*		Splendid alfonsino (Kinmedai)	Fatty acid esters (skin)	Mixtures of stereoisomers	0.9 in skin (*c.a*. 100% of total Car)	Accumulated from dietary crustaceans or Oxidative metabolite of *β*-carotene/ zeaxanthin?	[386]
*Pagrus malor*		Red sea bream (Madai)	Fatty acid esters (skin)	Mixtures of stereoisomers (38:0:62)	~2 in skin (wild) 0.25 in skin (firmed w/o. AX) / 0.98 (firmed with. AX) (~*c.a* 45% of total Car)	Accumulated from dietary crustaceans and supplementary pigment	[277,386,388]
*Carassius auratus*		Goldfish (Kingyo/Hibuna)	Free/ Fatty acid esters (skin)	3*S,*3′*S*	0.58 (whole body) (~47% of total Car)	Oxidative metabolite of *β*-carotene/ zeaxanthin	[307]
*Branchiostegus japonicus*		Red tilefish (Red amadai)	Fatty acid esters (skin)	Mixtures of stereoisomers (24:24:52)	0.39 in skin (35.8% of total Car)	Accumulated from dietary crustaceans	[389]
**Animals (Vertebrate), Amphibian**							
*Cynops pyrrhogaster*		Japanese newt	Free form/ Fatty acid esters	N/A	4.55 in skin (*c.a*.21% of total Car)	Oxidative metabolite of *β*-carotene/ zeaxanthin	[309]
*Salamandra salamandra*		Fire salamander	Free form/ Fatty acid esters	N/A	0.23 (37.5% of total Car)	Oxidative metabolite of *β*-carotene/ zeaxanthin	[390]
*Lissotriton vulgaris* (*Triturus vulgaris*)		Smooth newt/common newt	Free form	N/A	0.1 (−23.5% of total Car)	Oxidative metabolite of *β*-carotene/ zeaxanthin	[390]
*Ranitomeya sirensis*		Sira poison frog	Free form/ Fatty acid esters ?	N/A	N/A (-*c.a.* 40% of total Car)	Oxidative metabolite of *β*-carotene/ zeaxanthin	[294]
*Bufo bufo*		Common toad	Free form/ Fatty acid esters	N/A	N/D in muscle and liver 0.02 in skin 0.35 in intestine (25.8–57.4% of total Car) 0.23 in gonads (95.8% of total Car)	Oxidative metabolite of *β*-carotene/ zeaxanthin	[390]
*Bufotes viridis* *(Bufo viridis)*		European green toad	Free form/ Fatty acid esters	N/A	N/D in muscle and liver 0.02 in skin 0.35 in intestine (25.8–57.4% of total Car) 0.23 in gonads (95.8% of total Car)	Oxidative metabolite of *β*-carotene/ zeaxanthin	[391]
*Pelobates fuscus*		European common spadefoot toad	Free form	N/A	1.1 in liver (19.6% of total Car)	Oxidative metabolite of *β*-carotene/ zeaxanthin **	[390]
*Melanophryniscus rubriventris*		(Aposematic poison toad)	N/A (Free form/ ester form?)	N/A	Undefined	Oxidative metabolite of *β*-carotene **	[392]
**Animals (Invertebrate), Reptile**							
*Chlamydosaurus kingii.* (the western red-frilled form)		Frillneck lizard	N/A (Free form?)	N/A	Undefined	Oxidative metabolite of *β*-carotene	[312]
*Chrysemys picta*		Painted turtle	N/A	N/A	*c.a* 0.11 in leg skin	Oxidative metabolite of *β*-carotene	[393]
*Trachemys scripta*		Red-eared slider	N/A	N/A	*c.a* 0.06 in skin around eye spot	Oxidative metabolite of *β*-carotene	[393]
**Animals (Invertebrate), Aves *****							
*Lagopus lagopus scoticus*		Red grouse	Free form/ Fatty acid esters	N/A	317.8 in combs N/D in plasma (-*c.a.*81.6% of total Car)	Oxidative metabolite of *β*-carotene	[394]
*Pygoscelis papua*		Gentoo penguins	Free form	N/A	2.19 in blood, breeding adults and chicks	Accumulated from dietary crustaceans, fishes	[395,396]

The respective number was quoted from the reference(s), and it may vary depending on the collection location and season.^†^, the presence of binding forms to carotenoproteins would not be mentioned in this table; ^‡^, the identification method of the compounds remains uncertain; *, the biosynthetic pathways have not been fully characterized; **, based on the information on close species/genus; N/A; not available; N/D; not detected.*** Since astaxanthin is diversely found in the skin, feathers, and retinas of birds, only the characteristic reports are described. ?; based on the information on close taxa. For the details of distribution in avian species, see the other review [314].

### 2.3. Biosynthesis and Metabolism of Astaxanthin

#### 2.3.1. Overview of Carotenoid Biosynthetic Pathways in Bacteria, Fungi, and Higher Plants

The biosynthesis of AX is entirely based on β-carotene as the common precursor. Therefore, this document omits a detailed discussion of the metabolic pathway to β-carotene. Briefly, carotenoids belong to isoprenoids, which are the most diverse group of natural compounds. Isoprenoids are commonly biosynthesized from isopentenyl diphosphate (IPP). IPP is synthesized via the mevalonate pathway, which is present in almost all eukaryotes (the domain Eukarya) and archaea (the domain Archaea), as well as some actinobacteria (the phylum Actinomycetota). Alternatively, the MEP (2-C-methyl-D-erythritol 4-phosphate) pathway is used, which begins with pyruvate and glyceraldehyde 3-phosphate and proceeds through 1-deoxy-D-xylulose 5-phosphate (DXP) and MEP. This pathway is found in almost all bacteria and the chloroplasts of photosynthetic eukaryotes. IPP is isomerized to dimethylallyl diphosphate (DMAPP) through the action of IPP isomerase (Idi; IDI). Subsequently, DMAPP is converted to farnesyl diphosphate (FPP) and geranylgeranyl diphosphate (GGPP) through sequential condensation reactions with IPP. These reactions are catalyzed by FPP synthase and GGPP synthase, respectively [397].

Figure 10 illustrates the biosynthetic pathway of carotenoids from FPP in the leaves of higher plants, highlighting the enzyme-catalyzed reactions involved. Additionally, this figure presents the functions of carotenoid biosynthesis enzymes derived from bacteria and fungi [155]. According to past reports, the group of enzymes involved in the carotenoid synthesis pathway in green algae is composed of a set of genes that share homology with those of higher plants [398].

#### 2.3.2. Overview of Astaxanthin Biosynthetic Pathways in Bacteria, Algae and Plants

The β-C3-hydroxylation and β-C4-ketoxylation reactions involved in AX biosynthesis are present in a wide taxonomic range of organisms. These reactions are mainly catalyzed by membrane-integral, diiron, nonheme oxygenase superfamily enzymes, and occasionally heme-dependent cytochrome P450-type monooxygenase enzymes. Detailed information regarding these enzymes is provided below. 

In Proteobacteria, such as the genera *Paracoccus* and *Brevundimonas*, the genes responsible for AX biosynthesis are well understood (refer to Section 2.2.1). AX is synthesized from β-carotene through the coordinated actions of CrtZ-type β-C3-hydroxylase (β-carotene 3,3′-hydroxylase; CrtZ) (EC: 1.14.15.24) and CrtW-type β-C4-ketolase (β-carotene 4,4′-oxygenase; CrtW) (EC: 1.14.99.63; including an incorrect description on the substrate specificity) with β-carotene as the initial substrate (see Figure 8 and Figure 11) [37,399]. However, due to variations in enzyme reactivity among different species, AX-producing bacteria often accumulate significant amounts of precursors, especially adonixanthin. 

Heterologous expression of these enzyme genes is extensively conducted for astaxanthin production. Currently, CrtZ from *Pantoea ananatis* 20D3 (formerly known as *Erwinia uredovora* 20D3) or *Brevundimonas* sp. SD212, as well as CrtW from *Brevundimonas* sp. SD212, have usually been used for transgenic purposes, as they exhibit higher astaxanthin accumulation when expressed heterologously [400,401]. Based on conserved amino acid sequence regions and cofactors, CrtW and CrtZ display partial homology within their conserved domains and belong to the nonheme membrane-integrated diiron nonheme oxygenase superfamily [402,403], suggesting a possible evolutionary relationship from a common ancestor.

On the other hand, *Phaffia* yeast (*Xanthophyllomyces dendrorhous*), a basidiomycete yeast, has a gene encoding a P450 family enzyme, astaxanthin synthase (CrtS/Asy, fungus, EC 1.14.99.63/1.14.15.24; see also Section 2.2.2), that catalyzes the oxygenation of the C4 position and the hydroxylation of the C3 position. This is a single enzyme that catalyzes two reactions; however, the orientation of the hydroxyl group is reversed from that in bacteria, giving the product (3*R*,3’*R*)-AX. The reaction of ASY (CrtS) in *Phaffia* yeast is unique, starting with the C4 oxygenation first, followed by the C3 hydroxylation reaction (Figure 11).

In higher plants, organisms that accumulate AX are extremely uncommon, but fortunately, two unique enzymes involved in AX biosynthesis are found in the petals of the flowering plant *Adonis aestivalis* (the family Ranunculaceae). Generally, in higher plants, the hydroxylation of the C3 position of the β-end group is catalyzed by BCH/BHY (also described as CrtR-b or CHY, EC 1.14.13.129; transferred to EC 1.14.15.24 in 2017), a non-heme monooxygenase enzyme that is an ortholog of bacterial CrtZ (Figure 10). For this plant enzyme, the gene for β-carotene (β-carotenoid) 3,3′-hydroxylase, BHY, is an appropriate description according to the naming method of plant carotenoid biosynthesis genes. On the other hand, it is quite unusual for plants to oxygenate the C4 position. Therefore, studies were conducted to identify the enzyme gene. Among them, Cunningham et al. identified two enzymes that can introduce a hydroxyl group at the C4 position, named AdKeto1 and AdKeto2 (later changed their names to CBFD1 and CBFD2, respectively [264]), which were isolated as proteins homologous to the plant-type BHY from a cDNA library of the *A. aestivalis* flowers [404]. The reaction by this gene product converts the β-end group into a mixture with either a 4-hydroxy-β ring and/or a 3,4-didehydro-β ring, rather than a 4-keto-β ring. Initially, they envisioned a keto-enol isomerization reaction of the 3,4-didehydro-4-hydroxy-β-ring. In addition to CBFD, further enzymes named HBFD1 and HBFD2, which catalyze the dehydrogenation of the hydroxyl group at the C4 position, are essential for the conversion of *Adonis* to the keto group at the C4 position. Following this reaction, CBFD is able to further introduce a hydroxyl group at the C3 position only if a 4-keto β-end group is present. Therefore, CBFD is considered to have a dual function as not only a β-4-hydroxylase/3,4-desaturase to the β-end group but also as a C3-hydroxylase of the C4-keto-β-end group [264] (Figure 11).

In green algae, such as the genus *Haematococcus*, the genes responsible for AX biosynthesis are well understood. AX is synthesized from β-carotene by the coordinated actions of BHY-type β-C3-hydroxylase (β-carotenoid 3,3′-hydroxylase) and BKT-type β-C4-oxygenase (β-carotenoid 4,4′-ketolase) through eight β-carotene metabolites (Figure 8 and Figure 11) [37,399]. As mentioned earlier, BHY is the ortholog of CrtZ, while *BKT* has been found to exhibit significant amino acid sequence homology to CrtW. Therefore, it is likely that the *BHY* and *BKT* genes are evolutionarily derived from the *crtZ* and *crtW* genes, respectively, although the CrtZ (BHY) ortholog is not found in cyanobacteria.

*Haematococcus* alga retains at least two *BKT* paralogs that are functional [387,388]. Thus, the β-carotenoid 4,4′-ketolase genes initially designated *bkt* [387] and *crtO* [388] were renamed *BKT2* (*bkt2*) and *BKT1* (*bkt1*), respectively [389].

#### 2.3.3. How Can *Hematococcus* Algae Achieve Ultra-High Concentrations of Astaxanthin Biosynthesis? 

*Haematococcus* algae exhibit remarkable potential for accumulating up to 9.8% of AX in cell weight under specific culture conditions [328]. These AX compounds predominantly exist as mono- and diesters, gracefully stored within oil droplets housed in the resilient cyst cells, as discussed earlier (refer to Section 2.2.2). Notably, mature aplanospores (cysts) showcase the presence of AX primarily within intracytoplasmic oil droplets rather than plastid-derived chromoplasts, as illustrated in Figure 12 [223,247,248,249,250,251]. These specialized droplets serve a dual purpose, shielding the nucleus as antioxidants and acting as light filters, effectively reducing excitation pressure on photosynthetic subunits and mitigating the risk of photodamage [405].

How is it possible for *Haematococcus* algae to achieve such a high concentration of AX compared to other organisms? One significant reason for this is their ability to accumulate the majority of AX as oil droplets in oil vacuoles within the cytoplasm rather than in plastids. These oil droplets, protein-lined structures typically synthesized from the endoplasmic reticulum, efficiently accumulate lipids (Figure 13), as observed in mammalian adipocytes and other organisms. Another crucial factor is the ester form of AX. Free AX, in the absence of a carrier protein, can aggregate at a certain soluble concentration, even in the presence of lipids as a solvent, potentially imposing physical stress on the cells. However, as fatty acid esters, the solubility of AX in fats, particularly triglycerides, is greatly enhanced. The biosynthesis and accumulation of AX and triglycerides within oil droplets occur in a coordinated manner, with de novo synthesis of free fatty acids and AX esterification taking place sequentially along this pathway (Figure 13) [406,407,408,409,410]. Consequently, *Haematococcus* algae have the capability to accumulate AX within oil droplets in the cytoplasm while the volume of the chloroplasts decreases. 

There are intriguing reports on the biochemical and spectroscopic properties of the pigment-binding complexes responsible for light harvesting and energy conversion in these algae. Key characteristics of chlorophyll and carotenoid-binding complexes observed in higher plants and closely related species such as *Chlamydomonas reinhardtii* are also maintained in *Haematococcus* algae under controlled conditions. However, during the transition to the AX-rich cyst stage, the photosystem’s stability becomes compromised. During this phase, AX and its fatty acid esters have been shown to bind to both photosystems I and II, partially replacing β-carotene. Interestingly, the binding of AX to the photosystems does not enhance their photoprotective function; instead, it reduces the efficiency of excitation energy transfer to the reaction centers. Thus, the binding of AX may destabilize photosystems I and II to some extent [405]. When considering the ratio of AX to chlorophyll in total cells and isolated fractions, it is noteworthy that less than 1% of the total accumulated AX in *Haematococcus* cysts is bound to PSI or PSII, with the majority of AX accumulating in the cytoplasm. While AX is present in the photosynthetic pigment-binding complexes in *Haematococcus*, these AX molecules may be synthesized in the plastids or, more likely, synthesized in the cytosol and subsequently translocated back into the plastids. However, the mechanism of their transport remains unclear [405].

Under suitable conditions, *Haematococcus* algae, like other green algae, can autotrophically synthesize glucose de novo from carbon dioxide (CO_2_) as the primary inorganic carbon source through photosynthesis and accumulate carbohydrates, mainly starch, around the pyrenoids (Figure 12A,D). During encystment, astaxanthin (AX) biosynthesis occurs in the endoplasmic reticulum (ER), connected to the nucleus, rather than in the plastids. It generally utilizes endogenous carbohydrates as substrates and then accumulates in oil droplets in the cytoplasm. (Since it was not found at the level of phytoene desaturase (PDS) in the cytosol [411], it is commonly accepted that β-carotene, which is de novo synthesized in the plastid, undergoes oxidative modification in the ER to form AX [412]). Although *Haematococcus* algae can also biosynthesize AX under mixo- or heterotrophic conditions [413,414,415], it can utilize acetic acid as an exogenous substrate but not glucose [416]. Since heterologous expression of glucose transporters in this algae allows utilization of glucose [417]. Since heterologous expression of glucose transporters in *Haematococcus* algae allows utilization of glucose [418]. 

Similar to other carotenoids, the astaxanthin skeleton is synthesized from IPP (isopentenyl pyrophosphate) and its isomer, DMAPP (dimethylallyl pyrophosphate). Previous findings suggest that two separate pathways, the cytosolic mevalonic acid (MVA) pathway and the chloroplast MEP (methylerythritol phosphate) pathway, can provide these precursors (see Section 2.3.1). In the case of *Chlamydomonadales*, including *Haematococcus* algae, it has been reported that the precursors for astaxanthin are derived from the MEP pathway rather than the MVA pathway [419].

The biosynthetic pathway from β-carotene to astaxanthin (AX) in the genus *Haematococcus* involves enzymatic reactions that are very similar to those found in AX-producing bacteria. Specifically, focusing on the oxygenation reaction at the C4, C4′ position, two paralogous genes for β-carotenoid 4,4′-ketolase (*BKT*) were independently identified in *Haematococcus* algae during the same period. These genes were initially designated as bkt (*BKT*; from *H. pluvialis* Flotow (=*H. lacustris*) NIES-144) [38] and *crtO* (*CRTO*; from *H. pluvialis* (=*H. lacustris*) CCAP34/7) [420]. Subsequently, it was discovered that these genes were homologous to the nucleotide sequence of the *crtW* gene. It should be noted that the name “*crtO*” caused confusion, as it had already been assigned to a distinct bacterial β-C4-oxygenase gene. Therefore, *BKT1* and *BKT2* were later renamed from *CRTO* and *BKT*, respectively [421]. Further investigations revealed that *H. lacustris* typically possesses three *BKT* paralogs, namely *BKT1*, *BKT2*, and *BKT3*, with *BKT3* being non-functional [421]. Both *BKT1* and *BKT2* have the ability to accept β-terminal and C3-hydroxy-β-terminal groups as substrates, which is a similar function to that of CrtW [373,374,392,394]. Moreover, the whole genome sequencing and transcriptome analysis of *H. lacustris* strain SAG 192.80 led to the annotation of six *BKT* copies in the genome, distributed across different sites [422]. Among these six *BKT* genes, three showed the highest similarity to *BKT1* and were designated as *BKT1a*, *BKT1b*, and *BKT1c*. Two genes were similar to *BKT2* and named *BKT2a* and *BKT2b*, while the remaining gene was similar to *BKT3* [422]. However, in closely related species such as *Volvox carteri*, *Chlamydomonas reinhardtii*, and *Monoraphidium neglectum*, only one corresponding ortholog was identified for each [422]. 

During AX biosynthesis, β-carotene (β-carotenoid) 3,3′-hydroxylase (BHY) catalyzes the hydroxylation of the carbons at the 3 and 3′ positions of the β-terminal group. BHY enzymes in *Haematococcus* algae share structural and functional similarities with the plant-type BHY (BCH) [410,423]. The gene product of *BHY* in *Haematococcus* algae exhibits homology to the bacterial form of CrtZ rather than the CrtR found in cyanobacteria [189,410]. In *Haematococcus* algae, the majority of the AX that accumulates in mature cysts is in the form of mono- or diesters with fatty acids [64,424]. 

Recently, the genome sequence of another strain of *Haematococcus* algae, *H. lacustris* NIES-144, was also determined [425]. Based on this sequence, the gene set involved in carotenoid biosynthesis has not yet been fully resolved; however, further studies are expected in the future. 

In addition to gene expression, the analysis of the transcriptome and metabolome has provided valuable insights. Transcriptome studies have revealed the dynamic changes in gene expression profiles throughout different growth stages of *Haematococcus* algae. Meanwhile, metabolome analysis has enabled the identification and quantification of various metabolites involved in the carotenoid biosynthesis pathway, such as intermediate compounds and final products like AX. Those integrated studies have revealed that various factors, such as quality and strength of specific wavelengths of light [421,426,427,428,429,430,431,432,433,434,435,436,437,438,439,440,441,442,443,444,445,446,447,448,449,450,451,452,453,454,455,456,457,458,459,460], plant growth regulators (phytohormones) [453,455,458,461,462,463,464,465,466,467,468,469,470,471], osmotic pressure/salinity [328,409,434,454,460,472,473,474,475], pH change [476], temparature [409,477,478,479], microbiome [480,481,482,483], certain chemicals [454,468,484,485,486,487,488,489,490,491,492,493,494,495,496,497], exogenous carobon source [328,421,441,447,468,498,499,500,501,502,503,504,505,506,507,508], nitrogen concentration [328,409,426,431,433,438,506,509,510,511,512,513] and ROS (heavy metals) [409,441,447,478,514,515,516,517,518,519,520,521] characteristically regulate the AX biosynthesis pathway. Furthermore, receptors, signaling pathways, and transcription factors involved in the regulation of carotenogenic gene expression, which have been a black box to date, have also been gradually clarified [219,410,488,522,523,524,525,526,527,528,529,530,531,532,533]. The details of these factors are described in other reviews [534,535,536].

Overall, the integration of gene expression, transcriptome, and metabolome data has facilitated a comprehensive understanding of the regulatory mechanisms and metabolic pathways underlying carotenoid biosynthesis in *Haematococcus* algae.

#### 2.3.4. Metabolism of Astaxanthin in Animals

##### Overviews of Metabolism of Astaxanthin in Animals; General Remarks

In animals, the accumulation of carotenoids, including astaxanthin (AX), is the result of complex food chains and metabolism. As depicted in Figure 14, carotenoids are biosynthesized in organisms at the beginning of the food chain. These organisms serve as prey for higher-level organisms, which selectively accumulate and metabolize carotenoids, including precursor forms and AX. Concurrently, AX can undergo metabolic processes, leading to the formation of different carotenoids or degradation. This section aims to present the most recent findings concerning the metabolism of AX in non-mammalian animals.

Recent findings indicate that xanthophyll-derived hues, including ketocarotenoids found in animals, are associated with two important groups of genes. First, scavenger receptor class B family proteins play an important role in gastrointestinal absorption and translocation from blood to tissues. Second, a group of enzymes, collectively known as β-carotene oxygenases, which cleave polyene chains, play an important role in the breakdown of carotenoids in tissues, resulting in loss of color. Additionally, multiple protein carriers and transporters are known to be involved in the biological dynamics of carotenoids. The roles of these proteins in each animal are described in detail in this section and in Section 4.2.2 for mammals.

In the most upstream part of the food chain, carotenoid sources include microalgae and other green plants. However, carotenoid-producing fungi and bacteria can also serve as sources based on the feeding habits of predators. Certain arthropods, including crustaceans such as zooplankton, are also considered potential suppliers of carotenoids. While there is some knowledge about symbiotic bacteria [284,537,538] the exact extent of their contribution remains unclear.

Among arthropods, many Acari (spider mites) and aphids possess the ability to synthesize carotenoids de novo. This capability is believed to be the result of the lateral transfer of carotenoid biosynthesis genes from a carotenogenic fungus. Specifically, the bifunctional lycopene cyclase/phytoene synthase (*CrtYB*) and phytoene desaturase (*CrtI*) genes, whose functions are depicted in Figure 10 [295,296,297,298,299,300], were acquired by aphids and Acari through one or several lateral gene transfers from fungi. These genes are not naturally present in any known animal genome. Phylogenetic analyses suggest that the transfer of these carotenoid biosynthesis genes did not occur in a single event to a common arthropod ancestor. Instead, it is likely that the genes were independently transferred from related fungal donors after the divergence of major arthropod lineages [298]. In the case of these acarids, it is also possible that β-carotene or zeaxanthin is converted to AX (and related ketokarotenoids, i.e., echinonenone, 3-hydroxyechinenone, adnoixanthin, and adonirubin) by a cytochrome P450-type oxidase, as described in the next subsection (Section 2.3.3) [375].

In crustaceans such as shrimp and crab and echinoderms such as starfish, sea urchins, and sea cucumbers (as discussed in Section 2.2.4), dietary β-carotene can be utilized as a substrate and metabolically converted to AX [281,303]. I During this process, there is no stereospecificity of hydroxylation at the C3 (C3′) position, resulting in mixtures of optical isomers for 3-hydroxyechinenone, adonirubin, and AX. These organisms are also capable of converting (3*R*,3′*R*)-zeaxanthin to (3*S*,3′*S*)-AX, which has the same optical configuration due to pre-existing C3 and C3′ hydroxylation (Appendix A) [290]. 

Furthermore, several microcrustaceans located relatively upstream in the food chain, known as zooplankton, including copepods, may also convert β-carotene and xanthophylls (mainly zeaxanthin) to AX. These organisms utilize β-carotene and zeaxanthin as substrates. Interestingly, in laboratory studies of the copepod *Tigriopus californicus*, it was found that AX could be synthesized from β-carotene [290,356]. The stereoconfiguration of the hydroxyl group is yet to be definitively determined due to experimental limitations; however, the (3*S*,3′*S*) configuration was assumed [290]. When zeaxanthin was used as a substrate, most of the resulting AX appeared to inherit the stereo configuration of its base, zeaxanthin [290]. 

The prey of these organisms mainly consists of phytoplankton, including Cyanobacteria, Chlorophyta, Rhodophyta, Dinoflagellates, Cryptophyta, Euglenophyta, Chlorarachniophyta, Haptophyta, Diatoma, and Chlorophyta, many of which contain carotenoids such as β-carotene, diatoxanthin, peridinine, siphonaxanthin, fucoxanthin, and zeaxanthin [398]. It is likely that AX is converted from β-carotene or zeaxanthin, both of which are present in these phytoplankton. Most of these phytoplankton exhibit the (3*R*,3*’R*)-zeaxanthin conformation based on the reaction specificity of hydroxylase genes in plants and microalgae (see Section 2.3.2). However, Krill, in particular, has a very high ratio of (3*R*,3*’R*) stereoconfiguration of AX (see Section 2.2.4 and Table 2). Antarctic krill typically feed on diatoms, haplophytes, or dinoflagellate phytoplankton that occur under ice floes, known as ice algae. These organisms generally contain β-carotene and characteristic diatoxanthin, peridinine, and siphonaxanthin specific to each alga, as well as partial amounts of lutein and zeaxanthin. Therefore, considering the optical configuration of AX, it is possible that krill convert precursor carotenoids such as β-carotene, zeaxanthin, or lutein to AX *in vivo*, similar to copepods and Acari. Additionally, when AX is administered to prawns, for example, racemic conversion, including reductive metabolisms, may also occur *in vivo* [361,539]. However, currently, there is no information available regarding the enzyme responsible for this conversion. The ability of these arthropods to convert AX may be distributed among a diverse range of species, suggesting that they may possess P450-type β-C4 ketolase, as described in Section 2.3.3. The ratio of AX optical isomers in these species-specific patterns is thought to be formed through the accumulation and metabolic conversion induced by arthropod predation (see Section 2.2.4 and Table 2). 

The coiled shellfish mentioned in Section 2.2.4 are capable of metabolically converting dietary β-carotene and (3*R*,3′*R*)-zeaxanthin to (3*S*,3′*S*)-AX. In these snails, a specific chiral reaction involving hydroxylation at the C3 (C3′) position leads to the formation of (3*S*)-adonirubin and (3*S*,3′*S*)-AX [351]. Although these *in vivo* reactions have been extensively studied, the enzymes responsible for these reactions and their gene families are mostly unknown.

In recent years, interesting findings have emerged regarding the different colors observed in the adductor muscle of the noble scallop with brown or golden shells [284] (as discussed in Section 2.2.4). Some individuals of this bivalve species exhibit polymorphism in shell color, displaying beautiful hues, with the mantle, gill, gonad, and adductor appearing golden. This attractive golden color is attributed to the accumulation of carotenoids. Interestingly, in individuals with golden scallops, carotenogenic bacteria such as *Brevundimonas* and *Sphingomonas*, which are known to produce AX (refer to Section 2.2.1 and Section 2.3.1), were found to be significantly more abundant as gut bacteria (symbionts) compared to individuals with brown shells [284]. In addition to the presence of carotenogenic bacteria, the golden coloration of noble scallops may be influenced by other factors. It has been observed that the expression of StAR-like-3, a homolog of StARD3 (a lutein-binding protein found in the silkworm *Bombyx mori*), is significantly higher in the intestine and blood cells of golden scallops compared to brown scallops. StAR-like-3 is believed to be involved in the transport and accumulation of carotenoids [540]. Furthermore, it has been reported that the expression of stearoyl-CoA desaturase (SCD) and SRB-like-3, which shares homology with scavenger receptor class B member 1 (SR-BI/SCARB1), known for its role in the uptake of blood carotenoids in mammals (see Section 4.1.1), is upregulated in all tissues of golden scallops. Suppression of these gene expressions leads to a reduction in the coloration of blood and adductor muscle [541]. 

Furthermore, the gene expression of β-carotene dioxygenase 2 (BCDO2/BCO2), a mitochondrial carotenoid-degrading enzyme, was found to be suppressed under certain conditions in colored Yesso scallops and golden noble scallops compared to normal individuals. This suggests that carotenoid degradation in the tissues of these scallops is suppressed [542,543]. 

In summary, several mechanisms contribute to carotenoid accumulation in bivalves, similar to the general ADME (absorption, distribution, metabolism, and excretion) concept. These mechanisms involve processes such as absorption from the intestine, transport of proteins in the circulation, tissue accumulation, and degradation. Additionally, in bivalves, these processes may be regulated by peroxisome proliferator-activated receptors (PPARs) and retinoid X receptors (RXRs) [544,545], which are nuclear transcription factors well-known for their roles in mammalian lipid metabolism (refer to our review [130]).

In fish that contain astaxanthin (AX), two types of origins can be observed: those that can metabolically convert AX from its precursor within the body and those that are unable to perform so. While the details are not yet fully understood, it has been demonstrated that certain species within the *Cyprinidae* family, such as Nishikigoi (*Cyprinus carpio*) and golden fish (*Carassius auratus*), possess the ability to introduce hydroxyl groups at the C4 and C4′ positions of (3*R*,3′*R*)-zeaxanthin. Subsequently, the hydroxylated zeaxanthin is oxidized, resulting in the formation of a carbonyl group and the production of AX [24,307]. (For more information, please refer to Section 2.2.5 and Appendix A). Consequently, metabolic intermediates such as 4-hydroxyzeaxanthin, adonixanthin, and idoxanthin have been identified in these fishes [307]. 

Salmonids are among the most well-known fish species containing AX [539]. Unlike the aforementioned fish, salmonids are unable to convert AX within their bodies and therefore rely on their diet to obtain all of their AX from crustacean plankton, such as copepods and krill [26,28,539]. During the transition from riverine to marine migratory form, known as smolting, salmonids undergo a remarkable transformation accompanied by a shimmering silvery coloration of their integument. This process is also characterized by dietary changes and significant shifts in the composition and quantity of carotenoids. As the salmonids mature and return to their natal river, the integument exhibits distinct coloration changes resulting from carotenoids and other pigments. It is now evident that these processes involve metabolic adjustments within their bodies [546,547]. Notably, the intake of copepods and other marine zooplankton [26,28], during smolting increases the supply of AX, which appears to confer survival advantages such as maintaining immune function [546]. AX is typically stored in the muscles, particularly in migratory forms; however, during the spawning season, males transfer carotenoids to their body surface while females transfer them to their ovaries [26,28,548]. In post-smolting Atlantic salmon (*Salmo salar*), the transcriptome of the pyloric ceca (comparable to the mammalian small intestine) has been shown to be more responsive to dietary AX supplementation than other tissues. Noteworthy genes sensitive to AX supplementation include *cd36* in the pylorus, *agr2* in the liver, and *fbp1* in muscle. The pylorus exhibited the most regulated group of genes, specifically those associated with AX absorption and metabolism. Furthermore, genes linked to the upstream regulation of the ferroptosis pathway were significantly influenced in the liver, suggesting the involvement of AX as an antioxidant in this process [547]. Additionally, the transportation of AX into skeletal muscle, which contributes to muscle pigmentation, is believed to involve the recently discovered paralogs of SCARB1, particularly the SCARB1-2 transporter [549].

The carbonyl groups at the C4 and C4′ positions of AX undergo reductive metabolism on the body surface, resulting in the conversion to zeaxanthin via adonixanthin through a reductive pathway [539,550]. Further epoxidation leads to the formation of antheraxanthin (see Appendix A). Interestingly, it is possible that lutein follows a similar metabolic pathway, as it is metabolized to salmoxanthin, an epoxy carotenoid that characterizes the skin of salmonids [281]. Since certain fishes produce retinoids through reductive metabolism to the keto group at the C4 position, AX may serve as a partial provitamin A for these organisms [281,551]. As a result, the composition and in vivo distribution of carotenoids in salmonids vary throughout their life cycle.

Among the Percomorpha order, yellowtail (Japanese amberjack, *Seriola quinqueradiata*), sea-bream (*Pagrus major*), and salmonids are unable to convert the precursor to AX within their bodies. Therefore, all the AX present in these fishes is obtained exclusively from their diet [30]. In the case of yellowtail, as depicted in Appendix A, the accumulated AX in the body is rapidly reduced to zeaxanthin and subsequently metabolically converted to tunaxanthin, which imparts a vibrant lemon color. Interestingly, the yellow pigment found in marine fish can also be attributed to AX [281]. This metabolic pathway is observed in numerous highly evolved fish species belonging to Acanthopterygii, including both saltwater and freshwater species [21].

##### Metabolic Conversion to Astaxanthin by Cytochrome P450

In terrestrial animals, AX and its related “red” ketocarotenoids are commonly found in the retina and serve as ornamental coloration on the skin and plumage of reptiles and birds, as discussed in Section 2.2.5. The precise role of AX in these organisms remains hypothetical; however, it is clear that AX accumulation is derived from carotenoids present in their food through the food chain. In the case of fish, crustaceans, and flamingos, it has long been known that AX is converted from “yellow” carotenoids such as β-carotene and zeaxanthin, which can be precursors obtained from their diet. However, the specific enzymes involved in this conversion have remained unclear (Section 2.2.5). Recent studies have suggested an intriguing possibility that the red color characteristic of these organisms may also arise from the conversion of “yellow” carotenoids, such as β-carotene and zeaxanthin, by certain cytochrome P450 (CYP; P450) (refer to Table 3). The involvement of P450s in AX biosynthesis has been demonstrated through the analysis of ASY (CrtS) in *Phaffia* yeast, as discussed in Section 2.3.1**.** Notably, the P450s presumed to be involved in the metabolic conversion to AX in the animal kingdom are diverse and do not have direct orthologs to ASY (CrtS). 

In their study, Mundy et al. made a groundbreaking discovery by identifying a P450 gene (*CYP2J19*) that is potentially involved in the biosynthesis of “red” ketocarotenoids for the first time in the animal kingdom. They accomplished this through a comprehensive genetic analysis of the “yellowbeak” mutant of the zebra finch (*Taeniopygia guttata*) [552]. The yellowbeak mutant possesses a mutation in *CYP2J19* within the gene cluster encoding CYP2, where the wild type exhibits *CYP2J19*A and *CYP2J19*B, whereas the yellowbeak mutant has *CYP2J19^yb^*. Consequently, *CYP2J19*A is exclusively expressed in the retina, while *CYP2J19*B is expressed in the beak and tarsus, with varying levels of expression in the retina. Conversely, *CYP2J19^yb^* expression is barely detectable in the beak of yellowbeak birds. These findings establish the essential role of CYP2J19 in ketocarotenoid biosynthesis in zebra finches [552]. 

Furthermore, Lopes et al., who belong to the same research group as mentioned above, reported the whole genome sequencing of red siskins (*Spinus cucullata*, exhibiting red body color), common canaries (*Serinus canaria*, exhibiting yellow body color), and “red factor” canaries (a crossbreed of red siskins and canaries) to investigate the genetic basis of red coloration in birds [553]. Their research identified two genomic regions crucial for red coloration in “red factor” canaries, one of which contains the gene encoding CYP2J19. Transcriptome analysis revealed a significant upregulation of CYP2J19 in the skin and liver of red factor canaries, further suggesting that CYP2J19 functions as a β-C4-ketolase, catalyzing the conversion of ketocarotenoids in birds [553].

It is plausible to assume that these P450 proteins acquired their functions through convergent evolution within each respective order/suborder [554]. While the metabolic pathways of these AX-generating P450s are expected to be similar to those of *Phaffia* yeast, information regarding the substrates involved in each P450 reaction and the absolute configuration of the resulting C3, C3′positions remains limited. The available data suggests that Acari, an organism capable of producing AX, exhibits an optical configuration of (3*S*,3′*S*), which is the opposite of the (3*R*,3’*R*) configuration observed in *Phaffia* yeast, as reported in an earlier publication. Therefore, it is anticipated that future studies will shed light on the reaction mechanisms, the substrate specificity of each P450 involved in AX conversion, the optical configuration of the produced AX, and the intermediates formed during the reaction. 

The *CYP2J19* gene, generally present as a single copy in their genome, is widespread among avian lineages. This observation aligns with the notion of a conserved ancestral function in color vision, followed by subsequent co-selection for red epidermis coloration. Similar to several other CYP loci with conserved functions, CYP2J19 exhibits evidence of having undergone positive selection across bird species. Although there is no direct evidence indicating changes in selection pressure on CYP2J19 associated with co-selection for red pigment, it is possible that compensatory mutations related to selection on the adjacent gene *CYP2J40* may contribute to this phenomenon [321]. 

In contrast, as discussed in Section 2.2.4, certain birds, such as penguins, kiwi, and some owls, lack ketocarotenoid-containing cone oil droplets in their retinas. In these avian species, the *CYP2J19* gene has undergone pseudogenization [320]. Although penguins do contain AX in their body tissues, it is believed to be derived from their diet [395,396]. 

Interestingly, there have been reports indicating that some bird species may have lost and subsequently regained the function of CYP2J19 during the course of evolution, suggesting the potential advantageous significance of reddish body coloration and the acquisition of ketocarotenoids for birds [555]. The zebra finch, for which the function of the *CYP2J19* gene was first indicated, also has two copies of the *CYP2J19* gene that have probably been duplicated during evolution [552].

Among tetrapods, turtles are the only group that possesses red oil droplets in their retinas, and several turtle species exhibit red carotenoid coloration. Twyman et al. conducted a study on the evolution of *CYP2J19*, a gene associated with color vision and red pigmentation in reptiles, utilizing genomics and gene expression analysis. They discovered that turtles, but not crocodilians and lepidosaurs (including lizards, snakes, and tuatara), possess orthologs of *CYP2J19*, which originated from a gene duplication event prior to the divergence of turtles and archosaurs. CYP2J19 is strongly and specifically expressed in the retina and red outer skin of turtles, which include ketocarotenoids. The researchers propose that CYP2J19 initially played a role in the color vision of archosaurs, and they conclude that red ketocarotenoid-based coloration independently evolved in birds and turtles through genetic regulation changes involving CYP2J19. In other words, these intriguing findings suggest that red ketocarotenoids might have contributed to color vision and ornamental coloration in dinosaurs and pterosaurs [554]!

Therefore, the presence of *CYP2J19* genes plays a crucial role in the ketolation of carotenoids in avian and reptilian species. However, reptiles, particularly lepidosaurs, likely lack *CYP2J19* [554]. Despite this, several reptile species exhibit ketocarotenoids, such as AX, in their integumentary coloration (refer to Section 2.2.4. for more details). 

Subsequently, a genetic approach identified genes involved in the conversion of carotenoids to ketocarotenoids in spider mites and a species of zebrafish [375,556]. Interestingly, these P450-type C4-ketolase enzymes in spider mites and zebrafish are not encoded by *CYP2J19* but are shown to be associated with separate P450 enzyme genes. For instance, in spider mites (*Tetranychus kanzawai*), the P450 encoded by the CYP384A1 gene significantly contributes to red body coloration through ketocarotenoids [375]. Similarly, in the zebrafish species (*Danio albolineatus*), the *CYP2AE2* gene is involved in the accumulation of red pigment in the skin erythrophores [556,557]. In other organisms, genetic studies and homology with known carotenoid oxygenases indicate the involvement of various P450 enzymes in the β-C4 oxygenation of carotenoids (Table 3).

For example, *Anolis favillarum*, a reptile belonging to the lepidosauria (lizards), exhibits white, yellow, and orange skin. The orange skin coloration is attributed to pteridines and ketocarotenoids. As mentioned earlier, reptilian lepidosauria have lost *CYP2J19* during evolution [554]. Genetic and transcriptome analyses revealed that *CYP2J2* and *CYP2J6* were highly expressed in orange individuals compared to yellow individuals, suggesting that these P450s contribute to the conversion of carotenoids into ketocarotenoids [313].

**Table 3 marinedrugs-21-00514-t003:** Summary of P450 enzymes with possible C4 ketolase activity using zeaxanthin or β-carotene as substrates.

Name P450	Origin		Super-Family, Clan	Methodlogy of Functional Analysis	Reference
CYP2J19	Aves/Testudines		CYP2	Genetics Heterologous expression Homology	[321,552,553,557]
CYP2AE2	Zebra fish;	*Danio albolineatus*	CYP2	Genetics Heterologous expression	[556,557]
	(Actinopterygii: Cypriniformes)				
CYP2J2	Anole Lizards;	*Anolis favillarum*	CYP2 CYP2	Genetics	[313]
CYP2J6		(Reptilia: Iguania, (Lepidosauria))			
CrtS (CYP5139Q1)	*Phaffia* Yeast;	*Xanthophyllomyces dendrorhous*	CYP3	Heterologous expression	[212,213,558]
		(Fungi: Basidiomycetes)			
CYP384A1	Spider mites;	*Tetranychus kanzawai*	CYP3	Genetics	[375]
CYP383A1		(Arthropoda: Chelicerata)	CYP3	Putative (Closest homologue of CYP384A)	
CYP3A80	Sira poison Frog;	*Ranitomeya sirensis* (Amphibia: Anura)	CYP3	Genetics	[294]
CYP3-like	Anchialine Shrimp;	*Halocaridina rubra*	CYP3	Putative	[559]
		(Arthropoda: Crustacea: Decapoda)			
P450 like	Copeods;	*Acartia fossae*	N/D	Putative	[560]
		(Arthropoda: Crustacea: Copepoda)			
P450 like	Pelagic tunicate;	*Oikopleura dioica*	N/D	Putative	
		(Chordata: Tunicata: Appendicularia)			

N/D; not determined.

Australian frilled lizards, *Chlamydosaurus kingii* exhibit either yellow or orange coloration of their frills depending on their habitat. The orange coloration primarily relies on the presence and type of pteridines and ketocarotenoids [312]. Transcriptome analysis of these lizards revealed altered gene expression in pteridine biosynthesis and retinoid metabolic pathways in both color-differentiated frilled lizards. However, the expression of certain CYPs was higher in the yellow population compared to the orange population, indicating that the high accumulation of ketocarotenoids in orange individuals may be due to enzymatic degradation and ketolation, or simply differences in dietary habits that require further investigation [312].

Furthermore, the conversion of AX from β-carotene and zeaxanthin is not solely catalyzed by the P450 enzyme system. Recent studies have reported heterologous expression analyses of the *CYP2J19* gene in chickens [557]. When the chicken-derived *CYP2J19* gene was transfected into human HEK 293 cells, it was found that *CYP2J19* alone likely catalyzes the addition of a hydroxyl group to zeaxanthin. However, oxygenation of the C4 position by itself appears to be challenging, and simultaneous expression of 3-hydroxybutyrate dehydrogenase 1-like (BDH1L) is required. BDH1L, previously an enzyme of unknown function, shows sequence similarity to BDH1, an enzyme involved in interconverting acetoacetate and 3-hydroxybutyrate, major ketone bodies produced during caloric restriction in vertebrates [561]. 

*CYP2J19* and BDH1L preferentially produce (3*S*,3′*S*)-AX when an equimolar mixture of the three zeaxanthin stereoisomers is used as a substrate. When (3*R*,3′*R*)-zeaxanthin is used as a substrate, the optical configuration remains largely unchanged, resulting in the conversion to (3*S*,3′*S*)-AX. This enzymatic reaction is stereoselective and can also convert chirality [557]. It aligns with the fact that avian retinas predominantly consist of (3*S*,3′*S*)-AX [319]. Additionally, the combination of both enzymes converts β-carotene to canthaxanthin and lutein to α-doradexanthin, indicating that the primary reaction involves oxygenation of the C4 positions of the β-endogroups with minimal hydroxylation reaction. BDH1L is involved in ε-ring formation through the transfer of a double bond in the terminal ring of the carotenoid [557]. These functions are similar to the initially hypothesized reaction mechanism of AX biosynthesis by the two non-heme oxidases in *Adonis* plants [404]. 

Collectively, these findings suggest that most animals obtain AX from their diet or have acquired the ability to convert it to AX through the acquisition of distinct and convergent oxidative metabolic functions during their evolution. The precise biological significance of AX is still a topic of debate; however, it is believed to serve important functions.

As a final aside, fenretinide [*N*-(4-hydroxyphenyl)retinamide (4-HPR)], a synthetic analog of all-*trans* retinoic acid (ATRA), which exhibits cytotoxic activity against cancer cells, undergoes a metabolic reaction in mammals by CYP3A4 and CYP2C8 in liver microsomes that involves oxygenation of the C4 position of the β-ionone structure in the 4-HPR molecule [562,563]. Thus, oxygenation of the β-C4 position may be a relatively common phenomenon even in mammals, although the strength of activity and degree of substrate specificity remain unknown. 

### 2.4. Chemical Synthesis and Analysis of Astaxanthin

#### 2.4.1. Chemical Synthesis of Astaxanthin

In this document, while the main focus is on AX derived from biological sources, a brief overview of chemically synthesized AX will also be provided. In 1967, Surmatis et al. were the first to synthesize AX in its dimethyl ester form [564]. I In the same year, Leftwick and Weedon successfully synthesized the free form of AX from canthaxanthin [565]. In the early 1980s, the group at Hoffman-LaRoche achieved the total synthesis of astaxanthin through the Wittig reaction of C15-end-group phosphonium salts at both ends of the central C10-dialdehyde (Figure 15). Using this method, they synthesized (3*S*,3′*S*), meso, and (3*R*,3′*R*) optical isomers of AX, along with several geometric isomers [566]. This synthetic method was employed for industrial synthesis, and synthetic AX is still commercially available as a coloring agent in aquaculture for fish such as salmon, trout, and sea bream. It is important to note that commercial products (Carophyll Pink or Lucantin Pink) are coated with gelatin and starch, making them water-dispersible. 

More recently, a one-pot base-catalyzed reaction of (3*R*,3′*R*,6′*R*)-lutein esters from marigolds has been reported to readily yield up to 95% meso-zeaxanthin and, ultimately, (3*R*,3′*S*)-AX with an overall yield of 68%. Density functional theory (DFT) calculations of the reaction mechanism suggest that the isomerization involves base-catalyzed deprotonation of C-6′ followed by protonation of C-4′, while the oxidation occurs via a free radical mechanism [567]. AX bulk products with a distinctive optical isomer ratio, presumably derived from this reaction, have been sporadically found on the market since the middle of 2010 (data not shown). 

#### 2.4.2. Quantitative and Qualitative Analysis of Astaxanthin

Practical analysis of AX is primarily conducted using UV/VIS absorbance spectrophotometry and/or HPLC methods. The absorbance spectrophotometric method is simple and cost-effective when analyzing samples that predominantly contain AX as the carotenoid. The molecular absorption coefficient $E_{cm}^{1\%}$ = 2100 (in acetone, hexane, and EtOH) of AX is often utilized [4,68] (Appendix A). HPLC systems are commonly employed for the quantification and identification of AX in carotenoid mixtures. Reversed-phase systems using ODS or C-30 columns are frequently used for quantitative and LC-MS analysis [64,68,264,568]. However, normal-phase systems with silica gel columns or CN columns can also be employed. The normal-phase system offers superior separation efficiency for analogues with similar polarity and geometrical isomers [56,181,187,196,401]. Chiral columns such as Sumichiral OA-2000 can be used for the analysis of optical isomers [53]. Commercially available, highly purified AX can be used as a standard; however, if the origin of the standard is synthetic, it typically consists of a mixture of the three optical isomers, unless otherwise specified. Standard AX esters from *Haematococcus* algae can be purchased from the USP with validated concentrations [68]. Depending on the analytical environment, however, it has been reported that the quantitative value may not be consistent due to the influence of light and ambiguous other factors [569]. This is considered to be mainly due to the fact that 13- and 15-*cis* geometrical isomers, which are bent around the center of the molecule, are easily converted to all-trans isomers by light [56,148]. Therefore, it is considered necessary to standardize the conditions for quantification, such as performing the analysis under light-shielded conditions. For another reason, geometric isomers with two or more of the molecule’s double bonds arranged in the cis configuration (e.g., di-*cis* isomers) have not been included in the quantitative values by the current USP method. Since these “multi”-*cis* molecules are also affected by light, it is possible that the quantitative value is estimated to be low by the current method. Therefore, the development of an improved analytical method should accelerate in the near future. 

When AX presents as its fatty acid esters (See Section 2.1.2), it must be hydrolyzed by cholesterol esterase or lipase to lead to the free form for identification and/or quantification. For example, in the case of quantitative analysis of AX in *Haematococcus* algae, the analytical method was validated using cholesterol esterase from *Pseudomonas* sp. (EC 3.1.1.13, available from Merck (Sigma-Aldrich) or FUJIFILM Wako Chemicals) [68]. In some instances, an inexpensive lipase powder from *Candida cylindracea* (EC 3.1.1.3, Lipase OF, Meito Sangyo Co., Ltd., Japan) [265] was used for qualitative analysis, showing comparable activity to cholesterol esterase. Roche’s group employed alkali saponification under completely anoxic conditions to saponify the fatty acid esters of AX, leading to free AX; however, the experimental setup and operation were highly complex [570]. Preparative liquid separation is often required for the purification of post-saponification treatments and biological samples. In such cases, the use of an internal standard (IS) with similar physical properties is crucial. Preferred IS options include Ehinenone, Ethyl 8′-apo-β-caroten-8′-oate, and trans-β-apo-8′-carotenal [68,568,571,572]. HPLC allows for the separation of all-trans forms of AX and geometrical isomers such as 9-*cis*, 13-*cis*, and 15-*cis* AX. These isomers exhibit different UV-VIS spectra and can be easily identified by comparison with reference values if the HPLC system includes a PDA detector [4].

More recently, LC-MS with APCI and ESI ionization, along with UV-VIS detection, has been used for the detection of AX during separation using HPLC systems. AX exhibits a prominent protonated molecule (MH^+^) with *m/z* 597 in APCI and ESI-MS. The detection sensitivity of carotenoids by LC-APCI-MS and LC-ESI-MS has significantly improved to the sub-ng order. LC/MS (or MS/MS) is also a powerful tool for the analysis of complex mixtures such as AX fatty acid esters and AX glycosides [64,265]. 

While HPLC analysis is accurate and reliable, it involves complex procedures and requires skill for analysis. In recent years, the use of resonance Raman spectroscopy has been considered as a potential solution for more noninvasive, on-site analysis. The resonance Raman spectra of AX, when excited by visible lasers, exhibit dominant bands at approximately 1008 cm^−1^, 1158 cm^−1^, and 1520 cm^−1^ [573]. Raman spectra measured on samples of salmon fillets at fish markets have demonstrated the detection of AX in salmon skeletal muscle [129]. Portable Raman spectrometers have become available, allowing for on-site studies, and it has been reported that the signal intensity of the AX-specific Raman band at 1518 cm^−1^ (C=C stretching frequency) increases in an AX concentration-dependent manner. This signal can be distinguished from fish proteins and lipids, enabling the determination of AX levels in different parts of salmon fillets and different species of salmon. These findings indicate the effectiveness of Raman spectrometry for on-site AX quantification [574]. Polynomial approximation or multivariate curve resolution-alternating least squares (MCR-ALS) using the Raman spectrum has also been used for the quantification of salmon, *Phaffia* yeast, and *Haematococcus* algae [251,575,576,577]. Raman microscopy has shown great potential for diverse types of analysis. AX, due to its strong lipophilic properties, co-localizes with lipid fractions in cells and exhibits a characteristically strong Raman spectrum. Therefore, it has been investigated as a non-toxic, non-destructive Raman probe for organelles [578,579]. Pioneering studies have used resonance Raman microscopy to determine the localization of topically applied AX in skin tissue [580].

It appears that the methods mentioned in these reports can also detect analogues (such as canthaxanthin and adonirubin) and degradation products [581], although these are not major concerns when the purity of AX is relatively high and the variation in the profile of these impurities is minimal. In conclusion, the selection of an appropriate analytical method is crucial and depends on the specific requirements of the analysis. 

### 2.5. Relationship between Human Culture and Astaxanthin

#### 2.5.1. Historical Exposure of Human Societies to Astaxanthin Sources in Nature

Human societies have interacted with nature for thousands of generations and taken advantage of substances with medical and health benefits. 

Carotenoids are the most abundant family of pigments in nature. Aquatic resources create numerous opportunities for humans to experience various applications of natural pigments, including AX [582]. AXs namesakes, the crayfish and lobster, hold an important place in many culinary and cultural traditions, including those of Sweden and America. 

*Astacus astacus* is a European crayfish species native to Scandinavia, that become popular among Swedish nobles in the Middle Ages. By the 16th century, festive crayfish parties became a Swedish tradition, primarily featuring freshwater crayfish (flodkräfta) or marine lobster (*Nephrops norvegicus*) in the case of the Swedish west coast. Both crayfish and lobster turn bright red after boiling when AX is released from a protein complex in the crustaceans’ shell. The celebration was coined “kräftskiva” in the 1930s and is still celebrated to-day in the month of August [583,584]. The luxury and popularity of AX-rich crustacean meals were often depicted in the still-life paintings of the Dutch Golden Age. The bright oranges and reds of cooked crustaceans were centerpieces in paintings depicting light breakfasts called “ontbijtjes” and lavish banquets featuring objects of luxury, called “banketje” or “pronkstilleven” [584].

In the United States, the red swamp crayfish (*Procambarus clarkii*) is popular in the state of Louisiana, where crayfish are also known as crawfish, mudbugs, and crawdads. The crawfish is a central figure in the culture of Louisiana’s Native American Houma nation. He was once part of the Chakchiuma group, distantly related to both the Choctaw and Chickasaw. “Shâkti Humma”, or “red crawfish”, was a giant crawfish who created the world and formed living creatures out of mud. Houma warriors depicted the red crayfish as their war emblem because this tiny creature is known to brandish its pincers, never backing down, even in the face of larger enemies. The red crawfish was an integral symbol of resilience and strength, and even the word “Hou-ma” means “red” [585,586,587].

When the Acadians from the Canadian maritime provinces settled in the bayous of Louisiana in the 17th century, they may have observed the Houma harvesting crawfish. The Acadians, now called Cajuns, may have adapted their Canadian lobster recipes for making crawfish boils, which today form the backbone of Louisiana cuisine. In the 1980s, the crawfish officially became Louisiana’s state crustacean, further reinforcing the cultural significance of AX in this region [588].

Although crustaceans are a celebrated source of AX, the most abundant source of AX in the human diet is wild salmon. Salmon have been an important food source for coastal cultures across the Northern Hemisphere since prehistoric times and are widely featured in legends and depicted in art. The oldest known image of salmon may be a relief carving of a salmon in a cave in France from the Gravettian era (25,000–20,000 BCE) [589,590].

Salmon is revered in many parts of the world. In Native American art, salmon are a symbol of perseverance, ancestral knowledge, and spiritual journey. The First Salmon Ceremonies practiced across the American Northwest celebrate the arrival of spawning salmon each spring [591]. In Celtic mythology, Fionn mac Cumhaill, the legendary giant credited with constructing Northern Ireland’s famous Giant’s Causeway, gained knowledge of the world by tasting the Salmon of Wisdom [592]. One of the tales from the 6th century featuring King Arthur recounts his quest, during which he rode on the back of a salmon to Gloucester and freed a captive deity [593]. In Icelandic Norse mythology from the 13th century, the trickster deity Loki transformed into a salmon to escape his brother Thor. However, Thor caught Loki at the base of his tail, forming the iconic grip point for fishermen [593].

In the city of Murakami in Niigata Prefecture, Japan, salted and wind-dried salmon have been prepared using traditional methods since the Heian period (794 to 1185). The history of salmon consumption in the region can be traced back even further, with salmon bones found in archaeological remains from the Jomon era (pre-B.C.). Salmon has long been recognized as one of the most important foods in the region. During the Edo period (1603–1867), when Murakami served as a feudal domain under the Tokugawa shogunate, salmon held significant economic value, serving as a currency for paying taxes, presenting gifts to the Imperial Court and Tokugawa shogunate, and being distributed as salaries to government officials. In the late Edo period, when salmon catches in the Miomotegawa River flowing through the city declined, a samurai named Heiji Aotobu proposed a salmon propagation project to the domain government. River construction and resource conservation efforts were undertaken to create an environment conducive to salmon spawning, resulting in the recovery of salmon populations. This work was groundbreaking in terms of sustainable fishing practices. Presently, Murakami salmon continues to be presented as gifts and is featured in special meals during New Year celebrations [594]. Murakami’s salted salmon has also been used in the Daijosai, the first Niiname-sai (a Japanese harvest ritual) of a Japanese emperor following their enthronement. This once-in-a-generation festival involves the emperor serving the newly harvested grains to the gods of heaven and earth, “Amaterasu and Tenjin Chigi”, as well as partaking in them. The festival site, known as “Yuki” in the east and “Suki” in the west, receives harvested grains from all over Japan to be offered to the gods. Murakami’s salted salmon is among the products from Niigata Prefecture dedicated to Yuki [595].

The Ainu people, who have inhabited northern parts of Japan since ancient times, also consider salmon to be a vital dietary source. The Ainu refer to salmon as “kamuichep” (divine fish) or “shipe” (the true food). The Ainu hold a “new salmon prayer” called “Asiri Chep-nomi” when the salmon return, seeking a bountiful catch. They have various rituals and traditions associated with salmon, which skillfully reflect the salmon’s ecology and the wisdom of their ancestors. For example, the first salmon to return to the river is reserved for the fox gods, who protect the water source, and must not be caught. The subsequent salmon are designated for other gods and then shared among humans, who rely on the salmon for sustenance [596,597]. Moreover, salmon holds deep cultural and culinary significance not only for the Ainu but also for the indigenous peoples of the North Pacific [596].

While animal sources continue to inspire and sustain AX consumption, the increasing demand for AX as a nutritional supplement must not become a burden on aquatic ecosystems. As exemplified by samurai Heiji Aotobu, environmental conservation and stewardship must guide us towards a sustainable future. That is why we turn to *Haematococcus* algae as a sustainable source of natural AX.

#### 2.5.2. Human Culture Shift towards Sustainability: *Haematococcus* algae as a Promising Source of Natural Astaxanthin

As discussed, historical evidence suggests that indigenous or traditional knowledge, such as knowledge about food and medicine, was developed through human interaction with the environment and nature. This knowledge has been passed down through generations as an integral part of human culture. In addition to aquatic animals such as shrimp and salmon, the consumption of freshwater aquatic plants has ancient origins due to their abundance, unique sensory properties, and nutritional and health benefits [598]. Microalgae and seaweed have been used as food since medieval times and have established markets in Asia, with a growing market in Europe [599]. Archaeological findings from Monte Verde, Chile, indicate that the use of marine algae as food dates back around 14,000 years [600]. Not only were aquatic plants consumed as food, but plant-based medicinal extracts, including algae, herbs, and fungi, were also prevalent. This knowledge forms the foundation for many contemporary health-related discoveries. In fact, over the past century, eating and feeding practices have evolved from social and cultural customs to explicit healthcare and medical practices [601]. Modern nutritional science has shed light on the nutritional and therapeutic effects of various components found in aquatic plants and their role in biological processes.

Furthermore, it is essential not to overlook the connection between the health of human populations and the Earth’s natural food systems and biodiversity, as well as the role of microalgae in ecosystem preservation.

In the late 18th century, Alexander von Humboldt, an influential German scientist, was one of the first to note the negative impact of human-induced environmental alterations (although the terms “ecology” and “ecosystems” did not exist at that time) and extensive land use on human well-being [602]. Overexploitation (harvesting species from the wild at rates that exceed their reproductive capacity) and climate change have been identified as major drivers of biodiversity decline on Earth [603].

In one of the latest reports by the European Commission (EC), developed through collaboration between 50 experts from 25 EU Member States, a set of recommendations has been formulated to rethink the relationship between humans and nature, highlighting the role of human culture in achieving the Sustainable Development Goals (SDGs) [604]. Microalgae, in this regard, are being considered as a promising candidate that can directly or indirectly contribute to the SDGs [605], while also having historical roots in human culture. As carbon-hungry and nutrient-rich “sustainable biofactories”, microalgae offer potential solutions for global food security and mitigating environmental issues [606,607]. The historical roots of microalgae and its derivatives in human culture facilitate the reintroduction of algal biomass and algal bioactive functional ingredients, including AX, into global food and nutrition systems. 

Over the past few decades, studies on microalgae and their bioactive compounds, such as AX, have continuously revealed their role in promoting the circular economy and their positive effects on the health and well-being of the Earth and its flora and fauna. Although *Haematococcus* alga was discovered in the 18th century [222], its significance as a promising sustainable and abundant natural source of AX has garnered increasing attention in recent years. 

It is important to note that overexploitation of any flora and fauna, including algae and other carotenoids’ sources in nature (fruits and vegetables), may have an irreversible impact on the environment. In this case, well-designed and properly scaled commercial/artificial mass production of carotenoids seems to be more sustainable. 

In fact, sustainability potential includes the source from which AX and other carotenoids are obtained. Microalgae is considered a more sustainable source than other plant sources of carotenoids. For instance, the marigold flower is currently the predominant natural source of lutein. However, it has a lower growth rate than microalgae and requires arable land, and its harvest and extraction are only seasonal. Such limitations apply to other carotenoids from fruits and vegetables, such as zeaxanthin. There are some concerns about the sustainability of agricultural products as compared to algaculture, including the use of water, land, pesticides, and fertilizers. 

Microalgal cultivation has added environmental benefits over plants with higher carbon sequestration, a reduced water footprint, and no pesticide use. 

Both lutein and AX from microalgae can be considered sustainable active compounds, with even potentially complementary health benefits, especially for vision and eyes [608]. Currently, there is no commercial production of lutein from microalgae [609,610]. 

Moreover, although this review is focused on AX, it is noteworthy that this phytonutrient is only part of the algal meal. The whole microalgae biomass is consumable by humans and contains several high-value nutrients and bioactive molecules with health-promoting properties, including protein, essential fatty acids, antioxidant compounds, etc. For this reason, there is normally no or negligible amount of waste or residue produced from microalgae harvest, unlike other carotenoids extracted from agricultural products.

## 3. Industrial Use of Astaxanthin

### 3.1. Astaxanthin as a Pigment and Beyond: Astaxanthin in Aquaculture and Poultry Industries

#### 3.1.1. Applications in Aquaculture

In the late 1980s, crustaceans, particularly krill, and their extracts were utilized as a source of AX in the aquaculture industry. In Japan, Antarctic krill was used as a coloring agent for sea bream during the 1980s [388]. Since 1984, a synthetic AX formulation called Carophyll Pink, developed by Hoffmann-La Roche (now DSM), has been introduced primarily to enhance the color of salmon and trout muscles, as well as the body color of sea bream and shrimp. It remains the most widely used AX product in the fishing industry. AX derived from *Phaffia* yeast is also commonly employed as a coloring agent for sea bream [611]. The bacterium *Paracoccus carotinifaciens* produces AX and related carotenoids, and a product called Panafard-AX, manufactured by ENEOS in Japan, has been recently introduced for industrial use as a color enhancer for salmonid muscle [329]. Additionally, it is used as a color enhancer for prawns, where AX is metabolized and converted into yellow carotenoids to enhance their body color [361]. This product is preferred in countries where synthetic AX is not accepted and by growers who are particularly interested in organic products. Previously, AX fed to aquacultured fish was mainly used to improve color; however, the application of biologically derived AX has expanded in recent years, encompassing other objectives such as enhancing the health of aquacultured fish, including improvements in egg quality, hatching, development, and fry growth, as discussed in Section 2.1.3 [20]. A comprehensive review of the application of AX in commercial aquaculture fisheries is available for further details [612].

#### 3.1.2. Applications in Poultry and Livestock Farming

In the poultry industry, carotenoid pigments such as paprika pigments and lutein are commonly used to enhance the color of chicken egg yolks. AX derived from *Phaffia* yeast has a long history of use in this regard [613]. Additionally, synthetic products and those derived from the bacterium *Paracoccus* and *Haematococcus* algae are also employed in the poultry industry. AX is frequently utilized to improve meat quality and coloration in broilers and other poultry species [614,615,616,617,618]. Furthermore, AX has recently been used in poultry health management, with potential benefits in protecting against heat stress, improving overall health conditions, hatchability, and egg quality [615,619,620,621,622,623,624,625,626,627]. For more detailed information on the application of AX in the poultry industry, a comprehensive review is available [628].

In the livestock industry, the administration of AX has shown positive effects on the growth, fertilization, and development of calves [629,630,631,632,633,634]. It has also been found to improve growth, quality, and health in pigs under environmental stresses [635,636,637]. Moreover, AX has been reported to enhance follicle and oocyte development in cultured follicles [631,638,639,640,641,642,643,644]. Additionally, AX has been shown to mitigate ovarian damage caused by oxidative stress [645,646]. AX is also useful in cryopreservation and temporary preservation techniques in livestock animals [647,648,649,650]. These findings suggest that AX has potential benefits for reproduction in livestock animals. Based on this concept, clinical trials are being conducted to investigate the efficacy of AX on human infertility [651,652,653].

### 3.2. Sustainable Commercial Production of Astaxanthin by Haematococcus algae

Commercial cultures of *Haematococcus* algae have been established and conducted in various countries worldwide. This document provides a brief overview of the cultivation techniques, while a separate comprehensive review (references [654,655,656,657]) can be referred to for a detailed description of each technique.

Initially, outdoor cultures in raceway open ponds were employed for *Haematococcus* algae cultivation, similar to other cyanobacteria and green algae. This method is still widely used today due to its low cost and scalability. However, environmental factors such as climate change, contamination, and chemical pollution pose challenges to sustaining production volume and quality. *Haematococcus* algae, being normal freshwater algae, grow slower than other common microalgae and are susceptible to invasion by faster-growing microalgae. Encystment for AX accumulation is also limited. To overcome these limitations, outdoor tubular systems with long tubular structures were developed. This closed system minimizes contamination risks and enables high AX concentrations in the algal biomass through the addition of organic acids that serve as substrates for secondary metabolic pathways. However, this system is still influenced by climate conditions, and the areas suitable for cultivation are restricted to ensure stable production. Moreover, maintaining stable gas exchange, linear velocity during cultivation, and cleaning in case of contamination require significant expertise.

An indoor closed-culture system has emerged as a solution to mitigate weather and contamination risks. This system utilizes photobioreactors, optimizing the sequence of outdoor algae cultivation events in an indoor, closed environment. Stable biomass and high AX accumulation can be achieved in both batch and fed-batch systems. However, this system is the most costly as it relies on light energy and requires not only cultivation facilities but also cleanliness and manpower to maintain the environment. *Haematococcus* algae can be categorized into two cell stages (described in Section 2.3.3), and these stages can be cultured simultaneously in a single-stage continuous or batch system. However, a two-stage culture system is commonly adopted for various reasons.

Furthermore, *Haematococcus* algae primarily grow autotrophically. However, under dark conditions, they can be cultured “heterotrophically” using organic acids such as acetic acid as an energy source [413]. Cultivating *Haematococcus* algae heterotrophically allows for similar techniques as other common microorganisms and can produce AX as a secondary metabolite by modifying the cultivation conditions [414,658]. This method offers advantages such as a relatively easy increase in specific growth rate and high biomass yield using conventional fermentation engineering techniques. However, it has some disadvantages, including lower AX yield compared to two-step autotrophic culture under complete darkness and the potential for impurity changes. Challenges in *Haematococcus* algae cultivation include light-illuminated mixotrophic culture, hybrid systems combining heterotrophic culture and autotrophs with light irradiation, and breeding mutant strains with higher AX concentrations [413,451,508,659,660,661,662,663,664,665,666,667,668,669]. Transgenic methods have also evolved considerably [663,670,671,672], and newer technologies such as CRISPR-Cas9 are believed to be effective in improving these processes [673,674,675,676]. However, non-GMO options are generally preferred by consumers for most AX products on the market.

### 3.3. Human Uses of Astaxanthin 

#### 3.3.1. Ever-growing Interest in Human Applications of Astaxanthin 

AX has been extensively studied in preclinical and clinical research for its potential antioxidant activity against various diseases and disorders caused by oxidative stress. Clinical trials have explored the potential benefits of AX in the prevention and/or management of oxidative impairment, acute and chronic inflammatory conditions, metabolic diseases, cardiovascular diseases (CVD), central nervous system (CNS) disorders, intestinal health conditions, and even exercise and physical performance. The detailed findings of individual non-clinical studies are beyond the scope of this review. A summary of clinical findings on AX that were previously published by the authors [130], is updated here in Table 4 (Appendix A.

These health-promoting effects of AX include the following: Individual references are listed in Table 4. 

Oxidative Stress (Table 4A): AX has also evaluated oxidative stress markers in clinical trials on subjects with widely divergent backgrounds, including obesity, metabolic diseases such as type 2 diabetes, exercise, aging, fatigue, surgical procedures, and infertility, in addition to healthy subjects (Table 4A). Several studies have been designed to evaluate oxidaiton product concentrations in blood or urine (e.g., MDA/TBARS, d-ROM, protein carobonyl, isoprostane (ISP), and OHdG) via reaction by ROS, to evaluate the expression level of antioxidant enzymes *in vivo* (e.g., superoxide dismutase (SOD), catalase, PON-1, and Nrf-2), and to evaluate the protective capacity of the blood against oxidative damage (TAC, BAP, and TAS). Although not all studies have shown beneficial effects on all oxidative stress markers, the results of these studies suggest that AX contributes to a significant, albeit partial, reduction in oxidative stress, especially under conditions of low biological defense capacity against ROS. The reason for the differences in the results can be attributed to the different evaluation methods and the quantity and quality of the oxidative stress exposure of the subjects. Future research should explore which markers of oxidative stress reflect which health conditions and which markers are best suited for evaluation. 

Skin Health (Table 4B): Since the very early days of human use of AX, it has been expected to contribute to “skin health” due to its strong ^1^O_2_ quenching activity and lipid peroxidation inhibitory effect (Table 4B and Appendix A). In the very early days of human topical application studies, Yamashita evaluated the effects of krill-derived AX diester on sunburn-induced erythema and pigmentation and found it potentially beneficial (see Appendix A). Subsequently, the evidence for the prevention of inflammation and pigmentation against UV-induced erythema from various different research groups has been accumulating. Most of these actions could be expected to maintain barrier function in the epidermis through topical application or oral administration of AX. In parallel, several research groups have reported that astaxanthin may contribute to the maintenance of skin wrinkles and elasticity since its action in the dermis, where the impact of singlet oxygen seems to be strongest, is expected to protect the matrix that maintains the elasticity of the epidermis, such as collagen and elastin, from ^1^O_2_. Recently, Honda et al. found that AX, especially its *cis*-isomers, directly inhibits elastase [677], suggesting that there may be a pathway that is not mediated by ROS. However, little is known about the dynamics of AX in the skin, and future studies are expected. 

Eye Health (Table 4C): Unlike lutein and zeaxanthin, AX has not been reported to be detected in the macula under normal dietary conditions. However, it has been reported that it migrates into the retina, albeit partially, in studies using monkeys (*Macaca fascicularis*) treated with AX and its precursor, adonixanthin [678]. Although preclinical studies are not described because they are outside the scope of the presented article, it has been reported that AX prevents choroidal neovascularization and protects photoreceptor cells and retinal ganglion cells from hypoxia and light stress [679,680,681,682,683]. Therefore, its efficacy in retinal degeneration, including age-related macular degeneration (AMD) and glaucoma involving neuronal cell death, is expected. Unfortunately, in human clinical trials, AX has been evaluated in combination with lutein and zeaxanthin in AMD, and its efficacy remains unclear. However, preclinical studies are actively being conducted in these fields, and further research is expected in the future. In contrast, many clinical trials have been conducted on asthenopia (eyestrain), and all have shown significant benefits. Since eyestrain can also lead to stiff shoulders and other symptoms, AX has been shown to be useful for these expanded conditions. The main mechanism for this is thought to be the improvement of blood flow in microvasculature [684,685,686], including the uveal tissue such as the ciliary body and the iris, which regulate the focus of the eye. Moreover, in the same group, AX has been reported to be beneficial against oxidative damage and inflammation during cataract surgery. Thus, the authors believe that AX may have some protective effects, just as various xanthophylls have ocular benefits in other organisms. 

Cardiovascular health (Table 4D): This category includes dyslipidemia, glucose/lipid metabolism, chronic inflammation, and type 2 diabetes (T2DM). Since the early 2000′s, AX has been investigated in clinical as well as non-clinical studies. In particular, oxidative stress in obesity has been extensively analyzed (Table 4A), and inflammatory markers and beneficial adipokines have also been investigated. More recently, studies on insulin resistance in obese and T2DM patients have suggested that AX has partial beneficial effects. In these studies, the scope has been extended from its effects as a simple antioxidant to mitochondrial activity, and preclinical studies are discussed in another review by the authors [130]. At the same time, evaluation of the usefulness of the combined effects of exercise is being demonstrated, and this may lead to a beneficial compound that does not cause exercise resistance. (See other reviews [687] for more on “Exercise Resistance.”) Concurrently, the maintenance and activation of mitochondrial activity are expected to be beneficial for cardiac function. In fact, the administration of AX to patients with CVD has been reported to partially improve cardiac function. 

Exercise/Sports performance, skeletal muscle function (Table 4E): Early preclinical studies from Aoi et al. suggested that AX could efficiently prevent oxidative damage to skeletal muscle during strenuous exercise and may accelerate recovery [688]. Many studies have been conducted with athletes, and results vary considerably, with some studies finding benefit and others finding no benefit in ameliorating post-exercise injuries. The reasons for this are speculative but may depend on the protocol of administration, the characteristics of the subject’s skeletal muscles, and other physical functions. Skeletal muscles with sufficient endurance for exercise may not be as effective as expected. Future studies are needed to determine which subjects should be treated with AX and how it should be administered. Similarly, the same thing can be observed with regard to energy metabolism in athletes’ physical activity functions. However, in subjects with relatively low exercise habits and in the elderly, AX has provided beneficial effects, such as improvement of muscle function, including muscle strength, enhanced endurance, and changes in body composition. Therefore, the effects of AX may have beneficial effects on muscle weakness, including sarcopenia, in the elderly and obesity. 

Brain health (Table 4F): AX has shown partially beneficial effects when cognitive function was evaluated as a brain function in the elderly. In particular, it is thought to be very beneficial for improving cognitive function in subjects with moderate cognitive impairment (MCI). In addition to the antioxidant effect in the brain, this effect is thought to be related to the improvement of blood flow and mitochondrial function. It is expected to be very useful for improving muscle function in the elderly as well as preventing so-called frailty. Concurrently, AX has also been reported to have beneficial effects on fatigue during brain overload. Similarly, a subjective but stress-reducing effect has also been reported. Although there are still many scientifically unclear aspects of fatigue, it is expected that as the scientific analysis of fatigue progresses, it will become clearer what kind of effects AX has on what kind of fatigue.

Infertility (Table 4G): AX has been reported to have beneficial effects on polycystic ovary syndrome (PCOS), endometriosis, and hypermenorrhea in females. In males, AX has been reported to improve sperm dysplasia and activity. Additionally, as in the case of livestock animals (Section 3.1.2) Additionally, as in the case of livestock animals (section), there is a beneficial effect on sperm cryopreservation, and it is expected that beneficial effects on assisted reproductive technology (ART) will be reported in the future.), there is a beneficial effect on sperm cryopreservation, and it is expected that beneficial effects on assisted reproductive technology (ART) will be reported in the future. 

The pharmacokinetics of AX are also presented in Table 4H. These are presented in detail in Section 4.1. Other benefits are summarized in Table 4I. With the expansion of clinical evidence for the health benefits of AX, its applications in nutraceuticals and cosmetic formulations keep growing globally. These benefits reflect only a small portion of the findings from the preclinical studies. The authors believe that further benefits will be demonstrated in the future, especially if the species differences are small and AX dosing is sufficient to satisfy the PK/PD concept.

##### Market History and Evolution

The usage of AX as a supplement for human consumption has a history of approximately 30 years, primarily initiated in East Asia, where it has been utilized in food supplements and topical skincare products. The introduction of AX into each market occurred at different times due to varying regulations across countries.

In the 1990s, Japan was the first Asian country to permit the use of AX derived from *Haematococcus* algae in food additives and general foods [689]. During this period, other sources of AX, such as krill, were also explored for their potential application in skincare products in Japan [690]. The very first AX supplement for human use was launched in Europe in 1995. Subsequently, AX was introduced to the North American supplement market in the late 1990s, slightly later than in Europe and East Asia. Between 2004 and 2006, AX supplements became available in Malaysia, Indonesia, and Thailand. Singapore saw the launch of its first AX supplements in 2012, followed by the Philippines in 2018.

The demand for natural AX in health and well-being products has been steadily increasing, with the majority of products positioned within the dietary supplement category. In Europe, there is a growing focus on AX products in the sports nutrition and animal health categories. The targeted health areas for AX supplements in the EU include brain/cognitive health, mood, skin health, eye health, cardiovascular health, immune health, energy and stamina, bone and joint health, and digestive health. Positioning AX as a natural, organic, and vegan/vegetarian ingredient is prominent in the market. In South East and South Asia markets, AX products are primarily promoted for skin health, eye health, brain health, and cardiovascular health, and these categories continue to be on-trend. 

AX supplements have a wide range of product formats, from conventional hard or softgel capsules to topical skin care formulations (creams, lotions, and gels) to more novel confectionery and liquid tonics. In fact, consumer interest in more innovative delivery formats such as water-dispersible powders, liquid shots, and gummies has been growing in recent years. The market for AX products has expanded, and launches of tablets, powders, liquids, and gummies have increased. However, the incorporation of AX into general foods is limited due to strict regional food regulations and the relatively high cost of AX. 

AX supplements often carry structure-function claims related to eye health and skin health, with energy, brain, and skin support being the fastest-growing applications. AX is known to support dynamic focus/accommodation, reduce eye strain, modulate ocular hydration (dry eye), impact optic nerve health (glaucoma models), and potentially play a protective role against age-related macular degeneration. 

Natural AX is commonly available with Halal and Kosher certifications and is considered a vegan ingredient derived from green algae. The algae used for AX production that are grown in freshwater in enclosed photobioreactor systems have the least environmental impact and can address consumer concerns about sustainability and safety. 

##### Health Claims and Regulations 

In Japan and the US, AX-containing products can be granted Foods with Function claims (FFC) or DSHEA claims, respectively, if there is sufficient clinical substantiation supporting the claims. In Japan, FFC has been granted to AX for supporting vision and eye health, skin beauty, healthy aging, cognitive function, and mental health (stress and fatigue). 

In Europe, such as many other markets, synthetic AX is not permitted to be used in human dietary supplements or food. Given that the first natural AX supplement (*Haematococcus* algal biomass) for human use was launched before 1997, according to EFSA (The European Food Safety Authority), *Haematococcus* algal meal or AX biomass did not need to be authorized as a Novel Food and has been used in food supplements for almost three decades. Moreover, AX-rich oleoresin from *Haematococcus* algae is approved as a Novel Food, and included in the positive list (EU) 2017/2470. There is yet no authorization granted for the use of AX in human food products in the EU. EFSA has not yet approved any health claims for AX in Europe. 

In most Asian countries, for instance, Korea and India, according to regulations for nutraceuticals, natural AX must be derived from *Haematococcus* algae. This requirement ensures that AX is sourced from this particular algae species. 

In regions outside of Asia, including Europe, alternative sources of AX such as *Phaffia* yeast and *Paracoccus*-derived AX have also been introduced to the market. However, their market share remains relatively small compared to *Haematococcus*-derived AX. This suggests that *Haematococcus* algae is still the preferred and more widely accepted source of natural AX in these regions as well. The preference for *Haematococcus*-derived AX may be attributed to several factors, such as its historical use, availability, and perceived efficacy [1].

**Table 4 marinedrugs-21-00514-t004:** Human clinical study for astaxanthin: Update on efficacy from human clinical studies.

(A) Oxidative Stress.						
Author/Year/ Reference	Study Design	Subjects	Dose ^#,##^	Duration	Major Outcome ^†^	Description
Oxidative stress markers in “metabolic disorder”						
Rad NR. et al. 2022 [691]	Randomized, double-blind, placebo-controlled prospective study	50 Type 2 Diabetes Mellitus (T2DM) patients receiving metformin	0, 10 mg/day	12 weeks	✓ Cardiovascular health ✓ Metabolic syndrome ✓ Oxidative stress ✓ (T2DM)	*Investigation of additive synergistic effects on metformin (1000–2000 mg/day).* Significantly increased blood total antioxidant capacity (TAC) levels at the end of the intervention only in the AX-treated group, while MDA remained unchanged. Similarly, increased SOD and catalase activity in blood and increased Nrf2 protein in PBMCs were observed at the end of the intervention only in the AX-treated group. *No safety concerns were identified in this study.*
Ishiwata S. et al. 2021 [692]	Open-labeled, prospective study	17 patients with systolic heart failure	12 mg/day *	3 months	✓ Cardiovascular health ✓ Oxidative stress ✓ (Heart failure)	After 3 months of AX supplementation, significantly decreased serum d-ROM (Diacron-Reactive Oxygen Metabolites), *p* = 0.018, but no change in plasma biological antioxidant potential (BAP) or urinary ratio of 8-OHdG/Cr *No safety concerns were identified in this study.*
Shokri-Mashhadi, N. et al. 2021 [693]	Randomized, double-blind, placebo-controlled prospective study	44 patients with type 2 diabetes	0, 8 mg/day	8 weeks	✓ Oxidative Stress, ✓ (Type 2 diabetes) ✓ Type 2 diabetes	Decrease plasma levels of MDA (*p* < 0.05) *No safety concerns were identified in this study.*
Kato T. et al. 2020 [694]	Open-labeled, prospective study	16 patients with systolic heart failure	12 mg/day *	3 months	✓ Cardiovascular health ✓ Musculoskeletal ✓ function	Increased left ventricular ejection fraction (LVEF) from 34.1 ± 8.6% to 38.0 ± 10.0% (*p* = 0.031), and the 6-min walk distance increased from 393.4 ± 95.9 m to 432.8 ± 93.3 m (*p* = 0.023). Significant relationships were observed between percent changes in serum dROM level and those in LVEF. *No safety concerns were identified in this study.*
Coombes J.S et al. 2016 [695] Fassett, R.G. et al. 2008, [696]	Randomized, double-blind, placebo-controlled prospective study	58 renal transplant recipients	0, 12 mg/day	12 months	✓ Oxidative stress ✓ Vascular health ✓ Bioavailability, ✓ ADME ✓ (in renal transplantation)	There was no effect on oxidative stress in renal transplant recipients. (The XANTHIN trial) *No safety concerns were identified in this study.*
Takemoto M. et al. 2015 [697]	Case report	1 Werner syndrome patient	12 mg/day *	6 months	✓ Werner syndrome ✓ (Metabolic syndrome)	There was no significant changes after AX intervention in MDA-modified LDL. *No safety concerns were identified in this study.*
Choi H.D. et al. 2011 [698]	Randomized, two-arm, prospective study	23 obese and overweight subjects	5 and 20 mg/day	3 weeks	✓ Oxidative Stress ✓ (Obesity) ✓ Cardiovascular health ✓ Metabolic syndrome	5 mg/day: MDA decreased by 34.6%, isoprostane (ISP) decreased by 64.9%, SOD increased by 193%, and TAC increased by 121% after 3 weeks compared to baseline (*p* < 0.01). 20 mg/day: MDA decreased by 35.2%, ISP decreased by 64.7%, SOD increased by 194%, and TAC increased by 125% after 3 weeks compared to baseline (*p* < 0.01). *No safety concerns were identified in this study.*
Choi, H.D. et al. 2011 [699]	Randomized, double-blind, placebo-controlled, prospective study	27 overweight subjects	0, 20 mg/day	12 weeks	✓ Oxidative Stress ✓ (Obesity)	MDA reduced by 17.3% and 29% after 8 and 12 weeks compared to placebo (*p* < 0.01), ISP reduced by 40.2% and 52.9% after 8 and 12 weeks compared to placebo (*p* < 0.01), SOD increased by 124.8% after 12 weeks compared to placebo (*p* < 0.01), and TAC increased by 130.1% after 12 weeks compared to placebo (*p* < 0.05). *No safety concerns were identified in this study.*
Iwabayashi M. et al. 2009 [700]	Open-labeled, prospective study	35 healthy female subjects (with high oxidative stress, postmenopausal)	12 mg/day	8 weeks	✓ Oxidative stress ✓ Metabolic syndrome ✓ Mood/Stress ✓ (Unidentified complaints)	Increased blood biological antioxidant potential (Biological Antioxidant Potential (BAP); +4.6%, *p* < 0.05). *No safety concerns were identified in this study.*
Kim Y.K. et al. 2004 [701]	Open-labeled, prospective study	15 healthy postmenopausal females	0, 2, 8 mg/day	8 weeks	✓ Oxidative Stress ✓ Metabolic syndrome ✓ (Postmenopausal females)	Decreased plasma TBARS levels: 2 mg group from 1.42 ± 0.18 to 1.13 ± 0.18 nM/mg (*p* < 0.05). 8 mg AX group from 1.62 ± 0.14 nM/mg to 1.13 ± 0.12 nM/mg after 8 weeks (*p* < 0.05). Increased TAS from 0.85 ± 0.42 mM/L to 1.90 ± 0.58 mM/L in the 8 mg group. Urinary 8-isoprostane excretion did not decrease significantly. *No safety concerns were identified in this study.*
Oxidative stress markers in “skin”						
Chalyk, N. et al. 2017 [702]	Open-label, prospective study	31 subjects; 18 obese, 8 overweight, 5 healthy, over the age of 40	4 mg/day	92 days	✓ Oxidative Stress ✓ (Middle age, obesity) ✓ Skin health	Plasma MDA decreased with AX by 11.2% on day 15 and by 21.7% on day 29 (*N.S.*). *No safety concerns were identified in this study.*
Yoon HS. et al. 2014 [703]	Randomized, double-blind, placebo-controlled prospective study	44 healthy females with wrinkles grade ≥ 2 (≥40 yrs.)	0, 2 mg/day *	12 weeks	✓ Skin health ✓ Oxidative Stress ✓ (UV irradiation)	*AX (2 mg/day) combined with collagen hydrolysate (3 g/day)*. Skin biopsy after UV irradiation: no difference in oxidative markers (Thymine dimers, 8-OHdG) between the two groups in histological evaluation. Regarding mRNA expression, significantly upregulated expression of procollagen type I tended to upregulate fibrillin-1, while significantly downregulated MMP1 and MMP12 in the AX group compared to the placebo. *No safety concerns were identified in this study.*
Satoh A. et al. 2009 [704]	Randomized, single-blind, placebo-controlled prospective study	27 patients with atopic dermatitis	0, 12 mg/day	4 weeks.	✓ Oxidative stress ✓ Skin health ✓ Atopic dermatitis ✓ Immunity	The Th1/Th2 balance shifted significantly toward Th1, and urinary 8-OHdG concentrations decreased slightly but significantly. *No safety concerns were identified in this study* [Article in Japanese]
Oxidative stress markers in “ophthalmology; after cataract surgery “						
Hashimoto H. et al. 2021 [705]	Open-labeled, prospective study	35 subjects who underwent bilateral cataract surgery (intraocular lens implantation) (Mean *c.a* 71 yrs.)	6 mg/day	2 weeks	✓ Eye health ✓ Oxidative stress ✓ (Cataract surgery)	The antioxidant effect of AX was analyzed in relation to age. None of the parameters were correlated with age before AX intake; however, only total hydroperoxide values were significantly correlated after AX intake (*r* = 0.4, *p* < 0.05). Total hydroperoxide levels were similar in younger and older age groups (<70 vs. ≥70 years) before AX intake but significantly decreased in younger age groups after intake (−0.21 ± 0.18 vs. −0.05 0.31, *p* < 0.05), resulting in a significant difference (*p* < 0.05). Thus, the previously observed decrease in mean total hydroperoxide levels after AX intake was likely due to a greater response in the younger age group analysis associated with this study [706]. *No safety concerns were identified in this study.*
Hashimoto H. et al. 2019 [707]	Open-labeled, prospective study	35 subjects who underwent bilateral cataract surgery (intraocular lens implantation) (Mean *c.a* 71 yrs.)	6 mg/day	2 weeks	✓ Eye health ✓ Oxidative stress ✓ (Cataract surgery)	In this analysis, the effect of AX intake on the relationship between VEGF levels and ROS-related parameters before and after bilateral cataract surgery was analyzed by gender. VEGF, hydrogen peroxide, and total hydroperoxide levels in the aqueous humor, as well as O_2_•^−^ scavenging activity, were measured. For women only, VEGF levels and O_2_•^−^ scavenging activity before AX intake were negatively correlated (*r* = −0.6, *p* < 0.01) and positively correlated with total hydrogen peroxide levels before and after AX intake (*r* = 0.7, *p* < 0.01, respectively). Analysis associated with this study [706] *No safety concerns were identified in this study.*
Hashimoto H. et al. 2016 [708]	Open-labeled, prospective study	35 subjects who underwent bilateral cataract surgery (intraocular lens implantation) (Mean *c.a* 71 yrs.)	6 mg/day	2 weeks	✓ Oxidative Stress ✓ (during surgery) ✓ Eye health	Superoxide anion scavenging activity (U/mL): 18.2 ± 4.1 at 0 weeks reduced to 19.9 ± 3.6 after 2 weeks of supplementation compared to baseline, *p* < 0.05. Total hydroperoxides (d-ROM) from 1.16 ± 0.18 at 0 weeks reduced to 1.04 ± 0.31 after 2 weeks were of supplementation compared to baseline, *p* < 0.05. Analysis associated with this study [706]. *No safety concerns were identified in this study.*
Hashimoto, H. et al. 2013 [709]	Open-labeled, prospective study	35 subjects who underwent bilateral cataract surgery (intraocular lens implantation) (Mean *c.a* 71 yrs.)	6 mg/day	2 weeks	✓ Oxidative Stress ✓ Eye health ✓ (in aqueous humor of cataract patients)	Reduced total hydroperoxides (hydrogen peroxides, lipid peroxides, and peroxides of protein in aqueous humor; *p* < 0.05) increased superoxide scavenging activity (*p*< 0.05) Analysis associated with this study [706] *No safety concerns were identified in this study.*
Hashimoto H. et al. 2011 [706]	Open-labeled, prospective study	35 subjects who underwent bilateral cataract surgery (intraocular lens implantation) (Mean *c.a* 71 yrs.)	6 mg/day	2 weeks	✓ Eye health ✓ Oxidative Stress ✓ (in aqueous humor of cataract patients)	Reduced total hydroperoxides (hydrogen peroxides, lipid peroxides, and peroxides of protein in aqueous humor; *p* < 0.05). *No safety concerns were identified in this study.* [Article in Japanese]
Oxidative stress markers in “sports/musculoskeletal function “						
Kawamura A. et al. 2021 [710]	Randomized open-labeled, prospective study	26 healthy male subject (22.3 ± 0.3 yrs.)	N/A (1mg AX/100g salmon) *	10 weeks	✓ Oxidative Stress, ✓ (Exercise) ✓ Sports performance ✓ Musculoskeletal ✓ function	Serum carbonylated protein level as an oxidative stress marker tended to be lower immediately after exercise than before exercise in the intervention group only (*p* = 0.056). *No safety concerns were identified in this study.*
McAllister M.J. et al. 2021 [711]	Randomized, double-blind, placebo-controlled, crossover study	14 healthy young subjects, (23 ± 2 yrs.)	0, 6 mg/day	4 weeks	✓ Oxidative Stress, ✓ (Exercise) ✓ Sports performance	Glutathione was ∼7% higher following AX compared with placebo (*p* < 0.05). There was no effect on plasma hydrogen peroxide or MDA (*p* > 0.05). Advanced oxidation protein products (AOPP) were reduced by ∼28% (N.S.; *p* = 0.45). *No safety concerns were identified in this study.*
Baralic, I. et al. 2015 [712]	Randomized, double-blind, placebo-controlled, prospective study	40 healthy subjects (young soccer players)	0, 4 mg/day	90 days	✓ Oxidative Stress ✓ (Exercise) ✓ Immunity	Improved prooxidant-antioxidant balance (PAB; *p* < 0.05) *No safety concerns were identified in this study.*
Baralic I. et al. 2013 [713]	Randomized, double-blind, placebo-controlled prospective study	40 healthy subjects (soccer players)	0, 4 mg/day	90 days	✓ Oxidative Stress ✓ (Exercise)	Protected thiol groups against oxidative modification (increase in SH groups, *p* < 0.05; improved PON1 activity towards paraoxon and diazoxon, *p* < 0.05 and *p* < 0.01, respectively) *No safety concerns were identified in this study.*
Djordjevic B. et al. 2013 [714]	Randomized, double-blind, placebo-controlled prospective study	32 healthy subjects (soccer players)	0, 4 mg/day	90 days	✓ Oxidative Stress ✓ (Exercise)	Regular training significantly increased O_2_•¯ levels (main training effect, *p* < 0.01). TBARS and AOPP levels did not change throughout this study. Decreased post-exercise TAS levels only in the placebo group (*p* < 0.01). Increased total SH levels in both the AX and placebo groups (by 21% and 9%, respectively), and the effect of supplementation was marginally significant (*p* = 0.08). Decreased basal SOD activity in both the placebo and AX groups at the end of this study (main training effect, *p* < 0.01). *No safety concerns were identified in this study.*
Klinkenberg L.J. et al. 2013 [715]	Randomized, double-blind, placebo-controlled, prospective study	32 well-trained male cyclists (25 ± 5 years, V˙O_2_^peak^ = 60 ± 5 mL·kg^−1^·min^−1^, W_max_ = 5.4 ± 0.5 W·kg^−1^)	0, 20 mg/day *	4 weeks	✓ Oxidative Stress ✓ (Exercise) ✓ Sports performance	Not significant *(N.S.);* changes in markers of antioxidant capacity (trolox equivalent antioxidant capacity; TEAC, uric acid, and MDA). *No safety concerns were identified in this study.*
Res T. et al. 2013 [716]	Randomized, double-blind, placebo-controlled, prospective study	32 trained male cyclists or triathletes (25 ± 1 years, V˙O_2_^peak^ = 60 ± 1 mL·kg^−1^·min^−1^, W_max_ = 395 ± 7 W)	0, 20 mg/day	4 weeks	✓ Oxidative Stress ✓ (Exercise) ✓ Sports performance	*N.S.*; Plasma TAC (*p* = 0.90) or attenuated malondialdehyde levels (*p* = 0.63). Whole-body fat oxidation rates during submaximal exercise (from 0.71 +/− 0.04 to 0.68 ± 0.03 g.min and from 0.66 ± 0.04 to 0.61 ± 0.05 g.min in the Placebo and AX groups, respectively; *p* = 0.73), time trial performance (from 236 ± 9 to 239 ± 7 and from 238 ± 6 to 244 ± 6 W in the Placebo and AX groups, respectively; *p* = 0.63). *No safety concerns were identified in this study.*
Djordjevic B. et al. 2011 [714]	Randomized, double-blind, placebo-controlled, prospective study	32 male elite soccer players	0, 4 mg/day	90 days	✓ Oxidative Stress ✓ (Exercise) ✓ Sports performance	Changes in elevated O_2_•¯ concentrations after football exercise were statistically significant only in the placebo group (exercise × supplement effect, *p* < 0.05); TAS values decreased significantly after exercise only in the placebo group (*p* < 0.01). After the intervention, total SH content increased in the SH group (21% and 9%, respectively), and the effect of AX was marginally significant (*p* = 0.08). Basal SOD activity was significantly reduced in both the Placobo and AX groups at the end of this study (main effect of training, *p* < 0.01). .*No safety concerns were identified in this study.*
Bloomer, R.J. et al. 2005 [717]	Randomized, placebo-controlled, prospective study	20 resistance trained male subjects (25.1 ± 1.6 years)	0, 4 mg/day *	3 months	✓ Oxidative Stress ✓ (Exercise) ✓ Sports performance	*N.S.*; Muscle soreness, creatine kinase (CK), and muscle performance were measured before and through 96 h of eccentric exercise *No safety concerns were identified in this study.*
Oxidative stress markers in “geriatrics “						
Nakanishi R. et al. 2021 [718]	Randomized, double-blind, placebo-controlled prospective study	29 nursing home residents healthy elderly subjects (80.9 ± 1.5 yrs.)	0, 12 mg/twice a day * (0, 24 mg/day)	16 weeks	✓ Oxidative Stress, ✓ (Elderly subjects) ✓ Musculoskeletal ✓ function	Decrease in d-ROM values with the AX group (*p* < 0.01) but not with the placebo group; *No safety concerns were identified in this study.*
Petyaev I.M., et al. 2018 [719]	Randomized, blinded, four-arm, prospective study	32 subjects with oxidative stress, 8 subjects taking AX only, (60–70 yrs)	0, 7 mg/day * with DC	4 weeks	✓ Oxidative Stress ✓ (elderly subjects) ✓ Bioavailability, ✓ ADME	Reduced serum oxidized LDL by 55.4% after 4 weeks (*p* < 0.05). Reduced MDA by 52.7% after 4 weeks (*p* < 0.05). Increase in serum nitric oxide (NO) levels (*p* = 0.054). *No safety concerns were identified in this study.*
Kiko T. et al. 2012 [720]	Randomized, double-blind, placebo-controlled prospective study	30 healthy subjects (56.3 ± 1.0 yrs.)	0, 6, 12 mg/day	12 weeks	✓ Alzheimer’s disease ✓ (AD) ✓ Oxidative stress ✓ (PLOOH in erythrocytes)	Amyloid β (Aβ) 40 and Aβ42 concentrations were much higher in erythrocytes (RBC) than in plasma. RBC Aβ levels increased with aging. After AX supplementation, RBC Aβ concentrations decreased. The RBC Aβ levels were positively correlated with RBC PLOOH and inversely correlated with AX concentration in RBC. A study related to the ref. [721] *No safety concerns were identified in this study.*
Oxidative stress markers in “fatigues”						
Imai A. et al. 2018 [722]	Randomized, double-blind, placebo-controlled crossover study	42 healthy subjects	0, 6 mg/day *	4 weeks	✓ Oxidative stress ✓ (during mental and physical tasks) ✓ Fatigues ✓ Mood/Stress	Elevated PCOOH levels during mental and physical tasks were attenuated by AX supplementation. *No safety concerns were identified in this study.*
Hongo N. et al. 2017 [723]	Randomized, double-blind, placebo-controlled, prospective study	39 healthy subjects	0, 12 mg/day *	12 weeks	✓ Oxidative stress ✓ (during mental and physical tasks) ✓ Fatigues ✓ Mood/Stress	The rate of change in BAP values at week 12 was not significantly different between the AX group and the control group. *No safety concerns were identified in this study.* [Article in Japanese]
Oxidative stress markers in “other disorders and unhealthy condition”						
Ledda A. et al. 2017 [724]	Open-labeled, two-arm prospective study	59 patients with genitourinary cancers (prostate or bladder malignancies) who had undergone and completed cancer treatments (radiotherapy, chemotherapy or intravesical immunotherapy with increased oxidative stress and residual symptoms)	0, 8 mg/day *	6 weeks	✓ Improving QOL in ✓ cancer therapy ✓ Oxidative stress ✓ (Cancer therapy)	*Oncotris: containing 264 mg/day curcumin, 500 mg/day extract of cordyceps, and 8 mg/day AX (from EP217785227).* Oncotris supplementation reduced plasma d-ROM levels. *No safety concerns were identified in this study.*
Yagi H. et al. 2013 [725]	Case reports	34 OAB patients with anticholinergic agent-resistant (75.5 ± 8.0 years)	0, 12 mg/day *	8 weeks	✓ Oxidative Stress ✓ Overactive bladder (OAB)	Significantly improved international prostate symptom score (IPSS), QOL scores, benign prostatic hyperplasia impact index (BII) scores, and urinary 8-OHdG in patients AX could improve both urinary symptoms and QOL for anticholinergic agent-resistant OAB. *No safety concerns were identified in this study.* [Article in Japanese]
Kim, J.H. et al. 2011 [726]	Randomized, repeated measured, prospective study	39 heavy smokers, 39 non-smokers	0, 5, 20, or 40 mg/day	3 weeks	✓ Oxidative Stress ✓ (Smoking)	5 mg/day: MDA and ISP were significantly lower after 2 and 3 weeks compared to baseline in smokers (*p* < 0.05). SOD and TAC significantly increased after 1, 2, and 3 weeks compared to baseline in smokers (*p* < 0.05) 20 mg/day: MDA and ISP significantly were lower after 1, 2, and 3 weeks compared to baseline in smokers (*p* < 0.05). SOD and TAC significantly increased after 1, 2, and 3 weeks compared to baseline in smokers (*p* < 0.05). 40 mg/day: MDA and ISP were significantly lower after 1, 2, and 3 weeks compared to baseline in smokers (*p* < 0.05). SOD and TAC significantly increased after 2 and 3 weeks compared to baseline in smokers (*p* < 0.05). *No safety concerns were identified in this study.*
Yamada T. et al. 2010 [727]	Open-labeled, prospective study	6 healthy subjects and 6 Sjoegren’s syndrome (SS) subjects	12 mg/day	2 weeks	✓ Oxidative Stress ✓ Immunity	Reduced protein oxidation (−10%, *p* < 0.05) *No safety concerns were identified in this study.*
Oxidative stress markers in “healthy subjects”						
Chen JT, Kotani K. 2017 [728]	Randomized, double-blind, placebo-controlled, prospective study	29 healthy females	0, 12 mg/day	3 months	✓ Oxidative stress ✓ Liver protection ✓ Immunity	*N.S.*: Serum d-ROM levels, urinary 8-OHdG, and BAP following AX treatment. A Significant increase in blood leukocytes was also found in the AX-treated group. *No safety concerns were identified in this study.*
Balcerczyk A. et al. 2014 [729]	Randomized, double-blind, placebo-controlled prospective study	66 healthy females, (35–55 yrs.)	0, 15 mg/day *	12 weeks	✓ Oxidative Stress ✓ Antiaging	*Test supplement (NucleVital Q10): omega-3 acids (1350 mg/day)*, *ubiquinone (300 mg/day)*, *lycopene (45 mg/day), lutein palmitate (30 mg/day)*, *zeaxanthin palmitate (6 mg/day)*, *L-selenomethionine (330 mg/day)*, *cholecalciferol (30 µg/day)*, *α-tocopherol (45 mg/day)*, *and AX (15 mg/day).* Oxidative stress: significantly increased TAC of plasma and activity of erythrocyte SOD, with slight effects on oxidative stress biomarkers in erythrocytes: MDA and 4-hydroxyalkene levels. Antiaging effect: significant changes in mRNA expression of SIRT1 and 2 in PBMCs. *No safety concerns were identified in this study.*
Miyazawa T. et al. 2011 [730]	Randomized, double-blind, placebo-controlled, prospective study	30 middle- aged & senior subjects (mean: 50.6 yrs.)	0, 1, 3 mg/day	12 weeks	✓ Bioavailability, ✓ ADME	Erythrocyte AX concentrations There was no significant changes in erythrocyte phospholipid hydroperoxide concentration after astaxanthin intake in either the 1 mg/day or 3 mg/day groups. *No safety concerns were identified in this study.*
Nakagawa K. et al. 2011 [721]	Randomized, double-blind, placebo-controlled prospective study	30 healthy subjects	0, 6, 12 mg/day	12 weeks	✓ Oxidative Stress ✓ Bioavailability, ✓ ADME	6 mg/day: Reduction in total phospholipid hydroperoxides (PLOOH) at 12 weeks compared to baseline (*p* < 0.01) and compared to placebo (*p* < 0.05). Reduced phosphatidyl ethanolamine hydroperoxide (PEOOH) at 12 weeks compared to baseline (*p* < 0.05) and compared to placebo (*p* < 0.05). Increased plasma AX concentration at 12 weeks (86 nM) compared to baseline (*p* < 0.01, 6 to 9 nM) and compared to placebo (*p* < 0.01, 8 nM). 12 mg/day: 48% reduction in total PLOOH at 12 weeks compared to baseline (*p* < 0.01) and 35% less total PLOOH at 12 weeks compared to control (*p* < 0.05). The 12 mg/day group had 46% less phosphatidylcholine hydroperoxide (PCOOH) at 12 weeks compared to baseline (*p* < 0.01). *No safety concerns were identified in this study.*
Peng L. et al*.,* 2011 [731]	Randomized, placebo-controlled study	115 healthy subjects	0, 40 mg/day	90 days	✓ Oxidative Stress	Comparing with the control group, MDA contents in the test group decreased significantly (*p* < 0.01), and SOD and GSH-Px activities increased significantly (*p* < 0.01). *No safety concerns were identified in this study.*
Park J.S. et al. 2010 [732]	Randomized, double-blind, placebo- controlled, prospective study	42 healthy subjects	0, 2, 8 mg/day	8 weeks	✓ Oxidative Stress ✓ Immunity ✓ Bioavailability, ✓ ADME	2 mg/day: Concentrations of plasma 8-hydroxy-2′-deoxyguanosine reduced after 4 weeks and 8 weeks compared to placebo (*p* < 0.05). 8 mg/day: Concentrations of plasma 8-hydroxy-2′-deoxyguanosine reduced after 4 weeks and 8 weeks compared to placebo (*p* < 0.05). *No safety concerns were identified in this study.*
Karppi, J. et al. 2007 [733]	Randomized, double-blind, placebo-controlled, prospective study	39 healthy subjects	0, 8 mg/day	3 months	✓ Oxidative Stress ✓ Bioavailability, ✓ ADME	Decreased oxidation of fatty acids in healthy men *(p* < 0.05) *No safety concerns were identified in this study.*
Oxidative stress markers in “infertility”						
Jabarpour M. et al. 2023 [652]	Randomized, placebo-controlled prospective study	53 Patients with polycystic ovary syndrome (PCOS)	0, 6 mg/twice a day (0,12 mg/day)	60 days	✓ Women’s health ✓ Oxidative Stress ✓ (PCOS) ✓ ER stress ✓ Infertility ✓ Assisted reproductive ✓ technology ✓ (ART)	Antioxidant markers: Increased levels of total antixoidant capacity (TAC) in follicular fluid. ART outcomes: higher rates of high-quality oocytes, high-quality embryo, and oocyte maturity in the AX group (the oocyte number, fertilization rate, and fertility rate; N.S.). *No safety concerns were identified in this study.*
Rostami S. et al. 2023 [653]	Randomized, triple-blind, placebo-controlled prospective study	50 Patients of endometriosis (stage III/ IV)	0, 6 mg/day	12 weeks	✓ Women’s health ✓ Oxidative Stress ✓ (Endometriosis) ✓ Infertility ✓ ART	Antioxidant markers: Increased serum levels of TAC (*p* = 0.004) and superoxide dismutase (SOD, 13.458 ± 7.276 vs. 9.040 ± 5.155; *p* = 0.010) were observed in the AX intervention group after therapy. In addition, serum Malondialdehyde (MDA, *p* = 0.031) decreased significantly after AX treatment. *No safety concerns were identified in this study.*
Ghantabpour T. et al. 2022 [734]	N/A	The first phase; 10 semen samples from healthy men, the second phase; 25 semen samples from healthy men	0, 0.5, 1, 2 μM	N/A	✓ Infertility ✓ ART	Supplementation of sperm freezing medium with 1 µM AX was found to improve all parameters of sperm motility and viability (*p* ≤ 0.05). In addition, reduced levels of ROS parameters (intracellular hydrogen peroxide and superoxide) compared to the control group (*p* ≤ 0.05). AX also significantly reduced phosphatidylserine exogenous levels (*p* ≤ 0.05) and lipid peroxidation (*p* ≤ 0.05) after the freeze-thaw process. (*in vitro* study)
Gharaei R. et al. 2022 [651]	Randomized, double-blind, placebo-controlled prospective study	40 Patients with polycystic ovary syndrome (PCOS)	0, 8 mg/day	40 days	✓ Oxidative Stress, ✓ (PCOS) ✓ Infertility ✓ ART	AX supplementation resulted in significantly higher serum catalase and TAC levels in the AX group compared with the placebo group. However, there were no significant differences in serum MDA and SOD levels between groups. The expression of antioxidant genes such as Nrf2, HO-1, and NQ-1 was significantly increased in the granulosa cells (GC) of the AX group. *No safety concerns were identified in this study.*
Comhaire F.H. et al. 2005 [735]	Randomized, double-blind, placebo-controlled, prospective study	30 males with infertility of ≥ 12 months	0, 16 mg/day	3 months	✓ Infertility ✓ Oxidative stress	Significantly decreased ROS (chemiluminescence) in spermatozoa in the Astaxanthin group (*n* = 11), but not in the placebo group (*n* = 19). *No safety concerns were identified in this study.*
**(B) Skin Health**						
**Author/year/** **reference**	**Study Design**	**Subjects**	**Dose ^#,##^**	**Duration**	**Major Outcome ^†^**	**Description**
Evaluation of the effects on the skin under normal condition.						
Sudo A et al. 2020 [736]	Placebo-controlled prospective study	11 healthy subject (College floorball athletes for women in physical education	0.3 mg/day * in V7	30 days	✓ Fatigues ✓ Sports performance ✓ Eye health ✓ Skin health	*Test supplement (V7; astaxanthin, reduced coenzyme Q10, leucine, arginine, citrulline, DHA, Krill oil)* studied the efficacy of V7 on subjective fatigue, sports performance, and skin conditions in floorball athletes. Significant improvements in ‘firmness’ and ‘whiteness’, which are subjective measures of skin condition, were observed in the V7 supplementation group compared to pre-supplementation, while no significant changes were observed in the placebo group. *No safety concerns were identified in this study.* [Article in Japanese]
Sudo A et al. 2019 [737]	Placebo-controlled prospective study	19 healthy subject (College softball player for women in physical education	0.3 mg/day * in V7	30 days	✓ Fatigues ✓ Sports performance ✓ Eye health ✓ Skin health	*Test supplement (V7; astaxanthin, reduced coenzyme Q10, leucine, arginine, citrulline, DHA, Krill oil)* studied the efficacy of V7 on subjective fatigue, sports performance, and skin conditions in college softball players. Subjective symptoms were evaluated with the VAS. The change of VAS in PRE and POST in the V7 group showed statistically significant improved skin blemishes; a comparison of V7 and placebo POST showed statistically significant improved skin elasticity and whitening. The percent change between PRE and POST in V7 was statistically significantly higher in dull skin and total score. *No safety concerns were identified in this study.* [Article in Japanese]
Chalyk, N. et al. 2017 [702]	Open-label, prospective study	31 subjects; 18 obese, 8 overweight, 5 healthy, over the age of 40	4 mg/day	92 days	✓ Oxidative Stress ✓ (Middle age, obesity) ✓ Skin health	Morphological analysis of the residual skin surface components (RSSC; age-related changes in corneocyte desquamation, microbial presence, and lipid droplet size): decreased levels of corneocyte desquamation (*p* = 0.0075) and microbial presence (*p* = 0.0367); increase in lipid droplet size among obese (body mass index >30 kg/m^2^) subjects (*p* = 0.0214). *No safety concerns were identified in this study.*
Tominaga K. et al. 2017 [738]	Randomized, double-blind, placebo-controlled prospective study	65 healthy female (age, 35–60 years) with a wrinkle grade of 2.5 to 5.0	0, 6, 12 mg/day	16 weeks	✓ Skin health	Water content (cheeks): no significant difference from the placebo group regarding improvement. However, it was significantly worse in the placebo group over the period, but unchanged in the low and high AX-treated groups. TEWL (cheek): no significant change during the study. Wrinkle depth (eye rims): no significant difference. However, it worsened during the study period in the placebo group but did not change in the low and high AX-treated groups. Elasticity (cheeks): significantly improved in the two AX groups IL-1α: increased during the study but not in the high AX dose group (*p* < 0.05). *No safety concerns were identified in this study.*
Tsukahara H. et al*,* 2016 [739]	Randomized, double-blind, placebo-controlled prospective study	40 healthy subjects, those concerned about skin dullness or age-related skin deterioration.	0, 3 mg/day	8 weeks	✓ Skin health	TEWL (left cheek): significantly improved. Moisture content (left cheek): significant improvement. Melanin and color difference (left cheek): Melanin improved significantly. Elasticity (left cheek): Significant improvement in partial (R6). Facial image analysis (left part): “texture” significantly improved. *No safety concerns were identified in this study.* [Article in Japanese]
Phetcharat L. et al. 2015 [740]	Randomized, double-blind, placebo-controlled, prospective study	34 healthy subjects with wrinkles on the face (crow’s-feet) (35–65 yrs.)	4 mg/day	8 weeks	✓ Skin health	*The comparative control was rose hip powder. Therefore, the results of the evaluation before and after the intervention in the AX group are shown*. Moisture content (forehead): improved from the pre-treatment level at week 8 (*p* < 0.001) in the AX group. Elasticity (cheeks): improved from the pre-treatment level in the AX group at week 8 (*p* < 0.05). Crow’s-feet wrinkle depth (buttocks of the eyes): improved from pre-treatment levels at Weeks 4 and 8 in the AX group (*p* < 0.05). *No safety concerns were identified in this study.*
Yoon HS. et al. 2014 [703]	Randomized, double-blind, placebo-controlled prospective study	44 healthy females with wrinkles grade ≥ 2 (≥40 yrs.)	0, 2 mg/day *	12 weeks	✓ Skin health ✓ Oxidative Stress ✓ (UV irradiation)	*AX (2 mg/day)is combined with collagen hydrolysate (3 g/day)*. Skin condition of non-UV- irradiated skin: significantly improved TEWL (cheek) at week 12 (*p* = 0.045), tended to improve water content (cheek), tended to improve elasticity (cheek) at week 4, and significantly improved at week 12. *No safety concerns were identified in this study.*
Suganuma K. et al. 2012 [741]	Randomized, double-blind, placebo-controlled, prospective study	44 female subjects (Mean 37.26 yrs.)	0, 6 mg/day *	20 weeks	✓ Skin health	Water content (cheeks): increasing trend over the study period. The AX+VC+VE group showed an increase compared to the VC+VE (*p* < 0.10). Elasticity (upper cheekbones): no significant change. Fine wrinkles: significant improvement in the AX+VC+VE group (also improved compared to VC+VE group) *No safety concerns were identified in this study.*
Tominaga K. et al. 2012 [742] (Study 2)	Randomized, double-blind, placebo-controlled, prospective study	36 male subjects (20 to 60 yrs.)	0, 3 mg/twice a day * (0,6 mg/day)	6 weeks	✓ Skin health	TEWL: significantly decreased in AX (p < 0.01). Water content: increasing trend in the left cheek in the AX group (*p* = 0.08). Elasticity: significantly improved in the AX group (*p* < 0.05). Fine wrinkles: the total area ratio and volume of wrinkles in the AX group decreased (*p* < 0.05). Sebum production: decreased in the AX group (*p* = 0.085). *No safety concerns were identified in this study.*
Evaluation of protective effect against UV irradiation						
Ito N. et al. 2018 [743]	Randomized, double-blind, placebo-controlled, prospective study	22 healthy subjects with skin phototype was type II or III (30–56 yrs.)	0, 6 mg/day	10 weeks	✓ Skin health ✓ Oxidative stress ✓ (UV irradiation)	Subjective skin condition was assessed on a visual analog scale; the AX group showed an increase in minimum erythema dose (MED) compared to placebo. The AX group had a reduced loss of skin moisture in the irradiated area compared with the placebo. Subjective skin conditions for “improvement of rough skin” and “texture” in non-irradiated areas were significantly improved by AX. *No safety concerns were identified in this study.*
Carrascosa JM et al. 2017 [744]	Randomized, double-blind, placebo-controlled, prospective study	31 healthy subjects with skin phototypes II and III	4 mg/day *	56 days	✓ Skin health	*Intervention: Genosun oral a combination of AX (4 mg), β-carotene(4.8 mg), vitamin E (6 mg), vitamin C(40 mg), lutein (2.4 mg), and lycopene (2.4 mg).* MED at Days 1, 29, and 57 was evaluated.The Intervention group showed a significant increase over placebo in the tolerance to an erythemal dose of UVR. Increased UVR tolerance was reflected in an increase in MED of 12.4% and 20.51% over baseline after 29 and 57 days, respectively, with a significant difference between treatment and control groups at the end of the study. *No safety concerns were identified in this study.*
Yoon HS. et al. 2014 [703]	Randomized, double-blind, placebo-controlled prospective study	44 healthy females with wrinkles grade ≥ 2 (≥40 yrs.)	0, 2 mg/day *	12 weeks	✓ Skin health ✓ Oxidative Stress ✓ (UV irradiation)	*AX (2 mg/day) combined with collagen hydrolysate (3 g/day)*. Skin biopsy after UV irradiation: no difference in oxidative markers (Thymine dimers, 8-OHdG) between the two groups in histological evaluation. Regarding mRNA expression, significantly upregulated expression of procollagen type I tended to upregulate fibrillin-1, while significantly downregulated MMP1 and MMP12 in the AX group compared to placebo. *No safety concerns were identified in this study.*
Satoh A. et al. 2011 [745]	Randomized, single-blind, placebo-controlled prospective study	26 healthy subjects	0, 3 mg/day	4 + 4 weeks	✓ Skin health	After 4 weeks of administration, UV light was irradiated (2 MED), and the given test substance was administered for another 4 weeks. Skin color was evaluated with a colorimeter, a mexameter, and a skin color scale before administration, before UV irradiation, and 1, 7, 14, 21, and 28 days after UV irradiation. The results showed that the L value of the colorimeter and skin color scale scores were significantly higher in the Ax group than in the placebo group, and the amount of melanin after UV irradiation was significantly lower in the Ax group than in the placebo group. *No safety concerns were identified in this study.* [Article in Japanese]
Yamashita E. 2006 [746]	Randomized, single-blind, placebo-controlled, prospective study	49 female subjects (Mean 47 yrs.)	0, 2 mg/twice a day (0, 4 mg/day) *	6 weeks	✓ Skin health	Water content (left cheek): significant improvement in the 6-week treatment group (compared to the start of treatment). Elasticity (left eye rim): significant improvement in the placebo group at weeks 3 and 6 (*p* < 0.05). Inspection/Palpation by a dermatologist: improvement in fine lines, wrinkles, and elasticity (week 6). Skin surface observation: improvement in fine lines, wrinkles, and elasticity (week 6). *No safety concerns were identified in this study.*
Yamashita E. 2002 [747]	Randomized, double-blind, placebo-controlled, prospective study	16 healthy female subjects with dry skin	0, 2 mg/day *	4 weeks	✓ Skin health	Moisture content: trend of improvement in the AX group compared to placebo at week 2 on the left eye corner and at week 4 on the cheeks, with significant improvement at week 4 on the eye corner (*p* < 0.05). Wrinkle depth (eye corner): no significant difference. Subjective symptoms (questionnaire): subjective improvement in spots/freckles (week 2), acne/wipes (week 4). Inspection/Palpation by a dermatologist: improvement in smoothness, moistness, and firmness. *No safety concerns were identified in this study.* [Article in Japanese]
Evaluation of efficacy in skin diseases.						
Satoh A. et al. 2009 [704]	Randomized, single-blind, placebo-controlled prospective study	27 patients with atopic dermatitis	0, 12 mg/day	4 weeks.	✓ Oxidative stress ✓ Skin health ✓ Atopic dermatitis ✓ Immunity	Severity (SCORAD), pruritus (VAS), quality of life (Skindex-16, STAI), immune function (Th1/Th2, blood catecholamines), and antioxidant status (urinary 8-OHdG, isoprostanes) were assessed. There were significant differences in the degree of itching between the Ax and placebo groups. However, there was significant improvement in Skindex-16 symptoms and STAI status anxiety in the Ax group. In addition, the Th1/Th2 balance shifted significantly toward Th1. *No safety concerns were identified in this study* [Article in Japanese]
**(C) Eye health**						
**Author/year/** **reference**	**Study design**	**Subjects**	**Dose ^#,##^**	**Duration**	**Major Outcome ^†^**	**Description**
Evaluation of efficacy in “asthenopia (eyestrain)”.						
Sekikawa T. et al. 2023 [748]	Randomized, double-blind, placebo-controlled prospective study	59 healthy subjects with VDT operation (Mean 39 yrs.)	0, 9 mg/day	6 weeks	✓ Eye health ✓ (Eyestrain)	Visual acuity: In participants ≥40 yrs, AX had a higher protective effect of on corrected visual acuity of the dominant eye after visual display terminal (VDT) work at 6 weeks after intake in the AX group vs the control group (*p* < 0.05). In <40 yrs, no significant difference between the AX and control groups. Functional visual acuity and pupil constriction rate: No significant difference between the AX and control groups *No safety concerns were identified in this study.*
Kizawa Y. et al. 2021 [749]	Randomized double-blind, placebo-controlled, prospective study	40 healthy subjects with VDT operation	0, 6 mg/day *	6 weeks	✓ Eye health ✓ (Eyestrain)	*Intervention (active group): 72 mg anthocyanin from blueberry (bilberry) extract, 10 mg lutein and 6 mg AX.* After 6 weeks, there was a significant improvement in the active group compared to the placebo group in the average percentage of pupillary response in both eyes and in the dominant eye before and after operating the visual display terminal. In addition, the scores for “A sensation of trouble in focusing the eyes” and “Difficulty in seeing objects in one’s hand and nearby, or fine print” were significantly improved in the active group compared to the placebo group before and after ingestion. No statistically significant improvements were observed in tear degradation time, visual acuity, Schirmer test value, macular pigment optical density level, or muscle hardness. *No safety concerns were identified in this study.*
Sudo A et al. 2019 [750]	Open-labeled, prospective study	19 healthy females (Mean 47.3 yrs.)	0, 0.3 mg/day * in V7	30 days	✓ Fatigues ✓ Skin health ✓ (Eye health) ✓ (Cognitive function)	*Test supplement (V7; astaxanthin, reduced coenzyme Q10, leucine, arginine, citrulline, DHA, Krill oil)* studied the efficacy of V7 on subjective fatigue and skin conditions in typical middle-aged females. Subjective symptoms were evaluated with the VAS. The change in VAS in PRE and POST in the V7 group showed no statistically significant improved in eye strain. *No safety concerns were identified in this study.* [Article in Japanese]
Sudo A et al. 2019 [737]	Placebo-controlled prospective study	19 healthy subject (College softball player for women in physical education	0.3 mg/day * in V7	30 days	✓ Fatigues ✓ Sports performance ✓ Eye health ✓ Skin health	*Test supplement (V7; astaxanthin, reduced coenzyme Q10, leucine, arginine, citrulline, DHA, Krill oil)* studied the efficacy of V7 on subjective fatigue, sports performance, and skin conditions in college softball players. Subjective symptoms were evaluated with the VAS. A comparison of V7 and placebo POST showed statistically significant increases in eye strain. The percent change between PRE and POST in V7 was statistically significantly higher in eye strain. *No safety concerns were identified in this study.* [Article in Japanese]
Kono K. et al. 2014 [751]	Randomized, double-blind, placebo-controlled prospective study	48 healthy subjects who complained of eye strain	0, 4 mg/day *	4 weeks	✓ Eye health ✓ Fatigue ✓ ((Shoulder stiffness)	*Test supplement (Enkin; lutein (10mg), 20 mg of bilberry extract, and 26.5 mg of black soybean hull extract (a total of 2.3 mg of cyanidin-3-glucoside in both extracts), DHA (50mg), and AX (4mg).* The variation of the “near-point accommodation” of both eyes from baseline to 4 weeks after-intervention in the test supplement group (TS) was significantly higher than in the placebo (P) group (1.321 ± 0.394 diopter (D) in the TS group and 0.108 ± 0.336 D in the P group, *p* = 0.023, respectively). Regarding subjective symptoms, there was a significant improved on “stiff shoulders or neck” and “blurred vision” in the TS group compared to the P group (*p* < 0.05). *No safety concerns were identified in this study.*
Nagaki Y. et al. 2010 [46]	Randomized, single-blind, placebo-controlled prospective study	82 healthy subjects with VDT operation (6 h or more per day for more than 1 year) and frequently experienced eyestrain	0, 9 mg/day	4 weeks	✓ Eye health ✓ (Eyestrain)	(1) The post-treatment accommodation ability of the AX group with respect to value and rate of change was significantly higher than that of the control group. (2) The distribution of rate of change also showed significant improvement in post-treatment accommodation ability of the AX group when compared to control group. (3) Subjective questionnaire regarding 4 conditions (“eyestrain,” “hazy vision,” “flickering images,” and “my shoulders/back feel stiff”) showed that the AX group significantly improved compared to the of control group. *No safety concerns were identified in this study.* [Article in Japanese]
Kajita M et al. 2009 [47]	Open-labeled, prospective study	82 healthy males with presbyopia (Mean 53.9 yrs.)	6 mg/day	4 weeks	✓ Eye health ✓ (Eyestrain)	The pupillary constriction ratio before and after AX supplementation was measured by TriIRIS C9000. The change in subjective symptoms after supplementation was examined by a questionnaire. The results showed a significant increase in pupillary constriction ratio after supplementation with AX, therefore suggesting that astaxanthin may also improve the accommodation function of the eye and some subjective symptoms related to presbyopia in middle-aged and older people with complaints of eye strain. *No safety concerns were identified in this study.* [Article in Japanese]
Seya Y. et al. 2009 [752]	Open-labeled, prospective study	10 healthy subjects with VDT operation (Mean Age 24.6 yrs., VDT 6.9 h/day)	6 mg/day	4 weeks.	✓ Eye health ✓ (Eyestrain)	The effects of visual fatigue on reaction times measured in a visual pursuit task. Regardless of the duration of the intake period for AX, the reaction times at the early trials/blocks of the reaction time task were shorter than those at the late trials/blocks during a long-lasting, 500-trial experimental session. In addition, the reaction times at the late stages (14th and 28th days) of the AX intake were shorter than those at the early stage (1st day). *No safety concerns were identified in this study* [Article in Japanese]
Tsukahara H. et al. 2008 [753]	Open-labeled, prospective study	13 healthy subjects with shoulder Stiffness	6 mg/days *	4 weeks	✓ Fatigue ✓ ((Shoulder stiffness) ✓ Cardiovascular health ✓ EyeHealth ✓ Microcirculatory flow	*6mg of AX and 50mg of flaxseed lignin.* All patients completed the efficacy evaluation, which confirmed significant improvement in physical symptoms such as shoulder stiffness, physical fatigue, mental irritability, cold hands and feet, eye fatigue, and redness of the eyes. At the end of the treatment, laser Doppler graphics also confirmed a significant increase in blood flow in the shoulders. *No safety concerns were identified in this study.* [Article in Japanese]
Iwasaki T. et al. 2006 [44]	Randomized, double-blind, placebo-controlled crossover study	39 healthy females (Mea 20.5 yrs.)	0, 6 mg/day	2 weeks	✓ Eye health ✓ (Eyestrain)	Accommodative function and subjective symptoms relating to eyestrain were measured before and after the task and after the 10-min rest following the task. The data were then compared between the AX and Placebo groups by the double-blind cross-over method. After the task, accommodation contraction and relaxation times were extended in both the AX and Placebo groups. Comparison between the two groups showed that after the task, accommodation relaxation time was significantly extended in the Placebo group, in contrast to AX. Accommodative contraction and relaxation times were significantly prolonged after the 10-min rest in the P group as compared to AX. The symptoms of eye fatigue, eye heaviness, blurred vision, and eye dryness in the Placebo group increased; however, the AX group only showed increases in eye fatigue and eye heaviness. *No safety concerns were identified in this study.* [Article in Japanese]
Nagaki Y. et al. 2006 [45]	Randomized, double-blind, placebo-controlled, prospective study	48 healthy subjects with VDT operation (6 h or more per day for more than 1 year) and frequently experienced eyestrain	0, 6 mg/day	4 weeks	✓ Eye health ✓ (Eyestrain)	1. Significantly improved the magnitude of change in amplitude of accommodation before and after supplementation in the AX supplemented group compared with the control group. 2. Significantly better scores of the distribution of the percentage change in amplitude of accommodation after supplementation in the AX supplemented group compared with the control group. 3. In the subjective asthenopia evaluation, the AX supplemented significantly improved for the two items “dimness of sight” and “stiff shoulders and back” compared with the control group, and an improvement tendency was seen in “heavy head.” *No safety concerns were identified in this study.* [Article in Japanese]
Nitta T. et al. 2005 [41]	Randomized, placebo-controlled, prospective study	30 health subjects (Mean time consumed for close work (e.g., VDT work) was approx. 7 h./day)	0, 6, 12 mg/day	4 weeks	✓ Eye health ✓ (Eyestrain)	l. Significantly increased the objective accommodation power of the AX 12 mg group compared to that of pre-dosing. 2. Significantly shortened was the positive accommodation time in the AX 6 mg and the 12 mg groups compared to those of pre-dosing, and the negative accommodation time was significantly shortened in the AX placebo and the 6 mg groups compared to those of pre-dosing. 3. VAS; many parameters of subjective symptoms were improved in the AX 6 mg group. *No safety concerns were identified in this study.* [Article in Japanese]
Shiratori K. et al. 2005 [42]	Randomized, placebo- controlled, prospective study	39 healthy subjects who complained of eyestrain	0, 6 mg/day	4 weeks	✓ Eye health ✓ (Eyestrain)	1. Significantly higher sub-objective accommodation power (changing rate) in the AX group than that of the control group. 2. Significantly higher rate of positive and negative accommodation times (rate of change) in the AX group compared to those of the control group. 3. In the AX group, subjective degree of asthenopia (eye strain) measured by VAS showed significant improvement in two parameters, i.e., “My eyes get bleary” and “I get irritated easily”, compared to the control group. *No safety concerns were identified in this study.* [Article in Japanese]
Takahashi N. et al. 2005 [43]	Open-labeled, prospective study	10 healthy subjects	6 mg/day	2 weeks	✓ Eye health ✓ (Eyestrain)	Effects of astaxanthin on accommodative recovery derived from a rest after VDT work were studied. Evaluated (9 dominant eyes) by values for objective diopter, HFC (High Frequency Component in Accommodative micro-fluctuation), and accommodative reaction. Increased HFC after rest was significantly restrained by AX supplementation compared to the increase shortly after working. *No safety concerns were identified in this study.* [Article in Japanese]
Nakamura A. et al. 2004 [40]	Randomized, placebo- controlled, prospective study	49 healthy subjects	0, 2, 4, 12 mg/day	4 weeks	✓ Eye health ✓ (Eyestrain)	For far visual acuity (5 m), there was no significant difference in the results for uncorrected visual acuity between the 0 mg group and the 2 mg group before and after the start of peroral administration; however, uncorrected visual acuity improved significantly for the 4 mg group and 12 mg group (*p* < 0.05). For corrected visual acuity, no significant difference was found for any of the groups. No significant changes were found in refraction or flicker fusion frequency. For the accommodation test, positive accommodation time was shortened significantly for the 4mg group and the 12 mg group (*p* < 0.05). No significant change was found in the other items. For pupillary reflex, no significant difference was found in the miosis ratio (%), T1 (ms), T2 (ms), or VC (mm^2^/s). *No safety concerns were identified in this study.* [Article in Japanese]
Nagaki Y. et al. 2002 [39]	Randomized, placebo- controlled, prospective study	26 VDT subjects + 13 non-VDT subjects (control)	0, 5 mg/day	4 weeks	✓ Eye health ✓ (Eyestrain)	Group A: 13 non-VDT workers/no supplementation. Group B: 13 VDT workers with AX, 5 mg/day, for 4 weeks, Group C: 13 VDT workers with placebo, 5 mg/day, for 4 weeks. Accommodation amplitudes in Groups B and C before supplementation were significantly (*p* < 0.05) lower than in Group A. After AX supplementation, the accommodation amplitude in Group B was significantly (*p* < 0.01) larger than before supplementation, while the accommodation amplitude in Group C after placebo supplementation was unchanged. The CFFs in Groups B and C before supplementation were significantly (*p* < 0.05) lower than in Group A. The CCFs in Groups B and C did not change after supplementation. Amplitudes and latencies of P100 in PVEP in Groups B and C before supplementation were similar to those in Group A and did not change after supplementation. *No safety concerns were identified in this study.*
Evaluation of benefits in cataract surgery						
Hashimoto H. et al. 2021 [705]	Open-labeled, prospective study	35 subjects who underwent bilateral cataract surgery (intraocular lens implantation) (Mean *c.a* 71 yrs.)	6 mg/day	2 weeks	✓ Eye health ✓ Oxidative stress ✓ ((Cataract surgery)	We analyzed the antioxidant effect of AX in relationship to age. None of the parameters were correlated with age before AX intake, but only total hydroperoxide values were significantly correlated after AX intake (*r* = 0.4, *p* < 0.05). Total hydroperoxide levels were similar in younger and older age groups ( < 70 vs. ≥70 years) before AX intake, but significantly decreased in younger age groups after intake (−0.21 ± 0.18 vs. −0.05 ± 0.31, *p* < 0.05), resulting in a significant difference (*p* < 0.05). Thus, the previously observed decrease in mean total hydroperoxide levels after AX intake was likely due to a greater response in the younger age group. Analysis associated with the study [706]. *No safety concerns were identified in this study.*
Hashimoto H. et al. 2019 [707]	Open-labeled, prospective study	35 subjects who underwent bilateral cataract surgery (intraocular lens implantation) (Mean *c.a* 71 yrs.)	6 mg/day	2 weeks	✓ Eye health ✓ Oxidative stress ✓ ((Cataract surgery)	In this analysis, the effect of AX intake on the relationship between VEGF levels and ROS-related parameters before and after bilateral cataract surgery was analyzed by gender. VEGF, hydrogen peroxide, and total hydroperoxide levels in the aqueous humor, as well as O2 scavenging activity, were measured. For women only, VEGF levels and O2 scavenging activity before AX intake were negatively correlated (*r* = −0.6, *p* < 0.01) and positively correlated with total hydrogen peroxide levels before and after AX intake (*r* = 0.7, 0.8, *p* < 0.01, respectively). Analysis associated with this study [706] *No safety concerns were identified in this study.*
Hashimoto H. et al. 2016 [708]	Open-labeled, prospective study	35 subjects who underwent bilateral cataract surgery (intraocular lens implantation) (Mean *c.a* 71 yrs.)	6 mg/day	2 weeks	✓ Oxidative Stress ✓ (during surgery) ✓ Eye health	Superoxide anion scavenging activity (U/mL): 18.2 ± 4.1 at 0 weeks reduced to 19.9 ± 3.6 after 2 weeks of supplementation compared to baseline, *p* < 0.05. Total hydroperoxides (U CARR) from 1.16 ± 0.18 at 0 weeks were reduced to 1.04 ± 0.31 after 2 weeks of supplementation compared to baseline, *p* < 0.05. Analysis associated with this study [706]. *No safety concerns were identified in this study.*
Hashimoto, H. et al. 2013 [709]	Open-labeled, prospective study	35 subjects who underwent bilateral cataract surgery (intraocular lens implantation) (Mean *c.a* 71 yrs.)	6 mg/day	2 weeks	✓ Oxidative Stress ✓ Eye health ✓ (in aqueous humor of cataract patients)	Reduced total hydroperoxides (hydrogen peroxides, lipid peroxides, and peroxides of protein in aqueous humor; *p* < 0.05) increased superoxide scavenging activity (*p* < 0.05). Analysis associated with this study [706] *No safety concerns were identified in this study.*
Hashimoto H. et al. 2011 [706]	Open-labeled, prospective study	35 subjects who underwent bilateral cataract surgery (intraocular lens implantation) (Mean *c.a* 71 yrs.)	6 mg/day	2 weeks	✓ Eye health ✓ Oxidative Stress ✓ (in aqueous humor of cataract patients)	Reduced total hydroperoxides (hydrogen peroxides, lipid peroxides, and peroxides of protein in aqueous humor; *p* < 0.05) *No safety concerns were identified in this study.* [Article in Japanese]
Evaluation of efficacy in other ophthalmological categories.						
Yoshida K. et al. 2023 [754]	Randomized, double-blind, placebo-controlled prospective study	57 healthy subjects	0, 6 mg/day	8 weeks	✓ Eye health ✓ (Eye–hand coordination)	*Active group: 10 mg lutein, 2 mg zeaxanthin, and 6 mg of AX.* Significantly improved eye–hand coordination after visual display terminal (VDT) operation at 8 weeks in the active group. No clear improvement in the effect of the supplementation on smooth-pursuit eye movements. The active group also showed a significant increase in macular pigment optical density (MPOD) levels. *No safety concerns were identified in this study.*
D’Aloisio R. et al., 2022 [755]	Retrospective study	15 AMD patients treated with daily oral nutritional supplement with AX. 13 AMD patients treated w/o. daily oral nutritional supplement (control)	0, 12 mg/day *	6 months	✓ Eye health (AMD) ✓ (Microcirculatory flow)	*Nutritional supplement: 100 mg lutein, 80 mg bromelain, 120 mg VC, 30 mg VE, 400 μg folic acid, Zn, Cu, 1000 IU D3 vitamin, and 12 mg AX (Astazin 10).* There was a statistically significant difference in choriocapillary vessel density (CCVD) values between cases and controls at baseline (*p* < 0.001) and at follow-up (*p* < 0.001); choroidal thickness measurements were statistically significant between cases and controls (*p* = 0.002) and in cases at follow-up (*p* < 0.001). *No safety concerns were identified in this study.*
Tian L. et al. 2022 [756]	Open-labeled, prospective study	60 middle-aged and elderly patients with mild-to-moderate dry eye disease (DED)	6 mg/twice a day (12mg/day)	30 days	✓ Eye health ✓ (DED)	Significantly improved (*p* < 0.05) to varying degrees after treatment compared to pre-treatment for the ocular surface disease index (OSDI) score, non-invasive tear break-up time (NIBUT), fluorescein break-up time (BUT), corneal fluorescein staining (CFS) score, eyelid margin signs, MG expressivity, mibum quality, and blink frequency, but no differences were found for tear meniscus height, Schirmer I test, conjunctival hyperemia, tear fluid lipid layer thickness, meibum quality, meibomian gland dropout (MGDR), incomplete blink rate, Visual acuity (VA), intraocular pressure (IOP). *No safety concerns were identified in this study.*
Huang J.Y. et al. 2016 [757]	Randomized, double-blind, placebo-controlled prospective study	43 patients with dry eye disease (DED)	0, 2 mg/day *	16 weeks	✓ Eye health ✓ (DED)	*Supplement: Commercially available antioxidant supplements contain anthocyanosides, vitamins A, C, and E, and crudely extracted additives from several Chinese herbal extracts and AX.* Lower diastolic blood pressure in the treated group. There were no statistically significant differences in systolic blood pressure, dry eye symptoms, serum anti-SSA and anti-SSB, visual acuity, intraocular pressure, or fluorescein corneal staining between the two groups. Significantly improved tear film break time scores and Schirmer test without local anesthesia in the treatment group. Tear ROS levels differed between groups and decreased after treatment. The overall subjective impression was significantly improved by treatment compared to placebo. *No safety concerns were identified in this study.*
Piermarocchi S. et al. 2012 [758]	Randomized, two-arm, prospective study	145 patients with nonexudative (dry) age-related macular degeneration (AMD) (72.5 ± 7 yrs.)	0, 4 mg/days *	24 months	✓ Eye health (AMD)	Two-year results of the CARMIS study: *Interventions were vitamin C (180 mg), vitamin E (30 mg), zinc (22.5 mg), copper (1 mg), lutein (10 mg), zeaxanthin (1 mg), and AX (4 mg)*. The treated group showed stabilization of visual acuity (VA), with significantly (*p* = 0.003) better VA scores compared to the non-treated group at 24-month follow-up. An improvement in contrast sensitivity (CS, *p* = 0.001) and final mean National Eye Institute visual function questionnaire (NEI VFQ-25) composite scores at 12 and 24 months were higher in the treated group compared to the non-treated group (*p* < 0.001). *No safety concerns were identified in this study.*
Saito M. et al. 2012 [685]	Randomized, double-blind, placebo-controlled, prospective study	20 healthy subjects	0, 12 mg/day	4 weeks	✓ Eye Health ✓ Cardiovascular health ✓ ((Microcirculatory flow)	Significant increase in the macular square blur rate (SBR) after 4 weeks after AX (*p* = 0.018). No statistical difference in the macular SBR was detected in the placebo group (*p* = 0.598). *No safety concerns were identified in this study.*
Parisi V.. et al. 2008 [759]	Randomized, two-arm, prospective study	27 patients with nonadvanced AMD and visual acuity ≥0.2 logarithm of the minimum angle of resolution (69.4 ± 4.3 yrs.)	0, 4 mg/days *	12 months	✓ Eye health (AMD)	one-year results of the CARMIS study; *Interventions were vitamin C (180 mg), vitamin E (30 mg), zinc (22.5 mg), copper (1 mg), lutein (10 mg), zeaxanthin (1 mg), and AX (4 mg)*. In nonadvanced AMD eyes, selective dysfunction of the central retina (0°−5°) was ameliorated by carotenoid and antioxidant supplementation. There were no functional changes in the more peripheral (5°−20°) retina. *No safety concerns were identified in this study.*
Nagaki Y. et al. 2005 [684]	Randomized, placebo-controlled, prospective study	36 health subjects (*c.a.* 41 yrs.)	0, 6 mg/day	4 weeks	✓ Eye Health ✓ ADME ✓ Cardiovascular health ✓ ((Microcirculatory flow)	After 4 weeks of supplementation, retinal capillary blood flow in the AX group was significantly (p < 0.01) higher than before supplementation in both eyes, while retinal capillary blood flow in the placebo group after placebo treatment was unchanged. Intraocular pressures in both groups remained unchanged during the supplementation period. *No safety concerns were identified in this study.* [Article in Japanese]
Sawaki et al. 2002 [760]	Randomized, double-blind, placebo-controlled, prospective study	18 healthy male Subjects (College handball player)	0, 6 mg/day	4 weeks	✓ Sports performance ✓ (Muscle fatigue) ✓ Eye health ✓ (sports vision)	No changes in static and dynamic visual acuity measurements were observed before and after administration of AX. Regarding the deep vision measurements after administration of AX, the AX group showed better values compared to the placebo group. Flicker values after AX administration showed that visual acuity was significantly more acute in the AX group than in the placebo group. *No safety concerns were identified in this study.* [Article in Japanese]
**(D) Cardiovascular Health: Dyslipidemia, Glucose/Lipid Metabolisms, Chronic Inflammation and Type 2 Diabetes**						
**Author/year/** **reference**	**Study Design**	**Subjects**	**Dose ^#,##^**	**Duration**	**Major Outcome ^†^**	**Description**
Evaluation of efficacy in obesity, dyslipidemia, hypertension, glucose intolerance and T2DM.						
Ciaraldi T.P. et al. 2023 [761]	Randomized, double-blind, placebo-controlled prospective study	34 Obese subjects with prediabetes and dyslipidaemia	0, 12 mg/day	24 weeks	✓ Cardiovascular health ✓ Metabolic syndrome ✓ Oxidative stress	After 24 weeks, there was a significant decrease in low-density lipoprotein (−0.33 ± 0.11 mM) and total cholesterol (Chol, −0.30 ± 0.14 mM) (both *p* < 0.05) in the AX group. Reduced levels of the cardiovascular disease (CVD) risk markers fibrinogen (−473 ± 210 ng/mL), L-selectin (−0.08 ± 0.03 ng/mL), and fetuin-A (−10.3 ± 3.6 ng/mL) (all *p* < 0.05) in the AX group. The trend of improvement in insulin action was also observed in insulin-stimulated whole-body glucose treatment (+0.52 ± 0.37 mg/m^2^/min, *p* = 0.078), as well as in fasting [insulin] (−5.6 ± 8.4 pM, *p* = 0.097) and HOMA2-IR (−0.31 ± 0.16, *p* = 0.060). No consistent, significant differences from baseline were observed in any of these results in the placebo group. *No safety concerns were identified in this study.*
Saeidi A. et al. 2023 [762]	Randomized placebo-controlled prospective study	Obese subjects; 15 control group (CG), 15 supplement group (SG), 15 training group (TG), 15 training plus supplement group (TSG). BMI: 33.6 ± 1.4	0, 20 mg/day	12 weeks	✓ Cardiovascular health ✓ Metabolic syndrome ✓ Energy metabolisms	Intervention: AX with/without high-intensity functional training. BW,% of Fat, BMI, Fat-Free Mass: After 12 weeks, there was significant improvement in SG, TG, and TSG, but not in CG (*p* < 0.05). VO_2 peak_: Increases after 12 weeks of exercise were significant in the TG (*p* = 0.0001) and TSG (*p* =0.0001) but not in the CG (*p* = 0.32) and SG (*p* = 0.21). Lipid profile (HDL, LDL, Total Cholesterol(Chol.), and Triglycerides(TG)): After 12 weeks, there was significant improvement in SG, TG, and TSG, but not in CG (*p* < 0.05). Metabolic Factors (Insulin, Glucose, HOMA-IR): Glucose and Insulin levels decreased significantly in the SG (*p* <0.001), TG (*p* < 0.001), and TSG (*p* <0.001), but not significantly in the CG (*p* > 0.05), and decreased HOMA-IR following 12 weeks of training in the SG (*p* = 0.0001), TG (*p* = 0.0001), and TSG (*p* = 0.0001), while the difference in HOMA-IR in the CG was not significant (*p* = 0.17). Adipokines and Growth Differentiation Factors: [Cq1/TNF-related protein 9 and 2 (CTRP9 and CTRP2) levels, and growth differentiation factors 8 and 15 (GDF8 and GDF15)] were measured. There were significant differences in all indicators between the groups (*p* < 0.05). *No safety concerns were identified in this study.*
Wika A.A. et al. 2023 [763]	Randomized, double-blind, placebo-controlled prospective study	19 obese subjects (Mean, age: 27.5 yrs; BF%: 37.9; BMI: 33.4 kg/m^2^; VO_2peak_: 25.9 ml·kg−1·min^−1^)	0, 12 mg/day	4 weeks	✓ Cardiovascular health ✓ Metabolic syndrome ✓ Energy metabolisms	Subjects performed a graded exercise test on a cycling ergometer and were measured for changes in glucose and lactate levels, fat and carbohydrate (CHO) oxidation rates, heart rate, and rating of perceived exertion (RPE). Although there were no changes in fat oxidation rate, blood lactate or glucose, or RPE (all *p* > 0.05), a significant decrease in CHO oxidation rate was observed in the AX group only, before and after supplementation. In addition, in the AX group, heart rate decreased by 7% during the graded exercise stress test. *No safety concerns were identified in this study.*
Rad NR. et al. 2022 [691]	Randomized, double-blind, placebo-controlled prospective study	50 Type 2 Diabetes Mellitus (T2DM) patients receiving metformin	0, 10 mg/day	12 weeks	✓ Cardiovascular health ✓ Metabolic syndrome ✓ Oxidative stress ✓ (T2DM)	*Investigation of additive synergistic effects on metformin (1000–2000 mg/day).* T2DM: After the intervention, while FBS, HbA1c (probably maintained in the high-normal range by metformin even at baseline), and systolic blood pressure (in the normal range) tended to decrease in both groups, blood lipids (within normal range) remained unchanged. Significantly reduced FBS in the AX group rather than the placebo group. Oxidative stress: Significantly increased blood TAC levels at the end of the intervention only in the AX-treated group, while MDA remained unchanged. Similarly, increased SOD and catalase activity in blood and increased Nrf2 protein in PBMCs were observed at the end of the intervention only in the AX-treated group. *No safety concerns were identified in this study.*
Shokri-Mashhadi, N. et al. 2021 [693]	Randomized, double-blind, placebo-controlled prospective study	44 patients with T2DM	0, 8 mg/day	8 weeks	✓ Oxidative Stress, ✓ (Type 2 diabetes) ✓ Type 2 diabetes	Decrease plasma levels of MDA and IL-6 (*p* < 0.05) and decrease the expression level of miR-146a, associated with inflammatory markers (fold change: −1/388) (*p* < 0.05). *No safety concerns were identified in this study.*
Urakaze M et al. 2021 [764]	Randomized, double-blind, placebo-controlled prospective study	44 subjects including prediabetes (Av. 46–48 yrs.)	0, 12 mg/day	12 weeks	✓ Type 2 diabetes ✓ Metabolic syndrome ✓ Bioavailability, ✓ ADME	Glucose levels at 120 min after the 75 g oral glucose tolerance test (OGTT) were significantly lower than before supplementation. HbA1c (*p* < 0.05), apo E (*p* < 0.05), and MDA-modified LDL (*p* < 0. 05) also decreased, while total cholesterol, triglycerides, and HDL-C levels were unchanged. Matuda index, a measure of insulin resistance, was improved in AX-treated subjects compared to pre-treatment. Plasma AX levels were undetectable at baseline and increased to 122.69 ng/mL (*c.a.* 205 nM) after 4 weeks in the intervention group, and this level was maintained until 12 weeks. *No safety concerns were identified in this study.*
Birudaraju D. et al. 2020 [765]	Randomized, double-blind, placebo-controlled prospective study	22 healthy subjects (48.8 ± 16.0 yrs.)	0, 6 mg/twice a day * (0,12 mg/day)	4 weeks	✓ Cardiovascular health	*Combination with Cavacurcumin, Eicosapentaenoic acid (Omega-3s), AX and γ-linoleic acid (Omega-6) (CEAG)* The CEAG group had significantly lower mean systolic blood pressure at 4 weeks [4.7 ± 6.8 (*p* = 0.002)] compared to the placebo group. A significant decrease in high-sensitivity C-reactive protein (hsCRP) (−0.49 ± 1.9 vs. + 0.51 ± 2.5, *p* = 0.059) and a blunt increase in IL-6 (+0.2 vs. +0.4, placebo = 0.60) were observed compared to placebo. *No safety concerns were identified in this study.*
Chan K. et al. 2019 [766]	Randomized, double-blind, placebo-controlled prospective study	54 patients With type 2 diabetes	0, 6, 12 mg/day	8 weeks	✓ Type 2 diabetes ✓ Cardiovascular health ✓ Metabolic syndrome	Increased plasma AX levels and decreased fasting plasma glucose and HbA1c levels. In the 12 mg AX group, there was a reduction in plasma triglyceride, total cholesterol, and LDL levels. Lowered changes in plasma IL-6 and TNF- levels and plasma vWF level and higher changes in AT-III level. In 12 mg AX group, there were decreased changes in plasma FVII and PAI-1levels. *No safety concerns were identified in this study.*
Landi F et al. 2019 [767]	Randomized open-labeled, prospective study	47 subjects with higher level of serum Chol. (not needing statins or statin intolerant) (58.7± 8.7 yrs.)	0.5 mg/day * in Nutraceutical B	6 weeks	✓ Cardiovascular health ✓ Metabolic syndrome	*Nutraceutical B: policosanol (10 mg), red yeast rice (200 mg; 3 mg monacolin K), Berberine (500 mg), Astaxanthin (0.5 mg), folic acid (200 mcg), and Coenzyme Q10 (2 mg)* Both nutraceutical combinations improved the lipid profile including total Chol., HDL-Chol., LDL-Chol., and TG. *No safety concerns were identified in this study.*
Mashhadi N.S. et al. 2018 [768]	Randomized, double-blind, placebo-controlled, prospective study	44 participants with type 2 diabetes	0, 8 mg/day	8 weeks	✓ Type 2 diabetes ✓ Cardiovascular health ✓ Metabolic syndrome	Increased the serum adiponectin concentration, reduced visceral body fat mass (*p* < 0.01), serum triglyceride and very-low-density lipoprotein (VLDL) cholesterol concentrations, systolic blood pressure, fructosamine concentration (*p* < 0.05), and marginally reduced the plasma glucose concentration (*p* = 0.057). *No safety concerns were identified in this study.*
Sarkkinen ES., et al. 2018 [769]	Randomized, double-blind, placebo-controlled, prospective study	35 overweight subjects with mildly or moderately elevated blood pressure	N/A (0, 4 g of kirill oil powder /day)	56 days	✓ Safety ✓ Cardiovascular health ✓ Metabolic syndrome	Average values of hematological measurements were within the reference range for all subjects, and no significant changes were observed in blood pressure or lipid levels. *No serious adverse events were reported.*
Canas J. A. et al. 2017 [770]	Randomized, double-blind, placebo-controlled, prospective study	20 children with simple obesity (BMI > 90%)	500 μg/day * (MCS)	6 months	✓ Obesity ✓ Metabolic syndrome ✓ (Pediatric dosage)	Mixed-carotenoid supplementation (MCS) increased carotene, total adiponectin, and high-molecular-weight adiponectin in plasma compared to placebo; MCS decreased BMI z-score, waist-to-height ratio, and subcutaneous adipose tissue compared to placebo. AX was used as a part of MCS. *No safety concerns were identified in this study.*
Maki KC. et al. 2015 [771]	Randomized, double-blind, placebo-controlled, prospective study	102 subjects with TAG 150–499 mg/dL and LDL cholesterol (LDL-C) ≥70 mg/dL	0, 12 mg/day *	8 weeks	✓ Cardiovascular health ✓ Metabolic syndrome	*Test food (PDL-0101): 1.8 g/day eicosapentaenoic acid, 100 mg/day tocopherol-freeγ/δ tocotrienols enriched with geranylgeraniol, extracted from annatto, and 12 mg/day AX.* After 8 weeks of treatment, PDL-0101 significantly reduced median TAG compared to placebo (−9.5% vs. 10.6%, *p* < 0.001), but there was no significant change in mean LDL-C (−3.0% vs. −8.0% for PDL-0101 and placebo, respectively, *p* = 0.071), no significant change in mean high-density lipoprotein cholesterol (approximately 3% reduction in both groups, *p* = 0.732), or median oxidized LDL concentration (5% vs. −5% for PDL-0101 and placebo, respectively, *p* = 0.112). *No safety concerns were identified in this study.*
Takemoto M. et al. 2015 [697]	Case report	1 Werner syndrome patient	12 mg/day *	6 months	✓ Werner syndrome ✓ (Metabolic syndrome)	Improved blood transaminase concentrations before AX intervention and 3 and 6 months after initiation were AST 40 IU/L, 41 IU/L, and 20 IU/L; ALT 69 IU/L, 62 IU/L, and 34 IU/L; GGT 38 IU/L, 41 IU/L, and 35 IU/L; and cholinesterase 360 IU/L, 366 IU/L, and 331 IU/L, respectively. Liver-to-spleen Hounsfield units on CT were 0.41 before AX initiation, 0.71 at 3 months, and 0.94 at 6 months. No significant changes after AX intervention in hyaluronic acid, a marker of liver fibrosis; high-sensitivity C-reactive protein, a marker of inflammation; or MDA-modified LDL. *No safety concerns were identified in this study.*
Ni Y. et al. 2015 [772]	Randomized, single-blind, placebo-controlled, prospective study	12 NASH patients	0, 12 mg/day	24 weeks	✓ Metabolic syndrome ✓ (NASH)	Improved steatosis (*p* < 0.05), marginally improved lobular inflammation (*p* = 0.15), and NAFLD activity score (*p* = 0.08) *No safety concerns were identified in this study.*
Choi H.D. et al. 2011 [698]	Randomized, two-arm, prospective study	23 obese and overweight subjects	5 and 20 mg/day	3 weeks	✓ Oxidative Stress ✓ (Obesity) ✓ Cardiovascular health ✓ Metabolic syndrome	5 mg/day: MDA decreased by 34.6%, isoprostane (ISP) decreased by 64.9%, SOD increased by 193%, and TAC increased by 121% after 3 weeks compared to baseline (*p* < 0.01). 20 mg/day: MDA decreased by 35.2%, ISP decreased by 64.7%, SOD increased by 194%, and TAC increased by 125% after 3 weeks compared to baseline (*p* < 0.01). Decreased LDL cholesterol and ApoB. *No safety concerns were identified in this study.*
Choi, H.D. et al. 2011 [699]	Randomized, double-blind, placebo-controlled, prospective study	27 overweight subjects	0, 20 mg/day	12 weeks	✓ Oxidative Stress ✓ (Obesity)	MDA reduced by 17.3% and 29% after 8 and 12 weeks compared to placebo (*p* < 0.01), ISP reduced by 40.2% and 52.9% after 8 and 12 weeks compared to placebo (*p* < 0.01), SOD increased by 124.8% after 12 weeks compared to placebo (*p* < 0.01), and TAC increased by 130.1% after 12 weeks compared to placebo (*p* < 0.05). *No safety concerns were identified in this study.*
Yoshida H. et al. 2010 [773]	Randomized, double-blind, placebo- controlled, prospective study	61 non-obese subjects with fasting serum triglyceride of 120–200mg/dl and without diabetes and hypertension	0, 6, 12, 18 mg/day	12 weeks	✓ Cardiovascular health ✓ Metabolic syndrome	Multiple comparison: triglycerides were significantly decreased by 12 and 18 mg/day, and HDL-cholesterol was significantly increased by 6 and 12 mg. Serum adiponectin was increased by AX (12 and 18 mg/day), and changes in adiponectin were positively correlated with changes in HDL-cholesterol. *No safety concerns were identified in this study.*
Satoh A. et al. 2009 [774]	Open-labeled, prospective study	20 subjects at risk for developing metabolic syndrome (from 127 healthy subjects)	4, (8, 20) mg/day	4 weeks	✓ Cardiovascular health ✓ Metabolic syndrome	When subjects who met the diagnostic criteria for metabolic syndrome in Japan (SBP ≥ 130 mmHg, DBP ≥ 85 mmHg, TG ≥ 150 mg/dL, FG ≥ 100 mg/dL) at the start of the study were selected from the 4mg group, there was a significant decrease in SBP (*p* < 0.01). On the other hand, there was no significant decrease in DBP. Reduced TG after treatment (218 mg/dL) than the baseline value (292 mg/dL), marginally reduced fasting glucose after the intervention (*p* < 0.1). *No safety concerns were identified in this study*
Uchiyama A. et al. 2008 [775]	Open-labeled, prospective study	17 subjects at risk for developing metabolic syndrome	8 mg twice day	3 months	✓ Cardiovascular health ✓ Metabolic syndrome ✓ Bioavailability, ✓ ADME	Significant decreases in plasma Hb_Alc_ (*p* = 0.0433) and TNF-α levels (*p* = 0.0022) and an increase in adiponectin concentration (*p* = 0.0053). *N.S*: body weight, BMI, and waist circumference. The blood concentration reached the plateau after a month of treatment and was retained at that level until 3 months of treatment (0.2–0.25 μg/mL (0.34–0.42 μmol/L)). *No safety concerns were identified in this study.*
Evaluation of efficacy in cardiovascular diseases (CVDs) patients						
Heidari M. et al. 2023 [776]	Randomized, double-blind, placebo-controlled prospective study	44 CAD patients (40–65 yrs.), angiographic evidence of 50% stenosis in at least one of the major coronary arteries, 25 < BMI <35	0, 12 mg/day	8 weeks	✓ Cardiovascular health ✓ Metabolic syndrome ✓ Oxidative stress	12 mg of AX or placebo (microcrystalline cellulose) groups along with a low-calorie diet for a period of 8 weeks. Significant reductions in total cholesterol (−14.95 ± 33.57 mg/dL, *p* < 0.05) and LDL-C (−14.64 ± 28.27 mg/dL, *p* < 0.05) in the AX group with coronary artery disease (CAD). However, TG and HDL-C levels could not be affected by AX did not change “serum” levels of Sirtuin1 and TNF-α, Body composition, or glycemic indices. (Comments: BMI 25–27, mild obesity, non-insulin resistance with normoglycemia, normal TC, slightly elevated TG, and very low HDL-Chol. The effect on body composition seems to be more influenced by the low calorie diet. The clinical significance of the serum-free form of Sirt-1 remains unclear. *No safety concerns were identified in this study.*
Ishiwata S. et al. 2021 [692]	Open-labeled, prospective study	17 patients with systolic heart failure	12 mg/day *	3 months	✓ Cardiovascular health ✓ Oxidative stress ✓ (Heart failure)	After 3 months of AX supplementation, the ”Specific Activity Scale” score increased from a median of 4.5 (interquartile range, 2.0) to 6.5 (interquartile range, 1.1) metabolic equivalents (*p* = 0.001), and the physical and mental component summary scores increased from 46.1 ± 9.2 to 50.8 ± 6.8 (*p* = 0.015) and 48.9 ± 9.1 to 53.8 ± 4.8 (*p* = 0.022), respectively. There was a linear relationship between baseline heart rate or mental component summary score and rate of change in the “Specific Activity Scale” score (*r* = 0.523, *p* = 0.031, *r* = −0.505, *p* = 0.039, respectively). Furthermore, there was a direct relationship between ischemic etiology and the rate of change in the physical component summary score (r = 0.483, *p* = 0.049, respectively). There was also a linear relationship between the rate of change in the “Specific Activity Scale” score and the rate of change in the mental component summary score (*r* = 0.595, *p* = 0.012). *No safety concerns were identified in this study.*
Kato T. et al. 2020 [694]	Open-labeled, prospective study	16 patients with systolic heart failure	12 mg/day *	3 months	✓ Cardiovascular health ✓ Musculoskeletal ✓ function	Increased left ventricular ejection fraction (LVEF) from 34.1 ± 8.6% to 38.0 ± 10.0% (*p* = 0.031), and the 6-min walk distance increased from 393.4 ± 95.9 m to 432.8 ± 93.3 m (*p* = 0.023). Significant relationships were observed between percent changes in dROM level and those in LVEF. *No safety concerns were identified in this study.*
Marazzi G et al. 2017 [777]	Randomized, single-blind, placebo-controlled, prospective study	100 patients with CVD, percutaneous coronary intervention in the past 12 months, high-dose statin intolerance, and LDS treatment alone did not reduce LDL-C by more than 50%.	0, 0.5 mg/day * (Armolipid Plus)	12 months	✓ Cardiovascular health ✓ Metabolic syndrome	The aim was to compare the efficacy and tolerability of low-dose statin (LDS) therapy versus combined therapy of LDS plus a nutraceutical combination (Armolipid Plus: red yeast rice (200 mg), policosanol (10 mg), berberine (500 mg), folic acid (0.2 mg), AX (0.5 mg), and coenzyme Q10 (2 mg) After 3 months, LDL-C and total cholesterol were significantly lowered in the LDS + Almolypid Plus (n = 50) group (*p* < 0.0001), and 70% of this group achieved the treatment goal (LDL-C <70 mg/dL), while patients in the LDS group did not. *No safety concerns were identified in this study.*
Evaluation of efficacy in postmenopausal females						
Iwabayashi M. et al. 2009 [700]	Open-labeled, prospective study	35 healthy female subjects (with high oxidative stress, postmenopausal)	12 mg/day	8 weeks	✓ Oxidative stress ✓ Metabolic syndrome ✓ Mood/Stress ✓ (Unidentified complaints)	Increased blood biological antioxidant potential (Biological Antioxidant Potential (BAP); +4.6%, *p* < 0.05). After eight-week treatment with astaxanthin, significant improvement was observed in 5 of the 34 physical symptoms listed in the common questionnaire, including “tired eyes”, “stiff shoulders”, “constipation”, “gray hair”, and “cold skin”, and in 3 of the 21 mental symptoms, including “daily life is not enjoyable”, “difficulty falling asleep”, and “a sense of tension”. In addition, systolic (*p* = 0.021) and diastolic blood pressure (*p* < 0.001) significantly decreased. *No safety concerns were identified in this study.*
Kim Y.K. et al. 2004 [701]	Open-labeled, prospective study	15 healthy postmenopausal females	0, 2, 8 mg/day	8 weeks	✓ Oxidative Stress ✓ Metabolic syndrome ✓ (Postmenopausal females)	Decreased plasma TBARS levels: 2 mg group from 1.42 ± 0.18 to 1.13 ± 0.18 nM/mg (*p* < 0.05). 8 mg AX group from 1.62 ± 0.14 nM/mg to 1.13 ± 0.12 nM/mg after 8 weeks (*p* < 0.05). Increased TAS from 0.85 ± 0.42 mM/L to 1.90 ± 0.58 mM/L in the 8 mg group. Urinary 8-isoprostane excretion did not decrease significantly. Increase HDL-cholesterol levels in 2 mg and 8mg group increased significantly after 8 weeks from 50.6 ± 5.8 to 60.4 ± 7.1 mg/dL, 44.4 ± 10.7 to 49.4 ± 2.7 mg/dL respectively (*p* < 0.05). In the 2mg group, triglyceride decreased significantly from 171.6 ± 67.4 mg/dL to 145.8 ± 5.1 mg/dL (*p* < 0.05). *No safety concerns were identified in this study.*
Evaluation of efficacy in healthy subjects						
Takami M. et al. 2019 [778]	Open-labeled, prospective study	20 healthy young male subjects	*c.a*, 4.5 mg/day * from salmon	4 weeks	✓ Sports performance ✓ Energy metabolisms	Increased maximum work load by training in both groups (*p* = 0.009), while increased oxygen consumption during exercise in the antioxidant group only (*p* = 0.014). There were positive correlations between maximum work load and fat (*r* = 0.575, *p* = 0.042) and carbohydrate oxidations (*r* =0.520, *p* = 0.059) in the antioxidant group. Higher carbohydrate oxidation during rest in the post-training than that in the pre-training only in the antioxidant group. More decreased levels of serum insulin and HOMA-IR after training were observed in the antioxidant group than in the control group. *No safety concerns were identified in this study.*
Fukamauchi M. et al. 2007 [779]	Randomized, double-blind, placebo- controlled, prospective study	32 healthy subjects	0, 6 mg/day	6 weeks	✓ Sports performance ✓ Weight loss	The synergistic effects of AX intake (12 mg/day, 6 weeks) and aerobic exercise (walking) were studied. AX contributed to the reduction of body fat and suppressed the increase in blood lactate levels after exercise. *No safety concerns were identified in this study.* [Article in Japanese]
Others						
Miyawaki H. et al. 2008 [686]	Single-blind, placebo-controlled prospective study	20 healthy subjects	0, 6 mg/days	10 days	✓ Cardiovascular health ✓ (Microcirculatory flow)	The time for blood to pass through the microchannel array flow analyzer decreased from 52.8 ± 4.9 s to 47.6 ± 4.2 s in the AX group (*p* < 0.01), and a significant difference was found when comparing values in the AX group (47.6 ± 4.2 s) and placebo group (54.2 ± 6.7 s) (*p* < 0.05). *No safety concerns were identified in this study.*
**(E) Exercise/Sports performance, skeletal**						
**Author/year/** **reference**	**Study Design**	**Subjects**	**Dose ^#,##^**	**Duration**	**Major Outcome ^†^**	**Description**
Evaluation of efficacy in exercise/sports performance in healthy subjects/athletes (muscular damages)						
Barker G.A. et al. 2023 [780]	Double-blind, placebo-controlled prospective study	19 Resistance-trained males	0, 12 mg/day	4 weeks	✓ Sports performance ✓ (Muscle damage/recovery ✓ after eccentric exercise)	Significantly decreased in delayed onset muscle soreness (SORE score (*p* = 0.02)) for the AX group post-supplementation compared to pre-supplementation test. There is no effect on Placebo. Significantly decreased in VAS (visual analog scale; *p* = 0.01) for the AX group post-supplementation compared to the pre-supplementation test, whereas these had no effect in the Placebo group. No effect on performance. *No safety concerns were identified in this study.*
Nieman, D.C. et al. 2023 [781]	Randomized, double-blind, placebo-controlled crossover study	18 healthy subjects (Capable of running 2.25 h on laboratory treadmills at 70% maximal oxygen consumption rate (VO_2max_))	0, 8 mg/day	4 weeks	✓ Sports performance ✓ Immunity	The running bout for 2.25 h induced significant muscle soreness, muscle damage, and inflammation; AX supplementation had no effect on exercise-induced muscle soreness, muscle damage, or increases in 6 plasma cytokines and 42 oxylipins. However, AX supplementation inhibited exercise-induced decreases in 82 plasma proteins (at 24 h post-recovery). Most of these proteins were involved in immune-related functions. Twenty plasma immunoglobulins were identified that differed significantly between the AX and placebo groups; plasma levels of IgM were significantly reduced after exercise but recovered in the AX trial but not in the placebo trial after a 24-h recovery period after exercise. *No safety concerns were identified in this study.*
Sudo A et al. 2019 [737]	Placebo-controlled prospective study	19 healthy subject (College softball player for women in physical education	0.3 mg/day * in V7	30 days	✓ Fatigues ✓ Sports performance ✓ Eye health ✓ Skin health	*Test supplement (V7; astaxanthin, reduced coenzyme Q10, leucine, arginine, citrulline, DHA, Krill oil).* Studied the efficacy of V7 on subjective fatigue, sports performance, and skin conditions in college softball players. Subjective symptoms were evaluated with the VAS. The change of VAS in PRE and POST in the V7 group showed statistically significant improvement in 50 m running performance; a comparison of V7 and placebo POST showed statistically significant increases in leg fatigue, knee, and hip pain. The percent change between PRE and POST in V7 was statistically significantly higher in leg fatigue, hip and back pain, and total score. *No safety concerns were identified in this study.* [Article in Japanese]
Baralic, I. et al. 2015 [712]	Randomized, double-blind, placebo-controlled, prospective study	40 healthy subjects (young soccer players)	0, 4 mg/day	90 days	✓ Oxidative Stress ✓ (during exercise) ✓ Immunity	The increase in neutrophil count and hs-CRP level was found only in the placebo group, indicating a significant blunting of the systemic inflammatory response in the subjects taking AX. Improved prooxidant-antioxidant balance (PAB; *p* < 0.05) AX supplementation improves the sIgA response and attenuates muscle damage. *No safety concerns were identified in this study.*
Baralic I. et al. 2013 [713]	Randomized, double-blind, placebo-controlled prospective study	40 healthy subjects (soccer players)	0, 4 mg/day	90 days	✓ Oxidative Stress ✓ (Exercise)	Protected thiol groups against oxidative modification (increase in SH groups, *p* < 0.05; improved PON1 activity towards paraoxon and diazoxon, *p* < 0.05 and *p* < 0.01, respectively) *No safety concerns were identified in this study.*
Djordjevic B. et al. 2013 [714]	Randomized, double-blind, placebo-controlled prospective study	32 healthy subjects (soccer players)	0, 4 mg/day	90 days	✓ Oxidative Stress ✓ (Exercise)	Regular training significantly increased O_2_•¯ levels (main training effect, *p* < 0.01). O_2_•¯ concentrations. Not change TBARS and AOPP levels throughout the study. Decreased TAS levels post- exercise only in the Placebo group (*p* < 0.01). The increased total SH group content both in the AX and in the placebo groups (by 21% and 9%, respectively) and supplementation effect were marginally significant (*p* = 0.08). Decreased basal SOD activity both in the Placebo and in the AX group by the end of the study (main training effect, *p* < 0.01). Significant decrease in basal CK and AST activities after 90 days (main training effect, *p* < 0.01 and *p* < 0.001, respectively). Post-exercise CK and AST levels were significantly lower in the AX group compared to the Placebo group (*p* < 0.05) *No safety concerns were identified in this study.*
Klinkenberg L.J. et al. 2013 [715]	Randomized, double-blind, placebo-controlled, prospective study	32 well-trained male cyclists (25 ± 5 years, V˙O_2_^peak^ = 60 ± 5 mL·kg^−1^·min^−1^, W_max_ = 5.4 ± 0.5 W·kg^−1^)	0, 20 mg/day *	4 weeks	✓ Oxidative Stress ✓ (Exercise) ✓ Sports performance	*N.S;* effect on exercise-induced cardiac troponin T release (*p* = 0.24), changes in antioxidant capacity markers (trolox equivalent antioxidant capacity, uric acid, and malondialdehyde). Markers of inflammation (high-sensitivity C-reactive protein) and exercise-induced skeletal muscle damage (creatine kinase). *No safety concerns were identified in this study.*
Djordjevic B. et al. 2011 [714]	Randomized, double-blind, placebo-controlled, prospective study	32 male elite soccer players	0, 4 mg/day	90 days	✓ Oxidative Stress ✓ (Exercise) ✓ Sports performance	Changes in elevated O2-¯ concentrations after soccer exercise were statistically significant only in the Placebo group (exercise × supplementation effect, *p* < 0.05); TAS values decreased significantly only in the Placebo group after exercise (*p* < 0.01). After intervention, total SH group content increased (21% and 9%, respectively), and the effect of AX was marginally significant (*p* = 0.08). Basal SOD activity was significantly reduced in both the Placobo and AX groups at the end of this study (main training effect, *p* < 0.01).Post-exercise CK and AST levels were significantly lower in the AX group than in the Placebo group (*p* < 0.05) *No safety concerns were identified in this study.*
Bloomer, R.J. et al. 2005 [717]	Randomized, placebo-controlled, prospective study	20 resistance trained male subjects (25.1 ± 1.6 years)	0, 4 mg/day *	3 months	✓ Oxidative Stress ✓ (Exercise) ✓ Sports performance	*N.S*; Muscle soreness, creatine kinase (CK), and muscle performance were measured before and through 96 h of eccentric exercise *No safety concerns were identified in this study.*
Sawaki et al. 2002 [760]	Randomized, double-blind, placebo-controlled, prospective study	18 healthy male Subjects (College handball player)	0, 6 mg/day	4 weeks	✓ Sports performance ✓ (Muscle fatigue) ✓ Eye health ✓ (Eyestrain)	Although there was no difference between the two groups in post-exercise CK levels or heart rate, the AX group had significantly lower blood lactate levels 2 min after exercise. *No safety concerns were identified in this study.* [Article in Japanese]
Evaluation of efficacy in exercise/sports performance in healthy subjects/athletes (performance/energy metabolisms)						
Waldman H.S. et al. 2023 [782]	Randomized, double-blind, placebo-controlled crossover study	Resistance-trained males (Mean 23.4 yrs.)	0, 12 mg/day	4 weeks	✓ Sports performance ✓ Energy metabolisms	AX supplementation had no statistical effect on markers of substrate metabolism, Wingate variables, or markers of muscle damage, inflammation, or delayed-onset muscle soreness during the graded exercise test (GXT) compared to placebo (*p* > 0.05). However, 4 weeks of AX supplementation significantly reduced oxygen consumption during the final phase of the GXT compared to placebo (12%, *p* = 0.02), reduced systolic blood pressure (approximately 7%, *p* = 0.04), and significantly reduced baseline insulin levels (*c.a.* 24%, *p* = 0.05). *No safety concerns were identified in this study.*
McAllister MJ. et al. 2022 [711]	Randomized, double-blind, placebo-controlled crossover study	14 healthy subjects (23 ± 2 yrs)	0, 6 mg/day	4 weeks	✓ Oxidative Stress, ✓ (Exercise) ✓ Sports performance ✓ Energy metabolisms	A graded exercise test was performed after each treatment to measure substrate utilization during exercise at increased intensity. Glutathione was approximately 7% higher after AX treatment compared to placebo (*p* = 0.02, d = 0.48). Plasma hydrogen peroxide and malondialdehyde (MDA) did not differ between treatments (*p* > 0.05). Although not statistically significant (*p* = 0.45), highly oxidized protein products were reduced by approximately 28%. In the graded exercise stress test, mean fat oxidation rates did not differ between treatments (*p* > 0.05); however, fat oxidation was reduced from 50 to 120 W (*p* < 0.001) and from 85 to 120 W (*p* = 0.004) in both conditions. *No safety concerns were identified in this study.*
Brown, R.D. et al. 2021 [783]	Randomized, double-blind, placebo-controlled crossover study	12 recreationally trained male cyclists (27.5 ± 5.7 years, VO_2peak_: 56.5 ± 5.5 mL⋅kg^−1^⋅min^−1^, W_max_: 346.8 ± 38.4 W)	0, 12 mg/day	7 days	✓ Sports performance ✓ Energy metabolisms	Completion time of the 40-km cycling time trial improved by 1.2 ± 1.7% with AX supplementation, from 70.76 ± 3.93 min in the placebo condition to 69.90 ± 3.78 min in the AX condition (mean improvement time = 51 ± 71 s, *p* = 0.029, g = 0.21). Whole body fat oxidation rate was also greater in the AX group between 39 and 40 km (+0.09 ± 0.13 g ⋅ min^−1^, *p* = 0.044, g = 0.52) and respiratory exchange ratio was lower (−0.03 ± 0.04, *p* = 0.024, g = 0.60). *No safety concerns were identified in this study.*
Kawamura A. et al. 2021 [710]	Randomized open-labeled, prospective study	26 healthy male subject (22.3 ± 0.3 yrs.)	N/A (1 mg AX/100g salmon) *	10 weeks	✓ Oxidative Stress, ✓ (Exercise) ✓ Sports performance ✓ Musculoskeletal ✓ function	The skeletal muscle mass was higher after training than before training in both the control and intervention groups (*p* < 0.05). Increased maximal voluntary contraction after training in the intervention group (*p* < 0.05), but not significantly increased in the control group. Higher resting oxygen consumption after training in the intervention group only (*p* < 0.05). Serum carbonylated protein level as an oxidative stress marker tended to be lower immediately after exercise than before exercise in the intervention group only (*p* = 0.056). *No safety concerns were identified in this study.*
McAllister M.J. et al. 2021 [711]	Randomized, double-blind, placebo-controlled, crossover study	14 healthy young subjects, (23 ± 2 yrs.)	0, 6 mg/day	4 weeks	✓ Oxidative Stress, ✓ (Exercise) ✓ Sports performance	Glutathione was ~7% higher following AX compared with placebo (*p* < 0.05). There was no effect on plasma hydrogen peroxide or MDA (*p* > 0.05). Advanced oxidation protein products (AOPP) were reduced by ~28% (N.S.; *p* = 0.45) not affect substrate utilization during exercise. *No safety concerns were identified in this study.*
Fleischmann C. et al. 2019 [784]	Randomized, double-blind, placebo-controlled prospective study	22 healthy subjects (23.1 ± 3.5 yrs.)	0, 12 mg/day	30 days	✓ Sports performance ✓ Energy metabolisms ✓ Heat strain/stress	Decreased raised blood lactate caused by the VO_2 Max_ test in the AX group (9.4 ± 3.1 and 13.0 ± 3.1 mM in the AX and Placebo groups, respectively, *p* < 0.02). Change in oxygen uptake during recovery (−2.02 ± 0.64 and 0.83 ± 0.79% of VO_2 Max_ in the AX and Placebo groups, respectively, *p* = 0.001). *N.S*; anaerobic threshold or VO_2_ Max. physiological or biochemical differences in the heat tolerance test (HTT) (2 h walk at 40 °C, 40% relative humidity. *No safety concerns were identified in this study.*
Sudo A et al. 2020 [736]	Placebo-controlled prospective study	11 healthy subjects (College floorball athletes for women in physical education	0.3 mg/day * in V7	30 days	✓ Fatigues ✓ Sports performance ✓ Eye health ✓ Skin health	*Test supplement (V7; astaxanthin, reduced coenzyme Q10, leucine, arginine, citrulline, DHA, Krill oil).* Studied the efficacy of V7 on subjective fatigue, sports performance, and skin conditions in floorball athletes. Subjective symptoms were evaluated with the VAS. The change in VAS in PRE and POST in the V7 group showed that fatigue was significantly alleviated overall and in the torso. Significantly improved on a seated toe touch, increasing from 48.6 cm pre-intake to 51.8 cm post-intake. *No safety concerns were identified in this study.* [Article in Japanese]
Takami M. et al. 2019 [778]	Open-labeled, prospective study	20 healthy young male subjects	*c.a*, 4.5 mg/day * from salmon	4 weeks	✓ Sports performance ✓ Energy metabolisms	Increased maximum work load by training in both groups (*p* = 0.009), while increased oxygen consumption during exercise in the antioxidant group only (*p* = 0.014). There were positive correlations between maximum work load and fat (*r* = 0.575, *p* = 0.042) and carbohydrate oxidations (*r* =0.520, *p* = 0.059) in the antioxidant group. Higher carbohydrate oxidation during rest in the post-training than that in the pre-training only in the antioxidant group. *No safety concerns were identified in this study.*
Talbott I. et al. 2016 [785]	Randomized, double-blind, placebo-controlled prospective study	28 recreational runners (42 ± 8 yerars)	0, 12 mg/day	8 weeks	✓ Sports performance	Reduced average heart rate at submaximal endurance intensities (aerobic threshold, AeT, and anaerobic threshold, AT), but not at higher “peak” intensities. *No safety concerns were identified in this study.*
Res T. et al. 2013 [716]	Randomized, double-blind, placebo-controlled, prospective study	32 trained male cyclists or triathletes (25 ± 1 years, V˙O_2_^peak^ = 60 ± 1 mL·kg^−1^·min^−1^, W_max_ = 395 ± 7 W)	0, 20 mg/day	4 weeks	✓ Oxidative Stress ✓ >(Exercise) ✓ Sports performance	*N.S*; total plasma antioxidant capacity (*p* = 0.90) or attenuated malondialdehyde levels (*p* = 0.63). Whole-body fat oxidation rates during submaximal exercise (from 0.71 +/− 0.04 to 0.68 ± 0.03 g.min and from 0.66 ± 0.04 to 0.61 ± 0.05 g.min in the Placebo and AX groups, respectively; *p* = 0.73), time trial performance (from 236 ± 9 to 239 ± 7 and from 238 ± 6 to 244 ± 6 W in the Placebo and AX groups, respectively; *p* = 0.63). *No safety concerns were identified in this study.*
Earnest C.P. et al. 2011 [786]	Randomized, double-blind, placebo-controlled, prospective study	14 amateur endurance-trained subjects (18–39 years, V˙O_2_^peak^ = 52.84 ± 3.5 mL·kg^−1^·min^−1^, W_max_ = 330 ± 26 W	0, 4 mg/day	28 days	✓ Sports performance	Improved performance in the 20-km cycling time trial in the AX group (*n* = 7, −121 s; 95% CI, −185, −53), but not in the Placebo group (*n* = 7, −19 s; 95% CI, −84, 45). The AX group significantly increased power output (20 W; 95% CI, 1, 38), while the Placebo group did not (1.6 W; 95% CI, −17, 20). N.S: carbohydrate, fat oxidation, and blood indices indicative of fuel mobilization. *No safety concerns were identified in this study.*
Malmstena C.L.L. et al. 2008 [787]	Randomized, double-blind, placebo-controlled prospective study	40 young healthy subjects (17–19 years)	0, 4 mg/day	3 months	✓ Sports performance	The increased average number of knee bending (squats) increased by 27.05 (from 49.32 to 76.37, AX group) vs. 9.0 (from 46.06 to 55.06, placebo subjects), *p* = 0.047. *No safety concerns were identified in this study.*
Tajima T. et al. 2004 [788]	Randomized, double-blind, placebo-controlled, crossover study	18 healthy subjects (35.7 ± 4 years)	0, 5 mg/day	2 weeks	✓ Sports performance	Increases in CV_RR_ and HF/TF (Heart rate variability) were significant during exercise at 70% maximum heart rate (HR_max_) intensity (*p* < 0.05). Moreover, after the AX supplementation there was, decreased minute ventilation (V_E_) during exercise at 70% HRmax (*p* < 0.05). Decreased LDL cholesterol (*p* < 0.05) and respiratory quotient after exercise. *No safety concerns were identified in this study.* [Article in Japanese]
Evaluation of efficacy in exercise performance in elderly subjects (sarcopenia)						
Liu S.Z. et al. 2021 [789]	Randomized, double-blind, placebo-controlled, prospective study	42 elderly subjects (65–82 yrs.)	0, 12 mg/day *	12 weeks	✓ Oxidative Stress, ✓ (Elderly subjects) ✓ Musculoskeletal ✓ function ✓ Energy metabolisms	*Intervention: 6 mg Zn, 10 mg tocotrienol and 12 mg AX*. In endurance training (ET), specific muscular endurance was improved only in the AX group (Pre 353 ± 26 vs. Post 472 ± 41), and submaximal graded exercise test duration was improved in both groups (Placebo 40.8 ± 9.1% vs. AX 41.1 ± 6.3%). The increase in fat oxidation at low intensity after ET was greater in AX (Placebo 0.23 ± 0.15g vs AX 0.76 ± 0.18g) and was associated with reduced carbohydrate oxidation and improved exercise efficiency in men but not in women. Analysis associated with the study [790]. *No safety concerns were identified in this study.*
Nakanishi R. et al. 2021 [718]	Randomized, double-blind, placebo-controlled prospective study	29 nursing home resident’s healthy elderly subjects (80.9 ± 1.5 yrs.)	0, 12 mg/twice a day * (0, 24 mg/day)	16 weeks	✓ Oxidative Stress, ✓ (Elderly subjects) ✓ Musculoskeletal ✓ function	There was a decrease in d-ROM values with the AX group (*p* < 0.01) but not with the placebo group; the AX group had a therapeutic effect on 6-min walking distance compared with the placebo group (*p* < 0.05). The AX group had an increase in distance and number of steps in the 6-min walking test compared to the placebo group. Furthermore, the rate of increase in blood lactate levels after walking was lower in the AX group than in the placebo group (*p* < 0.01). *No safety concerns were identified in this study.*
Liu S.Z. et al. 2018 [790]	Randomized, double-blind, placebo-controlled, prospective study	42 elderly subjects (65–82 yrs.)	0, 12 mg/day *	12 weeks	✓ Musculoskeletal ✓ function ✓ Energy metabolisms	Administration of AX increased maximal voluntary force (MVC) by 14.4% (± 6.2%, *p* < 0.02), tibialis anterior muscle size (cross-sectional area, CSA) by 2.7% (±1.0%, *p* < 0.01), and specific impulse increased by 11.6% (MVC/CSA, ±6.0%, *p* = 0.05), respectively, whereas placebo treatment did not alter these characteristics (MVC, 2.9% ± 5.6%; CSA, 0.6% ± 1.2%; MVC/CSA, 2.4 ± 5.7%; all *p* > 0.6). *No safety concerns were identified in this study.*
Evaluation of efficacy in joint health (osteoarthritis (OA), carpal tunnel syndrome)						
Stonehouse W et al. 2022 [791]	Randomized, double-blind, placebo-controlled prospective study	235 Healthy adults (40–65 yrs, BMI >18.5 to <35 ) w/.clinically diagnosed with mild to moderate knee OA	0, 0.35 mg/day (in krill oil)	6 months	✓ Joint health	Knee pain scores (Western Ontario and McMaster Universities Osteoarthritis Index: WOMAC, numeric scale) improved in both groups, with greater improvement with krill oil than placebo. Knee stiffness and physical function also showed greater improvement with krill oil than placebo. NSAID usage, serum lipid profile, inflammatory markers, and safety markers did not differ between groups. *No safety concerns were identified in this study.*
Macdermid J.C. et al. 2012 [792]	Randomized, triple-blind, placebo-controlled prospective study	63 patients with carpal tunnel syndrome	0, 4 mg/days with splinting	9 weeks	✓ Joint health ✓ (Carpal tunnel syndrome)	The Symptom Severity Scale (SSS) over the course of treatment in both AX treated and placebo groups (*p* = 0.002) showed no differences between the groups (*p* = 0.18). The Disability of Arm, Shoulder, and Hand Questionnaire and the Short Form 36-item Health Survey showed no effects over time or between treatment groups. *No safety concerns were identified in this study.*
**(F) Brain Health**						
**Author/year/** **reference**	**Study design**	**Subjects**	**Dose ^#,##^**	**Duration**	**Major Outcome ^†^**	**Description**
Evaluation of efficacy in cognitive functions						
Hayashi M. et al. 2018 [793]	Randomized, double-blind, placebo-controlled prospective study	54 healthy subjects (45–64 yrs.)	0, 8 mg/day from *Paracoccus*	8 weeks	✓ Cognitive function ✓ Bioavailability, ADME	(After 8 weeks of AX administration, the serum AX concentration increased to 0.173 ± 0.058 μg/mL (0.29 μM).( Not detected in the placebo group.) Evaluation: word memory test, verbal fluency test, and Stroop test. AX group significantly larger increase in blood AX level. No significant intergroup differences in the results of the evaluations. Subgroup analysis (<55 yrs. old and ≥55 yrs. old): “words recalled after 5 min” in word memory test in <55 year old subjects showed significant improvement in the AX group than in the placebo group, which was not found in ≥55 year old subjects. *No safety concerns were identified in this study.*
Hongo N. et al. 2018 [794]	Randomized, double-blind, placebo-controlled prospective study	40 Healthy subjects aged 60–79 years reporting awareness of cognitive and/or physical decline (Mean 65.8 yrs.)	0, 12 mg/day *	12 weeks	✓ Cognitive function ✓ Fatigues ✓ Mood/Stress	*AX group (12 mg AX, 10 mg tocotrienols, 6 mg zinc, and 600 IU vitamin D) and control group (same formulation w/o. AX).* Cognitive functions/Fatigues: AX intake shortened the reaction time in working-memory tasks (particularly improved the speed of processing newly-provided visual information by comparing it with previously-provided information while appropriately retaining the previous information in the memory for several seconds). In addition, AX promoted efficacy in the improvement of self-assessed memory ability. Furthermore, in the subjects with improved endurance, AX enhanced the recognition memory function of nonverbal information, particularly the accuracy of memorizing faces and distinguishing them from newly presented faces. Moods/Stress: Improved the scores of “Anger-Hostility,” “Confusion-Bewilderment,” “Fatigue-Inertia,” and TMD in the POMS survey and the decrease in the score of “irritated” in the VAS survey in the AX group *No safety concerns were identified in this study.* [Article in Japanese]
Ito N. et al. 2018 [795]	Randomized, double-blind, placebo-controlled, prospective study	14 patients diagnosed with MCI (57–78 yrs.; MMSE scores 24–27).	0, 6 mg/day *	12 weeks	✓ Cognitive function	Evaluation of cognitive improvement in patients with mild cognitive impairment (MCI). The Central Nervous System Vital Signs (CNSVS, also known as ‘Cognitrax’) test significantly improved “psychomotor speed” and “processing speed” in the AX group compared to the placebo group. *No safety concerns were identified in this study.*
Zanotta D. et al. 2014 [796]	Open-labeled, prospective study	104 subjects diagnosed with mild cognitive impairment (Mean 71.2 yrs.)	*c.a*. 4.2 mg/day *	60 days	✓ Cognitive functions	*Test supplement (Illumina); 20mg Bacosides from* Bacopa monnieri*, 30 mg Phosphatidylserine, 30 mg V.E and 2 mg AX.* Significantly improved in the Alzheimer’s Disease Assessment Scale-cognitive subscale (ADAS-cog) total score from 13.7 ± 5.8 at baseline to 9.7 ± 4.9 at 60 days and in the clock drawing test from 8.5 ± 2.3 to 9.1 ± 1.9. The greatest improvement in each component of the ADAS-cog was in the memory task. The largest improvement in each component of the ADAS-cog was in the memory tasks. In multivariate analysis, larger improvements in ADAS-cog scores were associated with less deterioration in baseline Mini-Mental State Examination scores. Efficacy was rated “excellent” or “good” by 62% of subjects. The study compounds were well tolerated, with one non-serious adverse event (gastric disturbances in a subject who was taking concomitant oral corticosteroids) reported in the entire study population and tolerability rated as “excellent” or “good” by 99% of subjects.
Katagiri M. et al. 2012 [797]	Randomized, double-blind, placebo-controlled, prospective study	89 healthy middle-aged and elderly subjects who complained of age-related forgetfulness.	0, 6, 12 mg/day	12 weeks	✓ Cognitive functions	After 12 weeks, CogHealth battery scores improved in the high-dose group (12 mg AX/day). Improved Groton Maze Learning Test scores were seen earlier in the low-dose group (6 mg AX/day) and in the high-dose group than in the placebo group. However, the sample size was too small to show significant differences in cognitive function between the AX and placebo groups. *No safety concerns were identified in this study.*
Satoh A. et al. 2009 [774]	Open-labeled, prospective study	10 healthy male subjects (50–69 yrs.)	12 mg/day	12 weeks	✓ Cognitive function	In the CogHealth tasks (simple response, choice response, working memory, delayed memory, and divided attention), significant reductions in reaction time were observed in the “divided attention” task after 6 weeks and in all tasks after 12 weeks. Accuracy on the “working memory” task was significantly improved after 12 weeks of treatment; however, no such effect was observed on the “delayed recall” task. *No safety concerns were identified in this study*
Evaluation of efficacy in fatigue						
Yamazaki I. et al, 2022 [798]	Randomized, double-blind, placebo-controlled prospective study	257 subjects feeling fatigue with aging and on daily (Av. 44–45 yrs.)	0, 6 mg/day *	8 weeks	✓ Fatigues	*Combination with sesamins (sesame lignans).* Visual Analogue Scale (VAS) scores of the fatigue feeling at 4 weeks in the test food groups compared to the placebo food groups (*p <* 0.05), and these scores tended to be lower in the test food groups than the placebo food groups at 8 weeks (*p* < 0.1). Subgroup analysis: (A) There was a trend toward greater improvement in the group with a greater VAS value at screening than in the group with a smaller VAS value for fatigue at screening. (B)The frequency of exercise at screening was divided into two groups: “most/several times a month” and “at least once a week/every day.” The “at least once a week/every day” group, which exercised more frequently, showed more improvement in the VAS. *No safety concerns were identified in this study.* [Article in Japanese]
Sudo A et al. 2020 [736]	Placebo-controlled prospective study	11 healthy subjects (College floorball athletes for women in physical education	0.3 mg/day * in V7	30 days	✓ Fatigues ✓ Sports performance ✓ Eye health ✓ Skin health	*Test supplement (V7; astaxanthin, reduced coenzyme Q10, leucine, arginine, citrulline, DHA, Krill oil).* Studied the efficacy of V7 on subjective fatigue, sports performance, and skin conditions in floorball athletes. Subjective symptoms were evaluated with the VAS. The change in VAS in PRE and POST in the V7 group showed that fatigue was significantly alleviated overall and in the torso significantly improved on a seated toe touch, increasing from 48.6 cm pre-intake to 51.8 cm post-intake. *No safety concerns were identified in this study.* [Article in Japanese]
Sudo A et al. 2019 [750]	Open-labeled, prospective study	19 healthy females (Mean 47.3 yrs.)	0.3 mg/day * in V7	30 days	✓ Fatigues ✓ Skin health ✓ (Eye health) ✓ (Cognitive function)	*Test supplement (V7; astaxanthin, reduced coenzyme Q10, leucine, arginine, citrulline, DHA, Krill oil)*. Studied the efficacy of V7 on subjective fatigue and skin conditions in typical middle-aged females. Subjective symptoms were evaluated with the VAS. The change of VAS in PRE and POST in the V7 group showed statistically significantly improved general fatigue, leg fatigue, and the state of the lower back (*p* < 0.05, respectively); significantly improved dark spots, blotches, and the elasticity and appearance of the skin; no significant change in eye strain or memory loss. *No safety concerns were identified in this study.* [Article in Japanese]
Sudo A et al. 2019 [737]	Placebo-controlled prospective study	19 healthy subject (College softball player for women in physical education	0.3 mg/day * in V7	30 days	✓ Fatigues ✓ Sports performance ✓ Eye health ✓ Skin health	*Test supplement (V7; astaxanthin, reduced coenzyme Q10, leucine, arginine, citrulline, DHA, Krill oil).* Studied the efficacy of V7 on subjective fatigue, sports performance, and skin conditions in college softball players. Subjective symptoms were evaluated with the VAS. The change of VAS in PRE and POST in the V7 group showed statistically significant improvement in fatigue, shoulder pain, skin blemishes, and 50 m running performance; a comparison of V7 and placebo POST showed statistically significant increases in leg fatigue, knee and hip pain, skin elasticity and whitening, memory loss, and eye strain. The percent change between PRE and POST in V7 was statistically significantly higher in leg fatigue, hip and back pain, dull skin, eye strain, and total score. *No safety concerns were identified in this study.* [Article in Japanese]
Imai A. et al. 2018 [722]	Randomized, double-blind, placebo-controlled crossover study	42 healthy subjects	0, 6 mg/day *	4 weeks	✓ Oxidative stress ✓ (during mental and physical tasks) ✓ Fatigues ✓ Mood/Stress	Elevated PCOOH levels during mental and physical tasks were attenuated by AX supplementation. Improved recovery from mental fatigue compared to placebo. No differences were found between AX and placebo in other secondary outcomes such as subjective feelings, work efficiency, and autonomic activity. The treatment group showed a significant reduction (*p* < 0.05) in fatigue during recovery after the mental task compared to the placebo group. (Evaluated by VAS) *No safety concerns were identified in this study.*
Hongo N. et al. 2017 [723]	Randomized, double-blind, placebo-controlled, prospective study	39 healthy subjects	0, 12 mg/day *	12 weeks	✓ Oxidative stress ✓ (during mental and physical tasks) ✓ Fatigues ✓ Mood/Stress	Intent-to-treat (ITT) analysis; fatigue after physical and mental stress was significantly lower in the AX group than in the placebo at week 8; the change in POMS friendliness was significantly higher in the AX group than in the control group at week 8; the rate of change in BAP values at week 12 was not significantly different between the AX and control groups. The rate of change in BAP values at week 12 was not significantly different between the AX group and the control group. *No safety concerns were identified in this study.* [Article in Japanese]
Kono K. et al. 2014 [751]	Randomized, double-blind, placebo-controlled prospective study	48 healthy subjects who complained of eye strain	0, 4 mg/day *	4 weeks	✓ Eye health ✓ Fatigue ✓ (Shoulder Stiffness)	See Kono K. et al. (2014) in Table 4 (C). [Evaluation of efficacy in “asthenopia (eyestrain)”]
Tsukahara H. et al. 2008 [753]	Open-labeled, prospective study	13 healthy subjects with shoulder Stiffness	6 mg/days *	4 weeks	✓ Fatigue ✓ (Shoulder Stiffness) ✓ Cardiovascular health ✓ EyeHealth ✓ Microcirculatory flow	See Tsukahara H.et al. (2008) in Table 4 (C). Evaluation of efficacy in “asthenopia (eyestrain)”.
Evaluation of efficacy in stress/mood/sleep quality						
Hayashi M. et al. 2020 [799]	Randomized, double-blind, placebo-controlled prospective study	54 healthy subjects (Av. 45–46 yrs.)	0, 12 mg/day from *Paracoccus*	8 weeks	✓ Stress/Mood ✓ Sleep Quality ✓ Bioavailability, ✓ ADME	After 8 weeks of AX administration, the serum AX concentration increased to 0.171 ± 0.082 μg/mL (0.29 μM) not detected in the placebo group. Evaluated with the Profile of Mood States 2nd Edition for stress (POMS2) and the Oguri-Shirakawa-Azumi Sleep Inventory. I did not observe any significant intergroup differences in stress or sleep. Subgroup analysis (>65 and ≤65 in the ”Depression-Dejection”in POMS): sleep of subjects who scored >65 (”Depression-Dejection”) showed significant improvement in the AX group compared with the placebo group; however, no significant improvement was observed in stress or the other subjects. *No safety concerns were identified in this study.*
Hongo N. et al. 2018 [794]	Randomized, double-blind, placebo-controlled prospective study	40 Healthy subjects aged 60–79 years reporting awareness of cognitive and/or physical decline (Mean 65.8 yrs.)	0, 12 mg/day *	12 weeks	✓ Cognitive function ✓ Fatigues ✓ Mood/Stress	See Hongo N.et al*. (*2018*)* in Table 4 (G). [Evaluation of efficacy in cognitive functions]
Imai A. et al. 2018 [722]	Randomized, double-blind, placebo-controlled crossover study	42 healthy subjects	0, 6 mg/day *	4 weeks	✓ Oxidative stress ✓ (during mental and physical tasks) ✓ Fatigues ✓ Mood/Stress	See Imai A. et al. (2018) in Table 4 (G). [Evaluation of efficacy in cognitive functions]
Hongo N. et al. 2017 [723]	Randomized, double-blind, placebo-controlled, prospective study	39 healthy subjects	0, 12 mg/day *	12 weeks	✓ Oxidative stress ✓ (during mental and physical tasks) ✓ Fatigues ✓ Mood/Stress	See Hongo N.et al. 2017 in Table 4 (G). [Evaluation of efficacy in fatigue]
Saito H et al. 2017 [800]	Randomized, double-blind, placebo-controlled prospective study	120 healthy subjects	0, 1.5, 3 mg/day * (1.5 mg from krill, 3.0 mg from *Haematococcus* algae)	12 weeks	✓ Sleep Quality	Subjects were divided into four groups: (A) placebo, (B) zinc-rich food, (C) zinc- and AX-rich food (krill), and (D) placebo supplemented with zinc-enriched yeast and AX oil (from *Haematococcus* algae). Pittsburgh sleep quality index (PSQI): improved significantly within all groups at the endpoint. Total sleep time (TST): No significant difference was noted in any groups in comparison to group A. Sleep onset latency (SOL): significantly improved in group B and group D in comparison to the placebo group. Body positional changes: Although the number increased in the group A (3.74 ± 4.38), this increase was suppressed in group B ( 0.71 ± 5.01), group C (0.58 ± 5.75) and group D ( 0.95 ± 4.33) (*N.S*.). *No safety concerns were identified in this study.*
Iwabayashi M. et al. 2009 [700]	Open-labeled, prospective study	35 healthy female subjects (with high oxidative stress, postmenopausal)	12 mg/day	8 weeks	✓ Oxidative stress ✓ Metabolic syndrome ✓ Mood/Stress ✓ (Unidentified complaints)	Increased blood biological antioxidant potential (Biological Antioxidant Potential (BAP); +4.6%, *p* < 0.05). After eight-week treatment with astaxanthin, significant improvement was observed in five of the 34 physical symptoms listed in the common questionnaire, including “tired eyes”, “stiff shoulders”, “constipation”, “gray hair”, and “cold skin”, and in 3 of 21 mental symptoms, including “daily life is not enjoyable”, “difficulty in falling asleep”, and “a sense of tension”. In addition, systolic (*p* = 0.021) and diastolic blood pressure (*p* < 0.001) significantly decreased. *No safety concerns were identified in this study.*
**(G) Infertility**						
**Author/year/** **reference**	**Study design**	**Subjects**	**Dose ^#,##^**	**Duration**	**Major Outcome ^†^**	**Description**
Evaluation of efficacy in gynecological disorders and assisted reproductive technology (ART)						
Jabarpour M. et al. 2023 [652]	Randomized, placebo-controlled prospective study	53 Patients with polycystic ovary syndrome (PCOS)	0, 6 mg/twice a day (0,12 mg/day)	60 days	✓ Women’s health ✓ Oxidative Stress ✓ (PCOS) ✓ ER stress ✓ Infertility ✓ ART	ER stress (Granulosa cells): After the intervention, AX treatment reduced the mRNA expression levels of 78-kDa glucose-regulated protein (GRP78), CCAAT/enhancer-binding protein homologous protein (CHOP), and X-box-binding protein 1 compared to the placebo group, but activated transcription factor 6 (ATF6) was not statistically significant. However, AX significantly increased the ATF4 expression level. GRP78 and CHOP protein levels represented a considerable decrease in the treatment group after the intervention. Antioxidant markers: Increased levels of TAC in follicular fluid. ART outcomes: higher rates of high-quality oocytes, high-quality embryos, and oocyte maturity in the AX group (the oocyte number, fertilization rate, and fertility rate; N.S.) *No safety concerns were identified in this study.*
Rostami S. et al. 2023 [653]	Randomized, triple-blind, placebo-controlled prospective study	50 Patients of endometriosis (stage III/ IV)	0, 6 mg/day	12 weeks	✓ Women’s health ✓ Oxidative Stress ✓ (Endometriosis) ✓ Infertility ✓ ART	Antioxidant markers: Increased serum levels of total antioxidant capacity (TAC, *p* = 0.004) and superoxide dismutase (SOD, 13.458 ± 7.276 vs. 9.040 ± 5.155; *p* = 0.010) were observed in the AX intervention group after therapy. In addition, serum Malondialdehyde (MDA, *p* = 0.031) decreased significantly after AX treatment. Inflammation markers: Serum IL-1β (*p* = 0.000), IL-6 (*p* = 0.024), and TNF-α (*p* = 0.038) showed significantly lower levels after AX treatment. ART outcomes: AX supplementation resulted in a significantly improved number of oocytes retrieved (*p* = 0.043), mature (MII) oocytes (*p* = 0.041), and improved quality embryos (*p* = 0.024). *No safety concerns were identified in this study.*
Gharaei R. et al. 2022 [651]	Randomized, double-blind, placebo-controlled prospective study	40 Patients with polycystic ovary syndrome (PCOS)	0, 8 mg/day	40 days	✓ Oxidative Stress ✓ (PCOS) ✓ Infertility ✓ ART	AX supplementation resulted in significantly higher serum Catalase and TAC levels in the AX group compared to the placebo group. However, there were no significant differences in serum MDA and SOD levels between groups. The expression of antioxidant genes such as Nrf2, HO-1, and NQ-1 was significantly increased in the granulosa cells (GC) of the AX group.ART: MII oocyte and high quality embryo rates were significantly increased in the AX group compared to the placebo group. There were no significant differences in chemical and clinical pregnancy rates between groups. *No safety concerns were identified in this study.*
Nakayama T. et al. 2015 [801]	Open-labeled, prospective study	18 subjects with normal menstrual cycles who have severe menstrual cramps or hypermenorrhea	12 mg/day	12 weeks	✓ Women’s health ✓ Dysmenorrhea	VAS values for menstrual cramps: After 12 weeks of AX supplementation it, decreased significantly with duration of supplementation (*p* < 0.05 vs. pre; 60.2 ± 16.3, 4 weeks; 45.3 ± 17.0, 12 weeks; 38.5 ± 19.0). After12 weeks washout, it increased to 55.0 ± 18.3 (*p* < 0.05 vs. 12 weeks). VAS values for menstrual flow: Significantly decreased at 12 weeks (*p* < 0.05, pre; 67.0 ± 15.7 vs. 12 weeks; 45.1 ± 20.3), but increased again after 12 weeks of washout (55.6 ± 16.6, *p* < 0.05 vs. 12 weeks). Day-dysmenorrhea: Significantly decreased at 12 weeks (*p* < 0.05, pre; 2.5 ± 1.8 vs. 12 weeks; 1.3 ± 0.8), but increased again after 12 of weeks washout (2.5 ± 2.0, *p* < 0.05 vs. 12 weeks). Others: No significant changes in the frequency of NSAID dosing or the blood levels of CA125 and Hb over the study period. *No safety concerns were identified in this study.* [Article in Japanese]
Evaluation of efficacy in improving sperm quality and assisted reproductive technology (ART)						
Dede G et al. 2022 [802]	N/A	30 Semen samples from normozoospermic individuals	0, 50, 100, 500 μM	N/A	✓ Infertility ✓ ART	Purpose: the protective efficacy of AX against the damage that occurs during sperm cryopreservation. The loss of motility due to cryopreservation of sperm was highest in the control group, and the least loss of motility was seen in the 100 μM AX group. Chromatin condensation of spermatozoa showed that the number of condensed spermatozoa was higher in the 100 μM AX group than in the other groups. (*in vitro* study)
Ghantabpour T. et al. 2022 [734]	N/A	The first phase; 10 semen samples from healthy men, the second phase; 25 semen samples from healthy men	0, 0.5, 1, 2 μM	N/A	✓ Infertility ✓ ART	Supplementation of sperm freezing medium with 1 µM AX was found to improve all parameters of sperm motility and viability (*p* ≤ 0.05). In addition, there were reduced levels of ROS parameters (intracellular hydrogen peroxide and superoxide) compared to the control group (*p* ≤ 0.05). AX also significantly reduced phosphatidylserine exogenous levels (*p* ≤ 0.05) and lipid peroxidation (*p* ≤ 0.05) after the freeze-thaw process. (*in vitro* study)
Kumalic SI. et al. 2020 [803]	Randomized, double-blind, placebo-controlled prospective study	72 patients with oligo-astheno-teratozoospermia (Av. 35–36 yrs)	0, 16 mg/day	3 months	✓ Infertility	In theAX group, no improvements in the total number of spermatozoa, concentration of spermatozoa, total motility of spermatozoa, morphology of spermatozoa, DNA fragmentation, and mitochondrial membrane potential of spermatozoa, or serum follicle-stimulating hormone were determined in patients with oligo-astheno-teratozoospermia. *No safety concerns were identified in this study.*
Terai K. et al. 2019 [649]	Randomized, two-arm, open-labeled, prospective study	31 males with oligozoospermia and/or asthenozoospermia	0 (HE), 16mg/day *	12 weeks	✓ Infertility	*Intervention; Combination of antioxidant supplements (L-carnitine, Zn, CoQ10, vitamin C, vitamin B12, vitamin E, and AX) and a Chinese herbal medicine, hochu-ekki-to (HE).* There were no significant improvements in endocrinological findings in the supplement group. Although no statistically significant improvement was observed in semen volume, sperm concentration, or sperm motility, total motile sperm counts were significantly improved. On the other hand, the HE group tended to increase semen concentration, semen motility, and total motile sperm count, but not significantly improve any endocrinological factors or semen findings. *No safety concerns were identified in this study.*
Andrisani A. et al*.,* 2015 [804]	N/A	24 Semen samples from healthy male donors of proven fertility, 27 Semen samples from patients who had failed to conceive after at least one year of regular unprotected intercourse	0, 2 μM	N/A	✓ Infertility ✓ ART	Evaluation of sperm capacitation, which involves a series of modifications, including regulation of reactive oxygen species, depletion of cholesterol in the sperm outer membrane, and protein tyrosine phosphorylation (Tyr-P) processes in the head region, that acquire sperm for the essential functions for fertilization of oocytes. AX successfully triggered Lyn translocation in patient-derived sperm(PG), bypassing the impaired ROS-related mechanisms for raft and Lyn translocation. In this study, ROS generation, lipid raft, and Lyn relocation are interdependent, leading to a continuous cellular acrosome reaction (AR). AX could be potentially used to ameliorate male idiopathic infertility by improving PG sperm function. A study related to the ref. [805]. (*in vitro* study)
Dona G. et al. 2013 [805]	N/A	24 Semen samples from healthy male donors	0 to 2 μM	N/A	✓ Infertility ✓ ART	The AX improved Tyr-P and acrosome reaction cells (ARC) values in sperm heads without affecting the ROS generation curve, while diamide successfully increased Tyr-P levels in flagella but did not increase ARC values. This suggests that AX, when inserted into the membrane, causes membrane changes, such as capsulation, that allow for Tyr-P conversion of the head. In other words, an acrosome reaction can occur, and more cells can be involved. A study related to the ref. [804]. (*in vitro* study)
Comhaire F.H. et al. 2005 [735]	Randomized, double-blind, placebo-controlled, prospective study	30 males with infertility of ≥12 months	0, 16 mg/day	3 months	✓ Infertility ✓ Oxidative stress	Significantly decreased ROS and Inhibin B and sperm linear velocity increased in the Astaxanthin group (*n* = 11), but not in the placebo group (*n* = 19). The total and per cycle pregnancy rates among the Astaxanthin group (54.5% and 23.1%) were higher compared with 10.5% and 3.6%, respectively, in the placebo cases (*p* = 0.028; *p* = 0.036). *No safety concerns were identified in this study.*
**(H) Absorption Distribution Metabolism and Excretion (ADME)**						
**Author/year/** **reference**	**Study design**	**Subjects**	**Dose ^#,##^**	**Duration**	**Major Outcome ^†^**	**Description**
Continuous dosing pharmacokinetic evaluation						
Urakaze M et al. 2021 [764]	Randomized, double-blind, placebo-controlled prospective study	44 subjects Including Prediabetes (Av. 46–48 yrs.)	0, 12 mg/day	12 weeks	✓ Type 2 diabetes ✓ Metabolic syndrome ✓ Bioavailability, ✓ ADME	Plasma AX levels were undetectable at baseline and increased to 122.69 ng/mL (*c.a.* 205 nM) after 4 weeks in the intervention group, and this level was maintained until 12 weeks. *No safety concerns were identified in this study.*
Hayashi M. et al. 2020 [799]	Randomized, double-blind, placebo-controlled prospective study	54 healthy subjects (Av. 45–46 yrs.)	0, 12 mg/day from *Paracoccus*	8 weeks	✓ Stress/Mood ✓ Sleep Quality ✓ Bioavailability, ✓ ADME	After 8 weeks of AX administration, the serum AX concentration increased to 0.171 ± 0.082 μg/mL (0.29 μM). Not detected in the placebo group. *No safety concerns were identified in this study.*
Hayashi M. et al. 2018 [793]	Randomized, double-blind, placebo-controlled prospective study	54 healthy subjects (45–64 yrs.)	0, 8 mg/day from *Paracoccus*	8 weeks	✓ Cognitive function ✓ Bioavailability, ✓ BADME	After 8 weeks of AX administration, the serum AX concentration increased to 0.173 ± 0.058 μg/mL (0.29 μM). Not detected in the placebo group. *No safety concerns were identified in this study.*
Petyaev I.M., et al. 2018 [719]	Randomized, blinded, four-arm, prospective study	32 subjects with oxidative stress, 8 subjects taking AX only, (60–70 yrs)	0, 7 mg/day * with DC	4 weeks	✓ Oxidative Stress ✓ (Elderly subjects) ✓ Bioavailability, ✓ ADME	Plasma AX concertation was seen in the volunteers supplemented with a lycosomal formulation of dark chocolate (DC) containing 7 mg of co-crystalized AX (L-DC-ASTX). Increase in plasma AX concertation. Plasma AX concentrations after 4 weeks were higher in the L-DC-ASTX formulation than other groups (5.22–7.31 nmol/L vs. 17.34 nmol/L ). *No safety concerns were identified in this study.*
Coombes J.S et al. 2016 [695] Fassett, R.G. et al. 2008, [696]	Randomized, double-blind, placebo-controlled prospective study	58 renal transplant recipients	0, 12 mg/day	12 months	✓ Oxidative stress ✓ Vascular health ✓ Bioavailability ✓ ADME ✓ (in renal transplantation)	Plasma AX concentrations were 0.29 ± 0.18 μmol/L at 6 months and 0.28 ± 0.17 μmol/L at 12 months (Mean ± SD). (The XANTHIN trial) *No safety concerns were identified in this study.*
Miyazawa T. et al. 2011 [730]	Randomized, double-blind, placebo-controlled, prospective study	30 middle- aged & senior subjects (Mean: 50.6 yrs.)	0, 1, 3 mg/day	12 weeks	✓ Bioavailability, ✓ ADME	Erythrocyte AX concentrations after 4 or 12 weeks of supplementation (3 mg/day administration, 2.5 nM AX in packed cells for 4 weeks, and 2.9 nM for 4 weeks respectively) were significantly higher than after placebo or 1 mg AX supplementation. *No safety concerns were identified in this study.*
Miyazawa T. et al. 2011 [568]	Randomized, double-blind, placebo-controlled, prospective study	20 middle- aged & senior subjects (Mean: 50.6 yrs.)	1, 3 mg/day	12 weeks	✓ Bioavailability, ✓ ADME	Plasma AX concentrations were significantly higher in a dose-dependent manner after AX supplementation than before in both the 1 mg/day (0.4, 12.3, and 18.9 nM for 0, 4, and 12 weeks after administration, respectively) and 3 mg/day (0.7, 14.4, and 62.4 nM for 0, 4, and 12 weeks after administration, respectively) groups. *No safety concerns were identified in this study.*
Nakagawa K. et al. 2011 [721]	Randomized, double-blind, placebo-controlled prospective study	30 healthy subjects	0, 6, 12 mg/day	12 weeks	✓ Oxidative Stress ✓ Bioavailability, ✓ ADME	6mg/day: Increased AX concentration in plasma after 12 weeks (86 nM) compared to baseline (*p* < 0.01, 6 to 9 nM) and compared to placebo (*p* < 0.01, 8 nM). 12mg/day: Increased AX concentration in plasma after 12 weeks (109 nM) compared to baseline (*p* < 0.01, 8 to 9 nM) and compared to placebo (*p* < 0.01, 8 nM) *No safety concerns were identified in this study.*
Park J.S. et al. 2010 [732]	Randomized, double-blind, placebo- controlled, prospective study	42 healthy subjects	0, 2, 8 mg/day	8 weeks	✓ Oxidative Stress ✓ Immunity ✓ Bioavailability, ✓ ADME	Plasma AX concentration From 4 weeks after administration, *c.a.* 0.1 µM (2 mg AX/day), *c.a.* 0.12–0.14 µM (8 mg AX/day) *No safety concerns were identified in this study.*
Uchiyama A. et al. 2008 [775]	Open-labeled, prospective study	17 subjects at risk for developing metabolic syndrome	8 mg twice day	3 months	✓ Cardiovascular health ✓ Metabolic syndrome ✓ Bioavailability, ✓ ADME	The blood concentration reached the plateau after a month of treatment and was retained at that level unti1 3 months of treatment (0.2–0.25 μg/mL (0.34–0.42 μmol/L)). *No safety concerns were identified in this study.*
Karppi, J. et al. 2007 [733]	Randomized, double-blind, placebo-controlled, prospective study	39 healthy subjects	0, 8 mg/day	3 months	✓ Oxidative Stress ✓ Bioavailability ✓ ADME	AX supplementation elevated plasma AX levels to 0.032 nM (*c.a.* 0.02μg/mL, *p* < 0.001 for the change compared with the placebo group). *No safety concerns were identified in this study.*
Nagaki Y. et al. 2005 [684]	Randomized, placebo-controlled, prospective study	36 health subjects (*c.a.* 41 yrs.)	0, 6 mg/day	4 weeks	✓ Eye Health ✓ ADME ✓ Cardiovascular health ✓ (Microcirculatory flow)	The fasting plasma AX level in the AX group was significantly (*p* < 0.001, 35.6 ng/mL at 4 weeks) higher than before supplementation. The fasting plasma AX level in the placebo group after placebo treatment remained unchanged. *No safety concerns were identified in this study.* [Article in Japanese]
Single dosing pharmacokinetic evaluation						
Madhavi D. et al. 2018 [806]	Open-label, crossover study	6 healthy subjects	60 mg	Single dosing	✓ Bioavailability, ✓ ADME	*Test articles: AX-SR; 2.5% AX in a sustained-release matrix: AX (control); 10% AX (unformulated AX oil).* Dissolution study: AX-SR formulation formed a stable dispersion in simulated gastric and intestinal fluids. Single-dose pharmacokinetic study (clinical study); AX-SR formulation (AUC_0–24h_: 4393 ± 869 ng/mL-h) showed 3.6 times higher bioavailability than AX oil (AUC_0–24h_: 1227 ± 1328 ng/mL-h) (*p* < 0.0005). AX oil showed very low absorption in 3 of 6 subjects, whereas the AX-SR formulation appeared in the blood of all subjects. *No safety concerns were identified in this study.*
Okada Y. et al. 2009 [807]	Open-labeled, prospective study	Healthy subjects; before the meal to nonsmokers (n = 7), after the meal to nonsmokers (n = 6), and after the meal to smokers (n = 7)	48 mg	Single dosing	✓ Bioavailability, ✓ ADME	Dosing timing significantly affected AX bioavailability, including area under the curve (AUC_0–168h_, 2968 ± 959 μg·h/L in the pre-meal group and 7219 ± 3118 μg·h/L in the after-meal group), suggesting a higher availability in the after-meal group. Smoking also affected pharmacokinetic parameters, significantly reducing the elimination half-life (t1⁄2) of AX. *No safety concerns were identified in this study.*
Coral-Hinostroza G.N. et al. 2004 [808]	Open-labeled, prospective study	3 male subjects (41–50 yrs, BMI 27.7–27.8)	10, 100 mg	Single dose	✓ ADME ✓ (*R/S, E/Z,* esters)	*Test meal: Dissolved the oily AX diesters in warm olive oil (30% of meal weight) and served as a dressing to a pasta salad.* 100 mg (C_max_: 0.28 ± 0.1 mg/L, T_max_: 11.5 ± 1.2 hr, T_1/2_: 52.2 ± 39.5 hr. AUC_(0–∞)_ 11.0 ± 2.8 mg h/L) C_max_ at the low dose (10 mg): 0.08 mg/L (non-linear dose response). AX esters were not detected in plasma. *No safety concerns were identified in this study*
Mercke Odeberg J. et al. 2003 [809]	Open-labeled, prospective study	32 healthy male Subjects (nonsmoker/no medications)	40 mg	Single dose	✓ ADME ✓ (Lipid formulation)	*Test meal: Haematococcus algal meal and dextrin in hard gelatin capsules (reference) and with long-chain triglyceride (palm oil) and polysorbate 80 (formulation A), glycerol mono- and dioleate and polysorbate 80, (formulation B), and glycerol mono- and dioleate, polysorbate 80 and sorbitan monooleate (formulation C).* The highest bioavailability was observed with formulation B (3.7 times higher UC_(0–∞)_ compared to the reference control). *No safety concerns were identified in this study.*
Østerlie M. et al. 2000 [810]	Open-labeled, prospective study	3 male subjects (37–43 years, BMI 27.5–31.7)	100 mg	Single dose	✓ ADME ✓ (*R/S, E/Z*)	*Test meal: prepared by dispersing the AX beadlets in warm water (40 °C) and mixing with olive oil (50% of meal weight) and cereals.* C_max_: 1.3 ± 0.1 mg/L, Tmax: 6.7 ± 1.2 hr, T_1/2_: 21 ± 11 hr, AUC_(0–∞)_ 42 ± 3 mg h/L. Accumulated 13*Z*-AX selectively, whereas the steroisomer distribution was similar to that of the experimental meal. AX was present mainly in VLDL containing chylomicrons (36–64% of total AX), whereas LDL and HDL contained 29% and 24% of total AX, respectively. The AX isomer distribution in plasma, VLDL/CM, LDL, and HDL were not affected by time. *No safety concerns were identified in this study.*
**(I) Others**						
**Author/Year/** **Reference**	**Study Design**	**Subjects**	**Dose ^#,##^**	**Duration**	**Major Outcome ^†^**	**Description**
Continuous dosing pharmacokinetic evaluation						
Ledda A. et al. 2017 [724]	Open-labeled, two-arm prospective study	59 patients with genitourinary cancers (prostate or bladder malignancies) who had undergone and completed cancer treatments (radiotherapy, chemotherapy or intravesical immunotherapy with increased oxidative stress and residual symptoms)	0, 8 mg/day *	6 weeks	✓ Improving QOL in ✓ cancer therapy ✓ Oxidative stress ✓ (Cancer therapy)	*Oncotris: containing 264 mg/day curcumin, 500 mg/day extract of cordyceps, and 8 mg/day AX (from EP217785227).* Signs and symptoms (treatment-related) and the intensity of residual side effects were significantly reduced after 6 weeks in the supplemented group: minimal changes were seen in the control group. Oncotris supplementation was associated with significant improvements in blood cell counts and reductions in levels of plasma PSA and oxidative stress. *No safety concerns were identified in this study.*
Kaneko M et al. 2017 [811]	Open-label, prospective study	10 male subjects	24 mg/day	28 days	✓ Vocal fold health	Aerodynamic assessment, acoustic analysis, and the GRBAS scale (grade, roughness, breathiness, asthenia, and strain) were significantly worse in the no AX status (day 0) immediately after vocal loading but improved by 30 min after loading. On the other hand, in AST(+) (day 35), no statistical worsening of any of the phonatory parameters was observed when measured immediately after the vocal load. *No safety concerns were identified in this study.*
Yagi H. et al. 2013 [725]	Case reports	34 OAB patients with anticholinergic agent-resistant (75.5 ± 8.0 years)	0, 12 mg/day *	8 weeks	✓ Overactive bladder ✓ (OAB)	Significantly improved international prostate symptom score (IPSS), QOL scores, benign prostatic hyperplasia impact index (BII) scores, and urinary 8-OHdG in patients AX could improve both urinary symptoms and QOL for anticholinergic agent-resistant OAB. *No safety concerns were identified in this study.* [Article in Japanese]
Park J.S. et al. 2010 [732]	Randomized, double-blind, placebo- controlled, prospective study	42 healthy subjects	0, 2, 8 mg/day	8 weeks	✓ Oxidative Stress ✓ Immunity ✓ Bioavailability, ✓ ADME	Immunity Increased natural killer (NK) cell cytotoxic activity and increased total T and B cell subpopulations; however, did not influence populations of T_h_, T_cytotoxic_ or NK cells. No difference in TNF-α and IL-2 concentrations, but plasma IFN-γ and IL-6 increased after 8 weeks in subjects given 8 mg AX. *No safety concerns were identified in this study.*
Yamada T. et al. 2010 [727]	Open-labeled, prospective study	6 healthy subjects and 6 Sjoegren’s syndrome (SS) subjects	12 mg/day	2 weeks	✓ Oxidative Stress ✓ Immunity	Although the increased amount of salivary secretion after the intake of AX for 2 weeks was faint in the SS group (1.02 g/2 min to 1.04 g/2 min, *p* = 0.69), a significant increase was also observed in the normal group (6.23 g/2 min to 7.02 g/2 min, *p* = 0.03). reduced protein oxidation (−10%, *p* < 0.05) *No safety concerns were identified in this study.*
Kupcinskas L. et al. 2008 [812]	Randomized, single-blind, placebo-controlled prospective study	131 patients with functional dyspepsia with or without *Helicobacter pylori* infection	0, 8, 20 mg/twice a day (0,16, 40 mg/day)	4 weeks	✓ GI health	No statistically significant differences were observed between the three treatment groups at the end of treatment (week 4) for mean scores on the Gastrointestinal Symptom Rating Scale (GSRS) for abdominal pain, indigestion, and reflux syndrome, and similar results were observed at the end of follow-up (week 8). The reflux syndrome tended to improve at the higher dose (40 mg) compared to the other treatment groups (16 mg and placebo), and this response was more pronounced (*p* = 0.04) in *H.pylori*-infected patients. *No safety concerns were identified in this study.*
Andersen LP. et al. 2007 [813]	Randomized, double-blind, placebo-controlled, prospective study	44 patients with functional dyspepsia with or without *Helicobacter pylori* infection	0, 20 mg/twice a day (0, 40 mg/day)	4 weeks	✓ GI health	There were no significant changes in either H. pylori density or interleukins during or after treatment; significant upregulation of CD4 (*p* < 0.05) and downregulation of CD8 (*p* < 0.001) were observed in *H. pylori* patients treated with AX. *No safety concerns were identified in this study.*

This table was updated and prepared from Ref. [130] to May 2023. For statistical analysis of efficacy, most analyses were performed with “Per Protocol Set”, thus the number of subjects adopted for the final analysis is shown. This table sorts each clinical trial by category. As a result, some of the trial outcomes have been multiplied and listed. Tables without overlap are summarized in Appendix A. ^†^ With the exception of the study in which safety was the only primary outcome, the other outcomes were noted. * In addition to AX, other nutrients, such as antioxidants, were used in the study. # All the studies in the “Dose” column that did not indicate the origin were conducted using AX derived from *Haematococcus* algae. ^##^ AX concentrations were shown in free form.

## 4. Advantages of Astaxanthin from *Haematococcus* Algae

### 4.1. Absorption and Bioavailability

#### 4.1.1. General Aspects of AX Absorption Dynamics

Several studies summarized in Table 4 measured concentrations of AX in blood during clinical trials [568,696,719,721,730,732,733,764,775,793,799]. These results revealed that AX seemed to circulate in its free form in the blood, whether it took in esters with free fatty acids or the free form from a dietary source, similar to other xanthophylls. This is because the fatty acid esters of xanthophylls undergo hydrolysis of the side chains during digestion and absorption by esterases [814,815,816]. However, since there has been partial re-esterification of AX in studies with Caco-2, a cell line derived from human colon adenocarcinoma, further studies will be needed to determine where ester forms undergo hydrolysis [817]. Moreover, some xanthophylls are known to undergo reesterification in subcutaneous tissues [818]. However, there is no evidence of AX re-esterification in human skin tissue. 

These studies also revealed significant variation in the concentrations reported among them. This inconsistency could be attributed to variations in study protocols, such as differences in dietary conditions and time schedules for blood collection after the final intake. Race, dosage, and other genetic factors are also contributing factors to carotene absorption [819]. Nevertheless, even when accounting for these factors, the bioavailability of AX appears relatively low when considering the transition of AX concentration in the blood relative to the dosage. 

The absorption and bioavailability of AX differ between humans and other species. In humans, the AX *cis*-isomers are preferentially absorbed over the all-trans form, whereas the all-*trans* isomer is preferably absorbed in salmonid fish [808]. The rate of absorption also varies with the ingested dose and the co-ingestion of other nutrients, such as fat. Lower amounts of AX seem to be absorbed more efficiently than higher doses, as a saturation effect has been observed at higher doses [820]. As mentioned earlier, bioavailability depends not only on the dose but also on the presence of other nutrients. 

Another factor influencing AX bioavailability is certain lifestyle factors, such as smoking. Smokers generally exhibit lower levels of antioxidants due to high endogenous consumption, and long-term smoking is believed to deplete endogenous antioxidant levels. Smoking also reduces the half-life of AX in the blood [807].

#### 4.1.2. Molecular Mechanism of Intestinal Absorption of AX

When looking at the molecular mechanism of absorption of AX in mammals, although the mechanism specific to AX absorption is not yet known, it seems reasonable to consider similar modes of intestinal absorption as other carotenoids, cholesterol, and fat-soluble vitamins. AX from foods and dietary supplements undergoes digestion by the gastrointestinal tract, with a part of it considered absorbable in the small intestine and the remaining excreted through feces. Absorption of AX in the intestinal epithelium (enterocyte brush border) from the intestinal lumen includes both uptake by simple diffusion of micelles in solubilization with bile salts, digested lipids, and proteins (caused by concentration differences between the gastrointestinal lumen and cells) and passive diffusion (active transport) via lipid transporters such as Niemann-Pick C1 like intracellular cholesterol transporter 1 (NPC1L1), cluster of differentiation 36 (CD36), and SCARB1 like other animals. Polymorphisms in these genes are known to affect blood levels of carotenoids. [816,819,821] (Section 2.3.3). 

After incorporation into the enterocytes, carotenoids become packed into chylomicrons and thereafter distribute systemically [821]. This is consistent with the fact that in humans, orally administered AX appears more frequently in the chylomicron fraction of the circulating plasma [808,810]. Interestingly, polymorphisms in genes involved in the formation and secretion of chylomicrons are known to affect blood carotenoid concentrations (i.e., *ABCA1*, *LPL*, *INSIG2*, *SLC27A6*, *LIPC*, *MTP*, and *APOB*) [819,822,823]. Furthermore, with regard to intracellular to basolateral secretion, genetic polymorphisms in ABCA1 mediating the transfer of absorbed cholesterol to HDL have also indicated a possible role in carotenoid blood concentrations [819].

At this point, it is also important to consider the reverse transport (efflux transport) of fat-soluble vitamins and carotenoids from the enterocytes to the lumen. The ATP-binding cassette subfamily G members (ABCGs) are responsible for the major players. For example, the uptake of chicken lutein into egg yolk (*ABCG5*) [824] and genetic polymorphisms of several transporters in humans are known to affect blood xanthophyll levels (*ABCG5/8*) [819]. Recently, genetic polymorphisms in *ABCG2*, a typical efflux transporter, have also been implicated in blood levels of lutein [822]. Unfortunately, the relationship between the function of these ABCG transporters and the bioavailability of AX remains unclear. 

Generally, carotenoids that have provitamin A activity are taken up by intestinal epithelial cells in the above process; however, a partial portion of them are metabolized into retinoids by BCO1. Retinoids are metabolized to retinoic acid via enzymatic reactions, which bind to RARs and RXRs and activate the transcription factor intestine-specific homeobox (ISX), which suppresses transcription of BCO1 and SCARB1. Thus, the systemic concentration of carotenoids is also regulated by the function of these gatekeeper molecules [825]. 

Although xanthophylls such as AX are not substrates for BCO1, they are cleaved asymmetrically by BCO2 at the C9′ and C10′ positions of the double bond, producing apocarotenoids at mitochondria in various tissues [821]. BCO2 is strongly expressed in the enterocytes of the intestine and liver. Since AX would also be recognized as a substrate for BCO2 [821], it is possible that a portion of AX is degraded not only in peripheral tissues such as the liver but also in the intestinal tract during absorption. Especially in rodents such as rats, the expression of BCO2 in the enterocytes of the small intestine is remarkably higher than in humans. Therefore, it also seems that this is one of the reasons for the difference in blood concentration between humans and rats (unpublished data from Prof. Johannes von Lintig, Plenary Lectures, “Session 6: Carotenoid Metabolism and Function, Molecular Components Affecting Carotenoid Homeostasis in Mammalian Cells and Tissues” at the 19th International Symposium on Carotenoids, Toyama, Japan, 14 July 2023). Perhaps, like other carotenoids [826,827,828,829], the apocarotenoid degradation products of AX may have unique bioactivity. Therefore, research has been conducted in recent years [830]. Further studies are expected to help understand the extended biological function of AX. 

After absorption from the intestinal tract, intact carotenoids are transported to the liver, mainly by chylomicrons, where they are further metabolized and reorganized into lipoprotein complexes such as HDL, LDL, and VLDL, which circulate systemically. Therefore, polymorphisms in genes such as *LPL*, *APOA1, APOA4, APOB, LIPC*, and *CETP* that comprise them could also affect the concentration and distribution of carotenoids in the blood [819].

#### 4.1.3. Challenges in Improving Low Bioavailability

Generally, AX and all other carotenoids have low oral bioavailability due to their poorly soluble properties. Since AX has demonstrated superior antioxidant capacity, higher than other carotenoids (such as lutein or zeaxanthin), there is a growing interest in developing more bioavailable AX formulations. Technological advancements are being applied to enhance the bioaccessibility and bioavailability of this carotenoid, including oil-in-water emulsions, nano-emulsions, microencapsulation, and liposomes. It is believed that fat increases AXs intestinal absorption. Experimental and *in vitro* studies have shown increased bioavailability of AX when administered in emulsified lipid formulations, with reduced droplet size enhancing AX bioavailability in rats (*Rattus norvegicus*) [809,831,832]. Nano-dispersions containing selected emulsifiers and carriers, such as polysorbate, sucrose esters, whey protein, plant-based proteins, and/or lecithin, have the potential to enhance absorption and provide a promising solution to the issue of low bioavailability [833,834,835,836]. In addition to increased absorption in the intestine, small nanoparticles with emulsifiers likely improve cellular uptake and enhance AX bioavailability in target tissues. Some of these nanodispersions also offer slow intestinal release of AX, improving absorption among individuals with poor absorption capabilities [806].

The main challenge lies in selecting the optimal emulsifier with the appropriate properties to meet chemical and physiological requirements, including high emulsifying properties, digestibility, pH stability, biodegradability, and non-toxicity. Another challenge is to counteract the degradation of AX itself. The addition of other antioxidants, such as ascorbic acid and α-tocopherols, to nanodispersions has shown promise in improving the chemical stability of AX [834].

### 4.2. Safety of Haematococcus Astaxanthin

AX and its precursors are produced by bacteria, yeasts, mites, and algae, and they become incorporated into various edible organisms through the food chain, as discussed comprehensively in Section 2. Human beings, who commonly consume red-colored aquatic products, have extensive dietary exposure to naturally occurring AX. Furthermore, chemically synthesized AX has been used in some parts of the world for many years as a feed additive in the fisheries industry to color aquacultured fish. As aquacultured products are widely consumed by humans, the Food Safety Commission of Japan has submitted a report to the Minister of Health, Labour, and Welfare and the Minister of Agriculture, Forestry, and Fisheries, presenting the findings of an assessment regarding the health effects of AX in food. No serious safety concerns were observed in the safety evaluation of AX, either in the form that is naturally present and consumed in daily foods or as a feed additive. No Acceptable Daily Intake (ADI) levels have yet been established [689]. 

#### 4.2.1. Genotoxicity

In food toxicology evaluations, it is common to conduct *in vitro* studies when alternative methods are available and *in vivo* animal studies when *in vivo* evaluation is necessary. Commonly performed studies include genotoxicity assessments such as the Ames test, micronucleus test, and chromosomal aberration test, as well as acute toxicity studies in rodents and sub-chronic toxicity studies. Several animal studies have evaluated the potential toxicity of AX, whether as a synthetic or *Haemotococcus*-derived compound. AX, in its natural algal biomass, natural oil/extract, and synthetic form, has been evaluated for genotoxicity, acute toxicity, and sub-chronic toxicity using both *in vitro* and animal models. 

In the Ames test (reverse mutation test) using *E. coli* and *Salmonella typhimurium*, colony formation was observed upon the addition of AX extract (8.6–275 mg/plate as AX) from *Haematococcus* algae (5% AX oil, AstaReal oil 50F). No increase in colonies caused by mutation was observed [837]. Other studies have reported the use of five strains of *S. typhimurium* (TA1535, TA1537, TA1538, TA98, and TA100) in reverse mutation models, with AX concentrations ranging from 0.03 to 5 mg/plate. Regardless of metabolic activation by the addition of rat liver homogenate fraction (S9-Mix), no mutation was induced in any of the five strains. Therefore, no mutagenicity was observed [689]. An Ames study conducted under GLP compliance also exists. In this study, a dose-finding study was initially performed using *S. typhimurium* (TA1535, TA1537, TA98, TA100) and *E. coli* WP2uvrA, both in the presence and absence of 5% S9 fraction, with a maximum dose of 5000 μg/plate of AX. Due to precipitation of AX at 3333 and 5000 μg/plate, subsequent treatments with 10, 33, 100, 333, and 1000 μg/plate of AX in the presence and absence of 10% S9 fraction showed similar results to the solvent control. No biologically significant reversion mutations were observed at any concentration [838].

In the micronucleus test conducted on mice, a single oral dose of *Haematococcus* algal extract (25, 50, or 100 mg/kg body weight (bw) as AX) was administered. The bone marrow cells were assessed for the formation of micronuclei (lesion nuclei) at 24, 48, and 72 h after administration of AX. No micronucleus formation was observed [839]. Similarly, in a chromosome aberration test using cultured mammalian cells, treatment with *Haematococcus* algal extract and subsequent observation of chromosome aberrations showed no significant difference in the frequency of cells with chromosome aberrations compared to the control group [839]. In another study, AX was orally administered to Füllinsdorf albino mice at doses of 500, 1000, and 2000 mg/kg body weight, twice with a 30-h interval before sample preparation. No induction of chromosome breaks or mitotic non-disjunctions in bone marrow cells was observed [689]. 

A micronucleus test conducted under GLP compliance has also been reported. In the initial dose-finding study, AX precipitated in the culture medium at a concentration of 333 μg/mL. Subsequently, concentrations of 10, 33, and 100 μg/mL were selected for the first 3-h exposure assay. At a concentration of 100 μg/mL, significant or biologically increased precipitation of mononuclear cells with micronuclei and binuclear cells was observed, both in the absence and presence of S9. In a second experiment with a 24-h incubation time without metabolic activation, doses of 33, 100, and 333 μg/mL of AX were selected. Precipitation of AX was observed at 100 and 333 μg/mL. The test items induced a concentration-independent and statistically insignificant increase in micronuclei in binuclear cells, as well as a significant increase in micronuclei in mononuclear cells. However, the increase observed in mononuclear cells fell within the acceptable limits of the test. Due to the relatively high number of micronuclei observed in all groups (including solvent controls) in this cytogenetic assay, the experiment was repeated to confirm the results. The repeated experiment concluded that AX did not induce statistically significant or biologically relevant increases in the number of mononuclear and binuclear cells with micronuclei [838].

#### 4.2.2. Acute and Sub-Chronic Toxicity

An acute and sub-chronic toxicity study was conducted in rodents that were administered AX. To evaluate acute toxicity, Sprague-Dawley rats were administered a 20% intralipid solution with the biomass derived from *Haematococcus* algae at the maximum administrable dose of 12,000 mg/ kg-bw. This study reported an oral LD_50_ greater than 12,000 mg/kg-bw [840]. 

Additionally, Takahashi et al. reported in 2004 [837], that AX oil was administered by gavage to Sprague-Dawley rats at a dose of 2000 mg/ kg-bw (containing 5% AX and AstaReal oil 50F derived from *Haematococcus* algae). The rats were observed for a period of 14 days, and an LD_50_ greater than 2000 mg/kg was established, indicating low acute toxicity. In another study, Niu T. et al. reported that in a 14-day observation acute toxicity study in ICR mice of 10% AX oleoresin from *Haematococcus* algae, the oral LD_50_ of AX was greater than the maximum dose of 20,000 mg/kg-bw [841]. Furthermore, AX was suspended in peanut oil and orally administered to Füllinsdorf albino rats (gender information not provided, 10 rats per group) at doses of 0, 125, 250, 500, 1000, and 2000 mg/kg-bw once daily for 10 days. This study examined general symptoms, body weight changes, and mortality. The LD_50_ value of AX was determined to be greater than 2000 mg AX/ kg-bw, indicating low acute toxicity [689]. In summary, the acute toxicity of AX itself is quite low. 

In the sub-chronic toxicity study, Wistar rats were fed diets containing up to 20% of the biomass (weight/weight) for a duration of 90 days. The NOAEL (No Observed Adverse Effect Level) was determined to be 465 mg AX/kg-bw/day for male rats and 557 mg AX/kg-bw/day for female rats [840]. Takahashi et al., also reported the 90-day oral subchronic dose study in male and female Crj:CD(SD)IGS rats administrated at 0, 37.0, 185.2, and 925.9 mg (0, 2, 10, and 50 mg AX, respectively)/kg-bw/day (containing 5% AX and AstaReal oil 50F derived from *Haematococcus* algae) [837]. There were no changes related to exposure to the AX-rich extract in body weight gain, food consumption, or necropsy. In hematological analysis, there was an increase in prothrombin time or activated partial thromboplastin time in the male 925.9 mg group, but their differences were slight and not defined as an adverse effect. No abnormal changes related to the treatment were observed in the serum biochemical analyses, organ weight, or histopathological examinations. Hence, the NOAEL for oleoresin from *Haematococcus* algae was estimated to be 925.9 mg (50 mg AX)/kg-bw/day in this study.

Taken together, by applying a safety factor of 100, the estimated ADI (Acceptable Daily Intake) values of *Haematococcus* algae products were set at at least more than 30 mg AX/day for a nominal 60 kg person.

In a sub-chronic toxicity study involving other AX sources except *Heamatococcus* algae, oral dosages of AX extract from *Paracoccus carotinifaciens* were administered, and a slightly higher NOAEL of at least 1000 mg/kg-bw/day for a duration of 13 weeks was reported for male and female Sprague-Dawley rats, respectively [842]. Sub-chronic toxicity studies conducted with synthetic AX in rats have yielded similar outcomes. Wistar rats were fed a diet containing up to 50,000 ppm AX, resulting in a NOAEL of 700 mg AX/kg-bw/day for male rats and 920 mg AX/kg-bw/day for female rats [843]. Another study evaluating the oral administration of synthetic AX to rats showed consistent results. Hanlbm Wistar rats received synthetic AX at doses of up to 1033 mg/kg-bw/day for a duration of 13 weeks, and a NOAEL of 1033 mg AX/kg-bw/day was established [844]. In another study involving Füllinsdorf albino rats (16 males and 16 females per group), mixed feeding at doses of 0, 310, 620, and 1240 mg AX/kg-bw body weight/day was administered for 13 weeks. No abnormalities in organ weights or histological findings were observed. However, hair loss was observed in both the control group and the test group. The treatment group received AX in a granular formulation coated with water-soluble cornstarch and mixed in the diet, while the control group received placebo granules to maintain the same amount of feed given as the treatment group. The upper limit for both granule doses was 25% of the total diet, and the observed alopecia was attributed to malnutrition rather than the administration of AX. It was concluded that the alopecia observed in all groups was not related to AX administration [689]. In a 3-month repeated-dose study involving beagle dogs (3 dogs per group, both male and female), no abnormalities were observed except for orange-yellowing of adipose tissue derived from AX in the AX group after mixed feeding at all doses of 40, 80, and 160 mg AX/kg-bw/day for 3 months [689].

#### 4.2.3. Reproductive and Developmental Toxicity

Another study investigated the effects of AX from *Haematococcus* algae on prenatal developmental toxicity in Wistar rats. Female rats were given a diet containing 0.02% AX and mated with male rats, and the number of births and malformations was compared to the control group. Based on the body weight at the time of mating (about 200 g) and food intake (about 15 g/day), the estimated AX dose that did not affect pregnancy and delivery in rats was 15 mg/kg-bw [845]. Another study evaluated the effects of dietary AX on the reproductive and developmental toxicity of Hanlbm Wistar (SPF) rats. No effects were observed in the offspring of female rats exposed to up to 1.39% AX in the diet during organogenesis (gestation days 7–16). Fecal discoloration and yellow pigmentation of adipose tissue were observed in the 13-week study; however, these were determined to be specific properties of the substance and not signs of toxicity. The authors considered the highest dietary concentration of AX in each study as the NOAEL, which corresponded to 1.52% and an average intake of 1033 mg/kg-bw/day (range: 880–1240 mg/kg-bw/day) in the 13-week study and 1.39% and an average intake of approximately 830 mg/kg-bw/day (range: 457–957 mg/kg-bw/day) in the developmental toxicity study [844].

In a study on Füllinsdorf albino rats (40 females per group), doses of 0, 250, 500, and 1000 mg/kg-bw/day were administered via mixed feeding on gestation days 7 through 16. No abnormalities were observed in either maternal or fetal animals at the highest dose [689]. In a study with Swiss rabbits (20 females per group), no abnormalities were observed in either maternal or fetal animals after forced oral administration at doses of 0, 100, 200, and 400 mg/kg-bw/day on gestation days 7 through 19. Partial loss of the medulla oblongata was observed in some of the animals during the evaluation examination using the Wilson method. However, since the medulla oblongata is fragile and artificially damaged during head amputation and similar defects were observed in the control group, it was concluded that this was not caused by AX administration. Therefore, the NOAEL in this study was estimated to be >400 mg/kg-bw/day [689].

Taken together, no reproductive or developmental influences were observed at the highest doses in all of the studies. 

#### 4.2.4. Toxicokinetics and Liver Toxicity

The tissue distribution and accumulation of AX following oral administration are likely to be dose-dependent. A dose-response study conducted in rats involving oral administration of AX for two weeks indicated a rapid elimination or catabolism of AX. High concentrations of AX were observed in organs such as the spleen, kidneys, and adrenals. In line with previous reports, AX accumulated in the eye in a dose-dependent manner, similar to cantaxanthin. The highest concentrations of AX were detected in the hair and tail [846].

It is well known that AX and other carotenoids can modulate hepatic xenobiotic-metabolizing phase I enzymes in some species of rodents. While hepatic metabolization is considered the primary xenobiotic pathway, induction of Phase I enzymes in extrahepatic tissues, such as the lung and kidneys, has also been reported in rats [847,848]. Previous studies have indicated that AX induces the hepatic P450 enzyme CYP1A1/2 in rats. However, this CYP gene induction has not been observed in mouse models or human hepatocytes [326,847,849,850,851]. An *in vitro* study using pooled human microsomes confirmed these findings and showed only weak inhibition of the CYP 2C19 enzyme [852]. Consequently, AX supplementation is unlikely to affect Phase I drug metabolism in humans and is likely safe to use in clinical trials. The metabolism of AX and CYP induction appears to be species-specific, and concerns for humans are considered to be minor, although further confirmations remain required.

In 2016, Edwards et al. conducted a review of the genotoxicity and rat carcinogenicity investigations with AX [838]. The review discussed the results of three specific animal studies conducted by DSM in 2003, with a focus on a 2-year rat carcinogenesis study that examined the effects of synthetic AX. In this study, female rats in the AX-treated group exhibited an increased incidence of hepatocellular damage and hepatocellular adenoma at doses of 200 and 1000 mg AX/kg-bw/day [838]. Since the NOAEL could not be determined in this carcinogenicity study, based on this weak hepatic influence, the ADI in humans was set at 0.2 mg/kg-bw/day in the EU [853]. However, it was concluded that this effect was secondary to the slight hepatotoxicity and associated tissue regeneration at the respective doses. It is important to note that the hepatocellular adenomas were observed only in female rats, and no tumors were found in any of the other groups. Taking all these factors into consideration, the authors concluded that the hepatocellular adenomas were most likely species-specific and not relevant to humans [838]. In fact, this rodent-specific chronic hepatotoxic effect is also observed in other carotenoids, such as lycopene. ADI in humans was set at 0.5 mg/kg bw/day in the EU [854]. For other carotenoids for which there are no results of chronic toxicity studies, such as carcinogenicity in rodents, e.g., lutein and zeaxanthin, the NOAEL was dependent on the results of subchronic toxicity studies, and the ADI was set much higher, 0.75–1 mg/kg-bw/day [855,856]. Therefore, it is reasonable to assume that the chronic hepatotoxicity of AX would be similar in quality and level to that of other carotenoids ingested in large daily doses and that it is probably safe at the doses used in the clinical trials. 

Furthermore, there is evidence suggesting the inhibition of tumor cell invasion in the liver by AX and other carotenoids. *In vitro* studies have reported that AX inhibits proliferation and induces apoptosis in hepatic carcinoma cells [857,858,859]. This observed effect could be a significant mechanism in preventing the spread of hepatic cancer and systemic cancer invasion.

Taking the previous discussion into account, it is worth noting that most animal studies do not indicate the induction of liver toxicity following AX supplementation. On the contrary, the majority of studies demonstrate a hepatoprotective effect [860,861,862]. Studies conducted on rats have shown that AX enhances antioxidant status and safeguards against liver damage by counteracting lipid and protein free radicals generated by oxidative stress [860,863]. These findings suggest that AX provides protection by inhibiting lipid peroxidation and stimulating the cellular antioxidant system. The underlying mechanism is believed to involve the induction of hepatic antioxidants such as superoxide dismutase (SOD), catalase, and glutathione peroxidase while reducing pro-inflammatory mediators such as nitric oxide (NO), interleukin-6 (IL-6), and tumor necrosis factor-alpha (TNF-α) [864,865]. Moreover, natural AX complexes derived from microalgae may enhance hepatic antioxidant capacity more effectively than synthetic AX [862].

*In vitro* studies suggest that AX has the potential to protect against liver steatosis by preventing fat accumulation in hepatocytes [866,867], It has been reported that AX may reduce hepatic lipid accumulation by modulating the activity of nuclear receptors PPAR-α and PPAR-γ in a favorable manner, acting as a PPAR-α agonist and a PPAR-γ antagonist, respectively [866]. Additionally, AX has demonstrated greater efficacy in reducing lipotoxicity in hepatocytes *in vitro* compared to both vitamin C and *N*-acetylcysteine [867]. These findings suggest that AX could be a promising nutraceutical candidate for controlling plasma lipid concentrations and managing hepatic steatosis in the future. 

Mouse studies have demonstrated that AX has a protective effect against liver injury in mice with induced autoimmune hepatitis by downregulating apoptosis and autophagy of hepatic cells. This positive effect is likely attributed to the inhibition of the release of the pro-inflammatory cytokine TNF-α from damaged and stressed cells [868]. Additionally, the suppression of autophagy may play a crucial role in preventing the progression of liver fibrosis [869]. These findings suggest that AX holds potential for mitigating liver damage and promoting liver health by regulating cellular processes and reducing inflammation. 

#### 4.2.5. Human Safety

Human safety studies of AX have also been conducted on *Haematococcus*-derived AX oleoresin, summarized in Table 5. Short-term overdose administration studies involving daily supplementation of approximately 45 mg AX for up to 4 weeks have also reported no adverse safety effects. 

A comprehensive list of human clinical studies investigating the effects of AX includes over a hundred studies (Table 4). Although not all of these studies conducted blood count and blood biochemistry examinations, no clinical safety concerns have been reported in short-term studies using doses up to 24 mg or AX per day for 12 weeks and/or in long-term studies of more than 6 months with 12 mg AX per day in healthy subjects (Table 6). 

## 5. Prospective: The Future of Astaxanthin

### 5.1. Perspective: Health Promotion Potentials of Natural Astaxanthin

Microalgae-derived AX is gaining attention due to its unique potential. Thus, natural AX has been extensively studied for its various health benefits. It is known to be a powerful antioxidant and anti-inflammatory phytonutrient that contributes to healthy aging in several critical areas, including improved mobility, locomotive activities [718,790], enhanced vision [39,40,41,42,43,44,45,46,685,749,758,759], mental fitness, cognitive functions [700,722,723,774,793,794,795,796,797,798,799], and skin health [702,703,704,737,738,739,740,741,742,743,744,745,746] Refer to Section 3 and Table 4 for more details.

However, there is still much to explore and investigate regarding the full range of health-promoting and commercial potential of AX. One example is hyperuricemia, a condition associated with oxidative stress and complications such as gout and cardiovascular diseases. Decreased physical activity and prolonged sedentary behavior have been linked to an increased prevalence of hyperuricemia [875]. Recent studies have suggested that AX has the ability to attenuate inflammatory reactions triggered by the deposition of monosodium urate (MSU) crystals in specific joints [876]. 

Physical inactivity, coupled with age-related physiological changes, can contribute to a state of low-grade chronic inflammation known as “inflammaging.” This condition disrupts the balance between oxidants and antioxidants in the body, leading to increased production of reactive oxygen species (ROS) and proinflammatory cytokines such as IL-1β and TNF-α by chondrocytes. Consequently, it can disrupt cartilage homeostasis, contribute to articular cartilage degradation, and lead to alterations in synovial tissue, subchondral bone, and the formation of osteophytes, ultimately resulting in osteoarthritis [877]. *In vitro* and *in vivo* studies have indicated that AX may offer protection against osteoarthritis and potentially other inflammaging-related conditions, including frailty, atherosclerosis, Alzheimer’s disease, sarcopenia, type 2 diabetes, and osteoporosis. This protection is believed to occur through the inhibition of Nrf2 signaling, the alleviation of TNF-α-induced extracellular matrix degradation, and the prevention of chondrocyte apoptosis [877,878,879]. These findings suggest that AX has the potential to mitigate the progression of various inflammaging-associated pathologies.

The potential benefits of AX supplementation in improving physical performance, preventing frailty and sarcopenia, and promoting bone, joint, and muscle health are of particular interest among aging populations [718,789,790]. However, the benefits of AX are not limited to older individuals and can extend to various age groups with different professional needs. For example, white-collar workers who experience prolonged periods of sedentary behavior may benefit from AX supplementation. The lack of movement and improper body positioning in this group can contribute to the development of musculoskeletal disorders. Additionally, elevated stress levels, emotional strain, and an increased risk of obesity among white-collar workers can further raise the risk of chronic diseases [880]. While there are already promising results, further investigations are needed to determine whether incorporating AX supplementation into worksite health promotion programs could help alleviate the negative effects of sedentary behavior and improve the musculoskeletal health of deskbound professionals. 

As discussed in Section 3, AX has shown benefits in preventing or improving resilience towards physical/musculoskeletal complications, mental stress and anxiety, lack of focus or cognitive slowing, fatigue, poor sleep, and computer vision syndrome (refer to Section 3). These factors and their associated nutritional needs are relevant not only to office workers in desk-bound jobs but also to E-gamers, E-sports players, and anyone engaged in prolonged sedentary use of digital screens [881,882]. Further investigations are required to explore the efficacy of AX supplementation in managing the physical and mental needs of these groups, as well as its role in nutra-ergonomics and E-sports nutrition. 

In addition to the above, studies have shown promising results regarding the potential role of AX in enhancing wound healing, particularly in conditions such as diabetic foot ulcers or post-surgery recovery [883,884]. 

While limited, preliminary research suggests that AX supplementation may have a favorable impact on the gut microbiota, contributing to gut homeostasis. This can play an important role in suppressing gut inflammatory responses and strengthening intestinal immune function [885]. It is speculated that the unabsorbed AX remaining in the gastrointestinal tract may help protect against disturbances in gut flora caused by various factors, including a diet high in saturated fat [886].

Studies have indicated promising results regarding the potential of AX in alleviating chronic active inflammation caused by *Helicobacter pylori*, a bacterium known to be associated with gastric disorders. AX may serve as a beneficial remedy in the prevention or treatment of non-ulcer dyspepsia, chronic active gastritis, and gastric and duodenal ulcers [812,813,887]. These findings suggest that AX supplementation could have a positive impact on gastrointestinal tract disorders related to *H. pylori* infection.

Another intriguing area that could potentially benefit from natural AX supplementation is fertility and reproduction. Due to population aging and unfavorable lifestyle changes, infertility rates continue to rise, with only marginal improvements in pregnancy and birth rates after assisted reproduction treatment in developed countries [888]. The World Health Organization estimates that nearly 190 million people worldwide struggle with infertility, and the number of couples seeking medical assistance is steadily increasing [889]. 

In 2017, a meta-regression analysis of 185 studies reported a significant decline in sperm counts (measured by sperm concentration and total sperm count) between 1973 and 2011. This decline was particularly pronounced, with a 50–60% decrease among men not selected for fertility from North America, Europe, Australia, and New Zealand. On average, this represents a decline of approximately 1.6% per year in human testicular sperm production in most populations [890]. One possible contributing factor is the lack of molecular-level investigation into the causes of infertility.

Among the various types of male infertility, the most prevalent (30–50%) and troublesome one is idiopathic, which suggests a lack of an apparent cause for the decline in semen quality. The etiology of idiopathic male infertility can include environmental, dietary, medical, genetic, and physiological factors [891]. It is estimated that 30–80% of infertile men have elevated levels of seminal reactive oxygen species (ROS), a condition that can potentially be treated [892]. Spermatozoa, in particular, contain significant amounts of unsaturated fatty acids in their cell membranes, making them highly susceptible to the detrimental effects of ROS. This can lead to lipid peroxidation, resulting in increased intracellular oxidative burden, loss of membrane integrity, heightened permeability, reduced sperm motility, structural DNA damage, and apoptosis [893]. AX has recently gained attention in the management of male infertility due to its potential for maintaining mitochondrial energy output in spermatozoa and improving several fertility-related parameters. AX has been associated with lower oxidative stress, reduced sperm membrane lipid peroxidation and DNA fragmentation, increased sperm concentration and viability, improved sperm motility and membrane fluidity, enhanced sperm capacitation, as well as better sperm head-oocyte interaction and acrosomal reaction through increased Tyr-phosphorylation (Tyr-P) levels in the sperm head region [735,805,894].

A growing body of evidence from animal experiments and human observational studies suggests that long-term exposure to environmental and industrial pollutants, such as Bisphenol A (BPA), can have a negative impact on reproductive function in both men and women [895,896].

In a human clinical study, it was observed that infertile women with high concentrations of BPA in their urine exhibited a clear decrease in the number of primordial follicles [897]. In recent years, many antioxidants have been utilized to mitigate the oxidative reactions caused by BPA and to counteract reproductive failures resulting from excessive reactive oxygen species (ROS). In the case of female infertility or subfertility, preliminary yet promising results have been observed in studies involving AX. For example, AX may enhance follicle and oocyte development by increasing the antioxidant capacity of follicles and oocytes, as well as alleviating BPA-induced oxidative stress during follicular development and oocyte maturation [643]. Furthermore, AX supplementation has shown the ability to elevate serum total antioxidant capacity levels and activate the Nrf2 axis in granulosa cells of patients with polycystic ovary syndrome [651,898]. 

Metabolic syndrome is a cluster of risk factors, including hypertension, glucose intolerance, dyslipidemia, and abdominal obesity, that collectively increase the risk of coronary heart disease, diabetes, stroke, and other serious health problems. It is also referred to as insulin resistance syndrome [899,900]. One particular area that could benefit from the potential supportive and preventive effects of AX is the management of glucose and lipid levels in pre-diabetes and diabetes. By improving glucose and lipid metabolism, glycemic control, insulin sensitivity, and lipid profile, AX supplementation may provide support to individuals at risk of developing not only diabetes but also atherosclerosis and other cardiovascular complications [764,768]. 

AX itself modulates the action of insulin involved in glucose metabolism and improves physiological conditions that are fundamental to these pathological conditions, such as insulin sensitivity and insulin resistance [772,901,902]. Studies have shown that AX supplementation may increase serum adiponectin concentration and reduce visceral body fat mass, serum triglyceride and very-low-density lipoprotein cholesterol concentrations, systolic blood pressure, fructosamine concentration, and plasma glucose concentration. Additionally, it may have a favorable impact on high-density lipoprotein levels [768]. 

Preliminary pre-clinical and clinical data suggest that AX supplementation could serve as a novel complementary treatment option for the prevention of diabetes, cardiovascular diseases, and metabolic and weight management [176,698,762,772,901,903,904,905,906,907]. 

Furthermore, in the context of the NAD world, which revolves around the regulation of Sirtuins by NAD+, there has been a growing interest in their effects on longevity control and rejuvenation [908,909]. It has been reported that AX enhances the gene expression of Sirtuins, particularly mitochondrial Sirtuins [176], and increases the concentration of NAD+ in cells [130]. Additionally, AX has shown a protective effect on the microenvironment (niche) of stem cells and related cell populations, which play a role in the pathogenesis of chronic inflammation [910]. 

The possible mechanisms underlying these effects are discussed in more detail in a separate review [130]. The health-promoting potential of AX is continually being uncovered, highlighting its promising role in various aspects of well-being. In summary, the past and expected future benefits of astaxanthin are illustrated in Figure 16. 

### 5.2. Prospective: Global Demand and Social Requirements for Astaxanthin

According to the European Commission, the proportion of older people in the EU is expected to steadily increase in the coming decades. Currently, about 20% of the population is over 65 years old; however, it is projected to reach around 30% by 2070. The proportion of people above 80 years old is also expected to more than double, reaching 13% by that time [911]. Similar trends have been observed in Japan and other East Asian countries, making it a significant social concern. Japan has surpassed EU countries to become the country with the largest proportion of older people based on 2022 population statistics [912]. This demographic shift has a significant impact on people’s quality of life and society as a whole. In Japan, population projections indicate that the number of people aged 70 and over will exceed 25% by 2040, resulting in a shrinking workforce, which is recognized as a serious problem [913].

In addition, the crisis unleashed by the COVID-19 pandemic has been affecting every aspect of our lives. Given the scale, speed, and impact that societal and environmental alterations have across the globe, the Member States and the European Union are continuously reviewing, updating, and proposing new action plans to strengthen resilience and tackle uprising challenges that affect planetary health. 

There is a growing awareness among consumers of the direct link between nutrition, health, and sustainability, leading to increased interest in food products that encompass these dimensions [914]. A recent report by the United Nations highlights the harmful impact of air pollution, extreme weather conditions, and food insecurity on health. Without timely interventions, these environmental risk factors will lead to excess mortality and further endanger human physical and mental health and well-being. It emphasizes that healthy aging, both now and in the future, is closely tied to a healthy planet [915]. 

There are several pressing environmental concerns that necessitate serious preventative and sustainable actions to preserve the Earth and its inhabitants. These concerns include climate change, biodiversity loss, deforestation, traditional food scarcity, and resulting nutritional deficiencies, which particularly affect the prevalence and incidence of infectious and chronic health complications, especially among aging and vulnerable populations. Since 1950, the human population has nearly doubled, fossil fuel consumption has increased by over 550%, and marine fish capture has increased by over 350% [916].

The agriculture and aquaculture industries were originally established to enhance food security; however, their negative impact on the environment has been the subject of extensive debate. While modern policies and technological advancements have allowed for increased efficiency and reduced environmental footprints, some major concerns persist. For instance, in aquaculture, food-grade fish (and terrestrial food-grade resources) are still commonly used as feed [917]. However, consumer awareness and market demands are growing for more sustainable, natural, and health-promoting alternatives to fish meal, including “non-synthetic” feed ingredients. AX-containing algal meal, which is rich in AX, is considered a natural and appealing alternative. It has been shown to provide benefits to fish, such as enhanced pigmentation, reproduction, disease resistance, survival, and growth [612]. Moreover, natural AX is being explored as a sustainable and environmentally friendly feed additive that promotes the health and productivity of livestock and agriculture [625,918,919]. With the prohibition of antibiotics, AX has emerged as a potential green natural alternative due to its natural, non-residual, antioxidant, and immune-protective properties [626].

Microalgae and their bioactive compounds are gaining attention for their wide range of potentials in various applications, including nutraceutical, pharmacological, medicinal, bio-packaging, and biofuel production. Global climate change, particularly global warming, is another pressing issue with significant health impacts on humans [920] and animals [921], affecting productivity and the economy. The increase in deadly heatwaves due to climate change is projected to affect around 30% of the world’s population for at least 20 days per year, and this is expected to rise by 74% by 2100 without effective interventions [922]. Urgent action is needed to transition to a more sustainable lifestyle and production chain. AX has demonstrated the ability to enhance resilience against heat stress, including heat-induced muscle injuries, heat-induced reproduction, and developmental failures [615,619,621,631,638,923,924,925].

The European Commission has proposed an action plan to fully harness the potential of algae in Europe for healthier diets, lower CO_2_ emissions, and addressing water pollution. The plan includes 23 actions aimed at improving the algae business environment, increasing consumer awareness and acceptance of algae and algae-based products, and bridging knowledge, research, and technology gaps [926]. The Organization for Economic Co-operation and Development (OECD) also emphasizes the need for proper actions to avoid serious impacts on the climate, natural resources, ecosystems, and ultimately, our health and well-being [927].

As the deadline for the Sustainable Development Goals (SDGs) approaches in 2030, it is crucial to raise awareness and focus on algae and algae-derived bioactive compounds, such as AX, as sustainable and health-promoting solutions. As of September 2022, a total of 87 microalgae-producing enterprises based in 17 European countries have been identified [928]. The majority of producers utilize photobioreactors (PBR) for microalgae cultivation, which offer advantages such as reduced infection risks, minimal environmental pollution, and scalability without extensive land or maritime space requirements. 

Products based on microalgae are gaining popularity as they contribute to a greener planet and offer health-promoting effects. The intensive research on microalgae-derived compounds such as AX has contributed to the rapid expansion of their applications. AX, for example, shows potential for combating the spread of infectious diseases with pandemic potential. It has been shown to prevent oxidative damage, attenuate inflammatory responses, and regulate multiple signaling pathways involved in inflammation [130,929]. 

The global AX market is projected to experience significant growth, driven by new and existing applications in the nutraceutical and pharmaceutical sectors, with a projected compound annual growth rate (CAGR) of 16.2% from 2022 to 2027 [1,2,3]. Natural AX offers fertile ground in various areas, including sustainable health promotion, environmental protection, and innovative food and feed production. It is poised to contribute to the transformation of living conditions on Earth.

## 6. Conclusions

AX is a carotenoid with essentially no provitamin A activity in mammals and is found as a pigment in a wide variety of organisms, such as marine foods. From an industrial perspective, AX can be produced in a sustainable manner from unutilized resources such as algae and seafood residues. 

Nearly 100 years after its discovery, AX has been found for its diverse usefulness. Consequently, its use in applications other than pigments has been rapidly growing. Initially, AX was used for the preventative and therapeutic treatment of ROS-related inflammatory pathologies in humans because of its strong antioxidant properties. 

Recently, however, as its mechanism of action has been gradually elucidated, it has been found to have additive and synergistic benefits beyond its simple antioxidant function. Therefore, AX is one of the promised beneficial blessings of nature that could be one of the solutions to the social problems that humanity will face in the future.

## Figures and Tables

**Figure 1 marinedrugs-21-00514-f001:**
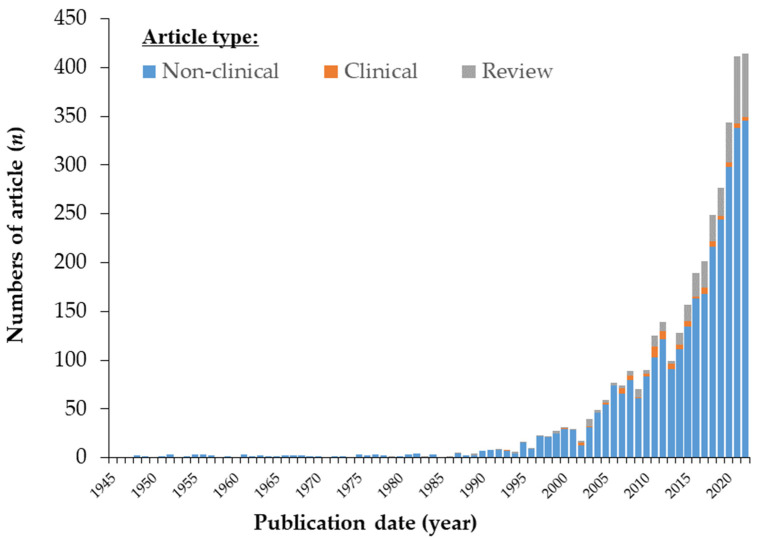
Number of scientific papers on astaxanthin (AX) by the end of 2022. Number of articles in PubMed (https://pubmed.ncbi.nlm.nih.gov/, accessed on 30 June 2023) by year. The keyword query “astaxanthin” was used to search the PubMed database. Note that “clinical trial” and “review” articles were selected as article type tags on PubMed, resulting in differences from the actual number of clinical reports shown in Section 3.3.1.

**Figure 2 marinedrugs-21-00514-f002:**
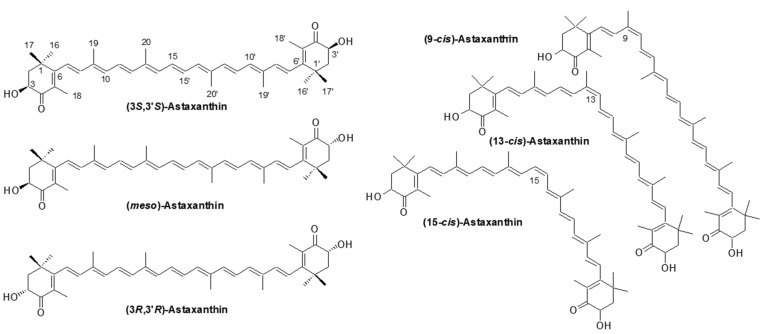
Astaxanthin; structure, optical isomers and major geometric isomers.

**Figure 3 marinedrugs-21-00514-f003:**
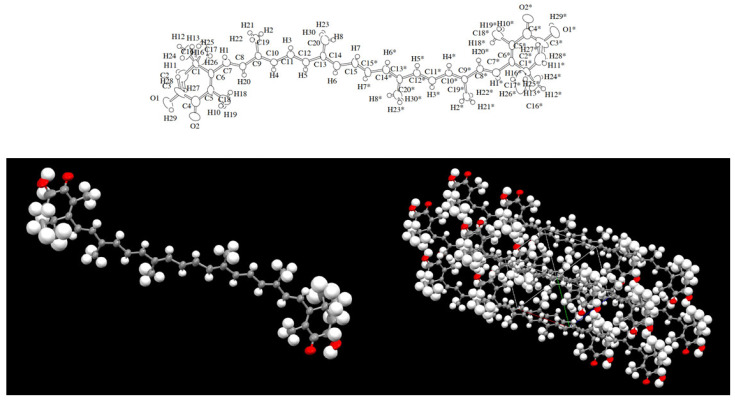
Oak Ridge Thermal Ellipsoid Program (ORTEP) diagram of a single molecule and the crystal structure obtained from a crystalline sample of all-trans astaxanthin. This figure was prepared based on reference [18].

**Figure 4 marinedrugs-21-00514-f004:**
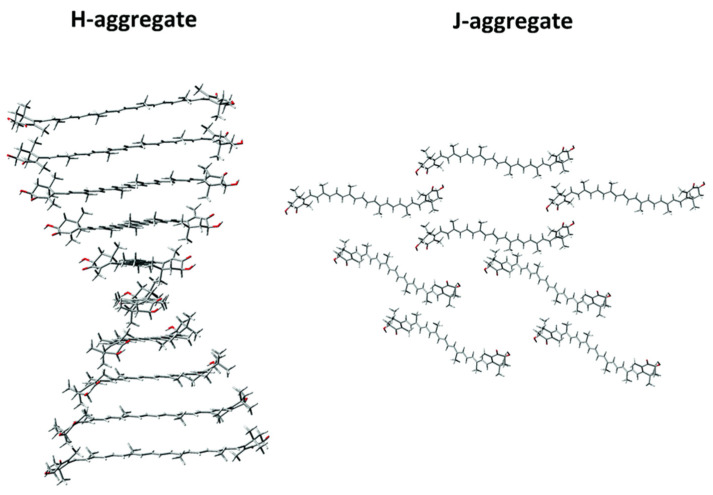
Predicted representative forms of astaxanthin aggregates in hydrated polar solvents. Reproduced from Ref. [74] with permission from the Royal Society of Chemistry.

**Figure 5 marinedrugs-21-00514-f005:**
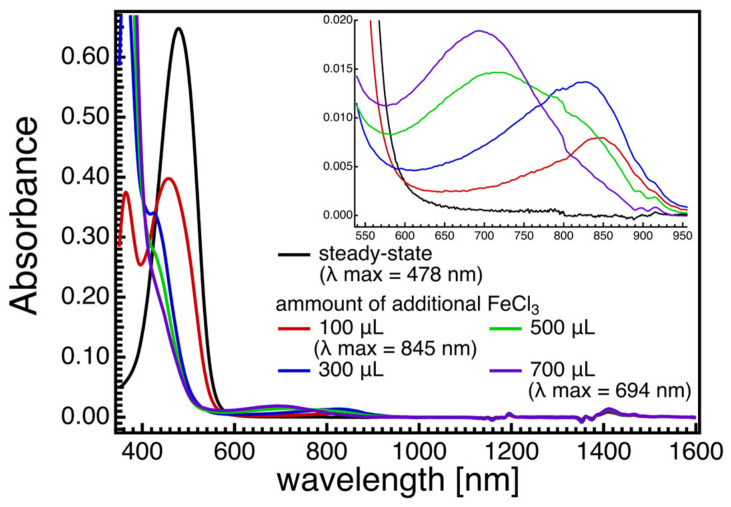
Steady-state absorption spectra of astaxanthin (AX) in acetone when different amounts of FeCl_3_ solutions (1 mM acetone solution) were added. According to the addition of the FeCl_3_ solution, the intensity of S_0_ → S_2_ absorption of AX around 500 nm decreases, and the new absorption bands appear in the 600–950 nm spectral region (see inset of Figure 5). With the small amount of FeCl_3_ solution added, the absorption band that is associable to the radical cation of AX appears to peak around 850 nm. With more addition of the FeCl_3_ solution, the radical cation of AX transforms to dication, peaking around 700 nm. The absorption band below 400 nm is due to the absorption of FeCl_3_.

**Figure 6 marinedrugs-21-00514-f006:**
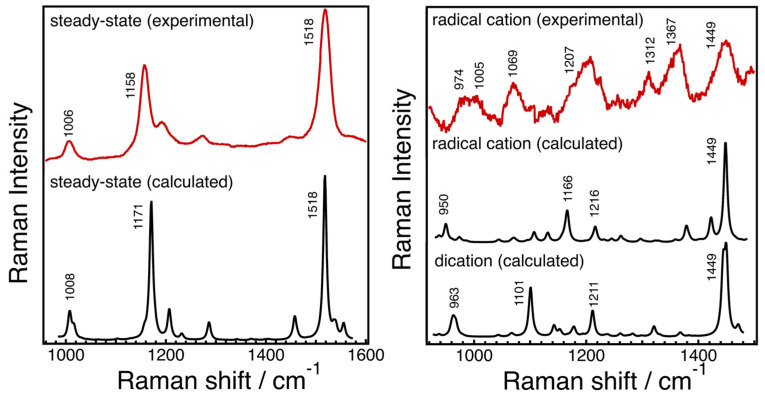
The steady-state resonance Raman spectrum of astaxanthin (AX) in acetone recorded with 532 nm excitation laser light at room temperature (solid red line in the left panel) and the resonance Raman spectra of radical species of AX recorded with 808 nm excitation laser light at room temperature (solid red line in the right panel). The results of DFT calculations of the ground (S_0_) species, radical cation, and dication of AX are also shown in each panel (solid black lines).

**Figure 7 marinedrugs-21-00514-f007:**
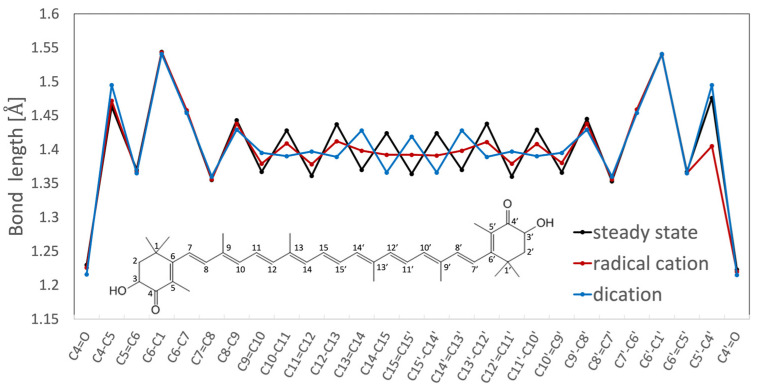
Comparison of the bond lengths of the ground (S_0_) state, radical cation, and dication of astaxanthin predicted theoretically by DFT calculations.

**Figure 8 marinedrugs-21-00514-f008:**
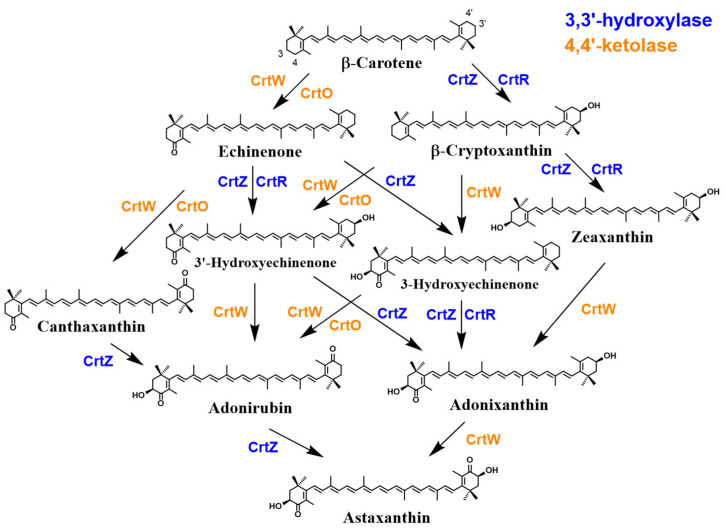
Biosynthetic pathway of astaxanthin from β-carotene with bacterial enzymes. β-Carotene (β-carotenoid) 3,3′-hydroxylase and 4,4′-ketolase are shown with blue and orange letters, respectively. In this figure, the maximal levels of catalytic activities are shown concerning CrtR and CrtO. Generally, the catalytic activity from adonixanthin to astaxanthin is weak, even with CrtW. This pathway is based on bacterial enzymes. However, the functions of green algal BHY and BKT are the same as those of CrtZ and CrtW, respectively.

**Figure 10 marinedrugs-21-00514-f010:**
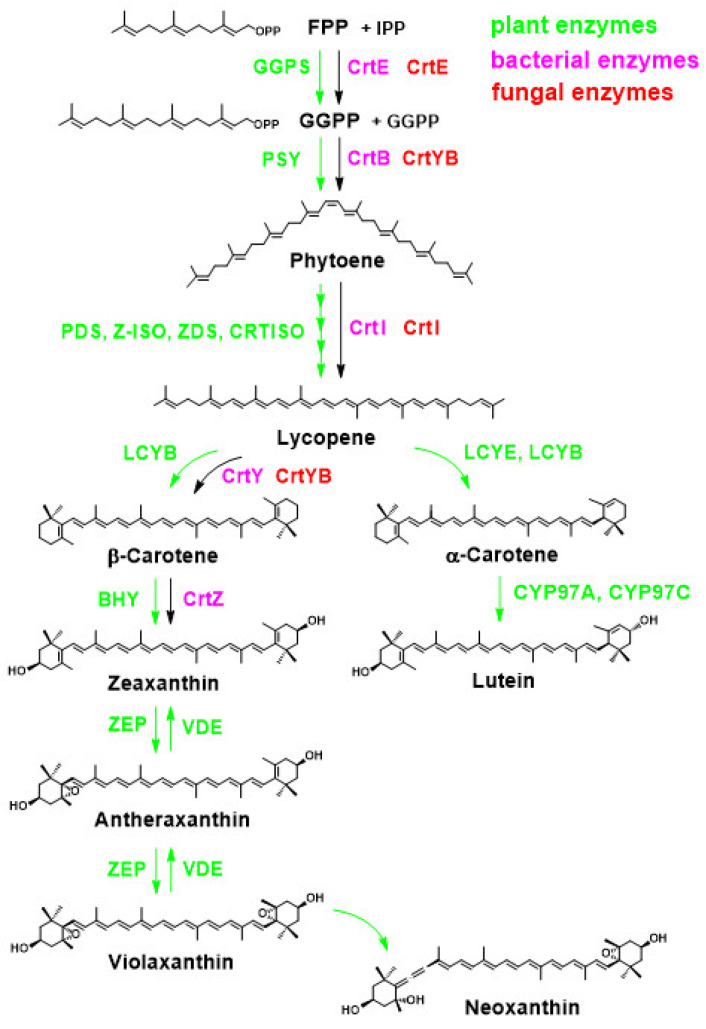
Biosynthetic pathway of carotenoids in the leaves of higher plants. Plant-type enzymes are shown with green letters, while bacterial enzymes and fungal enzymes that can catalyze in this pathway are written in pink and red, respectively.

**Figure 11 marinedrugs-21-00514-f011:**
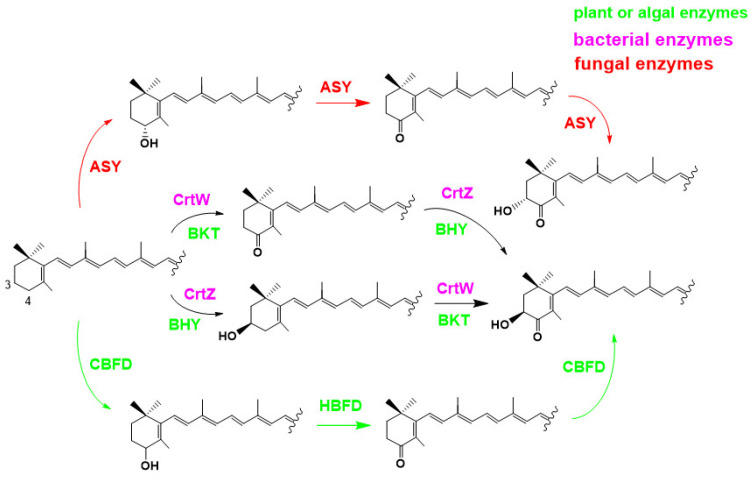
Conversion routes from β-ring to 3-hydroxy-4-keto-β-ring in AX-biosynthesizing organisms. ASY, also called CrtS, from *Xanthophyllomyces dendrorhous*; CrtW and CrtZ, from bacteria; *BKT* and BHY, from green algae; CBFD and HBFD, from *Adonis aestivalis*.

**Figure 12 marinedrugs-21-00514-f012:**
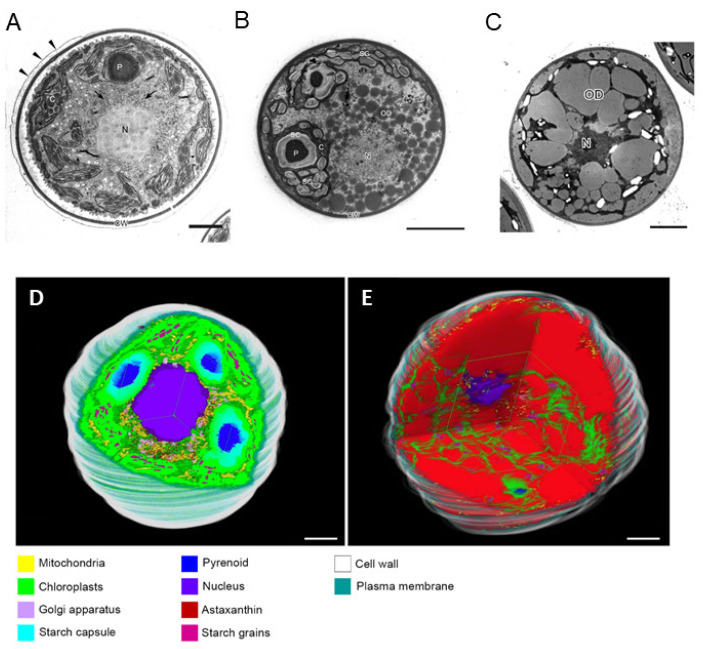
Cell morphology of *Haematococcus* algae at each stage. Transmission electron micrographs of (**A**) a green palmelloid cell, (**B**) an intermediate palmelloid cell, and (**C**) a mature aplanospore of *Haematococcus* algae during encystoment. In green palmellod cells, the cell wall is surrounded by an extracellular matrix (arrowheads). Arrows indicate AX granules. Cut-away image of 3D TEM images of whole cell of a green palemellod (**D**) and a cyst cell (**E**). This figure was adapted from the ref. [247,250] under the terms of the Creative Commons Attribution License. C, chloroplast; CW, cell wall; N, nucleus; OD, oil droplet; SC, starch capsule; SG, starch grain; P, pyrenoid. Scale bars: 5 µm.

**Figure 13 marinedrugs-21-00514-f013:**
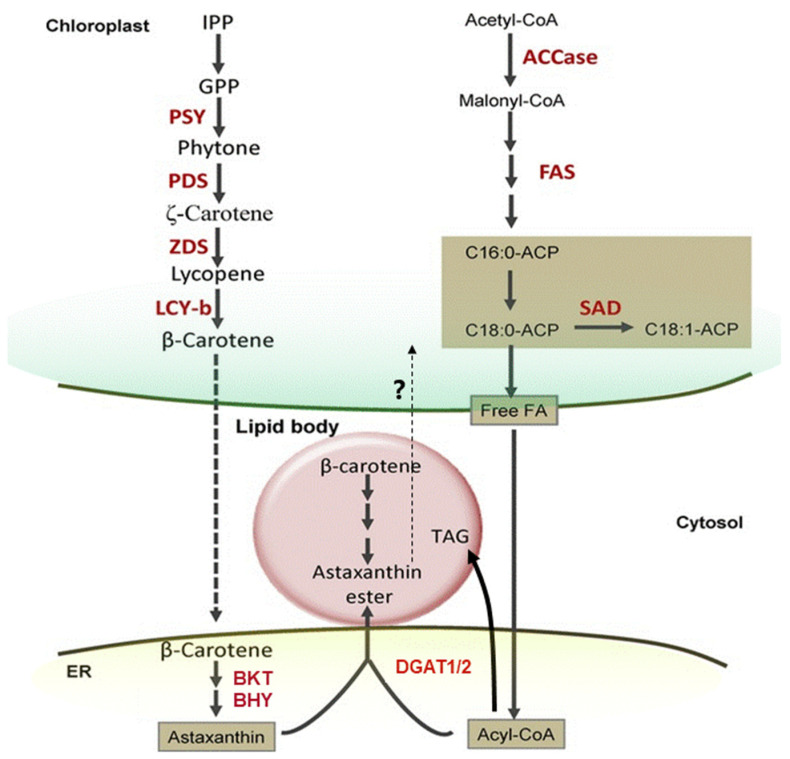
Pathways and localization of astaxanthin biosynthesis and esterification in *Haematococcus* algae. ACCase, acetyl CoA carboxylase; DGAT, diacylglycerol acyltransferase; FAS, fatty acid synthase; FA, fatty acid; SAD, stearoyl acyl carrier protein desaturase; TAG, triacylglycerol. Other enzyme abbreviations are listed in the main text. This figure was reproduced from ref. [410] with the permission of the publisher.

**Figure 14 marinedrugs-21-00514-f014:**
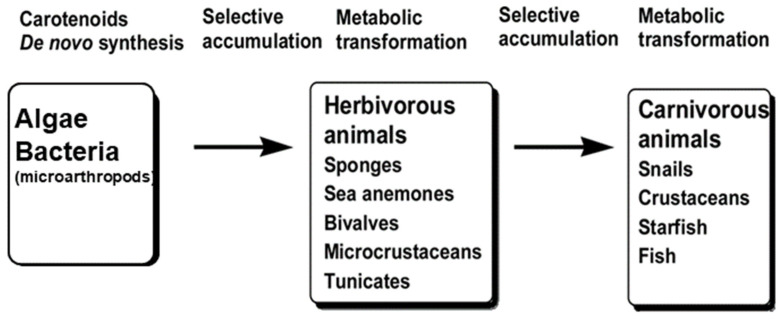
Relationship between food chain and metabolic conversion to astaxanthin in animals [280].

**Figure 15 marinedrugs-21-00514-f015:**
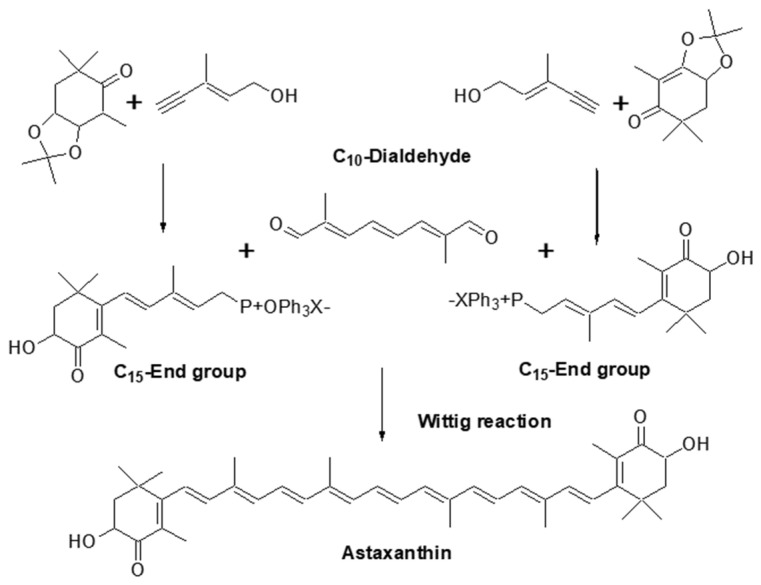
Industrial synthesis route for astaxanthin.

**Figure 16 marinedrugs-21-00514-f016:**
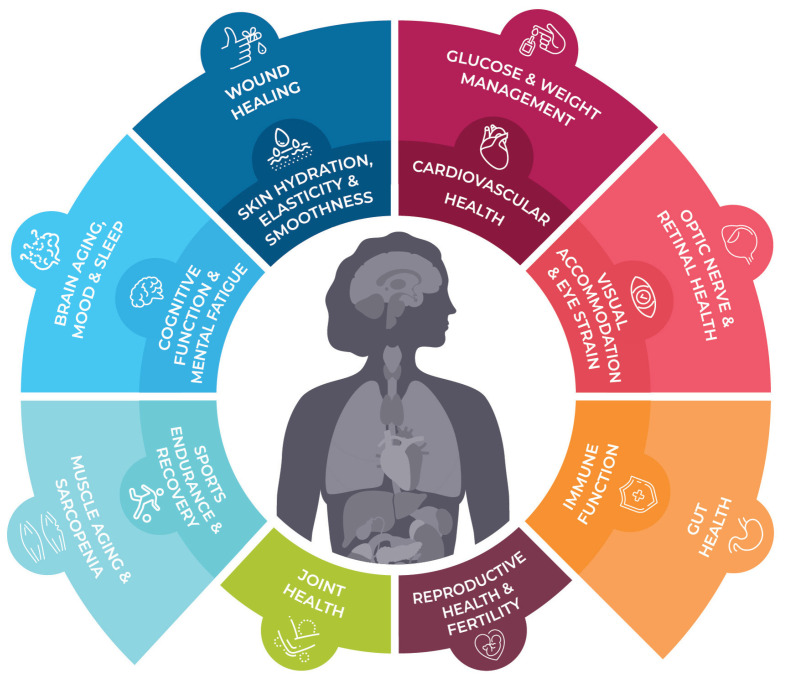
Summary of health benefits of astaxanthin in humans The inner circle summarizes the health benefits with substantial clinical evidence. The outer circle includes the areas with preliminary but promising pre-clinical and clinical data, suggesting potential future directions for clinical research on the beneficial effects of natural astaxanthin. References are discussed in more detail in Section 3, Table 4, and Section 5.

**Table 5 marinedrugs-21-00514-t005:** Human safety study of AX derived from *Haematococcus* algae.

Administration	Subject	Dose/ Number of Subjects	Duration	Outcomes	Ref.
Long-term	Healthy Japanese (Mean 33 yrs.)	6 mg/day (*n* = 15)	12 weeks	✓ No abnormalities in physical examination ✓ No abnormalities in blood tests and urinalysis ✓ No incidence of some kind of adverse event and side effect (subjective symptom/medical examination)	[870]
	Healthy Japanese (Mean 39 yrs.)	9 mg/day (*n* = 15)	12 weeks	✓ No abnormalities in physical examination ✓ No abnormalities in blood tests and urinalysis ✓ No incidence of some kind of adverse event and side effect (subjective symptom/medical examination)	[871]
Overdose	Healthy Japanese	20 mg/day (*n* = 16)	4 weeks	✓ No abnormalities in blood pressure measurement ✓ No abnormalities in blood tests ✓ No incidence of some kind of adverse event and side effect	[774]
	Healthy Japanese	30 mg/day (*n* = 10)	4 weeks	✓ No abnormalities in physical examination ✓ No abnormalities in blood tests and urinalysis ✓ No abnormalities in ophthalmic examination ✓ No incidence of some kind of adverse event and side effect (medical examination)	[872]
	Healthy Japanese (Mean 42 yrs.)	30 mg/day (*n* = 11) Placebo (*n* = 12)	4 weeks	✓ No abnormalities in blood pressure measurement ✓ No abnormalities in blood tests ✓ No abnormalities in ophthalmic examination ✓ No incidence of some kind of adverse event and side effect	[873]
	Healthy Japanese (Mean 41 yrs.)	45 mg/day (*n* = 15) Placebo (*n* = 7)	4 weeks	✓ No abnormalities in physical examination ✓ No abnormalities in blood tests and urinalysis ✓ No abnormalities in ophthalmic examination ✓ No incidence of some kind of adverse event and side effect (medical examination)	[874]

**Table 6 marinedrugs-21-00514-t006:** Safety confirmed in clinical trials: dosage and duration.

Dose ***^,^#*	*Repeated Intake Period * (Weeks)*								
	2	3	4	6	8	12	16	20	≥24
*<1 mg/day*			[737]	[767]	[710]	[791]			[770,777]
*2 mg/day*				[746]		[703]	[757]		
*3 mg/day*			[745]		[739]	[730,800]			
*4 mg/day*	[39,788]		[751,778,786,796]		[713,714,740,744,792]	[702,712,714,717,782,787]			[758,759]
*6 mg/day*	[43,44,706]		[42,45,47,684,711,719,722,752,753,756,760]	[742,749,779]	[652,743,754,798]	[653,705,795]		[741]	
*8 mg/day*			[651,781]	[724]	[693,701,732,768,793]	[733]			
*9 mg/day*			[46]	[748]					
*10 mg/day*						[691]			
*12 mg/day*	[727]		[40,41,685,704,763,765,780,784]		[700,725,729,766,771,776,785,799]	[692,694,721,723,728,764,774,790,794,797,801]	[738]		[695,697,755,761,772]
*16 mg/day*						[649,735,775,803]			
*18 mg/day*						[773]			
*20 mg/day*		[698]	[715,716,774,812]			[699]	[762]		
*24 mg/day*			[811]			[718]			
*40 mg/day*		[726]	[813]			[731]			

All information is based on Table 4. * For repeated intake durations not listed in this table, the nearest shorter intake duration listed in the table was adopted; ** For dosages not listed in the table, the dose on the nearest smaller listed in the table was adopted; # AX concentrations were shown in free form.

## Data Availability

All the data underlying the results is available as part of the article, and no additional source data are required.

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
