# Peer review of "Astaxanthin: Past, Present, and Future"

_marinedrugs, 2023, doi:10.3390/md21100514_

Round 1

Reviewer 1 Report

This review encompasses a comprehensive discussion of astaxanthin as a valuable functional and bioactive compound. While resembling a chapter of a book, it provides significant assistance to astaxanthin research and holds merit as a swift publication in Marine Drugs.    

Author Response

The authors express their utmost gratitude to the esteemed Reviewer for his/her invaluable and supportive comments. The primary objective behind crafting such an extensive review was indeed to furnish a comprehensive representation of the scientific and knowledge advancements concerning astaxanthin, with the aspiration of serving as a wellspring of inspiration for future research endeavors.

Reviewer 2 Report

This review of astaxanthin is very long and covers everything from its discovery, antioxidant properties, biosynthesis in living organisms and its genes, bioactivity, toxicity, bioavailability, industrial uses, traditional food culture, and legends and myths.

It must have been a lot of work for the authors to write such a long book, but it is hard for the readers to read it.

The reader is most likely interested in the science part of astaxanthin. The reviewer personally doubt that it is necessary to describe myths, food culture, laws, social requirements, etc. in the Marin Drugs. It seems that it would be better for the reader if those were omitted and the volume were more reduced and easier to read.

Also, the results of human trials, which the reader may want to know, are listed in Table 4, but it is hard for the reader to read all of Table 4. The reviewer wished the authors had included a summary and important points in the text.

The reviewer had to read the entire manuscript within a short period of time and was not able to scrutinize all of what was written, though not very much. The reviewer will only point out some of the areas of discrepancy as follows.

The authors link the production of AX by Hematococcus to environmental issues, which (to the reviewer) appear to be described as sustainable and environmentally friendly.

Section 2.2.5 states that the maximum content of AX per 100 g is 9,800 mg for hematococcus algae. This is only 10% of 100 g, even if it is about 10 g. What about the remaining 90%? If this remaining 90% is not effectively utilized, is it not good for the environment? This is because if it is disposed of, whether it is burned or decomposed by decomposers in nature, it will emit CO2.

Throughout the entire article, most of the information is on astaxanthin alone, so the reader has no idea whether the effects and benefits are specific to astaxanthin, carotenoids in general, or how they compare to other carotenoids. It may be possible to find out by examining all the references cited, but there are so many references (922 in total) that it would be quite difficult for the average reader to check them all.

In section 4.1.1 it is stated that "the bioavailability of AX appears to be relatively low", how low is this compared to other carotenoids? While the bioavailability is low, producing large quantities and consuming large quantities by humans may be good for human health, but is it also good for the global environment?

 For example, lutein is relatively bioavailable, is present in human blood, and accumulates specifically in the retina. Naturally, plants produce lutein both in the ocean and on land, so there seems to be no need for artificial mass production. Is astaxanthin still more beneficial in many ways than other carotenoids such as lutein?

 It seems to me that these should be mentioned by the authors in the text.

Author Response

The authors extend their sincere thanks to the esteemed Reviewer for dedicating his/her valuable time and providing insightful comments that have significantly contributed to enhancing the quality of the manuscript. These comments have been thoughtfully addressed and incorporated into the revised version of the manuscript.   

  • While the authors appreciate the suggestion of potentially dividing the review into two parts—one with a focus on basic science, including clinical data, and the other more oriented towards commercial and market aspects, encompassing nature, food culture, myths, and social considerations—they firmly believe that this comprehensive review offers a unique opportunity to encapsulate a wide range of information about astaxanthin. To facilitate readers' navigation and accessibility, the authors propose the inclusion of a table of contents immediately following the abstract, ensuring a user-friendly viewing experience.
  • The authors firmly believe that the discussions within this review pertaining to the history, mythology, culture, and social requirements hold considerable value in advancing scientific knowledge. As Professor William F. McComas from the University of Arkansas aptly noted, "there are historical, cultural, and social influences on the practice and direction of science." To gain a comprehensive understanding and anticipate the trajectory of research within specific domains, it is imperative to grasp the intricate interplay and influence of these factors on the evolution of scientific inquiry surrounding that particular topic. Science invariably adapts and evolves to better align with the needs and interests of populations, as is the case here, with a particular emphasis on sustainability.
  • Table 4 has been reorganized for clarity. Since the new Table 4 has been reorganized by outcomes, the original Table 4 is uploaded separately as Supplementary Table 4S because of duplication in the number of studies.
  • The highlights from Table 4, which serve as a summary of clinical findings, have been seamlessly integrated into the body of text within Section 3.3.1 and Section 5.1 for improved coherence and clarity.
  • The current commercial sources of lutein, such as marigold flowers and fruits and vegetables, may not be sustainable in the long term. Similar concerns about sustainability apply to other carotenoids derived from fruits and vegetables, including zeaxanthin. These concerns revolve around issues such as water and land use, the use of pesticides and fertilizers, and the environmental impact of traditional agricultural practices.

It's noteworthy that lutein can also be found in microalgae; however, as of now, there are no industrial facilities established for large-scale production of lutein from microalgae. A recent study by Muhammad et al. ("Sustainable production of lutein—an underexplored commercially relevant pigment from microalgae. Biomass Conv. Bioref. (2022)") highlights the potential for sustainable lutein production from microalgae. Microalgal cultivation offers several environmental advantages over traditional plants, including higher carbon sequestration capacity, reduced water footprint, and the elimination of pesticide use.

These insights underscore the significance of exploring alternative sources for valuable compounds like lutein to promote sustainability in the production of essential nutrients and pigments.

Indeed, both lutein and astaxanthin derived from microalgae offer the potential as sustainable and valuable active compounds, with the added possibility of complementary health benefits, particularly for vision and ocular health.

A study conducted by Cristaldi et al. ("Comparative Efficiency of Lutein and Astaxanthin in the Protection of Human Corneal Epithelial Cells In Vitro from Blue-Violet Light Photo-Oxidative Damage. Appl. Sci. 2022, 12, 1268.") suggests that when lutein and astaxanthin are used in combination, they may work synergistically to protect human corneal epithelial cells from the damaging effects of short (lutein) and longer (astaxanthin) wavelengths of light. This is particularly relevant in today's environment, where LED screens and solar light emit various wavelengths, some of which can be harmful to the eye.

Such research highlights the potential for a holistic approach to eye health by harnessing the benefits of multiple carotenoids like lutein and astaxanthin to provide comprehensive protection against light-induced oxidative damage to ocular cells.

The sustainability potential of microalgae as a source of astaxanthin and other carotenoids is indeed noteworthy. One of the key advantages of utilizing microalgae is that it represents a sustainable source that can yield multiple valuable nutrients and bioactive compounds with various health-promoting properties beyond astaxanthin alone. In the case of Haematococcus biomass, it is consumable by humans and contains a spectrum of beneficial nutrients and bioactives.

A significant aspect of this sustainable approach is the minimization of waste or residue production. This contrasts with some traditional production methods, which may generate substantial waste or byproducts. The utilization of microalgae, particularly Haematococcus, underscores the potential for a more eco-friendly and resource-efficient means of harnessing a wide array of health-promoting phytonutrients while minimizing environmental impact.  

  • In general, carotenoids, including astaxanthin and lutein have low bioavailability due to their molecular structure. Technological advancements are being applied to enhance bioaccessibility and bioavailability of health-promoting carotenoids, including oil-in-water emulsions, nano-emulsions, microencapsulation, and liposomes. In general, carotenoids, including astaxanthin and lutein, often exhibit low bioavailability due to their specific molecular structures. To address this limitation and improve the bioaccessibility and bioavailability of these health-promoting carotenoids, researchers are actively applying various technological advancements. These methods include the use of oil-in-water emulsions, nano-emulsions, microencapsulation, and liposomes.

    It's worth noting that while astaxanthin and lutein, along with other carotenoids, each offer distinct benefits, astaxanthin particularly stands out due to its exceptional antioxidant capacity. This property sets astaxanthin apart and makes it a noteworthy phytonutrient with the potential to confer a wide range of health advantages.

Reviewer 3 Report

This study is about the astaxanthin, its chemistry, biological activity, and importance in terms of human health were discussed.

An incredibly detailed literature review was performed and very well discussed.

The topic is not very novel since there are a number of many research and review articles about astaxanthin. However, the authors have approached from a quite different perspective. Hence, it is worth for publication.

I have some minor comments:

The abstract of this study does not reflect the scope and novelty. It is too general, and I think it should be revised. It should be re-written more specially.

Section 2.1 is well designed; however, it would be much better to insert another section for Astaxanthin fatty acid esters since they also exist as esters not in free form and it is not easy to saponify them as their structure can be degraded upon saponification.

Author Response

The authors express their sincere gratitude to the esteemed Reviewer for his/her thorough examination of the manuscript and constructive feedback. The comments provided have been thoughtfully incorporated into the manuscript, enhancing its overall quality and clarity.

  • The Abstract section has been revised to more accurately reflect the significance and novelty of the review work.

  • Section 2.1 has been restructured, and a new section has been added in response to the valuable comment from the Reviewer.